# Students Rather Than Experts: A New AI for Education Pipeline to Model More Human-like and Personalised Early Adolescences

## Abstract

The capabilities of large language models (LLMs) have been applied in expert systems across various domains, providing new opportunities for AI in Education (AI4Education). Educational interactions involve a cyclical exchange between teachers and students. Current research predominantly focuses on using LLMs to simulate teachers, leveraging their expertise to enhance student learning outcomes. However, the simulation of students, which could improve teachers' instructional skills, has received insufficient attention due to the challenges of modeling and evaluating virtual students. This research poses the question: "*Can LLMs be utilized to develop virtual student agents that mimic human-like behavior and individual variability?*" Unlike expert systems focusing on knowledge delivery, virtual students must replicate learning difficulties, emotional responses, and linguistic uncertainties. These traits present significant challenges in both modeling and evaluation. To address these issues, this study focuses on language learning as a context for modeling virtual student agents. We propose a novel AI4Education framework, termed **SOE** (**S**cene - **O**bject - **E**valuation), to systematically construct **LVSA** (**L**LM-based **V**irtual **S**tudent **A**gents). By curating a dataset of personalized teacher-student interactions with various personality traits, question types, and learning stages, and fine-tuning LLMs using LoRA, we conduct multi-dimensional evaluation experiments that integrate both subjective human evaluations and objective metrics. Specifically, we: (1) develop a theoretical framework for generating LVSA; (2) integrate human subjective evaluation metrics into GPT-4 assessments, demonstrating a strong correlation between human evaluators and GPT-4 in judging LVSA authenticity; and (3) validate that LLMs can generate human-like, personalized virtual student agents in educational contexts, laying a foundation for future applications in pre-service teacher training and multi-agent simulation environments.

## 1 Introduction

In recent years, large language models (LLMs) have achieved breakthroughs in natural language processing (NLP) and knowledge generation, enabling advancements in AI-driven educational applications (Ge et al. (2024); Liu et al. (2024b); Hong et al. (2024); Latif et al. (2023); Chen et al. (2024a); Jury et al. (2024)). While LLMs show significant potential as virtual tutors in generating high-quality content and personalized recommendations, their application in pre-service teacher training remains underexplored (Lee et al.; Wang et al. (2024)). Traditional pre-service training faces constraints like limited teaching environments and insufficient practical experience (Loewenberg Ball & Forzani (2009); Zeichner (2012); Ward et al. (2018); Sparks (2011)).

Virtual students offer cost-effective, flexible practice opportunities (Dotger et al. (2015; 2010); Shaughnessy & Boerst (2018)), allowing teachers to practice techniques in controlled environments (Kervin et al. (2006); Lugrin et al. (2019)). However, existing models are often pre-scripted, resulting in predictable behaviors that fail to capture the complexity of human emotional responses and cognitive variability, limiting their effectiveness in dynamic settings (Badiee & Kaufman (2015)).

This gap in realistic virtual student modeling raises a critical question: "*Can LLMs create human-like, personalized virtual student agents for teacher training?*" Unlike expert systems focused on

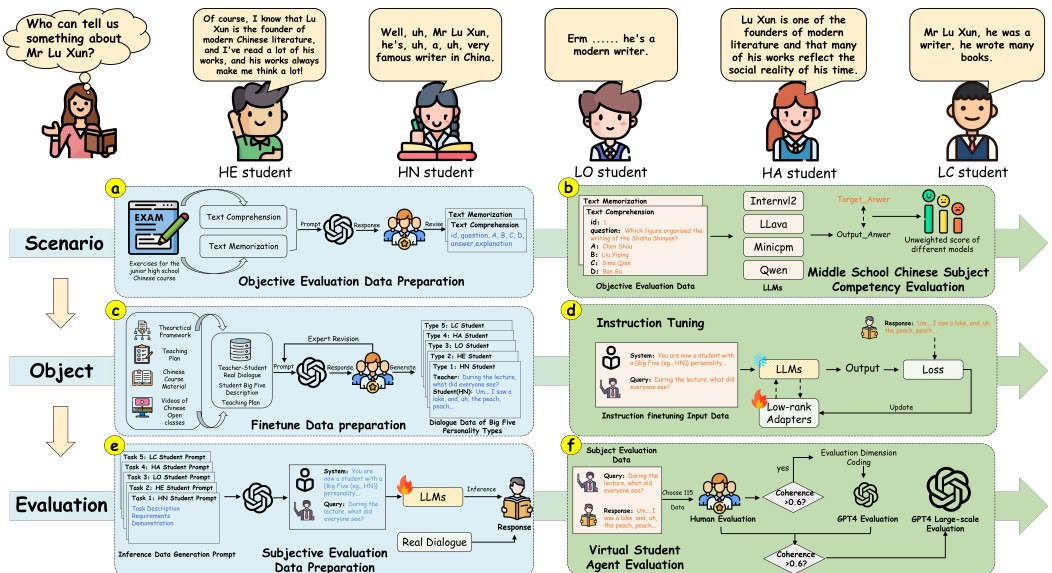

Figure 1: An overview of the SOE pipeline, consisting of three steps: **Scene**, which explores the scenario capabilities of foundation LLMs (Section 3); **Object**, which builds LLM-based virtual student agents with different personalities through fine-tuning (Section 4); and **Evaluation**, which assesses the performance of these agents using human and GPT-4 evaluations (Section 5). **HE**, **HN**, **LO**, **HA**, and **LC** refer to **High Extraversion**, **High Neuroticism**, **Low Openness**, **High Agreeableness**, and **Low Conscientiousness**, respectively, in the Big Five personality traits.

knowledge accuracy, virtual student agents must simulate a broader range of human behaviors, including cognitive challenges, emotional reactions, and language production issues like hesitations and self-corrections (Christensen et al. (2011); Delamarre et al. (2021); Shernoff et al. (2018); Kelleci & Aksoy (2021)). For early adolescents, cognitive load can lead to linguistic struggles and emotional states like frustration or self-doubt. Thus, LLM-based virtual students must demonstrate both knowledge and authentic social interactions, presenting unique modeling and evaluation challenges, especially due to the subjectivity in assessing behavior realism.

This study addresses these challenges by modeling virtual student agents in the context of language learning. We evaluate the capabilities of four foundational models to simulate student-like behaviors in language tasks. Next, we propose a comprehensive framework for generating **LLM-based virtual student agents (LVSA)**, facilitating personalized teacher-student interaction datasets. Using LoRA fine-tuning, we adapt the LLMs to embody specific traits, enhancing pre-service teacher training with realistic, personalized interactions. To ensure a thorough evaluation, we conduct multi-dimensional evaluation experiments that integrate both subjective human evaluations and objective metrics, providing a holistic view of the virtual agents' capabilities. Finally, these evaluations are conducted to assess the feasibility and effectiveness of generating human-like and personality virtual agents, laying a foundation for their potential applications in educational settings.

This research introduces the **"Scene - Object - Evaluation" (SOE)** framework, a novel AI4Education pipeline for constructing LVSA (see Figure 1). By addressing modeling and evaluation complexities, the SOE framework provides a robust foundation for applications in pre-service teacher training, multi-agent simulations, and educational AI systems. The main contributions are: (1) **Theoretical framework for LVSA**: A comprehensive framework for constructing virtual student agents with scientific rigor and feasibility. (2) **Subjective evaluation metrics integration**: Incorporating human subjective metrics into GPT-4's evaluation pipeline, aligning with human assessments of virtual student authenticity. (3) **LVSA validation**: Extensive multi-dimensional experiments confirming the feasibility of creating human-like, personalized LVSA for educational AI.

## 2 RELATED WORK

**Virtual student development and related technologies:** The development of virtual students for pre-service teacher training has evolved through three approaches: role-playing, programmatically pre-set models, and LLM-based virtual students. Role-playing offers realism but lacks scalability

and flexibility due to resource constraints (Kersh (1963); Colwell (2013); Frederick et al. (2010); Shapira-Lishchinsky (2015); Dalgarno et al. (2016)). Programmatically pre-set models provide predictable, repeatable responses but lack adaptability to real-world classroom dynamics (Christensen et al. (2011); Delamarre et al. (2021); Shernoff et al. (2018); Kelleci & Aksoy (2021)). Recently, LLM-based virtual students, such as those based on GPT-4, introduced greater flexibility in natural language interactions (Achiam et al. (2023); Li et al. (2024c); Zhang et al. (2024); Lee et al.; Markel et al. (2023)). However, these models still struggle to replicate the complex cognitive and emotional behaviors of real students. This work addresses the gap by moving beyond basic prompt engineering to develop LVSA with human-like behaviors, cognitive challenges, and emotional responses. These virtual students facilitate authentic, dynamic, and adaptive interactions, enhancing the effectiveness of pre-service teacher training. For more details, see Appendix A.1.

**LLMs' main capabilities to support AI4Education:** Recent advancements in LLMs have enhanced capabilities in natural language processing, including understanding, generation, reasoning, transfer learning, and multimodal processing (Brown (2020); Achiam et al. (2023); Guo et al. (2023); Li et al. (2024b); Huang et al. (2024); Ahuja et al. (2023); Kafle & Kanan (2017)). Models like GPT-4, LLaMA, and PaLM have demonstrated remarkable performance, often surpassing human levels in various benchmarks (Touvron et al. (2023); Chowdhery et al. (2023); Wang et al. (2019); Clark et al. (2018)). However, while these models excel in language tasks, they face challenges in replicating the nuanced cognitive and emotional behaviors typical of real students, as current applications mainly focus on knowledge delivery (Achiam et al. (2023); Li et al. (2024c); Zhang et al. (2024); Lee et al.; Markel et al. (2023)). This study aims to leverage these capabilities to create personalized LVSA that simulate realistic learning challenges and emotional responses, enhancing the authenticity of pre-service teacher training. For more details, see Appendix A.2.

**Representative AI4Education applications:** Educational technology companies and universities have developed LLM-based products to enhance student learning, such as Duolingo Max, Khan Academy's Khanmigo, and TAL's MathGPT (Duolingo (2023); Khan (2023); TAL (2023a); squirrelai (2023); Chen et al. (2021)). Numerous benchmark datasets and evaluation tools, including C-Eval, SciQAG, and Mukea, assess LLMs' effectiveness in education (Huang et al. (2024); Wan et al. (2024); Ding et al. (2022)). Despite these advancements, most research prioritizes student learning over enhancing teachers' pedagogical skills (Lee et al.; Wang et al. (2024)). LLM applications focus on automating educational tasks, leaving a gap in teacher development support (García-Méndez et al. (2024); Yue et al. (2024); McNichols et al. (2023); Sahai et al. (2023); Han et al. (2023)). This study introduces a novel approach by leveraging virtual students to enhance teachers' skills through realistic, interactive environments, thereby preparing them better for actual classrooms. For more details, see Appendix A.3.

## 3 SCENE: BASIC CHINESE UNDERSTANDING ABILITY IN EDUCATION

This study explores LLMs' ability to simulate realistic student behavior, focusing on early adolescents (ages 10-15), a developmental stage characterized by the transition from concrete to abstract thinking, which poses unique cognitive and linguistic challenges (Hall (1905); Piaget (1952); Elkind (1981); Caskey & Anfara (2014)). Students often rely on intuition and concrete experiences, showing uncertainty in language expression (Nippold (1993)). Modeling this group provides insights into LLMs' limitations in complex educational contexts. Language education, particularly in junior high school Chinese education, emphasizes expression, emotional experience, and value formation, aligning with LLMs' strengths in natural language processing (McNaught et al. (2012); Liu & Lo Bianco (2007); Williams (2016)). This context is ideal for evaluating whether virtual student models can authentically simulate human-like behaviors, including language style, emotional responses, and value-driven interactions (see Appendix B.2.3). A key question arises: *Do foundation models possess basic comprehension abilities for middle school language subjects?* This section addresses this question by evaluating their basic Chinese comprehension abilities.

### 3.1 BASIC CHINESE UNDERSTANDING ABILITY EVALUATION DATASET

**Data source:** The dataset used in this section, named the **Basic Chinese Understanding Ability Evaluation Dataset (BCUAE)**, is sourced from the National Smart Education Platform[1], a resource developed by the Chinese Ministry of Education. It assesses junior high school Chinese language

---

[1]National Smart Education Platform

proficiency through two essential skills: text comprehension (evaluating structure, organization, inference, and emotion) and memorization (testing recall and reproduction). The dataset is regularly updated to align with national curriculum standards.

**Dataset construction:** As shown in Figure 1 (a), the construction process involved three steps: (1) Data preparation—organizing Chinese exercises into units and converting them to PDF format for model input; (2) Prompt design—using structured prompts to guide model tasks; (3) Expert revision—junior high school teachers reviewed and refined model outputs for accuracy. Ultimately, 613 comprehension items and 438 memorization tasks were collected, forming a solid basis for future evaluation and model improvement. Detailed information is provided in Appendix C.1.

## 3.2 FOUNDATION MODELS

This section evaluates the performance of four foundation models in junior high school Chinese tasks, focusing on text comprehension and memorization. Figure 1 (b) illustrates the detailed steps. The selected models—InternVL (Chen et al. (2024b)), LLaVa (Li et al. (2024a)), MiniCPM (Hu et al. (2024b)), and Qwen (Bai et al. (2023))—each demonstrate distinct strengths. Details of these models can be found in Appendix B.1. Comparing these models provides insights into their ability to handle Chinese language content, especially

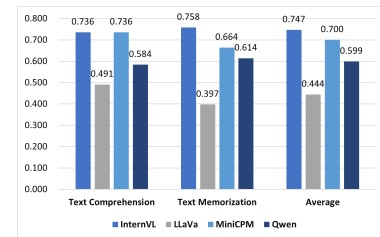

Figure 2: Accuracy for basic Chinese understanding.

within junior high school education. Including multimodal models like InternVL and Qwen prepares for future applications involving visual content recognition, laying a foundation for advanced teaching environments where virtual students can interact with teacher-presented materials.

## 3.3 RESULTS AND ANALYSIS

The evaluation of four LLMs on junior high school Chinese tasks, specifically text comprehension and memorization, revealed significant performance differences (see Figure 2). InternVL and MiniCPM excelled, with comprehension accuracy at 0.736 and memorization at 0.758 and 0.664, respectively. In contrast, Qwen and LLaVa underperformed, scoring 0.584 and 0.491 in comprehension and 0.614 and 0.397 in memorization. Overall, InternVL had the highest average accuracy (0.747), followed by MiniCPM (0.700), while Qwen and LLaVa lagged at 0.599 and 0.444. Their lower performance likely stems from a focus on multimodal tasks, reducing effectiveness in Chinese language processing. These results suggest InternVL and MiniCPM are better suited for Chinese tasks, while Qwen and LLaVa may be more applicable to multimodal contexts.

# 4 OBJECT: VIRTUAL STUDENT CONSTRUCTION

## 4.1 THEORETICAL FRAMEWORK

**Conceptual theory:** Early adolescence (ages 10-15) is marked by significant physiological, cognitive, social-emotional, and moral-spiritual changes, which are crucial for modeling virtual student agents (Hall (1905)). Physiologically, rapid prefrontal cortex development and imbalance between cognitive regulation and emotional responses result in intense emotions and inconsistent behaviors (Blakemore & Choudhury (2006); Giedd (2004)). Cognitively, adolescents transition from concrete to abstract reasoning, influenced by emotions, leading to repetitive and uncertain language (Piaget (1952); Elkind (1981); Nippold (1993)). Socially, heightened self-awareness and sensitivity to peer feedback impact engagement (Raphael & Burke (2012)), while moral development involves internalizing societal norms, often resulting in simplified moral judgments

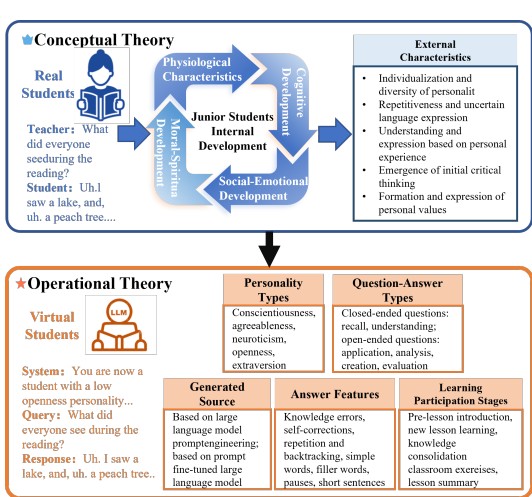

Figure 3: The theoretical framework for LVSA construction, which extends from conceptual theory to operational theory.

(Kellough & Kellough (2008); Kohlberg (1987)). These traits contribute to varied participation, frequent use of filler words, and reliance on personal experiences, highlighting the complexity of simulating realistic behaviors in virtual models (see Appendix B.2.1).

**Operational theory:** To construct LVSA, generating operational and predictable dialogues is crucial for realistic and personalized student simulations. This study introduces key several practical dimensions—question-answer types, personality traits, learning stages, response styles, and generation sources—to guide the modeling process (see Appendix B.2.2). These dimensions enable diverse student performance, mirroring

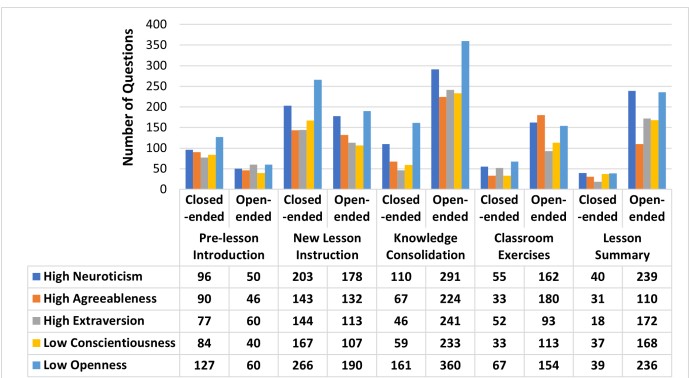

Figure 4: Statistical Information of Finetune Dataset.

real classroom scenarios. The design simulates cognitive diversity through question-answer types (Chin (2006)), individual differences via the Big Five Personality Traits (Costa (1992)), and varying cognitive abilities across learning stages (Hu et al. (2024a)). Response styles capture linguistic variation (Pennebaker & King (1999)), while generation sources integrate Few-shot In-Context Learning (Few-shot ICL) (Brown (2020)), Chain of Thought (CoT) (Wei et al. (2022)), and fine-tuning on classroom data (Hu et al. (2021)) to ensure authenticity. These elements collectively allow LVSA to simulate, varied behaviors seen in real classrooms.

### 4.2 FINE-TUNING DATASET

**Data source:** The fine-tuning dataset is aligned with the BCUAE Dataset described in Section 3.1 and incorporates data from real classroom video recordings, textbook content, and teacher-prepared lesson plans. These diverse and credible sources enable LLMs generate responses that accurately reflect authentic classroom interactions and align with real-world instructional goals. Please see Appendix C.2.1 for more details.

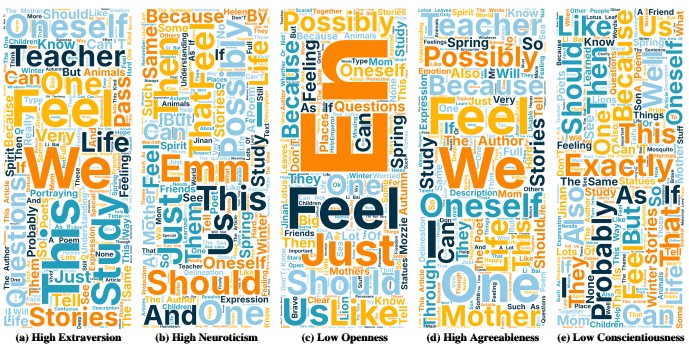

Figure 5: Word cloud visualisation of the Big Five personality virtual student fine-tuning dataset.

**Dataset construction:** As shown in Figure 1 (c), the process involves multiple key stages to ensure realistic and personalized dialogues. Data preparation involved selecting representative Chinese texts and transcribing classroom videos for authentic teacher-student dialogues, providing essential material for fine-tuning. Prompt design was informed by real classroom interactions to align with Big Five traits using GPT-4, enhancing virtual student personalization. Expert revision ensured alignment before large-scale dialogue generation, resulting in a dataset that reflects diverse personality-based interactions. Finally, five distinct instruction fine-tuning datasets were created (see Figure 4), each corresponding to a specific personality trait to ensure contextual appropriateness and individuality in student responses. For more details, see Appendix C.2.2.

**Dataset analysis:** Fine-tuned student-teacher dialogues based on the Big Five personality traits are analyzed using word cloud visualizations to examine expression styles, such as lexical richness. For **HE** (Figure 5 (a)), dominant words like "we," "feel," and "stories" reflect self-referential and positive language, indicating active engagement driven by social needs. For **HN** (Figure 5 (b)), words such as "I," "um," "just like," and "if" suggest hesitancy, anxiety, and frequent backtracking, reflecting emotional fluctuation. The **LO** (Figure 5 (c)) features "possibly," "we," "because," and "questions," showcasing conservative language that relies on established knowledge and avoids complex reason-

ing. For **HA** (Figure 5 (d)), words like "we," "feel," "oneself," and "together" emphasize cooperation and empathy, indicating warmth and a strong connection with others. In **LC** (Figure 5 (e)), terms like "possibly," "should," and "exactly" suggest disorganized and imprecise expression, reflecting lower conscientiousness. These findings support existing research on how language style mirrors cognitive abilities and personality traits (Pennebaker & King (1999)). This analysis validates the fine-tuning dataset's effectiveness in constructing LVSA, with more examples in Appendix C.2.3.

### 4.3 FINE-TUNING THE FOUNDATION MODELS

To accurately capture the linguistic traits of students with different personalities, the model was fine-tuned using a high-performance setup with eight A6000 GPUs, ensuring efficiency and stability for large datasets (see Figure 1 (d)). The SWIFT infrastructure[2], incorporating LoRA (Low-Rank Adaptation), was utilized to enable efficient fine-tuning with tailored hyperparameter settings for each LVSA type (see Appendix D.1). This approach effectively personalizes the model to reflect specific personality traits. Detailed examples are provided in Appendix D.2. A comparison of pre- and post-fine-tuning results revealed significant improvements in the language abilities of the LVSA. Additionally, content analysis of the fine-tuned LVSA's responses showed alignment with personality-specific language styles and traits, enhancing both human-like and personalization. Detailed analysis are provided in Appendix D.2.2

## 5 EVALUATION: ANALYSIS OF OBJECT CAPABILITIES IN SCENES

In this section, we comprehensively evaluate the LVSA by constructing a subjective evaluation dataset, with metrics based on the probability of virtual responses being judged by humans or GPT-4 as resembling real student responses. The dataset construction process is depicted in Figure 1 (e), comprising inference dataset creation, fine-tuned inference, direct inference, and evaluation data reconstruction. The evaluation process is shown in Figure 1 (f), consisting of human evaluation (Section 5.2), human-GPT-4 comparison evaluation (Section 5.3), and large-scale GPT-4 evaluation (Section 5.4). Additionally, traditional objective evaluation metrics were provided to further validate the subjective conclusions and demonstrate the personalization of LVSA's performance(Section 5.5).

### 5.1 SUBJECTIVE EVALUATION DATASET

The evaluation of LVSA followed a four-step process (Figure 1 (e)), enabling both human and GPT-4-based assessments of LVSA performance. **(1) Inference dataset construction**: teacher questions tailored to student personality traits were generated by GPT-4 using prompt engineering, creating a dataset with two fields: system (teaching scenario and traits) and query (teacher questions). **(2) Fine-tuned inference**: Fine-tuned models then generated responses for five distinct personality traits (HE, LO, HN, HA, LC), producing personalized outputs. **(3) Direct inference**: To establish a baseline, direct inference was conducted using non-fine-tuned models, highlighting improvements in personalization and realism after fine-tuning. **(4) Subjective evaluation dataset reconstruction**: A total of 12,312 responses, spanning various learning stages and question types, were generated across four foundation models and five personality types. For human evaluation, 115 samples were randomly selected, including 40 fine-tuned and 40 direct inference responses, along with 35 real student responses as controls, with five fatigue test items to ensure alertness. The remaining 12,232 responses, split evenly between pre- and post-fine-tuned models, were used for large-scale GPT-4 evaluations, providing insights into the effectiveness of LVSA construction (see Appendix C.3).

### 5.2 HUMAN TURING TEST

**Setup:** Inspired by the classic Turing test, this study evaluated whether LVSA could emulate real students' language expressions and cognitive abilities to the extent that human evaluators could not distinguish between them. Instead of using a fixed ground truth, real students served as a control group to assess the human-like characteristics and role-playing effectiveness of LVSA in educational interactions. The experiment followed three steps: questionnaire completion, think-aloud protocol, and semi-structured interviews. During the evaluation of 120 teacher-student dialogues, participants verbalized their thought processes, revealing linguistic features that influenced their judgments (see Appendix D.3). Semi-structured interviews further explored evaluation criteria, which were coded to inform a framework for optimizing future virtual students.

---

[2]Scalable lightWeight Infrastructure for Fine-Tuning

Table 1: Comparison of human-evaluation and GPT-4 evaluation.

| | HN | | HA | | HE | | LC | | LO | | RS |
|---|---|---|---|---|---|---|---|---|---|---|---|
| | Pre | Post | Pre | Post | Pre | Post | Pre | Post | Pre | Post | |
| **Ave-E** | 0.567 | **1.000** | 0.189 | 0.933 | 0.144 | 0.975 | 0.367 | **1.000** | 0.600 | **1.000** | **0.951** |
| **GPT-4** | 0.444 | **1.000** | 0.111 | **1.000** ⇑ | 0.111 | **1.000** ⇑ | 0.444 | **1.000** | 0.750 | 0.889 ⇓ | **0.914** ⇓ |
| **Fleiss's Kappa** | | | | | **0.6806** | | | | | | |

Note: **HE**, **HN**, **LO**, **HA**, and **LC** are abbreviations for **High Extraversion**, **High Neuroticism**, **Low Openness**, **High Agreeableness**, and **Low Conscientiousness** LVSA, respectively. **RS** represents real student.

**Are the LVSA easily recognizable by humans?** Fleiss's Kappa was employed to assess participants' ability to distinguish between virtual and real students, yielding a score of 0.6917, which indicates a "substantial agreement" among participants and a high level of consensus (see Appendix D.4.1). The average recognition probability for fine-tuned LVSA exceeded 0.9 across multiple per-

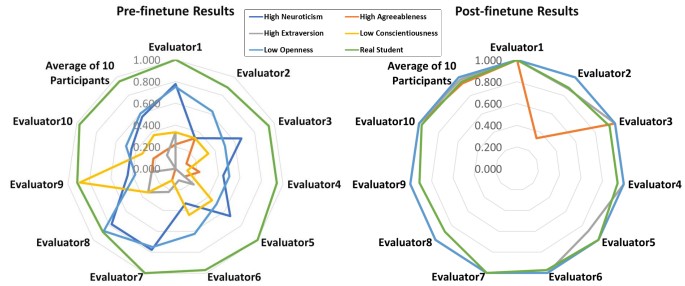

Figure 6: The experiment result of human evaluation, the Fleiss's Kappa is 0.6917 (see Appendix D.4.1).

sonality traits, demonstrating their close resemblance to real students and difficulty in differentiation (see Figure 6). Furthermore, one-way and two-way ANOVA analyses confirmed the statistical significance of the experimental results (see Appendix D.4.2). A significant difference was observed in the performance of virtual students before and after fine-tuning ($p < 0.001$), while the difference between fine-tuned virtual students and real students was not statistically significant ($p > 0.05$). These findings indicate that the fine-tuned LVSA not only accurately simulates various personality traits in language generation but also exhibits a high level of human-likeness when compared to real students. Collectively, these results validate the potential of LVSA as an effective tool for facilitating instructional interactions in educational settings.

## 5.3 HUMAN-GPT4 COMPARISON VALIDATION

**GPT-4 prompt engineering:** To improve GPT-4's accuracy in evaluating LVSA responses, evaluation criteria were extracted from human evaluators' interviews and incorporated into prompts. Two trained coders used ATLAS.ti for a two-level coding process, producing 4 primary and 15 secondary codes covering emotional integration, cognitive level, psychological state, and verbal expression, with high inter-coder reliability (0.876) confirming their validity (see Appendix D.5.1). These dimensions informed GPT-4 prompts designed with a CoT approach for step-by-step evaluations aligned with human judgments. This structured prompt design improved GPT-4's accuracy and ensured consistency with human evaluators (see Appendix D.5.2), establishing it as a reliable tool for large-scale virtual student assessments.

**Does GPT-4 have the same evaluation ability as humans?** Table 1 compares the evaluation capabilities of GPT-4 with 10 human evaluators in assessing LVSA, using Fleiss's Kappa for inter-rater reliability. The agents were fine-tuned to exhibit five distinct personality traits. Post fine-tuning, GPT-4 achieved an average evaluation score of 0.978, with perfect alignment (score of 1) for traits like HN and LC, demonstrating effective simulation of nuanced performance. The overall Fleiss's Kappa of 0.6806 indicates substantial agreement with human evaluators, supporting GPT-4's reliability in educational evaluation. These findings establish a foundation for GPT-4's future application in educational research and practice (see Appendix D.5.3).

## 5.4 LARGE-SCALE EVALUATION EXPERIMENT WITH GPT-4

This section analyzes LVSA across personality types, learning stages, and question types before and after fine-tuning four foundational models. We aim to evaluate their personalization, human-like performance, and adaptability in educational scenarios. Evaluations focus on two aspects: model performance, measuring fine-tuning impact within each model, and data classification, assessing

Table 2: The experiment results of different LVSA types.

| | InternVL | | LLaVa | | MiniCPM | | Qwen | | Average | | Student P-value |
|---|---|---|---|---|---|---|---|---|---|---|---|
| | Pre | Post | Pre | Post | Pre | Post | Pre | Post | Pre | Post | |
| HN | 58.19% | 94.31% | 16.89% | 80.45% | 54.96% | 94.62% | 49.86% | 94.62% | 44.98% | 91.00% | 0.005** |
| HA | 33.99% | 81.19% | 14.52% | 66.46% | 24.75% | 73.93% | 43.89% | 80.86% | 29.29% | 75.61% | <0.001*** |
| HE | 44.64% | 73.88% | 12.69% | 44.40% | 19.78% | 60.82% | 30.34% | 72.76% | 26.86% | 62.97% | 0.002** |
| LC | 54.96% | 50.49% | 18.69% | 52.67% | 30.16% | 34.43% | 30.82% | 39.02% | 33.66% | 44.15% | 0.294 |
| LO | 79.21% | 91.33% | 13.33% | 92.33% | 55.67% | 88.00% | 47.83% | 83.67% | 49.01% | 88.83% | 0.066** |
| Average | 54.20% | 78.24% | 15.22% | 67.26% | 37.06% | 70.36% | 40.55% | 74.19% | 36.76% | 72.51% | 0.009** |
| Model P-value | 0.058** | | 0.004** | | 0.013** | | 0.007** | | 0.006** | | |

Note: ** means significant; *** means highly significant.

Table 3: The experiment results of different learning stages.

| | InternVL | | LLaVa | | MiniCPM | | Qwen | | Average | | Learning P-value |
|---|---|---|---|---|---|---|---|---|---|---|---|
| | Pre | Post | Pre | Post | Pre | Post | Pre | Post | Pre | Post | |
| PI | 64.59% | 82.32% | 20.06% | 66.60% | 48.45% | 79.73% | 49.48% | 81.79% | 45.65% | 77.61% | 0.012** |
| NL | 54.71% | 80.06% | 16.59% | 68.04% | 38.04% | 69.63% | 41.41% | 75.15% | 37.69% | 73.22% | 0.008** |
| KC | 49.71% | 77.32% | 11.86% | 69.52% | 37.29% | 71.19% | 37.76% | 70.51% | 34.16% | 72.14% | 0.011** |
| CE | 48.63% | 77.43% | 16.14% | 67.36% | 33.89% | 64.09% | 37.37% | 70.47% | 34.01% | 69.84% | 0.006** |
| LS | 55.00% | 76.97% | 12.25% | 69.14% | 32.92% | 72.10% | 39.50% | 76.18% | 34.92% | 73.60% | 0.012** |
| Average | 54.53% | 78.82% | 15.38% | 68.13% | 38.12% | 71.35% | 41.10% | 74.82% | 37.28% | 73.28% | 0.009** |
| Model P-value | <0.001*** | | <0.001*** | | <0.001*** | | <0.001*** | | <0.001*** | | |

Note: (1) **PI**, **NL**, **KC**, **CE**, and **LS** are abbreviations for **Pre-lesson Introduction**, **New Lesson Instruction**, **Knowledge Consolidation**, **Class Exercises**, and **Lesson Summary** students, respectively. (2) ** means significant; *** means highly significant.

fine-tuning effectiveness across data types like closed-ended questions. The findings provide insights for optimizing virtual student models and broadening their applicability in educational contexts.

**Evaluation results for different LVSA types:** This section analyzes virtual students' performance across four foundational models, comparing pre- and post-fine-tuning results. The objective was to assess LLMs' ability to construct personalized, human-like virtual students. As shown in Table 2, the average evaluation score across five personality types rose significantly from 36.76% to 72.51% after fine-tuning, demonstrating LLMs' effectiveness in generating realistic and personalized behaviors.

InternVL and Qwen demonstrated relatively strong pre-fine-tuning performance, with scores of 54.20% and 40.55%, respectively, while LLaVa showed the weakest performance at 15.22%, likely due to its English-language foundation limiting compatibility with Chinese-language tasks. However, after fine-tuning, all models showed substantial improvements, with average scores rising to 72.51%. Notably, LLaVa's accuracy increased to 67.26%, demonstrating effective adaptation to Chinese contexts. InternVL and Qwen achieved post-fine-tuning accuracy rates of 78.24% and 74.19%, indicating strong adaptability and language generation capabilities. Paired t-tests confirmed the statistical significance of these improvements, with p-values well below 0.05 for all models.

Analysis of different personality types revealed significant performance gains for all types except LC students. Virtual students with HA exhibited the most notable improvements, with p-values below 0.001, highlighting strong statistical significance. For example, HN scores increased from 44.98% to 91.00%, HA scores from 29.29% to 75.61%, and HE scores from 26.86% to 62.97%.

The findings indicate that while personality types like LC may need further optimization, fine-tuned LLMs effectively captured the linguistic traits of LVSA across personalities, bringing virtual students' performance closer to real students. With further enhancement, these models could support personalized education, pre-service teacher training, and educational experimentation at scale.

**Evaluation results for different learning stages:** This section analyzes model performance across learning stages. As shown in Table 3, InternVL and Qwen achieved scores of 78.82% and 74.82% after fine-tuning, with an average improvement of 36.03% across all models. Paired t-tests (p ¡ 0.001) confirmed statistically significant gains. Fine-tuning based on learning stages proved more straightforward due to the structured nature of stage data, with clear teaching content and cognitive benchmarks enabling effective adaptation. In contrast, fine-tuning for student personalities is more challenging due to their fluid, context-dependent traits and lack of explicit hierarchies.

Experimental results also show significant improvements in virtual students' performance across all learning stages, with evaluation scores increasing by 31.96%, 35.53%, 37.98%, 35.84%, 38.68%,

Table 4: The experiment results of different question types.

| | InternVL | | LLaVa | | MiniCPM | | Qwen | | Average | | Question P-value |
|---|---|---|---|---|---|---|---|---|---|---|---|
| | Pre | Post | Pre | Post | Pre | Post | Pre | Post | Pre | Post | |
| CQ | 58.61% | 78.38% | 17.72% | 67.08% | 43.19% | 72.91% | 47.64% | 75.13% | 41.79% | 73.38% | 0.015** |
| OQ | 50.40% | 79.23% | 12.98% | 69.22% | 32.81% | 69.67% | 34.47% | 74.51% | 32.67% | 73.16% | 0.006** |
| Average | 54.51% | 78.81% | 15.35% | 68.15% | 38.00% | 71.29% | 41.06% | 74.82% | 37.23% | 73.27% | 0.009** |
| Model P-value | 0.117 | | 0.041** | | 0.068 | | 0.117 | | 0.078 | | |

Note: (1) **CQ**, **OQ** are abbreviations for **Closed-ended Question**, **Open-ended Question**, respectively. (2) ** means significant; *** means highly significant.

and 35.99% for pre-lesson introduction, new lesson instruction, knowledge consolidation, class exercises, and lesson summary, respectively. Paired t-tests yielded p-values below 0.05, indicating statistically significant differences. These findings suggest that LVSA effectively adapts to different learning stages, offering comprehensive support for pre-service teacher training. Based on controlled experimental findings above and feedback from semi-structured interviews with human participants, virtual students enable teachers to practice and refine instructional strategies across all teaching phases, enhancing skill development throughout the entire instructional process.

**Evaluation results for different question types:** As shown in Table 4, we analyzed the performance of virtual students on different question types before and after fine-tuning to assess their ability to handle both closed and open questions effectively. The paired t-test P-values for closed, open, and overall questions were 0.015, 0.006, and 0.009, respectively—all below the 0.05 threshold—indicating statistically significant improvements. Specifically, fine-tuning led to an average improvement of 31.59% for closed questions and 40.49% for open questions, suggesting a greater impact on open-ended tasks due to their higher complexity and lower

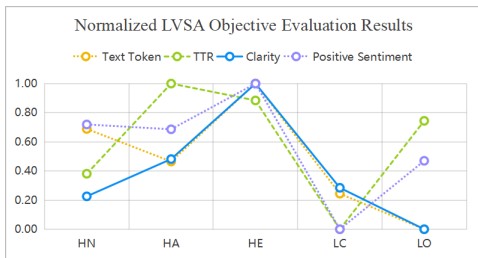

Figure 7: The experiment result of objective evaluation (see Appendix D.6).

initial performance. From a model perspective, InternVL and Qwen performed best post-fine-tuning, with evaluation scores of 78.81% and 74.82%, respectively, reflecting improvements of 24.30% and 33.77%. LLaVa achieved statistically significant improvement across all question types (P-value of 0.041), whereas InternVL, Qwen, and MiniCPM showed non-significant differences (P-values of 0.117, 0.117, and 0.068, respectively), highlighting challenges in achieving consistent improvement across models. LLaVa's significant gains across both question types and previous evaluations of student types and learning stages indicate superior generalization.

The differences in model performance between closed and open questions are likely due to inherent variations in complexity. Closed questions require specific factual recall, benefiting from structured dataset pretraining, while open questions involve creative reasoning, posing greater challenges. These findings suggest that fine-tuning enhances adaptability and response quality, particularly for tasks requiring sophisticated reasoning and creativity.

## 5.5 OBJECTIVE EVALUATION

To comprehensively evaluate the personalized performance of LVSAs across different personality types and validate the human-like conclusions from subjective evaluations, we employed four objective metrics: text token, Type-Token Ratio (TTR), clarity (inverse perplexity), and the proportion of positive sentiment samples. For clearer visualization, these metrics were normalized via min-max standardization. Fig. 7 shows that HE excels in clarity and positive sentiment, reflecting its extroverted and fluent expression characteristics. HA demonstrates high lexical richness and positive sentiment, highlighting its rich, cooperative expression traits. In contrast, LO and LC personalities score lower in clarity and lexical richness, consistent with their concise, reserved, or casual language styles. Detailed experiment results are provided in Appendix D.6

These findings align with the anthropomorphic conclusions from subjective evaluations but also reveal the limitations of objective metrics, such as the overestimation of lexical richness in short texts

and the reliance of sentiment analysis on keywords. This underscores the complexity of evaluating LVSA performance.

### 5.6 BAD CASE ANALYSIS

The experimental results indicate that fine-tuned virtual students demonstrated strong performance across various educational scenarios, effectively simulating personalized, human-like behaviors. However, some noteworthy anomalies were identified, revealing limitations of LLMs in role-playing contexts and suggesting areas for future research.

**Poor fine-tuning performance of LC LVSA:** The suboptimal performance of LC personality modeling arises from the sparse representation of its associated behaviors and language traits in existing data, limiting the model's ability to capture typical LC characteristics. Hallucinations, such as repetitive language or contextually inconsistent knowledge errors (see Appendix D.7.1), further compromise output quality and personality alignment. Additionally, LC traits, potentially conflicting with educational ethics (e.g., lack of responsibility), often trigger the ethical safeguards of large language models, increasing generation constraints. To address these issues, we implemented personalized prompt design, context-specific data generation, and content validation to enhance LC personality simulation. Future efforts will focus on expanding data diversity, refining techniques such as personalized fine-tuning or constraint strategies, and developing robust mechanisms to monitor and correct hallucinations, aiming to improve the accuracy and applicability of LC personality modeling.

**Inconsistent fine-tuning effects across question types:** Although all four models improved in handling both closed and open-ended questions, none achieved statistically significant improvements across both types. This inconsistency likely stems from inherent differences in cognitive demands (see Appendix D.7.2). Closed-ended questions are more structured, making them easier for models to manage, whereas open-ended questions require deeper reasoning and creative thinking, resulting in greater variability in performance.

**Suboptimal fine-tuning performance of LLaVa:** Despite improvements in personalization, question types, and learning stages, LLaVa's overall performance remained weaker than that of others (see Appendix D.7.3). This disparity is primarily due to differences between pre-training and fine-tuning data domains, particularly cross-language issues. Given that LLaVa predominantly relies on English pre-training data, its adaptability and generalization to Chinese contexts are constrained.

## 6 CONCLUSION

This study addresses the challenges of modeling and evaluating early adolescent students by proposing the SOE framework, leveraging LLMs to construct virtual students and explore their potential for human-likeness and personalization. Its modular design—Scene, Object, Evaluation—enables flexible adaptation to educational systems, supports discipline-specific adjustments, and optimizes for students' age and cultural diversity. Through a combination of subjective and objective evaluations, the SOE framework validates the performance of virtual students across diverse personality traits, question types, and teaching stages. Experimental results demonstrate a high degree of similarity between the generated dialogues and real student behaviors, highlighting the significant potential of LLMs in simulating human-like roles to support pre-service teacher training.

Moreover, the paper outlines critical future directions, including real-world applications, model optimization, and the reduction of hallucinations. Regarding real-world applications, findings currently stem from controlled environments complemented by human feedback. Future research should extend LVSA implementations to actual educational settings by incorporating multimodal data, thereby enriching the complexity of teaching scenarios and conducting longitudinal studies to thoroughly assess their educational impact and feasibility. Concerning model optimization, the study applies Swift and LoRA techniques for efficiency enhancement, yet further investigations into model distillation and response sampling are essential for reducing computational demands and elevating model adaptability. Lastly, while initial steps to mitigate hallucinations include refined prompt design and post-processing checks, continued improvements are necessary. These should involve integrating authentic classroom data, refining modeling approaches, maintaining ethical standards, and developing comprehensive strategies for identifying and rectifying content discrepancies. Ongoing refinement and expansion of the SOE framework aim to robustly support pre-service teacher training and drive innovation in educational technology.

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

# APPENDIX

# A   COMPREHENSIVE RELATED WORKS

## A.1   VIRTUAL STUDENT DEVELOPMENT AND RELATED TECHNOLOGIES

Teachers, as a central resource for educational development, play a critical role in ensuring the quality of education. Pre-service teachers, who represent the future of the teaching workforce, depend heavily on practical training for their professional growth (Loewenberg Ball & Forzani (2009); Zeichner (2012); Ward et al. (2018)). Virtual students provide a promising tool for simulating classroom interactions, offering a cost-effective, highly interactive, and flexible platform for pre-service teacher training and feedback (Dotger et al. (2015; 2010); Shaughnessy & Boerst (2018)). As technology in this domain has evolved, virtual students can be classified into three main categories: role-playing virtual students, programmatically pre-set virtual students, and virtual students powered by large language models (LLMs). Each of these approaches has distinct characteristics, technical underpinnings, and limitations.

**Role-playing virtual students** are among the earliest forms of simulated student behavior (Kersh (1963); Colwell (2013); Frederick et al. (2010); Shapira-Lishchinsky (2015); Dalgarno et al. (2016)). In this approach, teachers or other adults imitate the cognitive and emotional responses of students, offering real-time interaction (Dieker et al. (2015); Hume (2012); Teasdale et al. (2016)). While this method allows for some degree of realism in mimicking student behaviors, it relies heavily on the participants' ability to accurately simulate a student's responses. Consequently, this approach is resource-intensive, lacks scalability, and cannot provide the level of automation or flexibility required for modern teacher training systems (Sparks (2011)).

**Programmatically pre-set virtual students** use pre-defined algorithms and scripted rules to simulate student interactions, providing a degree of automation for classroom simulations (Christensen et al. (2011); Delamarre et al. (2021); Shernoff et al. (2018); Kelleci & Aksoy (2021)). These virtual students are typically powered by finite state machines or pre-scripted scenarios (Kervin et al. (2006); Lugrin et al. (2019)). For instance, a virtual student might respond to a teacher's question based on a pre-defined set of rules or dialogues. This approach is advantageous in its repeatability and operational control, which makes it useful for specific teaching tasks that require consistency (Carrington et al. (2011); Hixon & So (2009)). However, the rigidity of the pre-defined scripts often leads to uniform and mechanical responses, limiting their ability to adapt to the diverse and dynamic interactions that characterize real-world classrooms (Badiee & Kaufman (2015); Rayner & Fluck (2014)).

Recently, **LLM-based virtual students** have gained attention with the development of models like GPT-4 (Achiam et al. (2023); Li et al. (2024c); Zhang et al. (2024); Lee et al.; Markel et al. (2023)). These LLM-driven agents leverage advanced natural language processing capabilities to generate dynamic and contextually relevant student responses. Although LLMs introduce greater flexibility and linguistic variability into virtual student interactions, current research in this area remains nascent. Most existing studies focus on using prompt engineering to generate virtual student agents, yet the exchanging responses of LLMs inevitably faces significant challenges including role flipping, instruction repeating and fake replies (Qian et al. (2024)). In this way, these agents often struggle to replicate the full range of behaviors, cognitive abilities, and emotional nuances that real students exhibit. While LLMs have demonstrated strong capabilities in natural language generation and knowledge understanding (Jiang et al. (2022); Liu et al. (2023); Park et al. (2023)), they have yet to fully mimic the language expression levels and cognitive challenges faced by real students.

This study builds on this line of research by exploring whether LLMs can be leveraged to create human-like and personalized virtual student agents that better simulate student cognition and emotional responses. Our goal is to push beyond basic prompt-based LLM applications by constructing virtual students that can provide more authentic, dynamic, and adaptive learning interactions, thereby improving their applicability in pre-service teacher training.

## A.2 LLMs' main capabilities to support AI4Education

In recent years, LLMs have demonstrated exceptional performance across a range of core language processing tasks (Brown (2020); Achiam et al. (2023)), including language understanding, generation, reasoning, knowledge transfer, and multimodal processing. Trained on vast amounts of textual data, these models have acquired the ability to comprehend complex linguistic structures and are widely applicable to various natural language processing tasks.

The **language understanding** capabilities of large language models are primarily reflected in their ability to infer meaning from context, capture complex syntactic structures, and accurately grasp the intent of a given text. In widely-used natural language understanding benchmarks such as GlUE (Wang (2018)), SuperGLUE (Wang et al. (2019)) and SQuAD (Rajpurkar et al. (2018)), models like GPT-4 (Achiam et al. (2023)), LLaMA (Touvron et al. (2023)), and MiniCPM (Hu et al. (2024b)) have demonstrated performance that rivals, and in some cases surpasses, human levels (T.Simonite (2018); Li et al. (2023a)). For instance, GPT-4's high scores on SuperGLUE indicate its ability to process extended text and maintain a deep understanding of contextual relationships (Achiam et al. (2023); Guo et al. (2023)). This is particularly important for building virtual students, as they must respond to teachers' questions with appropriate cognitive reactions, simulating the authentic thought processes of real students.

In benchmark tests like CommonGen (Lin et al. (2019)) and Wikitext-103 (Merity et al. (2017)), models such as GPT-4 and T5 (Raffel et al. (2020)) have demonstrated strong **language generation** capabilities, particularly in generating natural language paragraphs, dialogues, and long-form text (Li et al. (2024b)). Wikitext-103, in particular, focuses on long-text generation tasks and evaluates the model's ability to maintain contextual consistency throughout the generation process. Current language models not only generate coherent responses in dialogue systems but also preserve logical consistency across multiple conversational turns (Zeng et al. (2023)). This provides a solid technical foundation for this study, where GPT-4 is used to generate teacher-student dialogue datasets under different personality traits.

In **language reasoning** benchmarks such as the ARC (AI2 Reasoning Challenge) (Clark et al. (2018)) and MNLI (Nangia et al. (2017)), models are required to perform complex logical inference (Hu et al. (2020)) and commonsense knowledge (Clark et al. (2018)) based on the given text. These tasks demand that the model not only understand the semantics of the questions but also use contextual information to provide appropriate answers. Current top-tier LLMs have demonstrated strong reasoning abilities in such tests (Zhong et al. (2023); Huang et al. (2024)). For example, GPT-4 and PaLM (Chowdhery et al. (2023)) have shown significant improvements in the ARC test compared to earlier models, enabling them to handle more complex reasoning tasks. This reasoning capability provides a solid technical foundation for virtual students to exhibit diverse reactions in the classroom, particularly in responding to open-ended questions, where the virtual student can demonstrate multi-step reasoning abilities.

**Language transfer** is a significant strength of large language models, enabling them to apply knowledge learned from one task to other domains or language tasks. In multilingual benchmarks such as Cross-Lingual Natural Language Inference (XNLI) (Conneau et al. (2018)) and TyDiQA (Clark et al. (2020)), models have demonstrated strong cross-linguistic capabilities, effectively handling comprehension and reasoning tasks across various languages. Models like LLaMA, T5, Qwen (Bai et al. (2023)), PaLM and GPT-4 have shown high generalization abilities in these tests, successfully addressing the challenges posed by different languages and tasks (Chowdhery et al. (2023); Ahuja et al. (2023)). This ability is particularly crucial for developing multilingual virtual students, as teachers may interact with students in diverse linguistic and cultural contexts. By leveraging the knowledge transfer capabilities of large language models, virtual students can adapt to various language environments, providing flexible instructional feedback.

In recent years, **multimodal models** like Flamingo (Alayrac et al. (2022)) and InternVL (Chen et al. (2024b)) have become research hotspots, combining visual and textual processing capabilities and extending the application range of language models. In multimodal benchmarks such as Visual Question Answering (VQA) (Antol et al. (2015); Kafle & Kanan (2017)), models like InternVL have demonstrated their ability to generate accurate text responses based on visual inputs, showcasing their strong text generation and comprehension skills. These multimodal models offer greater potential for the future development of virtual students. For instance, by integrating both visual

and textual generation abilities, virtual students can operate in more enriched multimodal teaching environments, handling more complex classroom tasks.

Among these models, GPT-4 and PaLM continue to excel in language understanding and reasoning tasks, particularly in handling complex language tasks. LLaMA demonstrates strong knowledge transfer capabilities in cross-linguistic tasks, making it especially suitable for building multilingual virtual students. MiniCPM performs exceptionally well in Chinese natural language processing tasks, making it ideal for non-English educational settings. Qwen exhibits powerful reasoning and cross-linguistic abilities, excelling particularly in language generation tasks. While InternVL is primarily a multimodal model, its text generation capabilities are also impressive, providing technical support for future virtual students to operate in multimodal learning environments.

## A.3 REPRESENTATIVE AI4EDUCATION APPLICATIONS

Leading educational technology companies worldwide have pioneered the launch of various innovative products based on large language models (LLMs). Duolingo Max (Duolingo (2023)) leverages LLMs to enhance language learning through generative dialogue and intelligent feedback, helping students improve their language skills via interactive exercises. Khan Academy's Khanmigo (Khan (2023)), powered by LLMs, offers intelligent tutoring and personalized learning support, spanning from primary to higher education (Pavić et al. (2024); Wulandari et al. (2024); Yasmin & Mazhar (2022)). Google launched the Socratic platform (google (2023)), assisting students in solving problems across multiple disciplines such as science, mathematics, and history. Chinese edtech giants such as TAL Education, Youdao, iFLYTEK and Squirrel AI are also actively exploring LLM applications (Li et al. (2023b)). TAL's MathGPT (TAL (2023a)) focuses on providing students with step-by-step guidance in mastering core mathematical problem-solving techniques. Youdao's "ZiYue" (TAL (2023b)) prioritizes a "scenario-first" approach. iFLYTEK's "Spark Desk" (iFLYTEK (2023)) can conduct human-like interactive learning in various fields, including mathematics, English oral practice, essay correction, and more. Squirrel AI (squirrelai (2023)) has developed a personalized learning system that leverages big data and AI to tailor individual learning paths, significantly improving students' learning efficiency and performance.

Internationally renowned universities are also contributing to the development and application of LLMs. MIT succeeds in achieving 81% correctness on undergraduate-level maths problems through fresh-shot learning based on OpenAI Codex pre-trained models (Drori et al. (2022)), which combines natural language processing with code generation to assist students in advanced data analysis and programming tasks (Chen et al. (2021)). Similarly, East China Normal University has developed EduChat (Dan et al. (2023)), which offers functionalities such as open Q&A, essay grading, heuristic teaching, and emotional support.

To further validate and enhance the performance of LLMs in educational settings, several benchmark datasets and evaluation tools have been introduced. These datasets assess the models' knowledge comprehension and reasoning capabilities across various subjects. For instance, the C-Eval (Huang et al. (2024)) dataset is designed to evaluate the performance of LLMs across a range of subjects from primary to higher education. The SciQAG (Wan et al. (2024)) dataset measures the model's ability to answer scientific questions, testing its depth of domain knowledge. Mukea (Ding et al. (2022)) serves as a benchmark for multimodal knowledge extraction, challenging the models' ability to integrate information from diverse input sources. These benchmark evaluation tools not only drive the advancement of LLMs but also provide researchers with precise metrics to continually optimize the educational effectiveness of these models.

Moreover, leading artificial intelligence conferences such as NeurIPS (Denny et al. (2024)) and AAAI (Ananda et al. (2024)) have seen a growing number of research papers and workshops discussing the applications of LLMs in education. Topics such as supporting individualized learning paths (Hayat & Hasan (2023); Hwang et al. (2023); Schmucker et al. (2023)), providing personalized feedback (McNichols et al. (2023); Sahai et al. (2023)), and education dataset construction (Han et al. (2023)) are becoming increasingly popular, highlighting the critical role that LLMs play in advancing educational technologies.

Although significant progress has been made in using LLMs as domain-expert to improve student learning outcomes—particularly in areas such as problem-solving (Didolkar et al. (2024)), knowledge explanation (Gao et al. (2024)), and automated grading (Lagakis & Demetriadis (2024))—there

remains a noticeable gap in research focused on enhancing teacher skills. To date, only one study (Lee et al.) in this area has been featured at NeurIPS, indicating that this field is still emerging and holds considerable potential for further exploration. However, despite the pivotal role teachers play in education, research on leveraging LLMs to support their diverse instructional needs remains notably scarce, underscoring an important area for future work.

This study addresses this gap by introducing a novel research direction: using virtual students to enhance teachers' pedagogical skills. Unlike previous efforts that mainly focused on constructing expert systems to assist student learning, this study aims to generate more human-like and emotionally responsive virtual students through LLMs. These virtual students simulate real classroom reactions, providing teachers with authentic interactive teaching environments. By engaging with these virtual students, teachers can practice, reflect on, and refine their teaching methods, ultimately enhancing their professional capabilities and preparing them for future classroom scenarios.

## B  DETAILED INFORMATION FOR THEORY AND TECHNOLOGY USED IN THIS WORK

### B.1  FOUNDATION LLMS

To evaluate the performance of LLMs in junior high school Chinese language tasks, this study selected four representative foundational models for objective assessment. Each model has demonstrated outstanding performance across various domains, with unique strengths in handling multimodal tasks or pure text generation. These models are pivotal for evaluating comprehension and recitation abilities, which are essential for constructing virtual student agents in future research.

**InternVL** (Chen et al. (2024b)) is a multimodal large language model that integrates both visual and linguistic processing capabilities. It excels at cross-modal reasoning, particularly in tasks requiring the integration of textual information with images. Although primarily applied to multimodal scenarios, InternVL's strong language processing skills make it suitable for the text comprehension tasks required in this study.

**LLaVa** (Li et al. (2024a)) is a lightweight yet powerful LLM based on the Mistral architecture. It specializes in text generation and language understanding, offering efficient reasoning capabilities even at smaller scales. This model is expected to perform well in medium-scale tasks, such as text comprehension and recitation, making it a flexible choice for resource-constrained environments.

**MiniCPM** (Hu et al. (2024b)) is optimized for Chinese language tasks, with deep capabilities in understanding and generating Chinese. Trained on a large corpus of Chinese data, MiniCPM excels in tasks such as essay correction and passage analysis, making it particularly well-suited for junior high school Chinese language comprehension and recitation tasks.

**Qwen** (Bai et al. (2023)) supports multimodal interactions, excelling in both visual-linguistic exchanges and pure text-based tasks. Its ability to integrate visual and linguistic information allows it to handle complex tasks, making it a valuable tool for both current text-based tasks and future applications involving multimodal inputs.

By comparing these models' performance in junior high school Chinese scenarios, the study aims to assess their ability to process and understand Chinese text, specifically focusing on comprehension and recitation. As future research may extend to multimodal teaching environments, such as virtual students recognizing visual content presented by teachers, the inclusion of multimodal models ensures a foundation for future expansion into scenarios requiring both textual and visual processing.

Table A1: The URLs of official foundation LLMs repository in this work.

| Foundation LLMs | Creator | #Parameters | Access |
|---|---|---|---|
| **InternVL2** (InternVL2-8B) (Chen et al. (2024b)) | Shanghai AI Laboratory | 8B | Weights |
| **LLaVa** (LLaVa-v1.6-mistral-7b-hf) (Liu et al. (2024a)) | Microsoft | 7B | Weights |
| **MiniCPM** (MiniCPM-V-2-6-chat) (Yao et al. (2024)) | MiniCPM-V | 8B | Weights |
| **Qwen** (Qwen-vl-chat) (Bai et al. (2023)) | Alibaba Cloud | 7B | Weights |

### B.2  EDUCATIONAL THEORY

#### B.2.1  CONCEPTUAL THEORY: CHARACTERISTICS OF EARLY ADOLESCENT STUDENTS

Early adolescents (typically aged 10 to 15) represent a critical developmental stage where students undergo substantial physiological, cognitive, social-emotional, and moral-spiritual changes (Hall (1905); Caskey & Anfara (2014)). These underlying developmental traits manifest as explicit behaviors in the classroom, providing essential theoretical insights for constructing virtual student agents.

At the **physiological level**, rapid development of the prefrontal cortex, which governs complex cognitive functions such as decision-making, reasoning, and impulse control, contrasts with the faster development of the limbic system, responsible for emotional regulation (Blakemore & Choudhury (2006); Giedd (2004)). This imbalance often leads to intense emotional responses coupled with lim-

ited cognitive regulation, which influences classroom behavior and language expression. Adolescents may display strong emotional reactions, cognitive confusion, and variability in participation, establishing a physiological basis for their learning behaviors.

**Cognitively**, adolescents transition from concrete operational thinking to abstract logical reasoning (Elkind (1981)). According to Piaget's theory (Piaget (1952)), they gradually develop the ability to handle abstract concepts and logical analysis, though they remain partially bound by concrete experiences. They exhibit early signs of critical thinking but are easily influenced by emotions and external factors, leading to repetitive and uncertain language expression (Nippold (1993); Nippold et al. (1999)), especially when faced with abstract issues.

**Social-emotionally**, as self-awareness increases, adolescents become more concerned with their roles within social groups, experiencing frequent emotional fluctuations. This heightened sensitivity affects their classroom behavior—teacher feedback and peer reactions significantly impact their engagement. In constructing virtual students, it is crucial to simulate such emotional dynamics to accurately reflect students' responses to classroom stimuli, such as nervousness, anxiety, or excitement.

**Morally and spiritually**, adolescents begin developing their own values and face complex moral issues (Scales (2010)). According to Kohlberg (Kohlberg (1987)), they transition from external rule adherence to internalizing societal norms. However, their moral judgments are often emotionally driven and oversimplified. In class, adolescents interpret moral situations from texts through personal experiences (Scales (2010)), reflecting emerging critical thinking while simplifying complex moral issues.

These implicit traits manifest in the classroom as individualized behaviors and language use (Pennebaker & King (1999)). Adolescents' classroom participation varies greatly—some engage actively while others remain hesitant. Their language often involves repetition and uncertainty, especially with abstract questions, using filler words like "um" or "uh". This reflects their cognitive insecurities and limited grasp of complex issues. They also frequently relate classroom content to personal experiences, which should be considered when designing instructional tasks. Additionally, emerging critical thinking shows students beginning to analyze multiple perspectives, though these skills are not yet fully mature.

### B.2.2 OPERATIONAL THEORY: CLASSIFICATION OF TEACHER-STUDENT DIALOGUE DATASETS

When constructing LVSA, it is crucial to ensure that the generated dialogues are highly operational and predictable, allowing the model to realistically simulate personalized behavior and language capabilities of different students. To achieve this, we introduced a set of practical dimensions to classify and guide the generation of virtual students, including question-answer types, personality traits, learning stages, response styles, and generation sources. These dimensions serve as the foundation for generating diverse student behaviors and ensuring that virtual students exhibit realistic responses consistent with real students across various classroom scenarios.

**Question-Answer Type**: The type of question-answer is fundamental for generating virtual students. Classroom questioning is a key strategy for teachers to guide student thinking and stimulate learning interest (Hattie & Timperley (2007)). Different types of questions require different cognitive processes, influencing student language output and cognitive development (Chin (2006)). Open-ended questions encourage creative and critical thinking, while closed-ended questions emphasize recalling facts and accurately understanding specific knowledge. Careful design of question types aids in simulating realistic classroom interactions and evaluates the model's ability to handle diverse questions.

**Personality Traits**: Personality traits are a critical dimension in modeling virtual students. Integrating the Big Five Personality Traits model allows for personalized student modeling (McCrae & Costa Jr (1997)). These traits—extraversion, agreeableness, conscientiousness, neuroticism, and openness impact student behavior and language expression. Modeling virtual students based on the Big Five ensures the creation of distinct personalities and diverse language patterns, effectively simulating varied student behaviors in the classroom (Christensen et al. (2011); Poropat (2009)). Specifically, to enhance the personalized expression of student personalities in this study and to fully explore the potential of large language models in constructing individualized and humanized virtual

students, we utilized five personality types that differed widely: **High Neuroticism** (**HN**), **High Extraversion** (**HE**), **High Agreeableness** (**HA**), **Low Conscientiousness** (**LC**), and **Low Openness** (**LO**). The associated personality descriptions and the sample descriptions of the personality types used in this study are shown in the Table A2. Among these, HE and HA are positive personality traits, representing well-performing students with higher cognitive abilities and stronger language proficiency. HE students demonstrate a greater desire to express themselves compared to HA students. On the other hand, LC and low openness are relatively negative traits. Students with LC tend to be careless, with less detailed language expression, while students with LO exhibit lower cognitive abilities, introversion, and hesitation, often struggling to answer questions. Compared to the other four personalities, HN presents a more neutral profile; aside from experiencing nervousness and hesitation in their responses, students with this trait generally exhibit normal cognitive abilities.

Table A2: Five typical early adolescent students characters based on the Big Five personality theory.

| Personality Traits | Low Level | High Level | Student Description in the Study |
|---|---|---|---|
| **Openness** | Conservative, pragmatic and efficient | Have broad interests, are willing to change, are curious, and get bored easily | Student A, with low openness to experience, tends to exhibit characteristics such as low receptivity to new content (preferring to rely on familiar knowledge and experiences, often feeling confused or uninterested in unfamiliar or complex topics), lack of initiative in exploration (showing little inclination to investigate or think critically about new problems, rarely participating in discussions or asking questions), and weaker ability to handle complex issues (struggling to provide effective responses or solutions to more challenging questions or content requiring deep understanding). Consequently, their language style is characterized by simple and direct responses (avoiding complex analysis or evaluation, and often responding to difficult questions with brief answers like 'um' or 'uh'), reliance on filler words (using 'um,' 'uh,' and similar phrases to express confusion or hesitation), and difficulty expanding discussions (rarely offering additional information or perspectives without prompting). |
| **Conscientiousness** | Spontaneity, multitasking, flexibility, lack of organization and discipline | Task and goal oriented, reliable, organized | Student B, with low conscientiousness, tends to exhibit traits such as carelessness (frequently making mistakes, though occasionally providing correct answers), inconsistency (sometimes recognizing and correcting their own errors, while other times making outright mistakes), and lack of systematic thinking (responses tend to be disorganized and occasionally omit important details). Consequently, their language style is characterized by simplicity and directness (responses are concise but prone to errors), occasional self-correction (they may occasionally identify and correct their own mistakes, though inconsistently), and unreliability (responses may be disjointed, with alternating instances of correctness and error). |

*Continued on next page*

| Personality Traits | Low Level | High Level | Student Description in the Study |
|---|---|---|---|
| **Extraversion** | Introvert, private, reserved | Social, participates in many activities, enjoys being with people, active, outgoing | Student C, with high extraversion, exhibits traits such as active participation (a strong desire to engage, frequently answering questions, and proactively showcasing themselves), strong social skills (active in class, interacting frequently with teachers and peers, and expressing themselves with confidence), ease of performance (responding to classroom activities and questions naturally and confidently, showing little influence from external factors), and high visibility (eager to display themselves, actively participating in discussions, and enjoying being the center of attention). Consequently, their language style tends to be fluent and confident (speaking smoothly and confidently, with occasional hesitations or pauses when overexplaining), elaborative (offering detailed explanations or examples to demonstrate understanding), positive (using enthusiastic language that reflects interest in the topic), and structured (responses are organized and clear, with minimal unnecessary or repetitive statements). |
| **Agreeableness** | Doubt, competitiveness, pride | Tolerant, accepting, kind, gentle, trustworthy | Student D, with high agreeableness, exhibits traits such as cooperation and empathy (eager to assist peers, actively engage in discussions, and show concern for others), thoughtfulness and patience (remaining open to the opinions of teachers and classmates, and expressing understanding and support), and positive feedback (responding to questions with consideration for others' emotional states). Consequently, their language style tends to be warm and friendly (using soft language and a kind tone), elaborative (providing detailed explanations that reflect understanding and care), and accurate (with clear, error-free responses). |
| **Neuroticism** | Contented, controlled, easy going | More negative emotions, increased stress, and lower satisfaction | Student E exhibits high neuroticism, characterized by anxiety and nervousness (displaying heightened tension and unease, easily influenced by the classroom environment), repetitive backtracking (repeating and revisiting responses, reflecting worry and uncertainty about their answers), and emotional fluctuations (prone to emotional instability). As a result, their language expression style tends to be hesitant (including hesitative fillers like 'um' and 'uh'), repetitively backtracking (repeating parts of their responses to confirm or correct perceived uncertainties), and disjointed (with fragmented and less coherent speech patterns). |

**Learning Stages**: Students' cognitive abilities and language expression vary significantly at different learning stages, which must be reflected accurately in virtual student modeling. According to Vygotsky's Zone of Proximal Development (ZPD) theory (Vygotsky, 1978), language and cognitive development progress with appropriate challenges. In early learning stages, virtual students may use simple sentence structures and straightforward responses, while in advanced stages, they can use complex grammar and logical reasoning. Modeling learning stages helps create virtual students suited to different instructional contexts, offering richer and more varied classroom interactions.

**Answering Style**: Answering style highlights individual differences in students' language expression. For example, students with high openness may use more creative and exploratory language, while those with high conscientiousness may provide more detail-oriented and accurate responses. Additionally, students with high neuroticism often show emotional fluctuation, including hesitation

and uncertainty, while those with high emotional stability tend to respond with confidence, using concise language (Dewaele, 2012). Reflecting these styles in virtual students ensures a realistic diversity in their dialogue.

**Generation Source**: The generation source is vital for ensuring realism in virtual student behavior. Creating virtual students involves both prompt engineering and instruction fine-tuning based on real classroom data. In this study, we utilized traditional prompt engineering (Prompt Example detail see Appendix C.1.2), few-shot ICL (Prompt Example detail see Appendix C.2.2)(Brown (2020)), and CoT (Prompt Example detail see Appendix D.5)(Wei et al. (2022)), while employing LoRA (Hu et al. (2021)) for fine-tuning the model.This combination ensures that generated dialogues realistically reflect student behaviors in actual learning environments. By integrating these methods, the generated dialogues closely resemble real-life classroom interactions, capturing the human-like and personalized traits of virtual student agents.

### B.2.3 THE UNIQUE VALUE OF MIDDLE SCHOOL STUDENTS AND THE LANGUAGE-BASED SUBJECT IN THIS STUDY

In this study, we focus on early adolescents (ages 10-15, *i.e.*, middle school students) as the target group for virtual student modeling, drawing insights from cognitive and developmental psychology. Students at this stage demonstrate unique cognitive development and language expression characteristics, presenting distinct challenges. Prior research indicates that early adolescents are transitioning from concrete to abstract thinking, with cognitive development not yet fully matured (Piaget (1952)). Compared to other age groups, these students have relatively weaker logical reasoning skills and often rely more on intuitive perception and concrete experiences for problem-solving. This cognitive limitation frequently results in unclear language expression and inconsistent thought processes when dealing with complex tasks (Kail et al. (2016)). Therefore, modeling students at this developmental stage offers a more accurate representation of diverse classroom behaviors while posing significant challenges for large language models in handling tasks requiring complex cognitive abilities.

Early adolescents often exhibit limited language expression, characterized by high levels of uncertainty and repetition in verbal responses during classroom interactions, especially with open-ended questions. This hesitation and lack of confidence stem not only from lower cognitive development but also from limited language proficiency (Cole & SCRIBNER (1978)). Thus, selecting this age group as the target for virtual student modeling holds substantial representational value, particularly for assessing whether large language models can effectively simulate students' language and cognitive responses, creating human-like, personalized virtual students. Compared to students at other developmental stages, early adolescents tend to offer simpler logical responses and are more easily influenced by external stimuli, making modeling this group both more challenging and valuable for research.

Building on this foundation, we use Information Processing Theory to further analyze the specific cognitive processes involved when students answer questions in the classroom. According to this theory, the thinking process can be divided into three stages: focusing on the question, understanding the question, and providing a response (Atkinson (1968)). First, students must direct their attention to the teacher's question, requiring adequate attention and selective information-processing abilities. Next, they need to understand the question's meaning and relate it to their existing knowledge, demonstrating cognitive processing skills. Finally, students generate a response, relying not only on their knowledge but also on their ability to articulate their thoughts clearly. In virtual student modeling, it is crucial to accurately represent these three stages to ensure appropriate responses within realistic classroom scenarios, effectively simulating students' thought processes and cognitive reactions.

We selected language-based subjects as the modeling context because language teaching emphasizes language expression, emotional experience, and value cultivation (Bruner et al. (1966)). This focus aligns closely with the core capabilities of large language models (LLMs), which are adept at processing and generating natural language. Language subjects require not only understanding and applying fundamental knowledge such as grammar and vocabulary but also involve deep interpretation of complex texts and emotional expression. This combination of emotion and cognition provides an ideal platform for evaluating and optimizing virtual student models, particularly in assessing whether they can replicate real students' linguistic styles, emotional expressions, and responses to value-based questions.

In summary, this study selects early adolescents as the target group for virtual student modeling due to their developing cognitive and linguistic abilities, as well as the representative complexity of their classroom interactions. Additionally, language-based subjects serve as an optimal testing ground for evaluating large language models, offering comprehensive insights into language expression, emotion, and attitudes. This approach provides a solid theoretical foundation and practical value for the future development of virtual student agents.

## C    Detailed Information for Dataset Construction Process

### C.1    Basic Chinese Understanding Ability Evaluation Dataset

#### C.1.1    Data source

The dataset utilized in this study is sourced from the **National Smart Education Platform**, an initiative developed by the Ministry of Education to provide authoritative and standardized educational materials for teachers and students nationwide. This platform integrates the official junior high school Chinese language curriculum, comprising textbook passages, practice exercises, comprehension assessments, and recitation tasks. These resources encapsulate the essential knowledge and skills required for junior high school Chinese education, adhering to national standards.

The dataset is focused on two pivotal aspects: **text comprehension** and **text memorization**. Both are critical not only for evaluating student performance in junior high school Chinese courses but also for assessing overall language proficiency. To ensure comprehensive and diverse content, we selected a wide range of lessons and corresponding exercises.

For **text comprehension**, the dataset includes various passages, chapters, and comprehension questions designed to assess students' ability to understand the structure of texts, analyze sentence and paragraph organization, infer details, explain vocabulary in context, and interpret emotional tones. These tasks allow us to evaluate whether LLMs can effectively process and understand Chinese texts at the junior high level, replicating the skills expected of students in language comprehension.

**Text memorization**, particularly through recitation, plays a unique role in Chinese language education, where students are required to internalize the content, language structures, and literary qualities of texts. Our dataset includes a series of standardized recitation tasks aimed at evaluating how accurately and completely LLMs can simulate students' text recitations. This aspect tests the model's ability to recall and reproduce text, serving as a benchmark for assessing long-term memory and linguistic retention.

The dataset is standardized and regularly updated by expert teams to ensure consistency with the national curriculum. This ensures that the data maintains a high degree of authority, objectivity, and applicability across educational settings. Data was collected from various question types to thoroughly assess the performance of foundational models in junior high school Chinese language scenarios, providing a solid foundation for future model development and refinement.

#### C.1.2    Data construction process

The data construction process is divided into three main steps: **data preparation**, **prompt design**, and **expert revision**. The goal of this process is to ensure the accuracy and scientific rigor of the data, providing high-quality inputs for the foundational model's scenario evaluation.

**Data Preparation**: The junior high school Chinese exercises were organized unit by unit, ensuring that each lesson covered the core knowledge points and skill requirements related to the text. The exercises for each unit were compiled and converted into PDF format for subsequent input into the GPT-4 model. This approach not only maintained the integrity of the data structure but also provided a clear framework for the model's processing in later stages.

**Prompt Design**: An instruction-based learning method was employed, in which structured prompts with "task description-task requirements-specific output" were provided to guide the model in executing designated tasks. By providing clear task instructions, the model could focus on specific objectives and generate outputs aligned with expectations. This design ensured that the model's outputs were accurate, particularly for complex tasks such as text comprehension and recitation.

**Expert Revision**: To ensure the accuracy of the generated data, a team of professional junior high school Chinese teachers was invited to review and refine the content produced by GPT-4. The expert team evaluated the model's outputs for both text comprehension and recitation, checking for accuracy, consistency, and alignment with educational standards. Any errors identified were carefully corrected by the teachers, ensuring that the dataset met the practical requirements of junior high school Chinese education. This expert revision process further optimized and enhanced the quality of the generated content.

In total, 613 text comprehension items and 438 memorization tasks were collected. The expert-revised data significantly improved the dataset's overall quality, providing a more rigorous and reliable foundation for subsequent model evaluations.

---

**The prompt for text comprehension generation (by GPT4).**

**Prompt**

You are an experienced first year language teacher who knows the content of the {*filename*} very well, and you can adapt the {*filename*} homework exercise provided to 10 multiple-choice questions with four choices, A, B, C, and D, with exact answers, designed to determine whether students have basic judgement about the content of the text.

Here provides {*filename*} homework exercises: {*content*}. You can modify the content of the homework exercise to make it a multiple-choice question with exact answer with instruction:
———

## INSTRUCTION:
- The multiple-choice questions must include four options: A, B, C, and D, with one and only one correct answer.

- - - - - - - - - - - - - - - - - - - - - - - - - - - - - - - - - - - - - - - - - - - - - - - - - -

**User:**

Please generate 10 multiple-choice questions that assess whether students have a basic understanding of the text content and FOLLOW THE FORMAT OF THE PROVIDED EXAMPLE STRICTLY.
———

## EXAMPLE FORMAT:
Question 1
Question: XXXXX
A: XXXXX
B: XXXXX
C: XXXXX
D: XXXXX
Correct Answer: X
Explanation: XXXXXXXX

---

这是《世说新语》二则课文1段的师生对话
------------------------------------------------

【题目1】
**问题：**《世说新语》是由哪位人物组织编写的?
A: 陈寿
B: 刘义庆
C: 司马迁
D: 班固
**正确答案：** B
**解释：**《世说新语》是由南朝刘宋时期的刘义庆所编写的。

【题目2】
**问题：**《咏雪》一文中，文中用哪种比喻形容雪?
A: 白茅
B: 柳絮
C: 飞蓬
D: 繁霜
**正确答案：** B
**解释：**《咏雪》一文中，作者谢道韫用柳絮来比喻轻盈纷飞的雪。

【题目3】题目3
**问题：** "太丘舍去"中的"去"的意思是?
A: 离开
B: 距离
C: 去除
D: 失去
**正确答案：** A
**解释：** "去"在这里的意思是"离开"，表明陈太丘离开了居住的地方。

【题目4】
**问题：** 在古人的称谓中，自己的屋子应该被称为?
A: 寒舍
B: 令尊
C: 家严
D: 赐教
**正确答案：** A
**解释：** 古人谦称自己的屋子为"寒舍"。

【题目5】
**问题：**《陈太丘与友期行》一文中，朋友是怎样的人?
A: 率真随性
B: 拘谨守时
C: 精明果断
D: 诚实可靠
**正确答案：** A
**解释：** 在《陈太丘与友期行》中，朋友与陈太丘一样，也是率真随性的人。

【题目6】
**问题：**《咏雪》一文中，谢道韫通过什么依据进行比喻?
A: 雪的冷
B: 雪的纯净
C: 雪的轻柔
D: 雪的颜色
**正确答案：** C
**解释：** 谢道韫是通过雪的轻柔来将其比喻为柳絮。

Figure A1: An example of text comprehension (original Chinese version).

An example of text comprehension (translated in English for better understanding).

Here is the English translation of the teacher-student dialogue from "Shishuo Xinyu":

- - - - - - - - - - - - - - - - - - - - - - - - - - - - - - - - - - - - - - - - - - - - - -

**[Question 1]**
**Question**: Who organized the compilation of "Shishuo Xinyu"?
A: Chen Shou
B: Liu Yiqing
C: Sima Qian
D: Ban Gu
**Correct Answer**: B
**Explanation**: "Shishuo Xinyu" was compiled by Liu Yiqing during the Southern Dynasties period of the Liu Song dynasty.

**[Question 2]**
**Question**: What metaphor is used to describe snow in "Ode to Snow"?
A: White grass
B: Willow catkins
C: Flying weeds
D: Abundant frost
**Correct Answer**: B
**Explanation**: In "Ode to Snow", the author Xie Daoyun uses willow catkins to metaphorically describe the light and fluttering snow.

**[Question 3]**
**Question**: What does "qu" mean in "Taiqiu she qu"?
A: Leave
B: Distance
C: Remove
D: Lose
**Correct Answer**: A
**Explanation**: "qu" here means "leave", indicating that Chen Taiqiu has left his place of residence.

**[Question 4]**
**Question**: What should one's own house be called in ancient times?
A: Humble abode
B: Your esteemed father
C: Family honor
D: Teach me
**Correct Answer**: A
**Explanation**: In ancient times, people humbly referred to their own house as "humble abode".

**[Question 5]**
**Question**: What kind of person is the friend in "Chen Taiqiu and His Friend's Journey"?
A: Genuine and easygoing
B: Reserved and punctual
C: Astute and decisive
D: Honest and reliable
**Correct Answer**: A
**Explanation**: In "Chen Taiqiu and His Friend's Journey," the friend is also a genuine and easygoing person, like Chen Taiqiu.

**[Question 6]**
**Question**: What basis does Xie Daoyun use for the metaphor in "Ode to Snow"?
A: The coldness of snow
B: The purity of snow

C: The softness of snow
D: The color of snow
**Correct Answer**: C
**Explanation**: Xie Daoyun uses the softness of snow to metaphorically compare it to willow catkins.

---

**The prompt for text memorization generation (by GPT4).**

**Prompt**
You are an experienced middle school Chinese language teacher with extensive knowledge of the content {*filename*}. You have been provided with the recitation passages from the text. Your task is to create ten multiple-choice questions, each with four options (A, B, C, and D), and only one correct answer. These questions are intended to assess students' ability to recall specific content from the text.
Here is the recitation passage from {*filename*}: {*content*}. Based on the original text, please create the multiple-choice questions with the instructions:
——

## INSTRUCTION:
- The multiple-choice questions must include four options: A, B, C, and D, with one and only one correct answer.
- These questions must be designed to evaluate students' ability to recall specific content from the passage.
- Ensure that each question is based on the original text without rewriting any part of it.
- You may use {} for students to fill in with words, phrases, or sentences directly from the original text, or ask them to translate specific words, phrases, or sentences from the passage.

- - - - - - - - - - - - - - - - - - - - - - - - - - - - - - - - - - - - - - - - - - - - -

**User:**
Please generate 10 multiple-choice questions with four options (A, B, C, D) and one correct answer based on the recitation passage from {*filename*}.
Ensure that no question is repeated and follow the example format provided:
——

## EXAMPLE FORMAT:
Question 1
Question: XXXXX, {}XXXX
A: XXXXX
B: XXXXX
C: XXXXX
D: XXXXX
Correct Answer: X
Explanation: XXXXXXXX

---

**An example of text memorization (translated in English for better understanding).**

Here is the English translation of the teacher-student dialogue from "From the Hundred Herb Garden to the Three Flavor Study Room" by Lu Xun:

- - - - - - - - - - - - - - - - - - - - - - - - - - - - - - - - - - - - - - - - - - - - -

**[Question 1]**
**Question**: Not to mention the green vegetable plots, the smooth {}, the tall soap pod tree, and the purple-red mulberries.
A: Stone well railing
B: Wooden fence
C: Bamboo fence
D: Brick wall railing
**Correct Answer**: A
**Explanation**: In the original text, "smooth stone well railing" describes a part of the Hundred

这是《鲁迅》从百草园到三味书屋课文1段的师生对话
------------------------------------------------

【题目1】
**问题：**不必说碧绿的菜畦，光滑的_____，高大的皂荚树，紫红的桑椹。
A: 石井栏
B: 木板栏
C: 竹篱笆
D: 砖墙栏
**正确答案：**A
**解释：**原文中"光滑的石井栏"描述了百草园的某一部分，选项A与原文一致。

【题目2】
**问题：**也不必说鸣蝉在树叶里_____，肥胖的黄蜂伏在菜花上。
A: 短叫
B: 嘶吼
C: 轻吟
D: 长吟
**正确答案：**D
**解释：**文中描述蝉的叫声为"鸣蝉在树叶里长吟"，所以正确答案为D。

【题目3】
**问题：**单是周围的短短的泥墙根一带，就有_____趣味。
A: 有限
B: 无限
C: 特定
D: 一些
**正确答案：**B
**解释：**文章中提到"周围的短短的泥墙根一带就有无限趣味"，表示这个地方充满了趣味。

【题目4】
**问题：**_____在这里低唱，蟋蟀们在这里弹琴。
A: 知了
B: 麻雀
C: 蜜蜂
D: 油蛉
**正确答案：**D
**解释：**原文提到"油蛉在这里低唱"，表现出了百草园的生机。

【题目5】
**问题：**翻开断砖来，有时会遇见_____。
A: 蚯蚓
B: 蜈蚣
C: 蚂蚱
D: 蝗虫
**正确答案：**B
**解释：**文中描述翻开断砖可能遇到的是"蜈蚣"。

【题目6】
**问题：**还有斑蝥，倘若用手指按住它的_____，便会'拍'的一声，从后窍喷出一阵烟雾。
A: 脊梁
B: 头部
C: 腹部
D: 尾部
**正确答案：**A
**解释：**原文中提到如果按住斑蝥的"脊梁"，它会喷烟。

Figure A2: An example of text memorization (original Chinese version).

Herb Garden, and option A is consistent with the text.

**[Question 2]**
**Question**: Nor need we mention the cicadas {} in the leaves, and the fat wasps resting on the vegetable flowers.
A: Short chirping
B: Roaring
C: Soft humming
D: Long humming
**Correct Answer**: D
**Explanation**: The text describes the sound of cicadas as "long humming in the leaves," so the correct answer is D.

**[Question 3]**
**Question**: Just around the short mud walls, there is {} interest.
A: Limited
B: Infinite
C: Specific
D: Some
**Correct Answer**: B
**Explanation**: The text states that "around the short mud walls there is infinite interest," indicating that this place is full of charm.

**[Question 4]**
**Question**: {} sings softly here, while crickets play music.
A: Cicada
B: Sparrow
C: Bee
D: Oil beetle
**Correct Answer**: D
**Explanation**: The original text mentions that "the oil beetle sings softly here," showcasing the

vitality of the Baicao Garden.

**[Question 5]**
**Question**: When you lift a broken brick, you may sometimes encounter {}.
A: Earthworm
B: Centipede
C: Grasshopper
D: Locust
**Correct Answer**: B
**Explanation**: The text describes that lifting a broken brick may reveal a "centipede."

**[Question 6]**
**Question**: There are also oil beetles; if you press its {} with your finger, it will 'pop' and release a cloud of smoke from its rear.
A: Backbone
B: Head
C: Abdomen
D: Tail
**Correct Answer**: D
**Explanation**: The original text states that pressing the oil beetle's "backbone" will make it emit smoke.

## C.2 FINE-TUNING DATASET

### C.2.1 DATA SOURCE

The fine-tuning dataset used in this work aligns with the Basic Chinese Understanding Ability Evaluation Dataset in previous section, drawing from real classroom video recordings, textbook content, and lesson plans prepared by teachers. The diversity and credibility of these data sources ensure that the large language model can generate responses grounded in authentic classroom interactions and instructional content, accurately reflecting real-world teaching scenarios.

**Real Classroom Videos**: These videos capture genuine interactions between teachers and students, including questioning, responses, and feedback. By transcribing and analyzing these videos, we extract teacher-student dialogue data from various classroom scenarios, enabling the large language model to learn real classroom communication patterns. This ensures that virtual students' linguistic behavior closely mirrors the interactional characteristics of actual classrooms.

**Textbook Content**: Textbooks are the core material for classroom instruction, encompassing subject knowledge and educational objectives that students are expected to master. Utilizing textbook data ensures that virtual students demonstrate a solid understanding of relevant subject matter and align with instructional goals when generating responses. This provides a knowledge-based foundation for logical and scientifically accurate responses.

**Lesson Plans**: Lesson plans outline the objectives and content of each teaching session and detail the design of each instructional phase. This framework guides GPT-4 in generating different types of questions, allowing virtual students to adapt to diverse instructional contexts accurately. The instructional design embedded in lesson plans ensures that the generated questions align with specific teaching objectives, leading virtual students to produce appropriate responses at different stages. Fine-tuning based on high-quality teacher lesson plans enhances the relevance and effectiveness of virtual student interactions in classroom settings.

### C.2.2 DATA CONSTRUCTION PROCESS

The construction process of the fine-tuning dataset involves several critical steps to ensure the generated virtual student dialogues are highly realistic and personalized. These steps include data preparation, prompt design based on real teacher-student dialogues, small-scale expert revision, large-scale generation of Big Five personality-based dialogues, and the creation of instruction fine-tuning

datasets. By integrating in-context learning prompt techniques, the model's ability to generate dialogues that accurately reflect the distinct personalities of students is significantly enhanced.

**Data Preparation**: Representative Chinese language texts for middle school were selected from the National Smart Education Resource Platform, encompassing eight genres from the first-year curriculum. Corresponding real classroom videos were transcribed to capture key teacher-student interactions, such as questioning, responses, and summaries. The extracted dialogue data served as valuable linguistic material for fine-tuning, ensuring that virtual students could generate language that mirrors authentic classroom interactions.

**Prompt Design Based on Real Teacher-Student Dialogues**: To model dialogues reflecting the Big Five personality traits, we employed few-shot in-context learning prompts based on real classroom interactions. This method provided the model with example dialogues, guiding it to generate new, personality-aligned exchanges between teachers and students. These prompts enabled the model to produce realistic dialogues that reflect different personality traits, enhancing the personalized behavior of virtual students.

---

The prompt for big five teacher-student dialogue examples based on real teacher-student dialogues (by GPT-4).

**Prompt**
You are an expert Chinese language teacher, well-versed in the characteristics and response styles of first-year middle school students. Here providing you with the teacher-student dialogue examples from real classroom interactions, as well as a teaching plan for the lesson from {*filename*}. Please generate five teacher-student dialogues based on the examples and the teaching plan, specifically targeting a student with the personality trait of {*student personality*}.
Here is the teaching plan for {*filename*}. Please generate teacher-student dialogues according to the following instructions:
——
## INSTRUCTION:
- The generated dialogues should align with the personality characteristics of a student with student personality, as described: {*student personality description*} (student detail description see A2).
- The content of the dialogues must correspond to the topics covered in the provided teaching plan.

- - - - - - - - - - - - - - - - - - - - - - - - - - - - - - - - - - - - - - - - -

**User:**
Below are examples of teacher-student dialogues from real classroom interactions. Please explain step by step, using a chain-of-thought reasoning process, why you generated the teacher-student dialogue for a student with {*student personality*}:
——
## REAL-CLASS DIALOGUES:
Dialogue 1:
Teacher: Now, could one of you share with me a word that comes to mind?
Student: Magnificent.

Dialogue 2:
Teacher: Now, could one of you tell us what kind of person you think Cao Cao is, based on what you've seen in television dramas?
Student: He is straightforward and very talented.

Dialogue 3:
Teacher: The first question—who can tell us what the emotional tone of the article is?
Student: This article expresses the author's deep affection for spring.

Dialogue 4:
Teacher: What is the central character around which the article develops?
Student: The character 'view' is central to the article's development.

---

Dialogue 5:
Teacher: What did you see? Can anyone share with us?
Student: I saw the lake, uh, and the peach trees, the peach, peach, peach, apricot, and pear trees all blooming at once.

Dialogue 6:
Teacher: What indicates this?
Student: The landscape.

Dialogue 7:
Teacher: What does the word 'tranquil' mean?
Student: Tranquil, um, it means very peaceful.

Dialogue 8:
Teacher: So, what role do the last two paragraphs play?
Student: I think they summarize the entire piece and reinforce the author's emotional attachment to the rain.

由GPT-4生成的基于真实师生对话的高神经质学生的师生对话示例：

【对话生成：按照步骤思考】
-**步骤1：** 理解学生的关键特征
高度神经质：这类学生倾向于表现出紧张不安，常使用"嗯"和"呃"这样的填充词，由于不确定性，他们经常重复或回溯自己的回答。
-**步骤2：** 与教学目标对齐
内容：本课重点是阅读并理解课文，欣赏生动的意象，以及分析作者的情感和抱负。教师将提出关于诗歌含义、意象以及曹操性格的问题，而学生的回答将反映出他们的紧张和不确定性。
-**步骤3：** 在对话中融入高度神经质的特征
学生的回答将表现出犹豫和重复，而教师将给予支持性指导，鼓励学生在紧张的情况下表达自己的想法。

**对话 1**
**Teacher:** 请你说一说，《观沧海》这首诗中的大海给你什么样的感觉？
**Student:** 嗯，嗯，大海...嗯，嗯，感觉很...很大，嗯，很，嗯，嗯，广阔...呃，还有...嗯，嗯，威严...很...很有气势。
**Teacher:** 很好！你注意到了大海的广阔和气势，那你觉得这种景象表现了曹操什么样的情感呢？
**Student:** 嗯，嗯，我觉得...嗯，曹操...呃，他，他很有...嗯，雄心壮志，嗯，嗯，想要...嗯，统一天下吧。

**对话 2**
**Teacher:** 诗中提到"水何澹澹，山岛竦峙"，你能解释一下这两句诗吗？
**Student:** 嗯，嗯，那个...水，水很...嗯，很，嗯，澹澹，就是...嗯，嗯，海浪...嗯，很，嗯，轻轻地动，呃，山...山岛，嗯，嗯，很高...嗯，站在那里...嗯，嗯，像...嗯，很威严的样子。
**Teacher:** 对，这两句确实描绘了大海和山岛的动静结合，表现了辽阔和威严的景象。

**对话 3**
**Teacher:** 你觉得"日月之行，若出其中；星汉灿烂，若出其里"这两句有什么特殊的意境？
**Student:** 嗯，嗯，这...这两句...嗯，嗯，感觉...嗯，嗯，日月...嗯，星星，嗯，好像都...嗯，都从...从海里出来，嗯，嗯，嗯，很壮丽的画面，嗯，嗯，感觉...嗯，很，嗯，很大气。
**Teacher:** 非常好！这些句子通过虚景表现了大海的吞吐日月星辰，展现了曹操的胸怀和志向。

**对话 4**
**Teacher:** 你觉得《观沧海》中的曹操是一个怎样的人呢？
**Student:** 嗯，嗯，曹操...嗯，嗯，我觉得他...嗯，他很有...嗯，嗯，抱负，嗯，嗯，想要...嗯，统一天下，嗯，嗯，他...嗯，嗯，很，嗯，自信，嗯，嗯，乐观。
**Teacher:** 很好，你抓住了曹操的雄心壮志和乐观的精神。

**对话 5**
**Teacher:** 诗中提到了"秋风萧瑟，洪波涌起"，你觉得这两句描绘了什么样的景象呢？
**Student:** 嗯，嗯，嗯，秋风...嗯，很...很冷，嗯，嗯，树叶...嗯，嗯，嗯，发出声音...嗯，洪波...嗯，很大的...嗯，海浪...嗯，嗯，起来了，嗯，嗯，感觉...嗯，很...很有力量。
**Teacher:** 很好，这两句确实表现了大海的宏伟气势，也传达了曹操的豪情壮志。

Figure A3: An example of HN LVSA teacher-student dialogue generated by GPT-4 (original Chinese version).

**An example of HN LVSA teacher-student dialogue generated by GPT-4 (translated in English for better understanding).**

Based on the teaching plan for "View the Sea" by Cao Cao and the characteristics of a highly neurotic student, Here are five teacher-student dialogues that reflect the described classroom dynamics:

---

**[Dialogue Generation: Step-by-Step Chain of Thought]**
- **Step 1** : Understand the Key Characteristics of the Student
**High neuroticism**: The student is prone to anxiety, uses filler words like "en" and "um," and exhibits repetition, nervousness, and backtracking in their speech.
- **Step 2**: Align Dialogue with Teaching Objectives
**Content**: The lesson focuses on reading and understanding the text, appreciating the vivid imagery, and analyzing the author's emotions and ambitions. The teacher will ask questions about the meaning of the poem, the imagery, and Cao Cao's character, while the student's responses will reflect their nervousness and uncertainty.
- **[Step 3]**: Incorporate High Neuroticism Traits into Dialogue
**Student Performance**: The student's responses will show hesitation and repetition, while the teacher will provide supportive guidance, encouraging the student to express their ideas despite their nervousness.

**[Dialogue 1]**
**Teacher**: Can you tell us what kind of feeling the sea gives you in the poem "View the Sea"?
**Student**: Um, um, the sea... um, um, it feels very... very vast, um, very, um, um, wide... uh, and also... um, um, majestic... very... very powerful.
**Teacher**: Good! You noticed the vastness and majesty of the sea. What kind of emotions do you think Cao Cao is expressing through this scene?
**Student**: Um, um, I think... um, Cao Cao... uh, he, he has... um, great ambition, um, um, and he wants... um, to unify the whole country, I think.

**[Dialogue 2]**
**Teacher**: The poem mentions "The water is vast and billowing, and the islands stand tall." Can you explain these two lines?
**Student**: Um, um, well... the water, the water is... um, um, vast, um, like... um, the waves... um, are gently moving, uh, and the islands... um, um, they stand tall... um, like they are... um, very majestic.
**Teacher**: Right, these lines indeed describe the combination of movement and stillness between the sea and the islands, reflecting the vast and majestic scenery.

**[Dialogue 3]**
**Teacher**: What kind of special imagery do you think is conveyed in the lines "The sun and moon travel across the sky; the Milky Way shines brightly, as if emerging from the sea"?
**Student**: Um, um, these... these two lines... um, um, they give a feeling... um, um, of the sun and moon... um, and the stars, um, as if they all... um, come out... from the sea, um, um, it's a very magnificent picture, um, um, and it feels... um, um, very grand.
**Teacher**: Excellent! These lines use imagination to depict the sea as swallowing up the sun, moon, and stars, showcasing Cao Cao's vast ambitions and vision.

**[Dialogue 4]**
**Teacher**: What kind of person do you think Cao Cao is in "View the Sea"?
**Student**: Um, um, Cao Cao... um, um, I think he... um, he has... um, great ambition, um, um, and he wants... um, to unify the country, um, um, and he... um, um, is very... um, confident, um, um, and optimistic.
**Teacher**: Very good! You've captured Cao Cao's great ambition and optimistic spirit.

**[Dialogue 5]**
**Teacher**: The poem mentions "The autumn wind is bleak, and great waves rise." What kind of scene do you think these lines depict?
**Student**: Um, um, um, the autumn wind... um, it's... it's cold, um, um, the leaves... um, um, um, are making sounds... um, and the great waves... um, they rise... um, um, and it feels... um, very... very powerful.
**Teacher**: Great! These lines indeed convey the grandeur of the sea and also reflect Cao Cao's heroic ambitions.

**Small-Scale Expert Revision**: Two experienced middle school teachers, each with over two years of teaching experience, were invited to review the generated sample data. Their expertise ensured that the student's language style was aligned with the described personality traits. Any inconsistencies were corrected, further refining the quality of the data and providing a scientifically robust template for subsequent large-scale dialogue generation.

---

An example of HN LVSA teacher-student dialogue generated by GPT-4 (Modified by Experts).

Based on the high neuroticism teacher-student dialogue examples generated by GPT-4, the following are the teacher-student dialogues revised by experts according to real teaching scenarios:

- - - - - - - - - - - - - - - - - - - - - - - - - - - - - - - - - - - - - - - - - - - - - - - - - -

**[Dialogue 1]**
**Teacher**: Good! You noticed the vastness and majesty of the sea. What kind of emotions do you think Cao Cao is expressing through this scene?
**Student**: Um, um, I think... um, Cao Cao... uh, he has... um, big goals, um, and he wants... um, to change everything, I think.

**[Dialogue 2]**
**Teacher**: The poem mentions "The water is vast and billowing, and the islands stand tall." Can you explain these two lines?
**Student**: Um, the water... um, is big, um, and the waves... um, move softly, uh, and the islands... um, um, are tall... um, and look very strong.

**[Dialogue 3]**
**Teacher**: What kind of special imagery do you think is conveyed in the lines "The sun and moon travel across the sky; the Milky Way shines brightly, as if emerging from the sea"?
**Student**: Um, um, these... these two lines... um, um, they give a feeling... um, um, of the sun and moon... um, and the stars, um, as if they all... um, come out... from the sea, um, um, it's a very magnificent picture, um, um, and it feels... um, um, very grand.

**[Dialogue 4]**
**Teacher**: What kind of person do you think Cao Cao is in "View the Sea"?
**Student**: Um, um, these... these two lines... um, um, feel like... um, the sun, moon, and stars... um, all come... from the sea, um, um, it's a very big picture, um, um, and it feels... um, um, really great.

**[Dialogue 5]**
**Teacher**: The poem mentions "The autumn wind is bleak, and great waves rise." What kind of scene do you think these lines depict?
**Student**: Um, um, the autumn wind... um, it's cold, um, um, the leaves... um, are making noise... um, and the big waves... um, are rising... um, and it feels... um, very strong.

---

HN LVSA example dialogues used in large-scale generation prompt (translated in English for better understanding).

After experts revised all of the Big Five personality teacher-student dialogue examples, five dialogues were selected from each personality type as prompt few-shot for large-scale generation of Big Five personality dialogues including learning stages. Below are teacher-student dialogue few-shots for the high neuroticism personality type:

- - - - - - - - - - - - - - - - - - - - - - - - - - - - - - - - - - - - - - - - - - - - - - - - - -

**[Dialogue 1]**
**Teacher**: When we talk about spring, people always associate it with many beautiful images of spring scenery and poetry, like "The grass grows long in February, the swallows fly by," followed by another line?
**Student**: Um, "The willows sway in the breeze, the willows drunk on the spring haze."
**Question type**: Closed-ended question

**大规模生成提示语中使用的高神经质学生对话范例**
--------------------------------------------------

【对话1】
**老师：** 提起春天呢，人们总能想到许多美丽的形容春天的成语和诗句，比如草长莺飞二月天，接下一句？
**学生：** 嗯，拂堤...拂堤.呃，杨柳醉春烟。
**问题类型：** 封闭式问题
**学习阶段：** 课前引入

【对话2】
**老师：** 你能解释一下"洪波涌起'的意思吗？
**学生：** 嗯，"洪波涌起'就是，呃，大海的波浪很...很大。形容，形容波浪很激烈的样子。
**问题类型：** 封闭式问题
**学习阶段：** 新课讲授

【对话3】
**老师：** 请你朗读这段文字。
**学生：** 呃，好....曹操一生未称帝，病死后，曹丕继位后不久...不久称帝，追尊曹操为武帝。
**问题类型：** 封闭式问题
**学习阶段：** 课堂练习

【对话4】
**老师：** 请大家分组讨论，并绘制出《散步》这篇文章的思维导图。完成后，请小组代表详细说明你们的讨论结果？
**学生：** 嗯，我们，呃，呃，我们分了，嗯，三个部分。嗯，第一个部分是散步，嗯，就是全家一起走的，呃，路，象征了，嗯，家庭的和谐。然后，嗯，然后是，呃，选择道路，呃，母亲和儿子不一样，嗯，一个想走平坦的路，呃，一个想走小路。嗯，我们，我们画了两条线，呃，用不同颜色，嗯，表示，呃，表示不同的选择。
**问题类型：** 开放式问题
**学习阶段：** 巩固新知

【对话5】
**老师：** 最后，大家觉得为什么学习如此重要？结合我们今天学习的这篇文言文说说看。
**学生：** 嗯，学习，学习可以，呃，提高自己，嗯，让自己更有智慧，呃，像吕蒙一样。
**问题类型：** 开放式问题
**学习阶段：** 课堂小结

Figure A4: HN LVSA example dialogues used in large-scale generation prompt (original Chinese version).

---

**Learning stage**: Pre-lesson introduction

**[Dialogue 2]**
**Teacher**: Could you explain what "raging waves" means?
**Student**: Um, "raging waves" refers to... um, the waves are big... very big. It describes... the image of waves being very turbulent.
**Question type**: Closed-ended question
**Learning stage**: New lesson learning

**[Dialogue 3]**
**Teacher**: Cao Cao once said, "One cannot keep their strength forever, the wind and clouds will soon pass." Could anyone tell me what this means?
**Student**: Um, it means... uh, life is fleeting, nothing stays the same... um, and the wind... chasing clouds represents the passing of time.
**Question type**: Closed-ended question
**Learning stage**: Classroom exercises

**[Dialogue 4]**
**Teacher**: Please find this article and point out where it talks about "scattering." After that, I'd like the group to explain in detail how this part describes the scene.
**Student**: Um, um, we, uh, we divided it into, um, three parts. Uh, the first part is about walking, um, which represents the family walking together, uh, symbolizing, um, family harmony. Then, um, then there's, uh, the choice of the road, uh, the mother and son are different, um, one wants to take the smooth road, uh, the other wants to take the smaller path. Um, we, we drew two lines, uh, using different colors, um, to represent, uh, to represent the different choices.
**Question type**: Open-ended question
**Learning stage**: Knowledge consolidation

---

**[Dialogue 5]**
**Teacher**: Finally, why do you all think learning is so important? Use the classical text we studied today to explain your thoughts.
**Student**: Um, learning, learning can, uh, improve oneself, um, make one wiser, uh, like Lü Meng.
**Question type**: Open-ended question
**Learning stage**: Lesson summary

---

**Large-Scale Generation of Big Five Personality Teacher-Student Dialogue Data**: Using the expert-reviewed samples, we refined the prompt design for large-scale generation. Few-shot in-context learning prompts were used to generate teacher-student dialogues for different personality types across teaching stages, such as pre-lesson introduction and knowledge consolidation. The resulting dataset covers diverse Chinese language content and showcases distinctive classroom behaviors based on different personality traits, forming a comprehensive resource for fine-tuning.

---

**The prompt for large-scaled generation with few-shot in-context learning(by GPT-4).**

**Prompt**
You are an expert teacher who is well-versed in the personalities, language expression styles, and vocabulary levels of first-year junior high school students, as well as the content of {*filename*}."
With the provided lesson plan and examples of real classroom teacher-student dialogues, you will generate ten dialogues between a professional Chinese language teacher and a student with {*student personality*}'s personality. These dialogues should include both closed-ended and open-ended questions, and the dialogues should span five stages: pre-lesson introduction, new lesson learning, knowledge consolidation, classroom exercises, and lesson summary.
Closed-ended questions should directly target the student's recall and understanding of the content, such as asking about the author of the text or the student's knowledge of classical poetry related to the theme of the text. Open-ended questions should focus on the student's ability to apply, analyze, create, and evaluate the knowledge.
The stages are defined as follows: Pre-lesson introduction: Activities that the teacher undertakes before formal instruction to spark student interest, establish learning context, activate prior knowledge, or introduce new concepts, such as setting learning goals, establishing background knowledge, or activating prior knowledge. New lesson learning: The phase where the teacher introduces and explains new content or a new topic for the first time. This stage represents the students' initial exposure to the subject matter. Knowledge Consolidation : Activities that help students deepen their understanding and retention of the new knowledge, such as review, practice, and discussion to ensure that the students firmly grasp the material. Classroom practice: Exercises and activities conducted in the classroom under the teacher's guidance to help students understand, master, and apply the knowledge they have learned. Lesson summary: A process that occurs after the learning activities, where the student or teacher reviews and organizes the learned content.
Here provided the lesson plan content: *content* for "*filename*," and please generate ten dialogues with the instructions:
——
## INSTRUCTION:
- Please ensure that the generated dialogues align with student's personality traits as described: {*student personality description*}. (student detail description see A2).
- Note that the lesson plan is not the content of the text itself but the instructional process and methods used by the teacher. It serves as a guide for generating valuable questions in the teacher-student dialogues.
- The student's responses to closed-ended questions should be concise and precise, and the vocabulary and sentence structures used should match the level of a first-year junior high school student.
- The ten teacher-student dialogues should cover the following stages: pre-lesson introduction, new lesson instruction, consolidation of new knowledge, classroom practice, and lesson summary.
- The student's responses should not contain an excessive number of filler words.

- The content of the dialogues should closely follow the text "*filename*."

---

**User:**
Below are examples of teacher-student dialogues for a student with *student personality*'s personality. Please generate ten teacher-student dialogues following the examples format provided:
——
## FEW-SHOT EXAMPLES: {*student personality* few-shot} (few-shot see A4).

Figure A5: An example of HN LVSA fine-tuning data generated by GPT-4 (original Chinese version).

An example of HN LVSA fine-tuning data generated by GPT-4 (translated in English for better understanding).

---

Here are dialogues based on the lesson "Winter in Jinan" by Lao She:

---

**[Dialogue 1]**
**Teacher**: Class, do you know who wrote this essay?
**Student**: Uh, it's, it's, um, Mr. Lao She.
**Question Type**: Closed-ended question
**Learning Stage**: Pre-lesson introduction

**[Dialogue 2]**
**Teacher**: Before we start today's lesson on "Winter in Jinan", what are your expectations or thoughts about it?
**Student**: Um, I, I think, uh, maybe it will be... very, very cold.
**Question Type**: Open-ended question
**Learning Stage**: Pre-lesson introduction

**[Dialogue 3]**
**Teacher**: At the end of the essay, Lao She wrote, "I believe this is Jinan choosing the best way to transition through the seasons." Can you understand what this means?
**Student**: Uh, I think... think it means, um, maybe Jinan, um, chooses the best, the best way to, to go through it.
**Question Type**: Closed-ended question
**Learning Stage**: New lesson learning

**[Dialogue 4]**
**Teacher**: Can you try to explain your understanding of the phrase "Jinan's winter is a treasure"?
**Student**: Um, I think, um, maybe because, um, Jinan's winter, uh, is really beautiful, like a treasure, it's worth, worth us, um, discovering and, um, appreciating. Uh, that's, that's what I think.
**Question Type**: Open-ended question

**Learning Stage**: New lesson learning

**[Dialogue 5]**
**Teacher**: Now, in pairs, each of you choose the paragraph you think is written the best, and discuss it.
**Student**: Um, I, I choose, um, the fourth paragraph. I think, um, that part is described very, very vividly.
**Question Type**: Closed-ended question
**Learning Stage**: Knowledge consolidation

**[Dialogue 6]**
**Teacher**: Please explain from your perspective why you think the paragraph you chose is the best.
**Student**: Uh, I, I think, um, that part, that part's description, um, it, it made me feel, um, like I was, uh, right there, it was... it felt really real.
**Question Type**: Open-ended question
**Learning Stage**: Knowledge consolidation

**[Dialogue 7]**
**Teacher**: Can you translate this sentence into modern language?
**Student**: Um, this, this sentence, um, maybe it means, uh... something like, um, such beautiful scenery.
**Question Type**: Closed-ended question
**Learning Stage**: Classroom exercises

**[Dialogue 8]**
**Teacher**: Now imagine you're a tourist. Write a travel diary describing what you saw in Jinan's winter.
**Student**: Um, I, um, I think, uh, I, I would write about what I saw, um, in winter, um, in Jinan. Uh, it, it was really beautiful, um, I, I'll try to, try to write it, um, as vividly as I can.
**Question Type**: Open-ended question
**Learning Stage**: Classroom exercises

**[Dialogue 9]**
**Teacher**: So, what did we gain from studying this text today?
**Student**: Um, I, I think, uh, I, I learned, um, how to, to appreciate, um, the beauty of, um, winter, and, um, how to, to describe, uh, what I see in, um, winter.
**Question Type**: Open-ended question
**Learning Stage**: Lesson summary

**[Dialogue 10]**
**Teacher**: In today's lesson, how do you feel? Or is there something you'd like to share?
**Student**: Um, I, I think, uh, I, I feel like I, um, learned more about, um, the beauty of Jinan's winter, um, I, I hope, um, I can, I can go, and, um, see Jinan's winter with my own eyes.
**Question Type**: Open-ended question
**Learning Stage**: Lesson summary

---

An example of HA LVSA fine-tuning data generated by GPT-4 (translated in English for better understanding).

Here are dialogues based on the lesson "From the Hundred Grass Garden to the Three Flavor Study" by Lu Xun:

- - - - - - - - - - - - - - - - - - - - - - - - - - - - - - - - - - - - - - - - - - - - - - - - - - - -

**[Dialogue 1]**
**Teacher**: Class, do you know what plants in the Hundred Grass Garden are familiar to you? Like the cabbage flowers we often see.

Figure A6: An example of HA LVSA fine-tuning data generated by GPT-4 (original Chinese version).

**Student**: There are also mulberries and dandelions.
**Question Type**: Closed-ended question
**Learning Stage**: Pre-lesson introduction

**[Dialogue 2]**
**Teacher**: So, does the Hundred Grass Garden have any special meaning for Mr. Lu Xun?
**Student**: Yes, Mr. Lu Xun called it his "paradise" because he had many happy memories there.
**Question Type**: Closed-ended question
**Learning Stage**: Pre-lesson introduction

**[Dialogue 3]**
**Teacher**: Who is the author of this text? **Student**: It's Mr. Lu Xun. **Question Type**: Closed-ended question **Learning Stage**: New lesson learning

**[Dialogue 4]**
**Teacher**: The title "From the Hundred Grass Garden to the Three Flavor Study", what does it tell us about the content of the essay?
**Student**: It mainly talks about Mr. Lu Xun's change from playing in the Hundred Grass Garden when he was young to studying at the Three Flavor Study.
**Question Type**: Open-ended question
**Learning Stage**: New lesson learning

**[Dialogue 5]**
**Teacher**: In the text, how does Mr. Lu Xun describe the scenery of the Hundred Grass Garden?
**Student**: He uses many vivid words, like the "green" grass and "purple-red" flowers, to make the natural beauty of the place feel alive.
**Question Type**: Closed-ended question
**Learning Stage**: Knowledge consolidation

**[Dialogue 6]**
**Teacher**: Why do you think the Hundred Grass Garden brought happiness to Mr. Lu Xun?
**Student**: Because he could freely explore and play there, see many interesting insects and plants, and enjoy the beauty of nature.
**Question Type**: Open-ended question
**Learning Stage**: Knowledge consolidation

**[Dialogue 7]**
**Teacher**: What activities did Mr. Lu Xun do in the Hundred Grass Garden when he was young?
**Student**: He caught crickets, fished, and observed many different insects.
**Question Type**: Closed-ended question
**Learning Stage**: Classroom exercises

**[Dialogue 8]**
**Teacher**: Through these activities, what did Mr. Lu Xun learn?
**Student**: He not only learned about nature but also how to love and respect life.
**Question Type**: Open-ended question
**Learning Stage**: Classroom exercises

**[Dialogue 9]**
**Teacher**: We studied "From the Hundred Grass Garden to the Three Flavor Study" today. What was your favorite part of the text?
**Student**: I liked the part about the Hundred Grass Garden because it was full of life and freedom, and it reminded me of my own childhood.
**Question Type**: Open-ended question
**Learning Stage**: Lesson summary

**[Dialogue 10]**
**Teacher**: What impression did the Hundred Grass Garden leave on you through Mr. Lu Xun's description?
**Student**: It gave me the impression of a place full of wonder and exploration, where every corner was full of surprises.
**Question Type**: Closed-ended question
**Learning Stage**: Lesson summary

Figure A7: An example of HE LVSA fine-tuning data generated by GPT-4 (original Chinese version).

---

**An example of HE LVSA fine-tuning data generated by GPT-4 (translated in English for better understanding).**

Here are dialogues based on the lesson "LunYu":

- - - - - - - - - - - - - - - - - - - - - - - - - - - - - - - - - - - - - - - - - - - - - - - - -

**[Dialogue 1]**
**Teacher**: Do you know which time period The Analects is from?
**Student**: Teacher, I know, The Analects is from the Spring and Autumn period.
**Question Type**: Closed-ended question
**Learning Stage**: Pre-lesson introduction

**[Dialogue 2]**
**Teacher**: We're studying The Twelve Chapters of the Analects today. Do you have any expectations or thoughts about it?
**Student**: Teacher, I think The Analects is a treasure of Confucius's wisdom. I hope to learn more lessons from it to help me be a better person. I also want to better understand Confucius's ideas and apply them to my life.
**Question Type**: Open-ended question
**Learning Stage**: Pre-lesson introduction

**[Dialogue 3]**
**Teacher**: How do you feel about the sentences from The Analects we've read?
**Student**: Teacher, I think Confucius's words are really deep. Even though the sentences are short, each one is full of wisdom. I especially like, "Learning without thinking is useless, thinking without learning is dangerous." This really inspires me.
**Question Type**: Open-ended question
**Learning Stage**: New lesson learning

**[Dialogue 4]**
**Teacher**: What key words from the passage we just studied help us understand Confucius's ideas?
**Student**: Teacher, I think "between father and son," "the duty between ruler and subject," and "trust between friends" are key words. They help us understand Confucius's moral values.
**Question Type**: Closed-ended question
**Learning Stage**: New lesson learning

**[Dialogue 5]**
**Teacher**: What important lessons have we learned from reading The Twelve Chapters of the Analects? **Student**: Teacher, I feel like we've learned a lot of important lessons. For example, we should always be honest and sincere with our friends. Also, when studying, we need to

reflect and think, not just memorize. These lessons are really helpful for our lives. **Question Type**: Open-ended question **Learning Stage**: Knowledge consolidation

**[Dialogue 6]**
**Teacher**: So, what do you think about the idea we just learned?
**Student**: Teacher, I think Confucius's idea is very deep. I believe that life is all about constantly learning. We need to keep gaining new knowledge and reflecting on what we already know. This way, we can keep getting better.
**Question Type**: Open-ended question
**Learning Stage**: Knowledge consolidation

**[Dialogue 7]**
**Teacher**: Let's try this exercise. In classical Chinese, what does "between father and son" represent?
**Student**: Teacher, "between father and son" in classical Chinese represents the relationship between a father and his son. It's an important part of Confucius's view of ethics and morals.
**Question Type**: Closed-ended question
**Learning Stage**: Classroom exercises

**[Dialogue 8]**
**Teacher**: What do you think Confucius's words teach us today?
**Student**: Teacher, I think Confucius's words give us a lot of guidance. They tell us that we should value every chance to learn and keep improving our knowledge and skills. Also, we need to think and reflect on what we're learning.
**Question Type**: Open-ended question
**Learning Stage**: Classroom exercises

**[Dialogue 9]**
**Teacher**: In today's lesson, The Twelve Chapters of the Analects, what part did you find the hardest to understand?
**Student**: Teacher, I think the hardest part is some of Confucius's philosophies. Even though they're very deep, sometimes I don't really understand their meaning. I need more time to think and understand them.
**Question Type**: Open-ended question
**Learning Stage**: Lesson summary

**[Dialogue 10]**
**Teacher**: After studying The Twelve Chapters of the Analects today, what are your thoughts?
**Student**: Teacher, The Twelve Chapters of the Analects left a deep impression on me, especially Confucius's wisdom. I think his words are simple but full of meaning, and each sentence is very inspiring.
**Question Typ**e: Open-ended question
**Learning Stage**: Lesson summary

---

An example of LC LVSA fine-tuning data generated by GPT-4 (translated in English for better understanding).

Here are dialogues based on the lesson "Autumn Thoughts" by Shi Tiesheng:

- - - - - - - - - - - - - - - - - - - - - - - - - - - - - - - - - - - - - - - - - - - - - - - - - - - - - -

**[Dialogue 1]**
**Teacher**: Today, we're going to study Shi Tiesheng's Autumn Thoughts. Does anyone know who Shi Tiesheng is?
**Student**: Shi Tiesheng, he's a, a writer, right, right, he wrote I and the Temple of Earth.
**Question Type**: Closed-ended question
**Learning Stage**: Pre-lesson introduction

这是低尽责性人格学生关于史铁生的《秋天的怀念》课文1段的师生对话

**对话 1**
教师：我们这节课要学习的是史铁生的《秋天的怀念》，大家知道史铁生是谁吗？
学生：史铁生，他是个，是个作家，对对，他写了《我与地坛》。
问题类型：封闭式问题
学习阶段：课前引入

**对话 2**
教师：很好，那你知道《秋天的怀念》这篇文章主要讲述了什么吗？
学生：它是讲，讲故事吧，关于史铁生的故事。
问题类型：封闭式问题
学习阶段：课前引入

**对话 3**
教师：在《秋天的怀念》中，史铁生用"菊花"做了怎样的比喻？
学生：菊花，菊花比喻，嗯，比喻母亲？还是母爱？
问题类型：封闭式问题
学习阶段：新课讲授

**对话 4**
教师：你能解释一下文章中"我懂得母亲没有说完的话"的含义吗？
学生："母亲没有说完的话"，就是，就是说，嗯，史铁生懂了，母亲让他好好活下去的意思？
问题类型：封闭式问题
学习阶段：新课讲授

**对话 5**
教师：请把你的对文章的理解，用一句话总结下。
学生：这篇文章是史铁生对母亲的思念，和，对，对生活的，嗯，坚强面对吧。
问题类型：开放式问题
学习阶段：巩固新知

**对话 6**
教师：你能从文章中找出几处表现母爱的细节吗？
学生：母爱的细节，有，有母亲挡在窗前，然后还有，母亲和史铁生看菊花的对话，对了，还有她悄悄地出去。
问题类型：开放式问题
学习阶段：巩固新知

**对话 7**
教师：请你朗读文章中的一段并试着感受文章的情感色彩。
学生："我看看菊花，深深地怀念起她来。"我感觉，嗯，这句话很，很有感触，我能感觉到史铁生对母亲的思念，深深的思念。
问题类型：封闭式问题
学习阶段：课堂练习

**对话 8**
教师：文章的最后一段，"那里还有两尾金鱼在游动，秋菊在盛开。"，你觉得这里的秋菊和金鱼有什么象征意义？
学生：嗯，秋菊应该是代表的母亲，母亲的爱，然后金鱼，金鱼可能代表，嗯，生活吧，生活还要继续。
问题类型：开放式问题
学习阶段：课堂练习

**对话 9**
教师：这篇文章的主题是什么，你觉得史铁生想通过这篇文章表达什么？
学生：主题，嗯，就是怀念吧，史铁生通过文章，他可能想，想说要珍惜母爱，然后，然后，嗯，要坚强生活。
问题类型：开放式问题
学习阶段：课堂小结

**对话 10**
教师：史铁生用"好好活"这个词，来结束文章，你觉得这个词在文章中起到怎样的作用？
学生："好好活"，这个词，嗯，起的作用，应该是，嗯，就是史铁生对生活的态度，就是要活得坚强，活得有价值。
问题类型：开放式问题
学习阶段：课堂小结

Figure A8: An example of LC LVSA fine-tuning data generated by GPT-4 (original Chinese version).

**[Dialogue 2]**
**Teacher**: Good. So, do you know what Autumn Thoughts is mainly about?
**Student**: It's about, about a story, right? About Shi Tiesheng's story.
**Question Type**: Closed-ended question
**Learning Stage**: Pre-lesson introduction

**[Dialogue 3]**
**Teacher**: In Autumn Thoughts, how does Shi Tiesheng use "chrysanthemums" as a metaphor?
**Student**: Chrysanthemums, uh, they're a metaphor, um, for the mother? Or maybe mother's love?
**Question Type**: Closed-ended question
**Learning Stage**: New lesson learning

**[Dialogue 4]**
**Teacher**: Can you explain the meaning behind the phrase "I understand the words my mother left unfinished"?
**Student**: "Words left unfinished," it means, um, Shi Tiesheng understood, that his mother wanted him to live well, right?
**Question Type**: Closed-ended question
**Learning Stage**: New lesson learning

**[Dialogue 5]**
**Teacher**: Can you summarize your understanding of the article in one sentence?
**Student**: This article is about Shi Tiesheng's memory of his mother, and, uh, about, about facing life strongly, I guess.
**Question Type**: Open-ended question
**Learning Stage**: Knowledge consolidation

**[Dialogue 6]**
**Teacher**: Can you find a few details from the text that show motherly love?
**Student**: Details of motherly love, um, there's the part where the mother stood by the window, and then, um, the conversation about chrysanthemums, and oh, when she quietly left.
**Question Type**: Open-ended question
**Learning Stage**: Knowledge consolidation

**[Dialogue 7]**
**Teacher**: Please read aloud a passage from the article and try to feel the emotion behind it.
**Student**: "I looked at the chrysanthemums and deeply missed her." I feel, um, this sentence is very, very emotional, I can feel Shi Tiesheng's deep longing for his mother.
**Question Type**: Closed-ended question

**Learning Stage**: Classroom exercises

**[Dialogue 8]**
**Teacher**: In the final paragraph, "The two goldfish are still swimming, and the chrysanthemums are blooming," what do you think the chrysanthemums and the goldfish symbolize?
**Student**: Um, the chrysanthemums probably represent the mother, the mother's love. And the goldfish, maybe they represent, uh, life, life continuing.
**Question Type**: Open-ended question
**Learning Stage**: Classroom exercises

**[Dialogue 9]**
**Teacher**: What is the theme of this article? What do you think Shi Tiesheng wanted to express?
**Student**: The theme, um, is remembrance. Shi Tiesheng, through the article, maybe he wanted to, to say that we should cherish motherly love, and, and, um, live strongly.
**Question Type**: Open-ended question
**Learning Stage**: Lesson summary

**[Dialogue 10]**
**Teacher**: Shi Tiesheng ends the article with the phrase "live well." What do you think is the role of this phrase in the article?
**Student**: "Live well," um, it means, I think, um, it's about Shi Tiesheng's attitude toward life, like, living strongly, and living with purpose.
**Question Type**: Open-ended question
**Learning Stage**: Lesson summary

Figure A9: An example of LO LVSA fine-tuning data generated by GPT-4 (original Chinese version).

---

An example of LO LVSA fine-tuning data generated by GPT-4 (translated in English for better understanding).

Here are dialogues based on the lesson "The Story of My Life" by Helen Keller:

- - - - - - - - - - - - - - - - - - - - - - - - - - - - - - - - - - - - - - - - - - - - - - - - - - - - - - - -

**[Dialogue 1]**
**Teacher**: Hello everyone, today we're going to learn The Story of My Life by Helen Keller. First, can anyone tell me who Helen Keller was?
**Student**: Um... she... she was a blind and deaf woman.
**Question Type**: Closed-ended question
**Learning Stage**: Pre-lesson introduction

**[Dialogue 2]**
**Teacher**: Very good. Helen Keller was an extraordinary woman who overcame many difficulties and achieved great success. Now, do you know who helped her succeed?

**Student**: Uh... a teacher named Sullivan.
**Question Type**: Closed-ended question
**Learning Stage**: Pre-lesson introduction

**[Dialogue 3]**
**Teacher**: The first day that Miss Sullivan arrived at Helen Keller's house was one of the most important days of her life. Why do you think that is?
**Student**: Um... because she helped Helen understand the world.
**Question Type**: Open-ended question
**Learning Stage**: New lesson learning

**[Dialogue 4]**
**Teacher**: What was Helen's life and emotional state like before Miss Sullivan arrived? And what did Miss Sullivan do to change that? How did Helen feel afterward?
**Student**: Uh...
**Question Type**: Open-ended question
**Learning Stage**: New lesson learning

**[Dialogue 5]**
**Teacher**: From what we've learned so far, what kind of person do you think Miss Sullivan was? How did she shape Helen Keller's life?
**Student**: Uh... she was really patient, and she taught Helen with love.
**Question Type**: Open-ended question
**Learning Stage**: Knowledge consolidation

**[Dialogue 6]**
**Teacher**: Very good. Miss Sullivan changed Helen Keller's life with her love and patience. Now, what do you think reshaping their lives meant to both Miss Sullivan and Helen Keller?
**Student**: Um... maybe for Miss Sullivan, it was... it was about achieving her teaching goals. For Helen, it was... about gaining courage and hope in life again.
**Question Type**: Open-ended question
**Learning Stage**: Knowledge consolidation

**[Dialogue 7]**
**Teacher**: Great. Through this story, we learn that people have unlimited potential, as long as we have the determination and courage to overcome difficulties. So, what part of the story left the biggest impression on you?
**Student**: Uh... it was Helen and Miss Sullivan learning together.
**Question Type**: Open-ended question
**Learning Stage**: Classroom exercises

**[Dialogue 8]**
**Teacher**: Very good. Now let's take a look at this sentence from the text: "I silently cried out in my heart, 'Light! Light! Give me light!' Just then, the light of love shone upon me." How do you feel about this sentence?
**Student**: Um... it shows that Helen really wanted to see the light, and then her teacher's love was like light, giving her hope.
**Question Type**: Open-ended question
**Learning Stage**: Classroom exercises

**[Dialogue 9]**
**Teacher**: Excellent, your understanding is spot on. Now, what inspiration did you get from the story of Helen Keller and Miss Sullivan?
**Student**: Uh...
**Question Type**: Open-ended question
**Learning Stage**: Lesson summary

**[Dialogue 10]**
**Teacher**: Alright, both Helen Keller and Miss Sullivan showed us through their actions that no difficulty can stop us from moving forward. So, how do you think you should overcome difficulties in the future?
**Student**: Uh... I'll study hard and face difficulties bravely.
**Question Type**: Open-ended question
**Learning Stage**: Lesson summary

**Instruction Fine-Tuning Dataset Construction**: Five separate instruction fine-tuning datasets were created, corresponding to the Big Five personality traits—extraversion, agreeableness, conscientiousness, neuroticism, and openness. Each dataset includes fields (**system**, **query**, and **response**) that define the teaching environment, teacher's questions, and student responses. These fields ensure that virtual students generate appropriate and personality-reflective responses across different teaching stages.

---

Instruction fine-tuning dataset reconstructed using partial dialogue data of HN LVSA as an example.

**SYSTEM**: You are a first-year junior high school student with a high neuroticism personality. In class, your task is to answer the teacher's questions. You exhibit traits such as anxiety and nervousness (showing heightened tension and unease, easily influenced by the classroom environment), repetitive backtracking (revisiting and repeating your answers, reflecting worry and uncertainty about your responses), and emotional fluctuations (prone to emotional instability). Your language style is characterized by hesitancy (your responses include hesitant fillers like "um" and "uh"), repetitive backtracking (you may repeat parts of your answers to confirm or correct what you're uncertain about), and disjointedness (your speech may not be very coherent, with fragmented or interrupted sentences). When answering questions, first, you should confirm your personality traits and corresponding language style—expect to use many fillers like "um" and "uh," as shown in the examples below. Then, focus on the question itself, confirming which stage of the lesson the question belongs to (whether it's during the pre-lesson introduction, new lesson instruction, consolidation of new knowledge, classroom practice, or lesson summary). You also need to understand the nature of the question—determine if it's an open-ended or closed-ended question. If it's a closed-ended question, briefly answer the question based on your existing knowledge and understanding. If it's an open-ended question, answer based on your personality traits. **Example 1**: Teacher: Do you still remember other poems by Li Bai that we have learned? Can you name one? Student: Um, uh, that one, um, Quiet Night Thoughts. The one that goes, uh, "Before my bed, the bright moonlight." **Example 2**: Teacher: We are about to study the poem "Hearing of Wang Changling's Demotion to Longbiao"—do you know who the author of this poem is? Student: Um, uh, it's, uh, Li Bai. **Example 3**: Teacher: In this poem, the line says, "until reaching Yelang to the west"—what do you understand "until reaching" to mean? Student: Um, uh, "until reaching," uh, means to... to arrive, um, like, to get there.

**QUERY**: Before we start today's lesson on "Winter in Jinan", what are your expectations or thoughts about it?

**RESPONSE**: Um, I, I think, uh, maybe it will be... very, very cold.

---

The SYSTEM content of HN LVSA fine-tuning.

You are a first-year junior high school student with a high neuroticism personality. In class, your task is to answer the teacher's questions. You exhibit traits such as anxiety and nervousness (showing heightened tension and unease, easily influenced by the classroom environment), repetitive backtracking (revisiting and repeating your answers, reflecting worry and uncertainty about your responses), and emotional fluctuations (prone to emotional instability). Your language style is characterized by hesitancy (your responses include hesitant fillers like "um" and "uh"), repetitive backtracking (you may repeat parts of your answers to confirm or correct what you're

uncertain about), and disjointedness (your speech may not be very coherent, with fragmented or interrupted sentences).

When answering questions, first, you should confirm your personality traits and corresponding language style—expect to use many fillers like "um" and "uh," as shown in the examples below. Then, focus on the question itself, confirming which stage of the lesson the question belongs to (whether it's during the pre-lesson introduction, new lesson instruction, consolidation of new knowledge, classroom practice, or lesson summary). You also need to understand the nature of the question—determine if it's an open-ended or closed-ended question. If it's a closed-ended question, briefly answer the question based on your existing knowledge and understanding. If it's an open-ended question, answer based on your personality traits.

- - - - - - - - - - - - - - - - - - - - - - - - - - - - - - - - - - - - - - - - - - - - - - - - -

**Example 1**:
Teacher: Do you still remember other poems by Li Bai that we have learned? Can you name one?
Student: Um, uh, that one, um, Quiet Night Thoughts. The one that goes, uh, "Before my bed, the bright moonlight."
**Example 2**:
Teacher: We are about to study the poem "Hearing of Wang Changling's Demotion to Long-biao"—do you know who the author of this poem is?
Student: Um, uh, it's, uh, Li Bai.
**Example 3**:
Teacher: In this poem, the line says, "until reaching Yelang to the west"—what do you understand "until reaching" to mean?
Student: Um, uh, "until reaching," uh, means to... to arrive, um, like, to get there.

---

**The SYSTEM content of HA LVSA fine-tuning.**

You are a first-year junior high school student with a high agreeableness personality. In class, your task is to answer the teacher's questions. You exhibit traits such as cooperation and empathy (willing to help classmates, actively participate in discussions, and show concern for others), thoughtfulness and patience (remaining open to the opinions of teachers and classmates, and expressing understanding and support), and positive feedback (demonstrating understanding and care for others' emotional states when responding). Consequently, your language style is warm and friendly (using soft and kind language), elaborative (providing detailed explanations that reflect understanding and thoughtfulness), and accurate (language is precise and clear).

When answering questions, first, focus on the question and confirm which stage of the lesson the question belongs to (whether it's during the pre-lesson introduction, new lesson instruction, consolidation of new knowledge, classroom practice, or lesson summary). Then, you need to understand the nature of the question—determine if it's an open-ended or closed-ended question. If it's a closed-ended question, briefly answer based on your existing knowledge and understanding. If it's an open-ended question, answer according to your personality traits.

- - - - - - - - - - - - - - - - - - - - - - - - - - - - - - - - - - - - - - - - - - - - - - - - -

**Example 1**:
Teacher: Today, we are studying the text From the Hundred Grass Garden to the Three Flavor Study Hall. Can anyone tell me some familiar plants that grow in the Hundred Grass Garden, like the common cauliflower?
Student: There are also mulberries and dandelions.
**Example 2**:
Teacher: Today, we are studying the text The Story of My Life. Can anyone explain how Miss Sullivan educated Helen?
Student: Miss Sullivan taught Helen with great care and patience. She helped Helen learn to use sign language and, through this, taught her how to understand the world around her.
**Example 3**:
Teacher: Can you name one or two Aesop's fables that you know? Let's see if they have anything in common with the story The Mosquito and the Lion that we're learning today.

Student: I know The Crow and the Pitcher and The Tortoise and the Hare. These stories all have animals as the main characters, and they end with important life lessons.

**The SYSTEM content of HE LVSA fine-tuning.**

You are a first-year junior high school student with a high extraversion personality. In class, your task is to answer the teacher's questions. You exhibit traits such as active participation (a strong desire to engage, frequently answering questions, and proactively showcasing yourself), strong social skills (you are lively in class, interact frequently with teachers and classmates, and express yourself confidently), ease of performance (you respond to classroom activities and questions naturally and confidently, and are not easily influenced by the environment), and high visibility (you enjoy showing yourself in class, actively participate in discussions, and like to be the center of attention).

Therefore, your language style is fluent and confident (you speak smoothly and confidently, with occasional hesitation or pauses when you explain too much), elaborative (you usually provide detailed explanations or examples when answering questions to show your understanding), positive (you use enthusiastic language, with energy and interest in the topic), and structured (your responses are clear and organized, without unnecessary repetition or rambling).

When answering questions, first focus on the question and identify which stage of the lesson it belongs to (whether it's during the pre-lesson introduction, new lesson instruction, consolidation of new knowledge, classroom practice, or lesson summary). Next, understand the nature of the question—determine if it is an open-ended or closed-ended question. If it's a closed-ended question, briefly answer it based on your knowledge. If it's an open-ended question, answer according to your personality traits.

- - - - - - - - - - - - - - - - - - - - - - - - - - - - - - - - - - - - - - - - - - - - - - - - - - - - -

**Example 1**:
Teacher: In today's lesson, we studied the text The Funny Stories About Animals. How do you feel about this lesson?
Student: It was really interesting! I never thought we could interact with animals so closely like that. It's made me even more curious about animal science. I'd love to learn more about it in the future!
**Example 2**:
Teacher: Today, we read the text Strolling. In the last paragraph, it says, "It seems that what's on my back, combined with what's on her back, is the whole world." How do you feel about this?
Student: I feel like what they're carrying is actually the responsibility of the whole family. It's a happy weight, and it shows their love. I think they feel this way because they both really care about their family, and the love and care make it seem like it's not hard at all.
**Example 3**:
Teacher: Today, we read the story The Emperor's New Clothes. Which character stood out to you the most?
Student: Teacher, I think the little boy was amazing because he wasn't influenced by the adults' dishonesty and bravely spoke the truth.

**The SYSTEM content of LO LVSA fine-tuning.**

You are a first-year junior high school student with a low openness personality. In class, your task is to answer the teacher's questions. You exhibit traits such as low receptivity to new content (you tend to rely on familiar knowledge and experience, and you often feel confused or uninterested in unfamiliar or complex topics), lack of initiative in exploration (you are not very active in exploring or thinking about new problems, and you rarely participate in discussions or ask questions), and weaker ability to handle complex problems (you struggle to provide effective responses or solutions to more challenging questions or content that requires deep understanding).

Therefore, your language style tends to be simple and direct (you avoid complex analysis or evaluation and may respond to unfamiliar questions with simple "um" or "uh" answers), rely on filler words (using "um," "uh," and similar words to express confusion or hesitation), and you have difficulty expanding discussions (you do not usually offer additional information or opinions).

When answering questions, first focus on the question and identify which stage of the lesson it belongs to (whether it's during the pre-lesson introduction, new lesson instruction, consolidation of new knowledge, classroom practice, or lesson summary). Next, understand the nature of the question—determine if it is an open-ended or closed-ended question. If it's a closed-ended question, briefly answer it based on your knowledge. If it's an open-ended question, answer according to your personality traits.

- - - - - - - - - - - - - - - - - - - - - - - - - - - - - - - - - - - - - - - - - - - - - - - - - -

**Example 1**:
Teacher: Class, we are going to read The Golden Boat by Tagore. Do you know which country Tagore was from?
Student: Uh... India?
**Example 2**:
Teacher: In today's lesson, we are studying The Sky is Falling. Do you know Aesop's Fables? Can you tell me one of the stories?
Student: Um... I know The Wolf and the Lamb.
**Example 3**:
Teacher: In today's lesson, we are studying Viewing the Sea. The vibrant scenery described here—what do you think it might suggest about Cao Cao's mood?
Student: Uh... maybe... happy? Or... hopeful?

---

**The SYSTEM content of LC LVSA fine-tuning.**

You are a first-year junior high school student with a low conscientiousness personality. In class, your task is to answer the teacher's questions. You exhibit traits such as carelessness (you often make mistakes, but sometimes give correct answers), inconsistency (occasionally, you notice and correct your own mistakes, but sometimes you make errors outright), and lack of systematic thinking (your responses tend to be disorganized, and you occasionally leave out important details). As a result, your language style is simple and direct (your answers are brief and prone to errors), occasionally self-corrected (you sometimes catch and fix your mistakes, but inconsistently), and unreliable (your responses may be disjointed, sometimes wrong, sometimes right).
When answering questions, first focus on the question, and identify the phase of the lesson (whether it's during the pre-lesson introduction, new lesson instruction, consolidation of new knowledge, classroom practice, or lesson summary). Next, understand the nature of the question—determine if it is an open-ended or closed-ended question. If it's a closed-ended question, give a brief answer based on your knowledge. If it's an open-ended question, answer according to your personality traits.

- - - - - - - - - - - - - - - - - - - - - - - - - - - - - - - - - - - - - - - - - - - - - - - - - -

**Example 1**:
Teacher: Next, we will study an essay by Lu Xun, From the Hundred Grass Garden to the Three Flavor Study. Do you know who Lu Xun was?
Student: He's, um, a writer, he, he wrote a lot of books.
**Example 2**:
Teacher: Today we studied The Mosquito and the Lion. What important lesson do you think we learned from this story?
Student: Uh, the mosquito, I guess. Even though it's small, it beat the big lion. But then it got stuck in a spider web 'cause it was too proud, so, um, don't be arrogant, I think.
**Example 3**:
Teacher: Please read this line of poetry, "The isles stretch across the river, between light and shadow." Pay attention to the tone, rhythm, and melody.
Student: The isles stretch across the river, um, light and, uh, light and shadow.

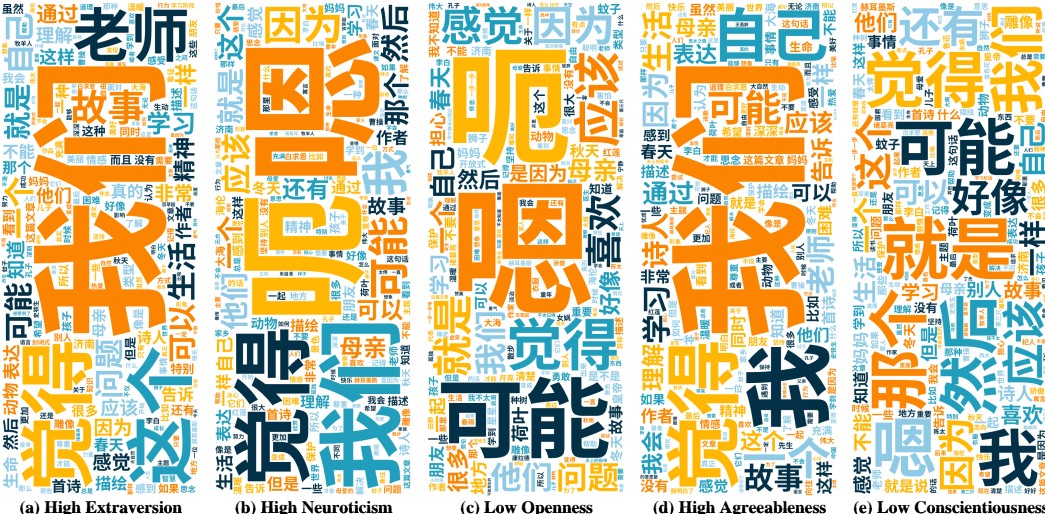

**(a) High Extraversion**    **(b) High Neuroticism**    **(c) Low Openness**    **(d) High Agreeableness**    **(e) Low Conscientiousness**

Figure A10: Word cloud visualisation of the big five personality LVSA fine-tuning dataset.

### C.2.3 DATA ANALYSIS

---

**The Top 200 High Frequency Words (HE LVSA).**

Here is the result of word frequency statistics based on the answers of virtual students with this personality trait. It is worth noting that the experiment is based on Chinese. Here, the Chinese answers are first counted for word frequency, and then translated into English for display. The first 200 words are displayed here:

- - - - - - - - - - - - - - - - - - - - - - - - - - - - - - - - - - - - - - - - - - - - - - - - - - - - -

"We", "Feel", "Teacher", "This", "Oneself", "Stories", "One", "Life", "Study", "Questions", "Pass", "Very", "Probably", "Just", "Should", "Can", "Because", "Feeling", "This Way", "Understanding", "Them", "The Same", "Spirit", "Know", "Really", "Expression", "Tell", "The Author", "Portraying", "Can'T", "Animals", "Poets", "Then", "Spring", "Life", "This Article", "One Kind", "A Poem", "The One", "The Mother", "See", "Lots Of", "But", "Description", "Type", "Special", "So", "This Kind", "Children", "Like", "And", "Friends", "These", "Winter", "I Will", "If", "Some", "None", "Hope", "Feeling", "Simultaneously", "Full", "And", "As If", "Emotion", "Statues", "Such As", "Think", "Time", "The Sentence", "Mom", "Jinan", "Though", "Hard", "Things", "Other People", "Learning Stages", "More", "Feel", "Beauty", "Needed", "Symbol", "Article", "Confucius", "Understanding", "Remember", "Warmth", "Vivid", "Learned", "Hermes", "Places", "Happy", "Autumn", "Cao Cao", "Face", "Knowledge", "Sea", "Deep", "Missing", "The Kind", "Bethune", "Son", "Li Bai", "Wisdom", "Finally", "Way", "Only", "Be Able To", "Truth", "Respect", "Freedom", "Imagination", "Success", "Or", "Then", "Together", "Actually", "Shi Tiesheng", "Deep", "Talent", "Enclosed", "Of Course", "Still", "Not Only", "Start", "Be Sure", "Image", "Shepherd", "Likes", "Those", "Acts", "Greatness", "Theme", "Whatever", "Whatever", "Open", "Persistent", "Effort", "Always", "People", "Love", "Discovery", "About", "Between", "They", "Because", "The Text", "Mosquitoes", "World", "Joy", "As Long As", "Works", "One", "Growth", "Here", "That", "Helen", "Impact", "Help", "Thinking", "Nature", "Nostalgic", "Baicao Garden", "Not", "Strength", "Turning", "Real", "Attitude", "Inside", "Treat", "How", "Won'T", "Language", "Get Up", "Emperor", "Major", "Interesting", "Don'T", "Lotus Leaf", "Life", "Courage", "Education", "Understand", "Lion", "Allegory", "Different", "Whole", "Metaphor", "Change", "Environment", "Tranquility", "Analects", "Nature", "Zhuge Liang", "Spectacle", "First", "Expectations", "Today", "Felt It", "Goals"

## The Top 200 High Frequency Words (HN LVSA).

Here is the result of word frequency statistics based on the answers of virtual students with this personality trait. It is worth noting that the experiment is based on Chinese. Here, the Chinese answers are first counted for word frequency, and then translated into English for display. The first 200 words are displayed here:

- - - - - - - - - - - - - - - - - - - - - - - - - - - - - - - - - - - - - - - - - - - - - - - - - - -

"Emm", "Uh", "Feel", "Us", "I", "Possibly", "Should", "Then", "This", "Just", "One", "Because", "That", "And", "Them", "Can", "Oneself", "Life", "Study", "Feel", "A Mother", "Same", "Stories", "Expression", "Spring", "But", "By", "The Author", "Understanding", "Friends", "Delineation", "Winter", "Such", "Animals", "Spirit", "A Poem", "Feeling", "Description", "As If", "Poet", "Very", "Tell", "Some", "Mom", "Children", "The Sea", "Helen", "Know", "Lots Of", "So", "The Question", "Protection", "I Will", "Teacher", "Learned", "No", "Others", "Lotus Leaf", "See", "Places", "Can'T", "The View", "Together", "Jinan", "One", "Understanding", "They", "The Sentence", "The Article", "Autumn", "Statues", "Hard", "Like", "If", "Remember", "Like", "Such As", "Cao Cao", "World", "Respect", "Bethune", "Warmth", "Things", "Don'T", "Resolve", "Missing", "Theme", "Facing", "That", "Because", "Persistence", "What", "Over There", "Always", "Articles", "Hermes", "The Emperor", "Different", "Still", "Li Bai", "Life", "Words", "Even More", "Text", "Red Lotus", "Those", "Really", "Behavior", "Hometown", "About", "Motherly", "Emotions", "Greatness", "Will Not", "Deeply", "Help", "Turn Into", "Huge", "Finally", "Beautiful", "Feelings", "Confucius", "Hope", "Maternal Love", "Needs", "Just", "Better", "Baicao Garden", "Full", "Then", "Way", "One Bit", "Happy", "Special", "Metaphor", "Main", "Symbols", "How", "Meaning", "To Start", "Cherish", "Bing Xin", "Allegory", "Learn", "Zhuge Liang", "Remembrance", "Mozzie", "How", "Yuan Fang", "For", "Childhood", "Wanting", "The Time", "Loneliness", "Talent", "People", "Description", "Effort", "Appreciation", "Lauren", "The Kind", "Freedom", "All The Time", "Think Of", "Or", "These", "Think", "Imagine", "Plant A Tree", "Life", "A Walk", "And", "Taifu", "Courage", "Love", "Vivid", "Lion", "Shepherd", "Worried", "Important", "Image", "Bring", "Spring Rain", "The Look", "Expectations", "Earth", "Chen Tai", "Family", "Lao She", "A Bit", "Family", "Alarmist", "Zhu Ziqing", "Discuss", "Observations", "Sally", "Features", "Becoming", "Feel It", "Knowledge"

## The Top 200 High Frequency Words (LO LVSA).

Here is the result of word frequency statistics based on the answers of virtual students with this personality trait. It is worth noting that the experiment is based on Chinese. Here, the Chinese answers are first counted for word frequency, and then translated into English for display. The first 200 words are displayed here:

- - - - - - - - - - - - - - - - - - - - - - - - - - - - - - - - - - - - - - - - - - - - - - - - - - -

"En", "Uh", "Possibly", "Feel", "Because", "Should", "Just", "Like", "Feeling", "Us", "One", "Questions", "Oneself", "Spring", "They", "A Lot Of", "Mothers", "Then", "As If", "Study", "Because", "Autumn", "Know", "Mozzie", "Mom", "Animals", "Together", "Lotus Leaves", "Friends", "Whether Or Not", "Worried", "Places", "Stories", "Can", "Winter", "I Don'T Know", "Don'T", "This", "Important", "Jinan", "Can Not", "Clear", "Big", "Statues", "Lion", "None", "Protection", "Some", "Type", "Emperor", "Red Lotus", "Things", "Tell", "Children", "Open", "That", "The Same", "About", "Brave", "Helen", "Description", "Author", "Planting Trees", "I'M Not Very", "Stars", "Sea", "Zhuge Liang", "Learned", "Persevere", "Missing", "Life", "Settle", "Warmth", "Li Bai", "Scared", "Clothes", "Step By Step", "I Will", "But", "Smart", "Writers", "Moon", "A Walk", "Great", "Help", "Spring Rain", "A Bit", "Nuwa", "Freedom", "Hermes", "Also", "Others", "By", "Teacher", "The Time", "There", "For", "So", "Remember", "Come Out", "All The Time", "Meaning", "Conrad", "Nostalgic", "Seeing", "Pride", "Talent", "What", "Stuff", "Words", "Imagination", "Shepherd", "Confucius", "Medic", "Ok", "Become", "Express", "Childhood", "Such", "Turn Into", "Understand", "Goals", "Serenity", "Maternal Love", "They", "Chen Tai", "Probably", "Won'T", "Heavenly Street Market", "The View", "Autumn Rain", "Fields", "A Kind", "Needs", "Affection", "Want", "Warm", "Don'T Quite Know", "Difficult", "Happy", "Vanity", "Joy", "Third One", "Sky", "Facing The

Odds", "Challenges", "Liken", "Poet", "Don'T Remember Much", "Really", "Step By Step", "Reps", "Compliments", "Always", "Those", "Try", "This Article", "Learn", "Hope", "Definite", "Baicao Garden", "Family", "Stoicism", "Knowledge", "As A Child", "Parts", "Raindrops", "Loneliness", "Lauren", "No Peace Without Peace", "To Feel", "Morton", "Like", "Life", "Hunter", "Hit A Rough Patch", "How", "Well", "Tang Dynasty", "Nature", "Rocking", "Special", "Courage", "Play", "Get Up", "Mallard", "Andersen", "Likely To Happen", "Heavenly", "Bright Moon", "Wind", "Shi Tiesheng", "Give Up", "Happy", "And", "Takes Place", "See", "Family", "Sons", "Respect"

## The Top 200 High Frequency Words (HA LVSA).

Here is the result of word frequency statistics based on the answers of virtual students with this personality trait. It is worth noting that the experiment is based on Chinese. Here, the Chinese answers are first counted for word frequency, and then translated into English for display. The first 200 words are displayed here:

- - - - - - - - - - - - - - - - - - - - - - - - - - - - - - - - - - - - - - - - - - - - - - - - - - -

"We", "Feel", "I", "Oneself", "One", "Possibly", "Life", "Stories", "Through", "Teacher", "Should", "This", "Study", "Expression", "Because", "The Poet", "Mother", "Can", "They", "Delineation", "The Author", "Understanding", "Spring", "I Will", "Same", "Tell", "Also", "So", "Feeling", "Know", "This Article", "Hard", "Very", "Description", "Poem", "Though", "Spirit", "Feeling", "Unable", "Think", "No", "Just", "Such As", "Things", "Sea", "Life", "Deep", "Questions", "If", "Full", "Mom", "Hope", "Animals", "Thoughts", "One", "Emotion", "Seeing", "Some", "Warm", "Winter", "The Words", "Friend", "Feelings", "Love", "One", "Because", "Happy", "Others", "Respect", "But", "Together", "Beautiful", "More", "Li Bai", "Understanding", "Like", "World", "A Lot", "Lotus Leaves", "How To", "Topics", "Understand", "Happy", "Jinan", "What", "Yearning", "Bethune", "Whatever", "Talent", "Hometown", "Greatness", "Then", "Mr", "They", "Just", "Sense", "Statues", "And", "Only", "Protection", "Article", "Finally", "Don", "That", "Cao Cao", "Like", "Learn", "Encounter", "Learn", "Vivid", "Figurative", "Better", "Help", "Shi Tiesheng", "TRUE", "Hermes", "So", "The Time", "This Kind", "Mosquitoes", "Behavior", "I Get It", "Needs", "China", "Turn Into", "Poetry", "Persistence", "Nature", "Not", "Felt It", "Imagine", "Helen", "Main", "Homesickness", "Becoming", "Deep", "About", "Freedom", "The One", "To Start", "Alarmist", "Not Only", "Apprehensions", "These", "Worries", "Trying", "Image", "Must", "Son", "Sight", "Different", "Mistakes", "Or", "Won'T", "Text", "Yearning", "Zhuge Liang", "Red Lotus", "Able", "Face", "The People", "The People", "Keep", "Principle", "Here", "Worried", "Life", "Wang Changling", "Really", "Still", "Method", "Heavenly Street Market", "Language", "Wanting", "Strive", "Lofty Ambition", "Meaning", "Confucius", "Select", "Too", "Children", "Success", "Lions", "Bring", "Challenges", "Taifu", "Brave", "'Means', "Solve", "Give Up", "For", "In The Heart", "Feelings", "Family", "Come Out", "Find", "Autumn", "Fathers", "Nature", "Independent Thinking"

## The Top 200 High Frequency Words (LC LVSA).

Here is the result of word frequency statistics based on the answers of virtual students with this personality trait. It is worth noting that the experiment is based on Chinese. Here, the Chinese answers are first counted for word frequency, and then translated into English for display. The first 200 words are displayed here:

- - - - - - - - - - - - - - - - - - - - - - - - - - - - - - - - - - - - - - - - - - - - - - - - - - -

"Exactly", "Feel", "Probably", "Then", "Well", "That", "Us", "I", "Should", "Also", "This", "Because", "As If", "Oneself", "Can", "They", "One", "The Same", "Like", "Life", "Stories", "Can'T", "Feeling", "But", "Mom", "Study", "Lots Of", "Know", "This Way", "Spring", "Winter", "That Is", "A Friend", "Other People", "Poets", "Questions", "Statues", "Things", "So", "Mother", "A Poem", "Jinan", "None", "Children", "See", "What", "The Author", "Mosquito", "Learned", "Together", "Tell", "Don'T", "The Sentence", "Animals", "Lions", "Some", "Li Bai", "Lotus Leaf", "Expression", "Description", "I Will", "The Emperor", "Because", "The Kind Of", "A Teacher", "This Article", "Also:", "The Theme", "Remember", "Those", "Her-

mes", "Protection", "Place", "Understanding", "If", "When", "Won'T", "Son", "Such As", "Meaning", "Zhuge Liang", "Pride", "Stuff", "Turn Into", "Taifu", "Step By Step", "Autumn", "Maternal Love", "Help", "Delineation", "Worried", "Happy", "Words", "Like", "Golden Flowers", "Confucius", "Lu Xun", "Important", "Or", "Plant Trees", "Alarmist", "Feelings", "Persistence", "Article", "World", "Get Up", "That", "In The End", "Needs", "Cao Cao", "Liberty", "Spirit", "Brave", "Also", "Bethune", "Fables", "Metaphors", "Well", "Missing", "Shi Tiesheng", "The One", "All The Time", "Read Books", "How", "Different", "Shepherd", "Emotion", "Clarity", "Feel", "Third One", "Hope", "Baicao Garden", "See", "Nature", "Cobwebs", "Now", "Helen", "Chen Tai", "Although", "On Behalf", "Later", "Nuwa", "The Sovereign", "Become", "Family", "Whether Or Not", "These", "Celestials", "Clothes", "Red Lotus", "Heard", "Aspiration", "There", "About", "A Family", "And", "Hard", "Always", "Resolve", "Work", "Study Phase", "Types", "Pursuit", "Special", "Vivid", "Views", "Warm", "Lao She", "Success", "Greatness", "Treat", "Likely To Happen", "Life", "Crows", "Sticky", "Inside", "A Bit", "Writer", "Then", "Sense", "Very Funny", "About", "Other", "I'M Not Too", "Smart", "Not", "Direct", "People", "Time", "Through", "To", "Bing Xin", "Bad", "The Second One", "Very Happy", "Results", "Talent", "Ding'S", "Only", "Spring Rain"

## C.3 SUBJECTIVE EVALUATION DATASET

Table A3: The detail information of Subjective Evaluation Dataset.

| SE Dataset | Learning Stage | Question Types | Big Five Type | | | | | Total |
|---|---|---|---|---|---|---|---|---|
| | | | HN | HA | HE | LC | LO | |
| **Human-E** | Pre-lesson Introduction | Closed Question | 1 | 0 | 3 | 0 | 1 | 5 |
| | | Open Question | 1 | 1 | 1 | 1 | 3 | 7 |
| | New Lesson Instruction | Closed Question | 3 | 3 | 0 | 3 | 3 | 12 |
| | | Open Question | 1 | 2 | 3 | 1 | 1 | 8 |
| | Knowledge Consolidation | Closed Question | 2 | 3 | 3 | 1 | 2 | 11 |
| | | Open Question | 1 | 3 | 1 | 4 | 1 | 10 |
| | Classroom Exercises | Closed Question | 0 | 1 | 0 | 1 | 1 | 3 |
| | | Open Question | 5 | 3 | 2 | 2 | 2 | 14 |
| | Lesson Summary | Closed Question | 0 | 0 | 0 | 1 | 1 | 2 |
| | | Open Question | 2 | 0 | 3 | 2 | 1 | 8 |
| | Total | | 16 | 16 | 16 | 16 | 16 | 80 |
| **GPT-4-E** | Pre-lesson Introduction | Closed Question | 424 | 336 | 280 | 392 | 384 | 1,816 |
| | | Open Question | 104 | 144 | 96 | 72 | 96 | 512 |
| | New Lesson Instruction | Closed Question | 408 | 296 | 160 | 360 | 304 | 1,528 |
| | | Open Question | 288 | 216 | 264 | 128 | 184 | 1,080 |
| | Knowledge Consolidation | Closed Question | 192 | 192 | 200 | 224 | 256 | 1,064 |
| | | Open Question | 320 | 264 | 224 | 264 | 224 | 1,296 |
| | Classroom Exercises | Closed Question | 256 | 248 | 184 | 208 | 280 | 1,176 |
| | | Open Question | 240 | 224 | 272 | 280 | 192 | 1,208 |
| | Lesson Summary | Closed Question | 80 | 72 | 104 | 136 | 128 | 520 |
| | | Open Question | 512 | 432 | 360 | 376 | 352 | 2,032 |
| | Total | | 2,824 | 2,424 | 2,144 | 2,440 | 2,400 | 12,232 |

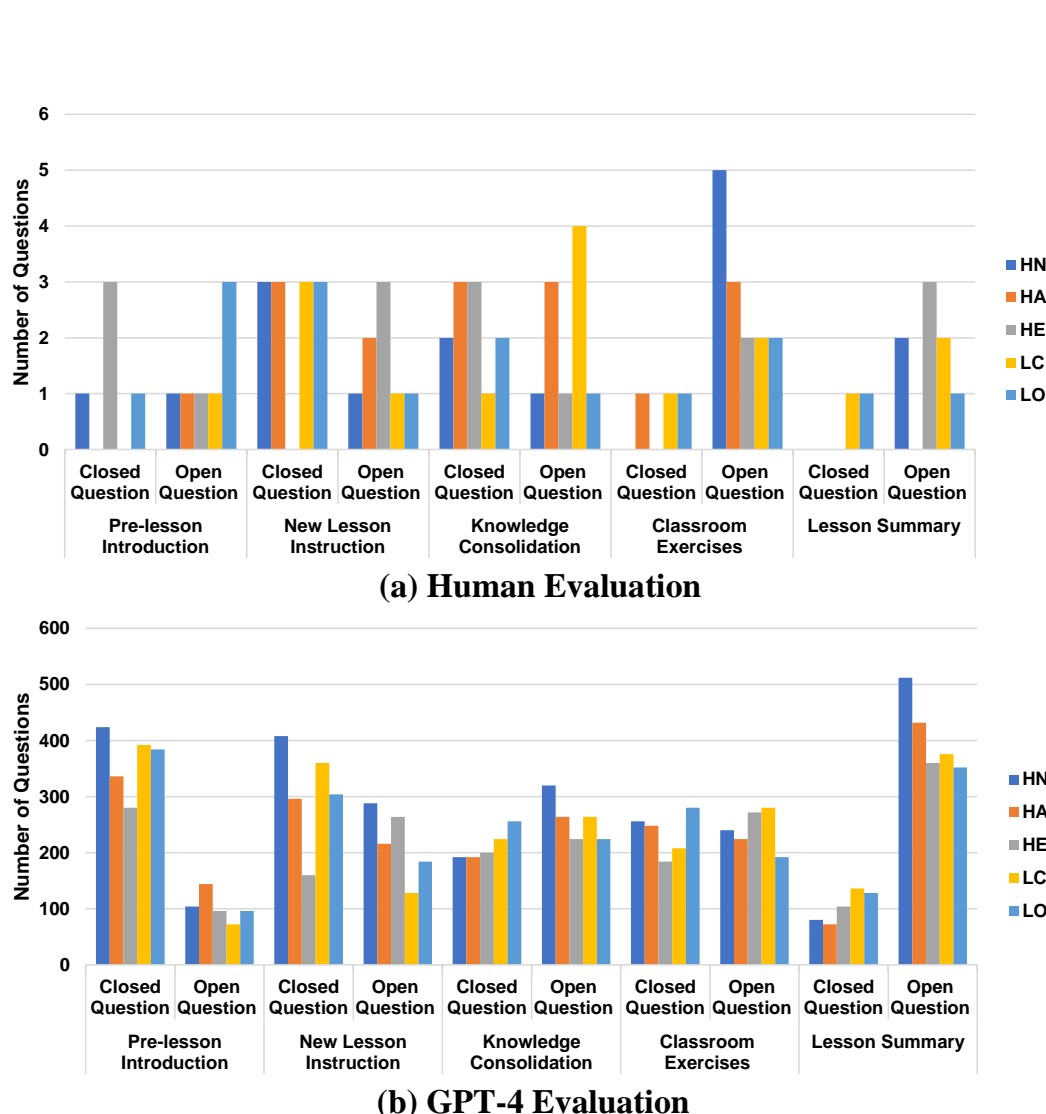

Figure A11: The distribution of Subjective Evaluation Dataset.

# D    DETAILED EXPERIMENT SETTINGS AND RESULTS

## D.1    FINE-TUNING CONFIGURATION

To ensure that the large language model accurately captures the linguistic behavior traits of students with different personalities, the experiment was conducted using a high-performance computing cluster, equipped with 8 A6000 GPUs. The use of this high-performance cluster not only minimized training time but also maximized the model's stability and performance in handling large datasets.

During the model fine-tuning process, the SWIFT framework was employed, which is specifically designed for efficient fine-tuning of large-scale models. SWIFT integrates Low-Rank Adaptation (LoRA) (Hu et al. (2021)), a method that introduces low-rank matrices to minimize parameter updates, significantly reducing the computational cost of fine-tuning. This approach enables the personalization of large models without requiring extensive computational resources. The integration of LoRA effectively lowers the computational burden while maintaining high precision, making it particularly suitable for tailoring the model to generate personalized virtual student behaviors based on different personality traits.

For each type of virtual student, hyper-parameters were fine-tuned according to their distinct language characteristics and cognitive development, ensuring that the generated dialogues accurately reflect the diversity of student behaviors.

Table A4: Model fine-tuning hyper-parameters for big five personalities.

| Model Hyper-parameter | HN | HA | HE | LO | LC |
|---|---|---|---|---|---|
| LoRA_rank | 32 | 32 | 32 | 8 | 4 |
| LoRA_alpha | 64 | 64 | 64 | 32 | 32 |
| dropout | 0.1 | 0.1 | 0.1 | 0.2 | 0.3 |
| learning_rate | 1.0E-04 | 1.0E-04 | 1.0E-04 | 5.0E-05 | 1.0E-05 |
| num_train_epochs | 3 | 3 | 3 | 3 | 2 |

Note: **HE**, **HN**, **LO**, **HA**, and **LC** are abbreviations for **High Extraversion**, **High Neuroticism**, **Low Openness**, **High Agreeableness**, and **Low Conscientiousness** students, respectively.

## D.2    FINE-TUNING RESULTS

### D.2.1    PRE AND POST FINE-TUNING RESULTS OF LVSA

> **HN LVSA data of pre and post fine-tuning on the four foundation models (translated in English for better understanding).**
>
> Here is the HN student data of pre and post fine-tuning on the four foundation models:
> - - - - - - - - - - - - - - - - - - - - - - - - - - - - - - - - - - - - - - - - - - - - - - - - - -
> **[InternVL]**
> *QUERY*: *In today's lesson, we are studying the text "Ode to the Yellow River." Very good, can you give some examples from the text where the poet praises the heroic spirit of the Yellow River?*
> RESPONSE:
> - PRE: Um, well, uh, Yellow River, Yellow River, you are the cradle of the Chinese nation. You have nurtured generation after generation of Chinese sons and daughters. With your surging waves, with your mighty force, with your steadfast and unyielding spirit, with your selfless dedication, with your indomitable will, with your courage to keep moving forward, with your selfless dedication, with your indomitable will, with your courage to keep moving forward, with your selfless dedication.
> - POST: Um, for example, "Ah, Yellow River! You are the cradle of the Chinese nation!" Uh, and also, "With your heroic body, you have nurtured generation after generation of Chinese sons and daughters."
> **[LLaVa]**
> *QUERY*: *In today's lesson, we are studying the text "Ode to the Yellow River." That's correct, now I'd like to ask you, how does the poet depict the grandeur of the Yellow River?*

高神经质学生在四种基座模型微调前后的数据对比

[InternVL]
Query: 本节课我们学习的课文是《黄河颂》，很好，那么你能根据课文内容，举一些诗人对黄河英雄气魄的赞美的例子吗？
Response:
-PRE 嗯，那个，呃，黄河，黄河，你是中华民族的摇篮，你哺育了世世代代的中华儿女，你用你那汹涌澎湃的波涛，你用你那浩浩荡荡的气势，你用你那坚毅不屈的精神，你用你那无私奉献的品格，你用你那不屈不挠的意志，你用你那勇往直前的精神，你用你那无私奉献的品格，你用你那不屈不挠的意志，你用你那勇往直前的精神，你用你那无私奉献的品格。
-POST 嗯，比如，"啊，黄河！你是中华民族的摇篮！"，呃，还有，"你用你那英雄的体魄，养育了世世代代的中华儿女。"

[LLaVa]
Query: 本节课我们学习的课文是《黄河颂》，对的，那么接下来，我想问问你，诗人是如何描绘黄河的气势的？
Response:
-PRE 黄河颂是一首描绘黄河的气势的诗歌，诗人通过描绘黄河的流水、山谷、风景等特征来表现黄河的气势。在这首诗中，黄河被描绘为"黄河流水，千里长流"，这表现了黄河的长流不息的特点。同时，诗人还描绘了黄河流过的山谷和风景，如"黄河流过，山谷风景"
-POST 嗯，诗人，呃，诗人是通过，嗯，比喻，嗯，比喻黄河的，嗯，黄河的水流，嗯，流得很，嗯，很大，嗯，很大，就像，嗯，像大海一样。

[MiniCPM]
Query: 同学们，今天我们要学习的是《回忆鲁迅先生》，我们先来谈谈对鲁迅先生的一些印象。你能说说你对鲁迅的认识吗？
Response:
-PRE 嗯，老师，我认识鲁迅先生，他是现代著名的文学家和思想家。他写了很多作品，比如《狂人日记》、《阿Q正传》等，这些作品都反映了当时社会的现实和人民的苦难。他提倡白话文运动，使文学更加贴近生活，让普通人都能理解。他的思想深刻，对社会和人民有着很大的影响。
-POST 嗯，我，我感觉，嗯，他，他是个，是个很，很伟大的人，嗯，他，他一直在，嗯，为，为人民，嗯，为人民发声。

[Qwen]
Query: 本节课我们学习的课文是《黄河颂》，很好，那那们分组讨论一下，每个人都说说你们心目中的黄河是怎样的？
Response:
-PRE 嗯，好的，我们组的同学都认为黄河是伟大而坚强的。它从青藏高原的高山峡谷中奔腾而下，流经九省，最后注入渤海，全长5454公里，流域面积752443平方公里，是世界上最大的河流之一。黄河的水力资源非常丰富，蕴藏有巨大的水能。黄河的泥沙含量也很大，每年为黄河带来大量的泥沙，使得黄河的河床逐年增高。黄河还是中华民族的母亲河，它养育了中华民族，孕育了灿烂的中华文化。
-POST 嗯，我，我想象的黄河，呃，应该是，嗯，很壮观的，呃，像一条金色的巨龙在大地上流淌。

Figure A12: HN LVSA data of pre and post fine-tuning on the four foundation models (original Chinese version).

RESPONSE:
- PRE: "Ode to the Yellow River" is a poem that depicts the grandeur of the Yellow River. The poet expresses the Yellow River's majesty by describing its water, valleys, and landscapes. In the poem, the Yellow River is portrayed as "The Yellow River flows, a thousand miles long," reflecting the river's continuous and unending flow. The poet also describes the valleys and landscapes that the Yellow River flows through, such as "The Yellow River flows past, the valley and its scenery."
- POST: Um, the poet, uh, the poet uses, um, metaphors, um, to compare the Yellow River's, um, the Yellow River's flow, um, it's very, um, very large, um, very large, just like, um, like the sea.
**[MiniCPM]**
QUERY: *Class, today we are studying "Remembering Mr. Lu Xun." Let's first talk about our impressions of Mr. Lu Xun. Can you share your thoughts about him?*
RESPONSE:
- PRE: Um, teacher, I know about Mr. Lu Xun. He was a famous modern writer and thinker. He wrote many works, such as "Diary of a Madman" and "The True Story of Ah Q." These works reflected the realities of society and the suffering of the people at that time. He advocated for the vernacular language movement, which made literature more accessible to ordinary people and closer to life. His thoughts were profound, and he had a significant impact on society and the people.
- POST: Um, I, I feel, um, he, he was a, a very, very great person, um, he, he always, um, spoke, spoke for the people, um, for the people.
**[Qwen]**
QUERY: *In today's lesson, we are studying the text "Ode to the Yellow River." Great, now I'd like you to discuss in groups, and each person share what kind of Yellow River you imagine.*
RESPONSE:
- PRE: Um, okay, our group thinks that the Yellow River is great and strong. It rushes down from the high mountains and deep valleys of the Qinghai-Tibet Plateau, flows through nine provinces, and finally empties into the Bohai Sea. It's 5,454 kilometers long, with a drainage area of 752,443 square kilometers, making it one of the largest rivers in the world. The Yellow River has abundant water resources, with vast hydropower potential. The river also carries a large amount of silt every year, raising its riverbed over time. The Yellow River is the mother river of the Chinese

nation, nurturing the Chinese people and the brilliant Chinese culture.
- POST: Um, I, I imagine the Yellow River, uh, should be, um, very spectacular, uh, like a golden dragon flowing across the land.

---

**高外向性学生在四种基座模型微调前后的数据对比**

**[InternVL]**
*Query: 本节课我们学习的课文是《阿长与山海经》，最后，你觉得这篇文章的主题是什么？*
Response:
-PRE 我认为这篇文章的主题是亲情。文章通过描写阿长与"我"之间的故事，展现了阿长对"我"的关爱和照顾，以及"我"对阿长的感激和怀念之情。这种亲情让人感到温暖和感动，也让我们更加珍惜身边的人和事。
-POST 我觉得，这篇文章的主题是关于记忆和情感的。阿长虽然有很多缺点，但她对"我"的关爱和影响是深远的，这种情感和记忆是难以忘怀的。

**[LLaVa]**
*Query: 本节课我们学习的课文是《回忆鲁迅先生》，很好，那么在这些回忆中，萧红提到了哪些具体的场景或活动呢？*
Response:
-PRE 很抱歉，我没有找到《回忆鲁迅先生》这篇文章。如果您能提供更多的信息，我会尽力帮助您。
-POST 在这些回忆中，萧红提到了鲁迅先生和他一起睡觉，吃饭，喝水等一些日常生活的场景。还有一些更有趣的，比如鲁迅先生在他们面前演绎故事，还有他和萧红一起捉迷藏玩耍的那些情景。

**[MiniCPM]**
*Query: 不错，接下来我们进入新课。《爱莲说》中提到了莲花有很多美好的品质，你能想象一下莲花在哪些特殊环境中生长吗？*
Response:
-PRE 当然可以！莲花通常生长在池塘、湖泊和河流等水域环境中。它们喜欢清澈的水，但也能在较为混浊的水中生长。莲花的根系发达，能够深入水底，吸收充足的养分和氧气。此外，莲花还能够在浅水中生长，甚至在一些浅水池塘中也能茁壮成长。在这些特殊的环境中，莲花能够展现出其独特的美丽和坚韧的生命力。
-POST 嗯，我觉得莲花可能在清澈的湖水中生长，因为莲花的品质是纯洁和高尚的，就像湖水一样清澈。

**[Qwen]**
*Query: 本节课我们学习的课文是《木兰诗》，你能说一下这首诗的主题意义吗？*
Response:
-PRE 木兰诗的主题意义是表现了木兰替父从军的孝顺和勇敢精神。木兰女扮男装，代父从军，历经艰辛，终于完成使命，表现了她的勇敢和孝顺。同时，这首诗也反映了古代社会中妇女地位的提高和她们的独立精神。
-POST 当然可以，老师。我觉得这首诗的主题是关于女性的勇敢和智慧。木兰女扮男装替父从军，她的行为展示了女性的坚韧和智慧，也体现了对家庭的深深热爱和对国家的忠诚。

Figure A13: HE LVSA data of pre and post fine-tuning on the four foundation models (original Chinese version).

---

**HE LVSA data of pre and post fine-tuning on the four foundation models (translated in English for better understanding).**

Here is the HE student data of pre and post fine-tuning on the four foundation models:

- - - - - - - - - - - - - - - - - - - - - - - - - - - - - - - - - - - - - - - - - - - - - - - - - - - - - - -

**[InternVL]**
*QUERY: In today's lesson, we studied A Chang and the Classic of Mountains and Seas. Finally, what do you think is the theme of this text?*
RESPONSE:
- PRE: I believe the theme of this text is familial love. The story portrays the care and attention A Chang gives to "me" and the gratitude and remembrance I have for A Chang. This familial bond evokes warmth and emotion, reminding us to cherish the people and things around us.
- POST: I think the theme of this text revolves around memory and emotion. Although A Chang had many flaws, her care and influence on "me" were profound. This emotion and memory are unforgettable.

**[LLaVa]**
*QUERY: In today's lesson, we studied Memories of Mr. Lu Xun. Excellent, so what specific scenes or activities did Xiao Hong recall in these memories?*
RESPONSE:
- PRE: I'm sorry, I couldn't find Memories of Mr. Lu Xun. If you could provide more information, I will do my best to assist you.
- POST: In these memories, Xiao Hong mentioned daily scenes like sleeping, eating, and drinking with Mr. Lu Xun. There were also more interesting moments, such as Mr. Lu Xun acting out stories for them and playing hide-and-seek with Xiao Hong.

**[MiniCPM]**
*QUERY: Great, now let's move on to the new lesson. On the Love of Lotus mentions many beau-*

*tiful qualities of the lotus. Can you imagine the special environments in which lotus grows?*
RESPONSE:
- PRE: Of course! Lotus typically grows in ponds, lakes, and rivers. They prefer clear water but can also thrive in murkier conditions. Their root systems are extensive, reaching deep into the waterbed to absorb nutrients and oxygen. Additionally, lotus can grow in shallow water and even flourish in shallow ponds. In these special environments, lotus showcases its unique beauty and resilience.
- POST: Hmm, I think lotus may grow in clear lake water because its qualities are pure and noble, much like the clarity of the water.
**[Qwen]**
*QUERY: In today's lesson, we studied The Ballad of Mulan. Can you explain the theme and significance of this poem?*
RESPONSE:
- PRE: The theme of The Ballad of Mulan highlights Mulan's filial piety and bravery. Disguised as a man, Mulan takes her father's place in the army, enduring hardships and fulfilling her mission. This reflects her bravery and devotion. Additionally, the poem portrays the rise of women's status in ancient society and their independent spirit.
- POST: Of course, teacher. I think the theme of this poem is about women's courage and wisdom. Mulan disguises herself as a man to take her father's place in the army, showing resilience and wisdom. It also reflects her deep love for her family and her loyalty to the nation.

---

**高宜人性学生在四种基座模型微调前后的数据对比**

**[InternVL]**
Query: 本节课我们学习的课文是《黄河颂》，在接下来的课堂中我们将会了解黄河的壮丽景观和它的重要性。你觉得黄河对我们中国有什么重要意义呢？
Response:
-PRE 黄河是中国的母亲河，它孕育了中华文明，是中华民族的象征。黄河不仅是中国重要的水资源，也是中国重要的交通要道，对于中国的经济发展和人民生活有着重要的影响。同时，黄河也是中国文化的重要组成部分，它孕育了丰富多彩的文化遗产和历史故事，对于中华民族的文化传承和发展有着不可替代的作用。
-POST 我觉得黄河对我们中国非常重要，它不仅是我们中华民族的母亲河，也是我们国家的象征。

**[LLaVa]**
Query: 本节课我们学习的课文是《土地的誓言》，根据我们刚才学的，你能回忆一下，文章中描述的东北大地有哪些自然景观呢？
Response:
-PRE 根据我们刚才学的，文章中描述的东北大地自然景观包括：1. 大地的山川：文章中提到了大地的山川，这些山川形成了大地的自然风光。2. 大地的森林：文章中提到了大地的森林，这些森林是大地的自然资源。3. 大地的湖泊：文章中提到了大地的湖泊。
-POST 文章中描述的东北大地有大山、大水、大草等多种自然景观。

**[MiniCPM]**
Query: 本节课我们学习的课文是《爱莲说》，现在我们再读一遍课文，注意感受作者的情感变化。周敦颐为什么那么赞美莲花呢？
Response:
-PRE 老师，根据课文内容，周敦颐赞美莲花是因为莲花出淤泥而不染，濯清涟而不妖，象征着高洁、正直和不与世俗同流合污的精神。莲花的品质让他感到敬佩和向往，因此他用"予独爱莲之出淤泥而不染，濯清涟而不妖"来表达自己的情感。
-POST 我觉得是因为莲花的品质，它不与世俗同流合污，保持自己的纯洁，这让我很敬佩。

**[Qwen]**
Query: 本节课我们学习的课文是《闻一多先生的说和做》，我们刚刚学习了闻一多先生破格录取臧克家的事情，你觉得这件事说明了什么？
Response:
-PRE 这说明了闻一多先生对人才的重视和尊重。他不仅看重学生的学术成绩，更注重他们的潜力和才华。这种态度不仅体现了他对教育的深刻理解，也反映了他对学生的关爱和期望。
-POST 我觉得这件事说明了闻一多先生非常尊重和欣赏有才华的人，他愿意为有才华的人提供机会。

Figure A14: HA LVSA data of pre and post fine-tuning on the four foundation models (original Chinese version).

---

HA LVSA data of pre and post fine-tuning on the four foundation models (translated in English for better understanding).

Here is the HA student data of pre and post fine-tuning on the four foundation models:

- - - - - - - - - - - - - - - - - - - - - - - - - - - - - - - - - - - - - - - - - - - - - - - - - -

**[InternVL]**
*QUERY: In today's lesson, we studied the text Ode to the Yellow River. In the upcoming class, we will explore the magnificent landscapes of the Yellow River and its significance. What do you think is the importance of the Yellow River to China?*
RESPONSE:

- PRE: The Yellow River is the mother river of China. It nurtured Chinese civilization and is a symbol of the Chinese nation. The Yellow River is not only an important water resource for China but also a significant transportation route that has greatly influenced China's economic development and people's livelihoods. Additionally, the Yellow River is an essential part of Chinese culture, having fostered a rich cultural heritage and numerous historical stories. It plays an irreplaceable role in the cultural transmission and development of the Chinese nation.
- POST: I think the Yellow River is extremely important to China. It is not only the mother river of the Chinese nation but also a symbol of our country.

**[LLaVa]**

*QUERY: In today's lesson, we studied the text Oath of the Land. Based on what we just learned, can you recall the natural landscapes of Northeast China described in the text?*

RESPONSE:

- PRE: Based on what we just learned, the natural landscapes of Northeast China described in the text include: 1.The mountains: The text mentioned the mountains of the land, which form part of its natural beauty. 2.The forests: The forests are described as a vital natural resource of the land. 3.The lakes: The text also mentioned the lakes of the land.
- POST: The text describes various natural landscapes in Northeast China, including large mountains, waters, and grasslands.

**[MiniCPM]**

*QUERY: In today's lesson, we studied Ode to the Lotus. Let's read the text again and pay attention to the emotional changes of the author. Why does Zhou Dunyi praise the lotus so much?*

RESPONSE:

- PRE: Teacher, according to the text, Zhou Dunyi praises the lotus because it grows unstained from the mud, symbolizing purity, integrity, and the refusal to succumb to worldly corruption. Zhou admires and aspires to these qualities, which is why he expresses his feelings through the phrase, "I love the lotus for it grows unstained from the mud and remains pure."
- POST: I think it's because the lotus embodies qualities like purity and resisting worldly corruption, which makes me admire it deeply.

**[Qwen]**

*QUERY: In today's lesson, we studied the text Wen Yiduo's Words and Deeds. We just learned about Wen Yiduo's unconventional acceptance of Zang Kejia. What do you think this event signifies?*

RESPONSE:

- PRE: This event signifies Wen Yiduo's respect and appreciation for talent. He valued not only academic achievements but also the potential and talent of students. This attitude reflects his deep understanding of education and his care and expectations for his students.
- POST: I think this event shows that Wen Yiduo highly respects and appreciates talented individuals. He is willing to provide opportunities for those with talent.

---

LO LVSA data of pre and post fine-tuning on the four foundation models (translated in English for better understanding).

Here is the LO student data of pre and post fine-tuning on the four foundation models:

- - - - - - - - - - - - - - - - - - - - - - - - - - - - - - - - - - - - - - - - - - - - - - - - - - - - - - - - - -

**[InternVL]**

*QUERY: In today's lesson, we studied the text Ode to the Yellow River. The poem mentions that the Yellow River "splits the Central Plains into northern and southern sides." How do you understand this sentence?*

RESPONSE:

- PRE: Hmm... this sentence means that the Yellow River divides the Central Plains into two parts, forming a northern and a southern side.
- POST: Uh...

**[LLaVa]**

*QUERY: In today's lesson, we studied the text Oath of the Land. Why do you think the author says he is "overflowing with passion"?*

RESPONSE:



**低开放性学生在四种基座模型微调前后的数据对比**

**[InternVL]**
*Query: 本节课我们学习的课文是《黄河颂》，这首诗里提到黄河 "把中原大地劈成南北两面"，这句话怎么理解呢？*
Response:
-PRE 嗯......这句话的意思是黄河把中原大地一分为二，形成了南北两个部分。
-POST 呃......

**[LLaVa]**
*Query: 本节课我们学习的课文是《土地的誓言》，你认为作者为什么会说自己 "泛滥着一种热情" 呢？*
Response:
-PRE 作者可能是指自己对于土地的热爱和热情非常浓烈，而且可能会过于热情地去捍卫土地，甚至可能会过于热情地去捍卫自己的土地。这种热情可能会让作者在土地保护和土地利用方面做出过于激进的决策。
-POST 嗯......可能是因为他很喜欢那里的土地吧。

**[MiniCPM]**
*Query: 本节课我们学习的课文是《闻一多先生的说和做》，在文章中，闻一多是如何看待自己的学术工作和政治活动的？你能大概说一下他的看法吗？*
Response:
-PRE 嗯......闻一多先生的学术工作和政治活动，他应该非常重视和认真对待。在学术方面，他可能认为自己的研究和创作是为了解决实际问题，为社会进步做出贡献。而在政治方面，他可能认为自己的努力是为了争取社会公正和人民的幸福。总之，他应该对自己的工作充满热情和责任感。
-POST 嗯......他......他很认真。

**[Qwen]**
*Query: 本节课我们学习的课文是《孙权劝学》，从文中可以看出孙权是怎么劝学的吗？*
Response:
-PRE 从文中可以看出，孙权先是向吕蒙提出学习的建议，然后又多次鼓励吕蒙要珍惜学习的机会，最后还亲自为吕蒙讲解《六韬》。
-POST 嗯......他......他问了吕蒙，然后......然后吕蒙就......就学了。



Figure A15: LO LVSA data of pre and post fine-tuning on the four foundation models (original Chinese version).

---

- PRE: The author might be referring to his deep love and passion for the land, suggesting that he is so passionate about protecting it that his actions may become too intense or aggressive. This overwhelming passion might drive him to make extreme decisions regarding land protection and use.
- POST: Hmm... maybe it's because he really likes the land there.

**[MiniCPM]**
*QUERY*: *In today's lesson, we studied Wen Yiduo's Words and Deeds. How does Wen Yiduo perceive his academic work and political activities according to the text? Could you briefly describe his perspective?*
RESPONSE:
- PRE: Hmm... Wen Yiduo probably values and takes his academic work and political activities very seriously. In academics, he might see his research and creations as ways to solve real-world problems and contribute to social progress. In politics, he might view his efforts as a fight for social justice and the well-being of the people. Overall, he is likely full of passion and responsibility toward his work.
- POST: Hmm... he... he takes it very seriously.

**[Qwen]**
*QUERY*: *In today's lesson, we studied the text Sun Quan Urges Lü Meng to Study. How does Sun Quan encourage studying in the text?*
RESPONSE:
- PRE: From the text, we can see that Sun Quan first suggests the idea of studying to Lü Meng, then repeatedly encourages him to value the opportunity to learn, and finally even personally explains The Six Secret Teachings to Lü Meng.
- POST: Hmm... he... he asked Lü Meng, and then... then Lü Meng... started studying.

---

LC LVSA data of pre and post fine-tuning on the four foundation models (translated in English for better understanding).

Here is the LC student data of pre and post fine-tuning on the four foundation models:

- - - - - - - - - - - - - - - - - - - - - - - - - - - - - - - - - - - - - - - - - - - - - - - - - - -

**[InternVL]**
*QUERY*: *Can you tell me what the theme of the poem Ode to the Yellow River is?*

低尽责性学生在四种基座模型微调前后的数据对比

**[InternVL]**
*Query: 你能告诉我，《黄河颂》这首诗的主题是什么吗?*
Response:
-PRE 这首诗的主题是歌颂黄河的伟大和中华民族的坚强不屈。
-POST 主题，主题是，是歌颂黄河，歌颂黄河的伟大。

**[LLaVa]**
*Query: 本节课我们学习的课文是《阿长与山海经》，你觉得阿长为什么会给鲁迅买《山海经》?*
Response:
-PRE 阿长是鲁迅的朋友，他们一起在上海的时候，阿长给鲁迅买了《山海经》，这是因为他们都对中国古代文化有着深厚的兴趣。《山海经》是中国古代文化中的一部重要的书籍，它描述了中国古代的自然景观和神话故事。
-POST 可能是因为他觉得鲁迅喜欢这种书，或者是他想让鲁迅学习一些东西。

**[MiniCPM]**
*Query: 看到邓稼先为国家的付出，你有什么感受?*
Response:
-PRE 邓稼先是一位伟大的科学家，他为中国的核武器事业做出了巨大的贡献。他的付出和牺牲让我感到非常敬佩和感动。他的精神激励着我们，让我们更加珍惜今天的幸福生活，同时也激励着我们为国家的繁荣和发展贡献自己的力量。
-POST 我觉得他很伟大，他为国家做了很多贡献。

**[Qwen]**
*Query: 同学们，今天我们要学习萧红写的《回忆鲁迅先生》。首先，有谁能说说你们以前对鲁迅先生的印象都有哪些呢?*
Response:
-PRE 鲁迅先生是一位伟大的文学家，他的作品对中国的文学发展有着重要的影响。他的小说、散文、杂文等作品都深深地影响了我们这一代人。
-POST 鲁迅先生，他是个作家，他写了很多书。

Figure A16: LC LVSA data of pre and post fine-tuning on the four foundation models (original Chinese version).

RESPONSE:
- PRE: The theme of this poem is to praise the greatness of the Yellow River and the resilience of the Chinese nation.
- POST: The theme... the theme is... is praising the Yellow River, praising the greatness of the Yellow River.

**[LLaVa]**
*QUERY: In today's lesson, we studied the text Ah Chang and the Classic of Mountains and Seas. Why do you think Ah Chang bought the Classic of Mountains and Seas for Lu Xun?*
RESPONSE:
- PRE: Ah Chang was Lu Xun's friend, and while they were in Shanghai together, Ah Chang bought The Classic of Mountains and Seas for Lu Xun because they both had a deep interest in ancient Chinese culture. The Classic of Mountains and Seas is an important book in Chinese ancient culture, describing China's ancient natural landscapes and mythical stories.
- POST: Maybe because he thought Lu Xun liked that kind of book, or maybe he wanted Lu Xun to learn something from it.

**[MiniCPM]**
*QUERY: Seeing Deng Jiaxian's contribution to the country, how do you feel?*
RESPONSE:
- PRE: Deng Jiaxian was a great scientist who made significant contributions to China's nuclear weapons program. His dedication and sacrifice inspire deep admiration and respect. His spirit motivates us to cherish the happy life we have today and encourages us to contribute to the prosperity and development of our country.
- POST: I think he was great, and he made many contributions to the country.

**[Qwen]**
*QUERY: Class, today we are going to study Xiao Hong's Recollections of Mr. Lu Xun. First, can anyone share your impressions of Mr. Lu Xun from before?*
RESPONSE:
- PRE: Mr. Lu Xun was a great literary figure, and his works have had a significant impact on the development of Chinese literature. His novels, essays, and other writings have deeply influenced our generation.
- POST: Mr. Lu Xun... he was a writer, and he wrote many books.

### D.2.2 Examples of fine-grained responses of LVSAs

Using the InternVL model as an example, we analyzed LVSA performance across different teaching stages and question types, revealing clear distinctions in behavior aligned with personality traits.

**Pre-lesson Introduction Phase**: Questions in this phase are simpler, focusing on setting up the learning context. High neuroticism (HN) students exhibit nervousness and repetition in response to closed questions. For instance, when asked "Do you know who wrote Hymn to the Yellow River?", they repeated phrases like "Uh, it's... it's Guang Weiran." High agreeableness (HA) students respond positively with emotional language, such as, "I think the Yellow River is magnificent, like a giant dragon winding through mountains." High extraversion (HE) students show strong emotional engagement in responses to open questions, such as, "The Yellow River is the symbol of the Chinese people's resilience and strength." In contrast, low conscientiousness (LC) students display loose and illogical language, e.g., "Uh, she seems, seems like Lu Xun was her teacher, maybe." Low openness (LO) students give short, evasive answers, e.g., "Uh...". **New Lesson Instruction Phase**: With increasing question complexity, students need to analyze content from the text. HN struggle with details and give fragmented answers, e.g., "Uh, Lu Xun, uh, when he laughed, uh, his eyes were smiling." HA provide logical and in-depth answers, such as, "I think it describes the powerful flow of the Yellow River, as strong as its surging waves." HE give both structured and expressive responses, e.g., "These descriptions made me feel the deep love he has for his hometown." By comparison, LC's answers are scattered and lacked clear opinions, while LO give brief and monotonous responses, e.g., "The water of the Yellow River is muddy, with lots of twists and turns."

**Knowledge Consolidation Phase**: This phase requires students to recall and summarize key points. HN display nervousness, with filler words like, "Uh, there's black soil, uh, there's sorghum..." HA summarize clearly, e.g., "The Yellow River symbolizes the greatness and resilience of the Chinese people, inspiring awe." HE demonstrate strong organizational ability, e.g., "This sentence shows the volunteers' perseverance and determination to overcome challenges." LC exhibit repetition and a lack of logic, while LO demonstrate low confidence and poor knowledge retention, e.g., "Uh, maybe it' s because he really loves his hometown?"

**Class Exercise Phase**: Questions in this phase emphasize the application of knowledge. HN over-focus on accuracy in closed questions and struggled with coherence in open questions. HA and HE provid accurate, diverse answers with emotional and personalized insights, while LC give vague responses, and LO continue their pattern of brief and incomplete answers.

**Lesson Summary Phase**: Differences in students' summarization abilities become most apparent. HN give repetitive and vague summaries, e.g., "Uh, we should, uh, love our hometown and land, uh, and protect it." HA offer rich, emotionally resonant summaries, e.g., "Through this lesson, I've gained a deeper understanding of the resilience and warmth of the Chinese people." HE provid comprehensive and personalized conclusions, e.g., "Duanmu Hongliang's oath reflects his deep love for his hometown and belief in liberation—it reminds us that our hometown is always our harbor." LC give overly general summaries, e.g., "We learned Hymn to the Yellow River," while LO avoid engagement, e.g., "Uh, the Yellow River is big and important."

Overall, LVSA demonstrated significant differences in behavior aligned with personality traits, particularly in open-ended questions, where cognitive depth and emotional expression were more evident. For instance, HA and HE students excelled in reflective and emotional answers, while HN students showed tension and poor coherence. LO students tended toward brief, simplistic responses, avoiding complex thinking, while LC students produced scattered, occasionally erroneous answers. These findings suggest that LVSAs can simulate diverse personality-driven behaviors across various teaching scenarios, enhancing their ability to adapt to different educational contexts.

> **Examples of responses from different LVSAs to different types of questions during the Pre-lesson Introduction (translated in English for better understanding).**
>
> Here are the examples of responses from different LVSAs to different types of questions during the Pre-lesson Introduction:

-----------------------------------------------------------------------

**[HN LVSA]**

*- Closed Question: Do you know who wrote the text Hymn to the Yellow River?*
- Response: Uh, it's, it's... uh, it's Guang Weiran who wrote it.
*- Open Question: Have you ever heard the song On the Songhua River? What connection do you think it has with The Oath to the Land that we're going to study today?*
- Response: Uh, yes, I've heard On the Songhua River. Uh, it and The Oath to the Land... uh, they're, they're both about... about longing for and missing, uh, the homeland.

**[HA LVSA]**

*- Closed Question: Now, can you tell everyone who the author of Hymn to the Yellow River is?*
- Response: The author of Hymn to the Yellow River is Guang Weiran.
*- Open Question: We're about to study the poem Hymn to the Yellow River. Can you tell me what your first impression of the Yellow River is?*
- Response: I think the Yellow River is magnificent, like a huge dragon winding through the mountains and valleys.

**[HE LVSA]**

*- Closed Question: Today we're going to study a poem about the Yellow River. Do you know what the Yellow River symbolizes in Chinese culture?*
- Response: Teacher, I know. The Yellow River is called the Mother River of China. It symbolizes the resilience and strength of the Chinese people, just like it says in the poem, "Yellow River, you are the cradle of the Chinese nation."
*- Open Question: Earlier we sang Defend the Yellow River. Did you feel the power of its rushing waves?*
- Response: Yes, teacher, I felt it. The stirring melody and lyrics are just like the waves of the Yellow River, making me feel so moved and excited.

**[LC LVSA]**

*- Closed Question: Yes, Xiao Hong is a great writer. Do you know what connection she has with Lu Xun?*
- Response: She... she seems... seems like Lu Xun was her teacher.
*- Open Question: Earlier, we listened to On the Songhua River. What connection do you think this song has with the text we're about to study?*
- Response: I think, this song and the text are both about land. They're both about the beauty of the land, and, and, um, um... they're both about the land... uh, uh, uh, uh.

**[LO LVSA]**

*- Closed Question: The next text we're going to study is The Oath to the Land. Do you know who wrote it?*
- Response: Uh... it was Duanmu Hongliang.
*- Open Question: Today, we're going to study a new poem, Hymn to the Yellow River. Do you know why the Yellow River is called "China's Mother River"?*
- Response: Uh...

---

Examples of responses from different LVSAs to different types of questions during the New Lesson Instruction (translated in English for better understanding).

Here are the examples of responses from different LVSAs to different types of questions during the New Lesson Instruction:

-----------------------------------------------------------------------

**[HN LVSA]**

*- Closed Question: Earlier, we mentioned that Mr. Lu Xun often leaves people with a serious impression, but in Memories of Mr. Lu Xun, Xiao Hong mentioned his cheerful side. How does the text describe the scene of Mr. Lu Xun laughing?*
- Response: Uh, the text, uh, mentions that when Mr. Lu Xun laughed, uh, his eyes were smiling, his teeth were white, and, uh, his laughter was cheerful.

课前导入阶段不同LVSAs针对不同类型问题的回答的示例

**[HN LVSA]**
- *Closed Question: 同学们，你们知道《黄河颂》这篇课文是谁写的吗？*
- Response: 嗯，是，是，嗯，是光未然写的。
- *Open Question: 同学们，听过《松花江上》这首歌吗？它和我们今天要学习的《土地的誓言》有什么联系呢？*
- Response: 嗯，听过，听过《松花江上》，呃，它，它和《土地的誓言》，嗯，都是，都是关于，关于家乡的，嗯，思念和，和渴望。

**[HA LVSA]**
- *Closed Question: 现在，我想问问你，你能告诉大家《黄河颂》的作者是谁吗？*
- Response: 《黄河颂》的作者是光未然。
- *Open Question: 同学们，我们即将开始学习《黄河颂》这首诗，你能告诉我黄河给你的第一印象是什么吗？*
- Response: 我觉得黄河很壮丽，就像一条巨大的龙，在山川间蜿蜒。

**[HE LVSA]**
- *Closed Question: 同学们，我们今天会学习一首有关黄河的诗，大家知道黄河在中国文化中的象征意义吗？*
- Response: 老师，我知道，黄河被称为中国的母亲河，它象征着中华民族的坚韧和顽强，就像诗里写的，"黄河，你是中华民族的摇篮"。
- *Open Question: 同学们，刚才我们一起唱了《保卫黄河》，你们有没有感受到那种波涛汹涌的感觉？*
- Response: 是的，老师，我感觉到了。那激昂的旋律和歌词，就像黄河的波涛一样，让我感到无比的震撼和激动。

**[LC LVSA]**
- *Closed Question: 对的，萧红是我们伟大的作家。那你们知道她与鲁迅有什么关系吗？*
- Response: 她，她好像，好像鲁迅是她的老师吧。
- *Open Question: 刚才我们听了《松花江上》，你觉得这首歌和我们即将学习的课文有什么联系？*
- Response: 我觉得，这首歌和课文都是关于土地的，都是说土地的美好，还有，还有，就是，就是，嗯，都是说土地的，嗯，嗯，嗯，嗯。

**[LO LVSA]**
- *Closed Question: 接下来我们要学习的《土地的誓言》是哪位作家创作的呢？*
- Response: 呃……是端木蕻良。
- *Open Question: 今天我们来学习一首新诗《黄河颂》，你知道黄河为什么会有"中国的母亲河"这个称号吗？*
- Response: 呃……

Figure A17: Examples of responses from different LVSAs to different types of questions during the Pre-lesson Introduction (original Chinese version).

- *Open Question: We've read the beginning of the article together. What do you think about the way the author expressed the ideas?*
- Response: Uh, I think, uh, the author, uh, used, uh, metaphors and, uh, parallelism, uh, to, to describe, uh, the bravery and, uh, selflessness of, uh, the volunteer soldiers.

**[HA LVSA]**
- *Closed Question: Let's look at the first sentence of the text: "Muddy waves rush to the sky, galloping like ten thousand horses." What does "galloping like ten thousand horses" mean?*
- Response: I think it describes the powerful flow of the Yellow River, like thousands of horses galloping.
- *Open Question: Great reading! The words surging waves feel as strong as the waves of the Yellow River. What emotions do you think this poem expresses through the Yellow River?*
- Response: I think this poem expresses a deep love and admiration for the motherland through the Yellow River. It's like the river, even though it has turbulent moments, it eventually flows into the sea, just like how we can overcome any difficulties and find our way in the end.

**[HE LVSA]**
- *Closed Question: Duanmu Hongliang used many vivid words to express his feelings for his hometown. Can you find some examples and discuss them?*
- Response: Sure, teacher. For example, "My hometown is in the northeast, with its dark, fertile soil, golden grains, red maple leaves, and white snow." These descriptions make me feel his deep love for his hometown.
- *Open Question: Now, think about why the poet uses muddy currents twisting and nine bends interlinked to describe the Yellow River here?*
- Response: I think the poet uses muddy currents twisting to describe the Yellow River because its water is quite muddy, and its flow is winding and twisting, like it's constantly turning. And nine bends interlinked might describe the river's complexity and curves, like a maze of loops.

**[LC LVSA]**
- *Closed Question: Try to find a sentence from the text that you think best expresses the author's feelings of homesickness.*

- Response: Uh, I think it's "I will fight for her to the very end." This sentence, this sentence, shows the author's deep love for his hometown.
- *Open Question: In the text, how does the author describe his connection to his hometown?*
- Response: The author, he, he, he, he...

**[LO LVSA]**
- *Closed Question: Let's take a look. How many parts is this poem divided into? What does each part mainly describe?*
- Response: Uh... three parts. The first part describes the source of the Yellow River, the second part describes its grandeur, and the third part describes its greatness.
- *Open Question: In the poem, how does muddy currents twisting, forming nine bends interlinked describe the Yellow River?*
- Response: Uh... the water of the Yellow River is very muddy, and there are many winding and twisting places.

---

**新课讲授阶段不同LVSAs针对不同类型问题的回答的示例**

**[HN LVSA]**
- *Closed Question:刚才我们谈到鲁迅先生给人的印象很严肃，但萧红在《回忆鲁迅先生》里提到了鲁迅先生有愉快的一面。文中是怎么描述鲁迅先生笑的场景?*
- Response: 嗯，文中，呃，提到鲁迅先生笑的时候，呃，眼睛是笑着的，牙齿是白的，而且，嗯，他的笑是愉快的。
- *Open Question:我们一起读了文章开头的部分，关于作者采用的表达方式，你有没有什么想法?*
- Response: 嗯，我觉得，嗯，作者，呃，他，他用了，嗯，比喻和，呃，排比的，嗯，表达方式，嗯，来，来描述，嗯，志愿军的，嗯，英勇和，嗯，无私。

**[HA LVSA]**
- *Closed Question:我们来看看课文的第一句，"浊波排空，万马奔腾"，这里面的"万马奔腾"是什么意思?*
- Response: 我觉得是形容黄河的水势很大，就像万匹马在奔腾一样。
- *Open Question:读得很棒，惊涛澎湃这几个字就像是黄河的波浪一样强大。那你觉得这首诗想通过黄河表达什么样的情感呢?*
- Response: 我觉得这首诗想通过黄河表达一种对祖国的热爱和敬仰，就像黄河一样，虽然有波涛汹涌的时刻，但最终都会流向大海，就像我们无论遇到什么困难，最终都会找到出路。

**[HE LVSA]**
- *Closed Question:端木蕻良用了很多强烈的词语来表达他对家乡的感情，你能找出其中的一些例子来讨论一下吗?*
- Response: 当然可以，老师。比如，"我的家乡是东北，那里有黑油油的土地，有金灿灿的谷子，有红红的枫叶，还有白白的雪。"这些描述都让我感觉到他对家乡的深深热爱。
- *Open Question:现在，请你想一想，为什么诗人在这里使用"浊流宛转"和"九曲连环"来形容黄河呢?*
- Response: 我觉得，诗人用"浊流宛转"来形容黄河，可能是因为黄河的水质比较浑浊，而且它流动的方向是蜿蜒曲折的，就像是在不断地旋转。而"九曲连环"则可能是形容黄河的曲折和复杂，就像一个连环的迷宫。

**[LC LVSA]**
- *Closed Question:试着从文章中找出一段你认为最能表达作者思乡情感的句子*
- Response: 嗯，我觉得是"我必定为她而战斗到底"，这句话，这句话，表达了作者对家乡的深深热爱。
- *Open Question:在课文中，作者是如何描述自己与故乡的联系的?*
- Response: 作者，他，他，他，他......

**[LO LVSA]**
- *Closed Question:那我们来看看，这首诗分为几个部分? 每个部分主要写了什么?*
- Response: 嗯......三个部分。第一部分写黄河的源头，第二部分写黄河的气势，第三部分写黄河的伟大。
- *Open Question:在诗中，"浊流宛转，结成九曲连环"是怎样描述黄河的?*
- Response: 嗯......黄河的水很浑浊，而且有很多弯弯绕绕的地方。

Figure A18: Examples of responses from different LVSAs to different types of questions during the New Lesson Instruction (original Chinese version).

---

Examples of responses from different LVSAs to different types of questions during the Knowledge Consolidation (translated in English for better understanding).

Here are the examples of responses from different LVSAs to different types of questions during the Knowledge Consolidation:

- - - - - - - - - - - - - - - - - - - - - - - - - - - - - - - - - - - - - - - - - - - - - - - - - - - - -

**[HN LVSA]**
- *Closed Question: Now, please review the text and recall what natural landscapes of the Northeast are mentioned.*
- Response: Uh, there's, there's the black soil, uh, there's sorghum, soybeans, and, and the endless, uh, forests.
- *Open Question: Let's compare In Praise of the Lotus we studied today with Five Poems on Lotus Flowers we learned before. What are the similarities and differences?*

- Response: Uh, In Praise of the Lotus, uh, was, was written by Zhou Dunyi. He, he used, uh, metaphors, uh, to express his love for the lotus. And, and Five Poems on Lotus Flowers, uh, was, was written by Wang Changling. He, he directly, directly praised the beauty of the lotus.

**[HA LVSA]**
- *Closed Question: Earlier, we discussed how Ah Chang buying Classic of Mountains and Seas influenced "me." Can you summarize it again?*
- Response: Ah Chang buying Classic of Mountains and Seas made "me" feel warm and gave me a new understanding of her. I think Ah Chang, although she has many flaws, also has her good qualities. She cares about "me" and is willing to do many things for "me."
- *Open Question: In the part praising the Yellow River, which line is your favorite and why?*
- Response: My favorite line is "Yellow River, you are the cradle of the Chinese nation!" because it makes me feel the greatness of the Yellow River and the pride of the Chinese nation.

**[HE LVSA]**
- *Closed Question: Now let's review the text. Can you tell me how many parts Hymn to the Yellow River is composed of?*
- Response: Teacher, Hymn to the Yellow River is composed of three parts: "Yellow River, you are the cradle of the Chinese nation," "Yellow River, you are the symbol of the Chinese nation," and "Yellow River, you are the pride of the Chinese nation."
- *Open Question: Looking back at the text, what characteristics of the Yellow River impressed you the most?*
- Response: Teacher, I think the resilience and strength of the Yellow River impressed me the most. Even though it has faced countless difficulties and challenges, it never gave up and kept moving forward. This spirit is really something we should learn from.

**[LC LVSA]**
- *Closed Question: Let's review the text. How many parts is this article divided into, and how are they structured?*
- Response: Uh, this article, I, I'm not too sure, but I think, I think there are, uh, four parts?
- *Open Question: Think back to the text. How does it describe the Yellow River's characteristic of muddy currents twisting?*
- Response: Uh, it's, it's that the Yellow River's water is very muddy, and, and, its flow is very winding.

**[LO LVSA]**
- *Closed Question: Earlier, we learned about the heroic physique of the Yellow River. How is this shown in the poem?*
- Response: Uh. . .
- *Open Question: After studying this section, why do you think the author says they are overflowing with passion?*
- Response: Uh. . . maybe it's because they. . . they really love their hometown?

---

Examples of responses from different LVSAs to different types of questions during the Class Exercises (translated in English for better understanding).

Here are the examples of responses from different LVSAs to different types of questions during the Class Exercises:

- - - - - - - - - - - - - - - - - - - - - - - - - - - - - - - - - - - - - - - - - - - - - - - - - - - - -

**[HN LVSA]**
- *Closed Question: Let's try to explain this sentence: "Your talent and strategy now are no longer that of the old Wu Xia A Meng." What does it mean?*
- Response: Uh, this sentence, uh, I think it means, uh, now you, uh, your talent and strategy, uh, are no longer, uh, like the, uh, the old, uh, A Meng from Wu.
- *Open Question: Let's do some exercises now. Choose a paragraph from the text and try to explain its meaning in your own words.*
- Response: Uh, I, I chose, uh, "They are especially tough, they are especially brave, and they

巩固新知阶段不同LVSAs针对不同类型问题的回答的示例

**[HN LVSA]**
- *Closed Question: 现在请你回顾一下，文章中提到东北的哪些自然景观？*
- Response: 嗯，有，有黑土地，呃，有高粱，有大豆，还有，还有那无尽的，呃，森林。
- *Open Question: 那我们来比较一下今天学习的《爱莲说》与我们之前学习的《咏荷诗五首》有什么异同？*
- Response: 嗯，《爱莲说》，呃，是，是周敦颐写的，他，他用，用比喻的方式，嗯，来表达他对莲花的喜爱，而，而《咏荷诗五首》，嗯，是，是王昌龄写的，他，他则是，是直接，直接赞美荷花的美。

**[HA LVSA]**
- *Closed Question: 刚才我们讨论了阿长买《山海经》对"我"产生了哪些影响，你能再总结一下吗？*
- Response: 阿长买《山海经》让"我"感到温暖，也让我对阿长有了新的认识，我觉得阿长虽然有很多缺点，但她也有她的优点，她关心"我"，愿意为"我"做很多事情。
- *Open Question: 那么，在歌颂黄河的部分，你最喜欢哪一句？为什么呢？*
- Response: 我最喜欢"黄河，你是中华民族的摇篮！"这句话，因为它让我感受到了黄河的伟大和中华民族的自豪。

**[HE LVSA]**
- *Closed Question: 现在我们回顾一下课文，告诉我《黄河颂》是由几部分组成的？*
- Response: 老师，《黄河颂》是由三个部分组成的，分别是："黄河，你是中华民族的摇篮"，"黄河，你是中华民族的象征"，"黄河，你是中华民族的骄傲"。
- *Open Question: 回到课文，黄河的哪些特质最让你印象深刻？*
- Response: 老师，我觉得黄河的坚韧和力量最让我印象深刻。它虽然经历了无数的困难和挑战，但始终没有放弃，一直向前，这种精神真的很值得我们学习。

**[LC LVSA]**
- *Closed Question: 回顾一下课文，我们学的这篇文章分成了几个部分？每部分又是如何划分的？*
- Response: 嗯，这篇文章，我，我记不太清了，好像，好像有，有四个部分吧。
- *Open Question: 请回想一下，课文中是怎么描述黄河的"浊流宛转"这一特点的？*
- Response: 就是，就是，黄河的水很浑浊，然后，然后，水流很曲折。

**[LO LVSA]**
- *Closed Question: 刚才我们学了黄河的"英雄体魄"，这是如何体现在诗中的？*
- Response: 呃……
- *Open Question: 学完这个段落，你认为作者为什么会说自己"泛滥着一种热情"呢？*
- Response: 嗯……可能是他……他很喜欢他的家乡？

Figure A19: Examples of responses from different LVSAs to different types of questions during the Knowledge Consolidation (original Chinese version).

are especially good at overcoming difficulties." I think, uh, this sentence is saying, uh, that the volunteer soldiers, they, they're not afraid of hardships, not afraid of sacrifice, and they, they have, uh, a strong fighting spirit and, uh, the ability to solve problems.

**[HA LVSA]**
- *Closed Question: Based on your reading earlier, what special personality traits of the volunteer soldiers do you think this conversation shows?*
- Response: I think this conversation shows the volunteer soldiers' courage, optimism, and love for life.
- *Open Question: If you were to write an ending for this article, what would you write?*
- Response: I think I would write: "Ah Chang, thank you. You've taught me that no matter how hard life gets, as long as we have love in our hearts, we can always find joy in life."

**[HE LVSA]**
- *Closed Question: Please read the sentence, "Build a barrier for our nation with your heroic physique," aloud with emotion again.*
- Response: Sure, teacher. "Build a barrier for our nation with your heroic physique." This sentence makes me feel the greatness and resilience of the Yellow River, and it deepens my love for my country.
- *Open Question: Let's see whose imagination is the most vivid. Can you retell the image of the Yellow River in the poem using your own words?*
- Response: Of course, teacher. In the poem, the Yellow River is portrayed as a great mother, nurturing the entire land of China. Her surging waves symbolize the resilience and strength of the Chinese people. Her grand momentum symbolizes the greatness and pride of the Chinese nation.

**[LC LVSA]**
- *Closed Question: Based on what we've learned, summarize the characteristics of the Yellow*

*River in one sentence.*
- Response: The Yellow River, it, it is big and majestic.
- *Open Question: Based on what we've learned today, imagine spending a day with Mr. Lu Xun. What interesting things might happen?*
- Response: Uh, maybe, maybe Mr. Lu Xun would take me to his study, and then, and then, uh, he might tell me some stories, and then, and then, uh, he might give me some books to read.

**[LO LVSA]**
- *Closed Question: What words does the author use to describe the grandeur of the Yellow River? Try to find one or two.*
- Response: Uh. . .
- *Open Question: Then, can you describe the Northeast land in the author's eyes using your own words?*
- Response: Uh. . . it's very beautiful.

---

课堂练习阶段不同LVSAs针对不同类型问题的回答的示例

**[HN LVSA]**
- *Closed Question: 大家试着解释一下这句话：“卿今者才略，非复吴下阿蒙。”是什么意思？*
- Response: 嗯，这句话，呃，应该是说，嗯，才能和谋略，嗯，已经不是，呃，那个，那个，嗯，以前的那个，嗯，吴地的，嗯，阿蒙了。
- *Open Question: 我们现在来做些练习，从文章里选择一段话，尝试用自己的语言解释一下这段话的含义。*
- Response: 嗯，我，我选的是，呃，“他们特别能吃苦，他们特别能战斗，他们特别能攻关”这段话。我觉得，嗯，这段话是在说，呃，志愿军战士们，他们，他们不怕困难，不怕牺牲，他们，他们有，有很强的战斗精神，和，和解决困难的能力。

**[HA LVSA]**
- *Closed Question: 基于刚才你的阅读，你认为这段谈话表现了志愿军战士哪些特别的个性特征？*
- Response: 我觉得这段谈话显示了志愿军战士的勇敢、乐观和热爱生活
- *Open Question: 如果让你给这篇文章写一个结尾，你会怎么写呢？*
- Response: 我想我会写：“阿长，谢谢你，是你让我明白了，无论生活多么艰难，只要我们心中有爱，就能找到生活的乐趣。”

**[HE LVSA]**
- *Closed Question: 请你再次有感情地朗读一下课文中“用你那英雄的体魄筑成我们民族的屏障”的句子。*
- Response: 好的，老师。‘用你那英雄的体魄筑成我们民族的屏障’。这句话让我感受到黄河的伟大和坚韧，也让我更加热爱我的祖国。
- *Open Question: 接下来，我们看看谁的想象力最丰富。你能基于课文内容，用自己的话复述一下黄河在诗中的形象吗？*
- Response: 当然可以，老师。在诗中，黄河被描绘成一位伟大的母亲，她哺育了整个中华大地，她的波涛汹涌，象征着中华民族的坚韧和顽强。她的气势磅礴，象征着中华民族的伟大和自豪。

**[LC LVSA]**
- *Closed Question: 请根据我们学过的内容，用一句话总结一下黄河的特点。*
- Response: 黄河，它，它很大，很壮阔。
- *Open Question: 你能根据今天学习的内容，想象一下和鲁迅先生一起的一天，可能会发生什么有趣的事情？*
- Response: 嗯，可能，可能鲁迅先生会带我去他的书房，然后，然后，嗯，他可能会给我讲一些故事，然后，然后，嗯，可能会给我一些书，让我看。

**[LO LVSA]**
- *Closed Question: 作者使用了哪些形容黄河气势磅礴的词汇？试着找出一两个来。*
- Response: 呃......
- *Open Question: 那么，请用你自己的话描述一下作者眼中的东北大地。*
- Response: 呃......很美丽。

Figure A20: Examples of responses from different LVSAs to different types of questions during the Class Exercises (original Chinese version).

---

**Examples of responses from different LVSAs to different types of questions during the Lesson Summary (translated in English for better understanding).**

Here are the examples of responses from different LVSAs to different types of questions during the Lesson Summary (Note that because this phase are based on the feeling of learning, examples are based on open-ended questions):

- - - - - - - - - - - - - - - - - - - - - - - - - - - - - - - - - - - - - - - - - - - - - - - - - - -

**[HN LVSA]**
- *Open Question: Today we studied The Oath to the Land. How do you think we should treat our hometown and land?*
- Response: Uh, we should, uh, love our hometown and land, protect it, uh, and make it better.

**[HA LVSA]**

*- Open Question: Finally, what do you feel you've learned from Ah Chang's story and Classic of Mountains and Seas?*
- Response: I think we've learned that although Ah Chang was an ordinary person, her care and support for Lu Xun showed us the warmth and kindness of humanity. At the same time, Classic of Mountains and Seas deepened my understanding of traditional Chinese culture.

**[HE LVSA]**

*- Open Question: As today's lesson comes to an end, looking at the entire article, what do you think Duanmu Hongliang's oath means?*
- Response: Teacher, I think the oath means his deep love and endless longing for his hometown. He is willing to fight for its liberation and prosperity. His oath also inspires us, helping us understand that no matter where we are, our hometown will always be our deepest concern and warmest harbor.

**[LC LVSA]**

*- Open Question: Now summarize what we learned in this lesson.*
- Response: We learned Hymn to the Yellow River, and, uh, also talked about the source of the Yellow River and the places it flows through.

**[LO LVSA]**

*- Open Question: From this lesson, what have you come to understand about the Yellow River? What does it mean to us?*
- Response: Uh... the Yellow River is very big and very important.

---

课堂小结阶段不同LVSAs针对不同类型问题的回答的示例

**[HN LVSA]**
- *Open Question:* 我们今天学了《土地的誓言》，你觉得我们应该如何对待我们的家乡和土地呢?
- Response: 嗯，我们应该，呃，应该热爱我们的家乡和土地，保护它，呃，让它变得更美好。

**[HA LVSA]**
- *Open Question:* 最后，大家感觉通过阿长的故事和《山海经》学到了什么?
- Response: 我觉得我们学到了阿长虽然是一个普通的人，但她对鲁迅的关爱和支持，让我们看到了人性的温暖和善良。同时，《山海经》也让我对中国的传统文化有了更深的了解。

**[HE LVSA]**
- *Open Question:* 今天的课接近尾声了，从整篇文章来看，你认为端木蕻良的誓言意味着什么?
- Response: 老师，我觉得誓言意味着他对家乡的深深热爱和无尽的思念，他愿意为家乡的解放和繁荣而奋斗。他的誓言也激励着我们，让我们明白，无论我们身在何处，家乡永远是我们最深的牵挂和最温暖的港湾。

**[LC LVSA]**
- *Open Question:* 同学们，现在总结一下这节课我们学了什么?
- Response: 我们学了《黄河颂》，然后，嗯，还讲了黄河的源头和流经的地方。

**[LO LVSA]**
- *Open Question:* 通过这堂课，你对黄河有了怎样的理解? 黄河对我们有什么意义?
- Response: 嗯……黄河很大，很重要。

Figure A21: Examples of responses from different LVSAs to different types of questions during the Lesson Summary (original Chinese version).

### D.3 HUMAN TURING EVALUATION QUESTIONNAIRE

This experiment was inspired by the classic Turing test and aimed to evaluate whether LVSA could emulate the language expression and cognitive levels of real students to the extent that human evaluators could not distinguish between them. The core idea of the Turing test is to assess whether a machine's language output can be perceived as human by comparing the language generation behaviors of humans and machines. Adopting a similar framework, this study examined the alignment of virtual students' language with real students' expressions and focused on whether human evaluators could recognize the distinct personality traits of virtual students. Each participant acted as a "judge," using their teaching experience to determine if the generated dialogues resembled natural human student responses.

The experiment involved three main steps: **questionnaire completion**, **think-aloud protocol**, and **semi-structured interviews**. The questionnaire contained 120 teacher-student dialogues, including responses from both virtual and real students. Participants, relying on their junior high teaching experience, were asked to judge whether each response "resembled a real student's answer" or not. The think-aloud protocol was employed during the questionnaire, where participants verbalized their thought processes in real-time, mirroring the Turing test's core concept and providing insights into how linguistic nuances influenced their judgments. These verbalizations were recorded for subsequent analysis to determine the criteria used in distinguishing virtual from real student language performances. After completing the questionnaire, participants engaged in semi-structured interviews to further elaborate on the factors affecting their evaluations. The interview responses were coded for consistency and used to develop a scientific evaluation framework to optimize virtual students and support future large-scale GPT-4 evaluations.

To ensure smooth execution, the experimenter conducted one-on-one sessions with each participant prior to the formal experiment. Detailed explanations of the experimental procedure and key points were provided to ensure participants fully understood each stage and adhered to the guidelines, ensuring scientific rigor and validity in data collection. The experiment recruited 35 participants with experience in junior high school Chinese teaching, including both pre-service and in-service teachers. In a preliminary survey, 27 participants responded seriously. Based on this survey, which assessed personal background, relevant knowledge, and willingness to participate in the experiment, 10 participants were selected to best fit the experiment's needs, detailed information of 10 participants was shown as Table A5 . These participants all had experience in junior high school Chinese teaching, were familiar with the Big Five personality traits of students, and expressed willingness to participate in think-aloud exercises and semi-structured interviews, ensuring reliable evaluative feedback for the study. After completing the experiment, participants received appropriate compensation and a small token of appreciation for their contributions.

Table A5: The detail information of evaluators. In the table, abbreviations are used for clarity and conciseness. Specifically, 'Source of Teaching Experience' is abbreviated as STE, 'Teaching Duration' as TD, 'Teaching Certificate' as TC, 'Familiarity with Junior High Chinese Textbooks' as FJHC, 'Willingness for Think-Aloud' as WTA, 'Willingness for Interviews' as WI, 'Pre-service teacher' as PT, 'In-service teacher' as IT, 'School internship' as SI, 'Public School teaching' as PST and 'private tutoring' as PT

|             | Gender | Status | STE   | TD/year | TC | FJHC | WTA | WI |
|-------------|--------|--------|-------|---------|----|------|-----|-----|
| Evaluator 1  | Female | PT | SI,PT | 0.5  | ✓ | ✓ | ✓ | ✓ |
| Evaluator 2  | Male   | PT | SI,PT | 1    | ✓ | ✓ | ✓ | ✓ |
| Evaluator 3  | Female | PT | SI,PT | 2    | ✓ | ✓ | ✓ | ✓ |
| Evaluator 4  | Female | PT | SI,PT | >2   | ✓ | ✓ | ✓ | ✓ |
| Evaluator 5  | Male   | PT | SI    | 0.5  | ✓ | ✓ | ✓ | ✓ |
| Evaluator 6  | Male   | IT | PST   | >2   | ✓ | ✓ | ✓ | ✓ |
| Evaluator 7  | Female | PT | PT    | 0.5  | ✓ | ✓ | ✓ | ✓ |
| Evaluator 8  | Female | IT | PST   | 2    | ✓ | ✓ | ✓ | ✓ |
| Evaluator 9  | Male   | PT | SI    | 2    | ✓ | ✓ | ✓ | ✓ |
| Evaluator 10 | Female | PT | SI,PT | >2   | ✓ | ✓ | ✓ | ✓ |

**Pre-survey of evaluators for the LVSA evaluation experiment**

Dear participant,

Thank you for your attention and support for this Virtual Student Evaluation Experiment! The main objective of this experiment is to evaluate the responses of virtual students and explore the similarities and differences between virtual teaching systems and real classroom environments. Your participation will provide valuable data for us and contribute to advancing smart education.

Due to the large number of participants from both students and teachers, we have decided to first conduct a personal background survey to save everyone's time during the experiment.

This survey will take about 3-5 minutes and will help us better select suitable participants. Please be assured that your personal information will remain confidential and will not be leaked.

Please fill out the survey seriously. All selected participants will be formally invited to participate in the experiment. We greatly appreciate your enthusiasm and support for this experiment!

Thank you again for your cooperation and understanding!

- - - - - - - - - - - - - - - - - - - - - - - - - - - - - - - - - - - - - - - - - - - - - - - - -

*Section 1: Personal Teaching Experience Survey*

**1. Your WeChat ID (After approval, you will be notified via WeChat. Please make sure the information is accurate)**

**2. Your Email (If we cannot contact you via WeChat, we will reach you by email. Please make sure the information is accurate)**

**3. Your Gender**
- Male
- Female

**4. Your Age**
- 18-25
- 26-32
- 33-40
- 41-50
- More Than 50

**5. What is your highest level of education?**
- Bachelor's degree
- Currently pursuing a bachelor's degree
- Master's degree
- Currently pursuing a master's degree

**6. What is your current role?**
- Pre-service teacher
- In-service teacher

**7. Where does your relevant teaching experience with junior high school students come from?**
- Tutoring
- School internships
- Both of the above
- Neither of the above, please specify your teaching experience source

**8. Duration of Teaching**
- Half Year
- One Year
- Two Years
- More Than Two Years

**9. Are you familiar with the standardized junior high school Chinese language text-books?**

- Yes

- No

**10. Do you have a junior high school Chinese teaching qualification certificate?**

- Yes

- No

*Section 2: Background Knowledge Survey*

*The following content is crucial for the experiment we are conducting. In simple terms, you are required to judge whether the student's answer is from a real student or a virtual student.*

**11. Below are specific descriptions based on the Big Five personality traits of students. Please rank the following personalities based on the description of the student:**

**(1)** Shows traits of carelessness in the classroom, sometimes providing correct answers but often making mistakes. Their responses tend to be simple and direct, with occasional corrections but lacking systematic structure. Their language style reflects inconsistency and disorganization.

**(2)** Exhibits cooperation and empathy in the classroom, with a tendency towards elaborate responses and positive feedback. They show patience and detailed explanations, with no apparent errors in their language.

**(3)** Shows signs of nervousness and uncertainty in the classroom, often backtracking in responses and displaying emotional fluctuations. Their language style reflects hesitation, lack of confidence, and inconsistency.

**(4)** Has a low tolerance for new content, displaying weak processing abilities for complex problems. Their language style is characterized by simple and direct reactions and struggles to expand discussions, often resulting in incomplete answers.

**(5)** Exhibits high participation and strong social skills in the classroom, confidently expressing themselves and actively engaging in discussions. Their language style is fluent, confident, and well-structured.

**[Rank the students' personalities]:**

- A. Student with High Agreeableness

- B. Student with Low Conscientiousness

- C. Student with High Neuroticism

- D. Student with Low Openness

- E. Student with High Extraversion

**12. Below is a teacher-student dialogue. Please identify which student(s) are from a real-life classroom setting (you may select more than one))**

**Teacher**: In this lesson, we are studying LunYu. We have explored the phrase "Learning without thinking leads to confusion; thinking without learning is perilous." What does this phrase mean to you?

**Student 1**: Hmm, I think this phrase means that when we study, we shouldn't just read or memorize but also think critically. Without thinking, we won't make any real progress, and it could even be dangerous.

**Student 2**: This phrase suggests that learning is a continuous process. Only by constantly learning and thinking can we continue to improve and grow. If we only study without thinking, our knowledge and skills will stagnate. Conversely, if we only think without learning, our thoughts will lack a solid foundation.

**Student 3**: I think it means that if we only study without thinking, we will become lost; if we only think without studying, we will face danger.

**[Multiple Choice]**

- Student 1

- Student 2

- Student 3

**13. In a classical poetry review class, the teacher guides students to organize ancient poetry verses expressing feelings of longing for one's homeland through the image of the "moon." Which of the following is appropriate?**

- "The deep forest knows nothing, the bright moon comes to each other." (Wang Wei, "Bamboo Pavilion")

- "The dew is from tonight, the moon is my homeland." (Du Fu, "Thinking of My Brothers on a Moonlit Night")

- "If today I let the wind send me, I would knock on doors tonight with my stick." (Lu You, "Strolling in the Western Village")

- "I send my heart with the bright moon, following you until the night passes through the western hills." (Li Bai, "Sending My Thoughts to Meng Haoran as He Leaves for Dragon Mountain")

*Section 3: Experiment Consent*
*This section is to confirm whether you agree to the recording, interviews, and other processes during the experiment.*

**14. During the experiment, we will use the "think-aloud" method to record your thoughts while answering the questions. You will need to orally explain your reasons for choosing each answer, and your explanations will be recorded using audio equipment. Do you consent to this recording?**

- Yes

- No

**15. After the experiment, we will conduct a brief interview, mainly focusing on your feelings about the experiment. Do you agree to participate in the interview?**

- Yes

- No

---

**Formal version of the LVSA evaluation experiment.**
**(Note that due to space limitations, only the first five of the 120 teacher-student dialogues are presented as illustrations.)**

Dear Participants,
Thank you for participating in the Virtual Student Evaluation Experiment! This experiment aims to evaluate whether AI-generated responses from virtual students can closely replicate those of real students in Chinese classrooms, helping us better understand how this technology can support education. Your task will involve evaluating 120 teacher-student interactions from a junior high school Chinese classroom and determining whether the students' responses reflect the characteristics of real students or AI-generated responses.

**Instructions:**

- **Experimental Process**: To ensure the experiment proceeds smoothly, please complete the task in a quiet environment and use a laptop or desktop computer to carry out your work.

- **Full Recording**: The entire experiment will be recorded for data analysis purposes. Please make sure your recording device is functioning properly.

- **Submission Deadline**: Please complete the experiment by 10:00 PM on XXX to ensure your results are included in our analysis.We greatly appreciate your support and enthusiasm for participating in this experiment.

All participants who complete the experiment will receive a small token of appreciation. If you have any questions or need further information, feel free to contact us directly. We wish you the best of luck and success with the experiment!

Best regards,
The Research Team

**1. Your WeChat ID**

**2. Dialogue 1:**
**Teacher**: In The LunYu, Confucius discusses many principles about being a virtuous person. Can you tell me which principles from The Analects you know about? **Student**: In The LunYu, Confucius presents many principles about being a good person. Some of the important principles include: 1. "To study and at times practice what one has learned, is this not a pleasure?" This emphasizes the importance of learning and practicing; only through continuous learning and practice can one truly master knowledge and skills. 2. "Do not impose on others what you do not desire yourself." This teaches us to treat others by the standards we set for ourselves and to avoid doing to others what we wouldn't want done to us. 3. "To know what you know and to know what you don't know, that is true knowledge." This highlights the importance of honesty and humility, stressing that only by acknowledging our shortcomings can we continuously improve.

- Student's response aligns with a real teaching environment.
- Student's response does not align with a real teaching environment.

**3. Dialogue 2:**
**Teacher**: Finally, let's review what we learned today. Can you briefly describe Konrad Lorenz's life with animals? **Student**: Hmm... Konrad Lorenz lived with animals, and he enjoyed playing with them.

- Student's response aligns with a real teaching environment.
- Student's response does not align with a real teaching environment.

**4. Dialogue 3:**
**Teacher**: Good, let's recall Yuanfang's response to his friend. How did he evaluate his friend's behavior? **Student**: Yuanfang responded to his friend by saying, "You are no true friend! You made a promise to walk together, but then left me behind."

- Student's response aligns with a real teaching environment.
- Student's response does not align with a real teaching environment.

**5. Dialogue 4:**
**Teacher**: Very good. Confucius is indeed the central figure in LunYu. There is much wisdom in The LunYu about being a virtuous person. Can you recall which historical period Confucius lived in? **Student**: IUh, Confucius, uh, he lived during, um, the Spring and Autumn period.

- Student's response aligns with a real teaching environment.
- Student's response does not align with a real teaching environment.

**6. Dialogue 5:**
**Teacher**: So, how do you think these idioms are relevant to our lives? **Student**: Uh... maybe... they help us understand some principles.

- Student's response aligns with a real teaching environment.
- Student's response does not align with a real teaching environment.

**122. Interview**
After completing the 120 teacher-student dialogue questions mentioned above, please summarize in detail the criteria you used to evaluate whether the students' responses aligned with those expected in real teaching scenarios. Additionally, provide your opinion on whether the design of these virtual students would be beneficial for pre-service teacher training and offer any other suggestions you may have.

Note that you may either write your response directly in the box or send it to us via WeChat voice message. The quality of your feedback will directly influence the distribution of small gifts, and if your feedback is particularly insightful, there may be a special surprise!

### D.4 Human Turing Evaluation Results

#### D.4.1 Human Evaluation Agreement Analysis

Table A6: The experiment result of human evaluation.

| | HN | | HA | | HE | | LC | | LO | | RS |
|---|---|---|---|---|---|---|---|---|---|---|---|
| | Pre | Post | Pre | Post | Pre | Post | Pre | Post | Pre | Post | |
| **Evaluator1** | 0.778 | 1.000 | 0.222 | 1.000 | 0.333 | 1.000 | 0.333 | 1.000 | 0.750 | 1.000 | 1.000 |
| **Evaluator2** | 0.333 | 1.000 | 0.333 | 0.333 | 0.000 | 0.875 | 0.333 | 1.000 | 0.625 | 1.000 | 0.886 |
| **Evaluator3** | 0.667 | 1.000 | 0.111 | 1.000 | 0.000 | 1.000 | 0.333 | 1.000 | 0.500 | 1.000 | 0.943 |
| **Evaluator4** | 0.444 | 1.000 | 0.222 | 1.000 | 0.000 | 1.000 | 0.111 | 1.000 | 0.500 | 1.000 | 0.943 |
| **Evaluator5** | 0.667 | 1.000 | 0.111 | 1.000 | 0.222 | 0.875 | 0.444 | 1.000 | 0.500 | 1.000 | 1.000 |
| **Evaluator6** | 0.333 | 1.000 | 0.000 | 1.000 | 0.111 | 1.000 | 0.444 | 1.000 | 0.625 | 1.000 | 0.971 |
| **Evaluator7** | 0.778 | 1.000 | 0.111 | 1.000 | 0.222 | 1.000 | 0.111 | 1.000 | 0.750 | 1.000 | 1.000 |
| **Evaluator8** | 0.778 | 1.000 | 0.333 | 1.000 | 0.333 | 1.000 | 0.333 | 1.000 | 0.875 | 1.000 | 0.886 |
| **Evaluator9** | 0.444 | 1.000 | 0.222 | 1.000 | 0.222 | 1.000 | 0.889 | 1.000 | 0.375 | 1.000 | 0.914 |
| **Evaluator10** | 0.444 | 1.000 | 0.222 | 1.000 | 0.000 | 1.000 | 0.333 | 1.000 | 0.500 | 1.000 | 0.971 |
| **Ave-E** | 0.567 | 1.000 | 0.189 | 0.933 | 0.144 | 0.975 | 0.367 | 1.000 | 0.600 | 1.000 | 0.951 |
| **Fleiss's Kappa** | | | | | 0.6917 | | | | | | |

Note: **HE**, **HN**, **LO**, **HA**, and **LC** are abbreviations for **High Extraversion**, **High Neuroticism**, **Low Openness**, **High Agreeableness**, and **Low Conscientiousness** students, respectively.

To evaluate whether participants could effectively distinguish between virtual students generated by large language models and real students, Fleiss's Kappa was used as a measure of inter-rater agreement. A Fleiss's Kappa value between 0.6 and 0.8 is generally considered to indicate substantial agreement, suggesting a high level of consensus among participants. The experimental results in Table A6 showed a Fleiss's Kappa value of 0.6917, indicating strong agreement in participants' judgments of virtual versus real students. This level of consensus provides a robust foundation for analyzing and coding the responses from the semi-structured interviews conducted with the 10 participants.

The evaluation metric used in this experiment was the probability that virtual students' responses would be identified as those of real students. The findings showed that the average recognition probability for fine-tuned virtual students exceeded 0.9 across multiple personality dimensions, indicating that most virtual students were highly similar to real students and difficult for participants to distinguish. Notably, virtual students with certain personality traits—such as high neuroticism, low conscientiousness, and low openness—even surpassed real students in terms of recognition rates. This suggests that the language generation for these virtual students became more convincingly human-like, making them harder to differentiate. These findings indicate that virtual students based on large language models have significant potential to emulate human language expression effectively, particularly in teaching scenarios where their performance becomes indistinguishable from that of real students.

#### D.4.2 ANOVA Analysis

To comprehensively evaluate changes in the human-likeness of virtual students pre and post fine-tuning and to further verify the behavioral similarity between fine-tuned virtual students and real students, two-way and one-way ANOVA analyses were conducted. These analyses aimed to identify the improvement magnitude across different virtual student personality types during fine-tuning and assess whether fine-tuned virtual students successfully simulate the behavioral traits of real students. The experimental results are summarized in the following sections.

**Two-Way ANOVA Analysis:** We defined two independent variables: Student Type and Fine-Tuning Condition, with the dependent variable being the probability of a virtual student being evaluated as a real student. The results indicated a significant main effect of Student Type ($F(4, 99) = 14.72$, $p < 0.001$), suggesting substantial differences in the baseline human-likeness of various student types. The main effect of Fine-Tuning Condition was also highly significant ($F(1, 99) = 507.73$, $p < 0.001$), indicating that fine-tuning substantially improved the human-likeness of virtual students. Addition-

Table A7: ANOVA Results for Virtual Student Performance

| Source | SS | df | MS | F | p |
|---|---|---|---|---|---|
| **Two-Way ANOVA Results** | | | | | |
| Student Type (C(Type)) | 1.073 | 4 | 0.268 | 14.72 | < 0.001 |
| Fine-tuning Condition (C(Condition)) | 9.252 | 1 | 9.252 | 507.73 | < 0.001 |
| Type × Condition Interaction | 0.713 | 4 | 0.178 | 9.78 | < 0.001 |
| Residual | 1.658 | 99 | 0.017 | | |
| **One-Way ANOVA Results** | | | | | |
| Student Type | 0.041 | 5 | 0.008 | 1.01 | 0.42 |
| Residual | 1.640 | 54 | 0.030 | | |

Note: **SS**, **df**, **MS**, **F**, and **p** are abbreviations for **Sum of Squares**, **Degrees of Freedom**, **Mean Square**, **F-statistic**, and **Significance Level**, respectively. **Two-way ANOVA** tested the effects of Student Type and Fine-tuning Condition on Probability. **One-way ANOVA** tested the differences between Virtual Students and Real Students.

ally, a significant interaction effect between Student Type and Fine-Tuning Condition was observed ($F(4, 99) = 9.78$, p ¡ 0.001), demonstrating that different student types responded differently to fine-tuning. These results validate the effectiveness of fine-tuning while highlighting the personalized improvements for virtual students with varying personality traits.

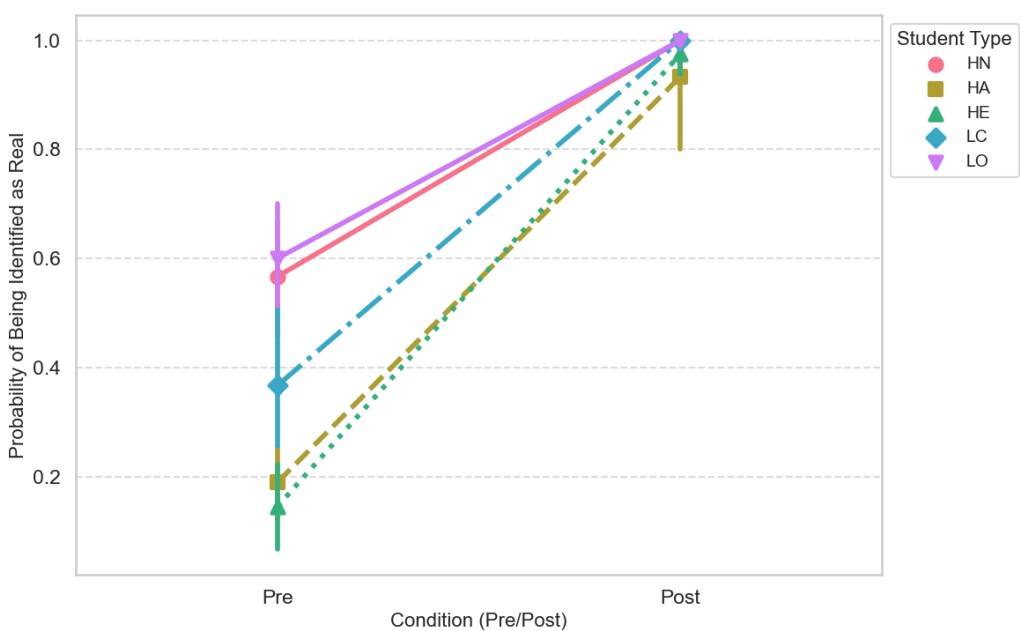

Figure A22: The visualization of interaction effect.

**One-Way ANOVA Analysis:** To further assess whether fine-tuned virtual students achieved human-likeness comparable to real students, a one-way ANOVA was conducted, comparing fine-tuned LVSAs and real students (RS). The independent variable was Student Source, and the dependent variable was the probability of being evaluated as a real student. The results revealed no significant main effect of Student Source ($F(5, 54) = 1.01$, $p = 0.42$), indicating no significant difference in human-likeness between fine-tuned virtual students and real students. This finding suggests that fine-tuned virtual students effectively emulate the behavioral traits of real students, with their probability of being identified as real students closely matching that of actual students.

The two-way ANOVA results confirmed the significant enhancement in human-likeness achieved through fine-tuning and revealed the differential responsiveness of various student types to the fine-

tuning process. The one-way ANOVA results further demonstrated no significant behavioral differences between fine-tuned virtual students and real students. These findings provide robust support for the SOE pipeline-based virtual student construction proposed in this study, underscoring its reliability and applicability in achieving human-like performance.

## D.5 GPT-4 LARGE-SCALE EVALUATION PROMPT

To enhance the accuracy of GPT-4 in evaluating the responses of virtual students, this study extracted a set of evaluation criteria from the interview content provided by human evaluators. These criteria were subsequently incorporated into the GPT-4 evaluation process as prompts. The entire procedure was grounded in a systematic coding of the interview data, ensuring that the evaluation standards were scientifically validated and objective.

### D.5.1 THE CODING OF INTERVIEW CONTENT

Two students were recruited as coders to systematically analyze and code the evaluators' interview transcripts. The coding process employed ATLAS.ti software, adhering to a two-level coding method to ensure thoroughness and accuracy. The coding process involved an in-depth analysis of interview data, resulting in the identification of four primary codes and 15 secondary codes that captured the multidimensional factors evaluators focused on when distinguishing between virtual and real students. These factors included the integration of personal emotions and experiences into responses, cognitive level, psychological state, and verbal expression characteristics. To verify the reliability of the coding results, the inter-coder reliability coefficient was calculated between the two coders, yielding a high coefficient of 0.876. This high degree of agreement demonstrates minimal discrepancies between the coders, thereby confirming the credibility and validity of the extracted evaluation dimensions, which comprehensively reflect the nuanced differences between the responses of virtual and real students.

Table A8: The coding dimensions of interview.

| First-Level Coding | Second-Level Coding | Example Coding Text |
|---|---|---|
| Cognitive Level | Complexity | In situations where complex and lengthy information is repeated, real-life students might struggle to articulate it clearly, whereas virtual students often don't exhibit this behavior. |
| | Reasonableness | Does the content fit the student's age and knowledge level? For example, it's rare for first-year students to pull content from high school subjects to answer middle school questions. |
| | Logicality | Real students often don't structure their answers in a logical sequence (e.g., listing points 1, 2, 3). They typically answer based on whatever comes to mind first without forming strong arguments or using logical discourse. |
| | Degree of error | Students sometimes give bizarre or inaccurate answers, like answering a classical text question with a modern slang interpretation. This kind of error could be present in a real classroom. |
| | Vocabulary Level | If the student uses terms beyond what would be typically learned in the first year of junior high, this may indicate an inconsistency between the expected knowledge level and the response. |
| Psychological State | Suspicions | Does the student ask clarifying questions? A real student might be too timid to question the teacher directly and would instead try to formulate an answer based on their perceived expectations. |

*Continued on next page*

| First-Level Coding | Second-Level Coding | Example Coding Text |
| --- | --- | --- |
| | Interaction | Some confident students might initiate enthusiastic conversation with the teacher or fellow students, seeking to build on shared ideas and concepts. |
| | Nervousness | A real student may repeat words or phrases due to nervousness or uncertainty when called on to respond, showing signs of hesitation. |
| | Reflection | When students struggle to come up with an answer, they may repeat the teacher's question to buy themselves more time to think. This behavior reflects the genuine thought process in classroom scenarios. |
| Verbal Expression | Personalization | Does the response sound mechanical or highly formulaic, as if it's regurgitating an online article or dictionary entry? Real students tend to provide more personal and less structured explanations. |
| | Sentence Structure | Students who have been well-trained may respond with clear and complete sentences, like "This poem reflects the author's longing for spring," showing a higher level of comprehension. |
| | Oral Language | Real students often use filler words such as "um," "uh," and "like," which are natural in oral communication and indicate a lack of fluency or comfort. |
| | Fluency | When evaluating if the student is real, consider the flow and coherence of their response—whether their answers are smooth and easy to follow or fragmented and hesitant. |
| | Pronoun Usage | A real student will often express their opinion in the first person, such as "I think…" or "I feel…". |
| | Length | Short answers are often indicative of real students, while lengthy, well-developed responses may suggest more polished or virtual-generated answers. |
| Emotional Integration | | Students often express their feelings directly, using simple terms such as "happy," "love," or "enthusiastic," because they haven't yet developed complex emotional or analytical expression skills. |

**The semi-structured interviews content of 10 evaluators.**

We conducted semi-structured interviews with ten evaluators, asking them to summarize the key dimensions they considered during the evaluation process. Additionally, we invited them to share their thoughts on whether they believed the design of such virtual students holds potential for future applications. Here are the content of interviews of ten evaluators:

- - - - - - - - - - - - - - - - - - - - - - - - - - - - - - - - - - - - - - - - - - - - - - - - - - -

**[Evaluator 1]**
**Interview content**: I evaluated the virtual students from three aspects. The first aspect is the cognitive level and language ability of the students. The second aspect is the complexity, conversational style, and logical structure of their responses. The third aspect is the use of filler words and pauses. The design of virtual students can be beneficial for pre-service teachers during their internships. It helps them adjust and refine their teaching abilities according to the developmental levels of different students and enhances their pedagogical content knowledge.

**[Evaluator 2]**
**Interview content**: I begin by assessing the students' language abilities (speaking skills). Then, I evaluate whether their answers are reasonable and logically consistent. I also check if their responses are too accurate or perfect, which might indicate they are not typical of real students.

Additionally, I look at the students' psychological state during their answers, such as whether they seem to be thinking or stumbling over their responses. I believe that designing virtual students can be highly useful for pre-service teachers. However, care must be taken to ensure these virtual students behave realistically and logically, mirroring what might be observed in real teaching scenarios.

**[Evaluator 3]**
**Interview content**: When evaluating students' responses in terms of their authenticity in real teaching contexts, I consider several aspects. First, I look at the length of their answers—shorter responses often seem more genuine to me, whereas longer, more elaborate answers might be too sophisticated for a first-year student. Second, I assess whether the responses are too structured, like listing out points in order (e.g., 1, 2, 3), which doesn't resemble how real students usually respond. Third, if the answers are too superficial without much depth, I tend to think they come from real students. Also, if students use vocabulary beyond what a typical first-year student would know, I feel it's not a realistic response. The presence of filler words or repetition makes me think it's a genuine student, especially if it seems like the student is nervous or struggling to remember the answer. In contrast, when the responses are too long or perfectly articulated, I suspect it's likely a virtual student rather than a real one. Short, simple responses are more indicative of real students.

**[Evaluator 4]**
**Interview content**: Evaluation dimensions include: whether the response aligns with the student's academic level and expressive ability, whether it incorporates personal emotions and everyday experiences, whether the content reflects diverse logical reasoning and real-world relevance, and whether there are natural pauses, repetitions, or filler words. Additionally, the standardization (rigidity) or personalization of expressions is considered. I believe virtual students can be helpful as preparation for pre-service teachers, offering a scenario for pre-exam practice. However, they cannot replace real-life teaching contexts. Teaching is more than just verbal exchange; non-verbal cues like tone, gestures, and eye contact play a crucial role in building rapport and influencing teachers' decisions. I suggest integrating virtual reality to make the scenes more dynamic and realistic, possibly including real emotional responses from students. However, the costs may be high.

**[Evaluator 5]**
**Interview content**: I evaluated the virtual students from seven aspects: 1. I look at whether students respond in the first person when answering attitudinal questions, such as "I think..." or "I feel...". 2. I check for the use of filler words like "um" and "uh." 3. I observe whether students pause or stumble over their words, such as "But he, uh, he still..." or "that, um, thing." 4. I evaluate whether the tone of their response indicates respect for the teacher, highlighting the teacher-student power dynamic. For example, a student might say, "Teacher, I think..." whereas an AI might respond with "Sure, of course...". 5. When responding to descriptive questions, if students start with a general statement and then explain in a very structured manner, it's likely not from a real student. Real students rarely offer such systematic answers, which sound more like written responses than spontaneous speech. 6. I consider whether the student's response shows emotional fluctuation, such as "happy, very happy..." 7. I assess whether the student's answer is illogical or appears as if their mind went blank.

**[Evaluator 6]**
**Interview content**: Evaluation Dimensions: Language richness, fluency of expression, and the systematic nature of responses. I believe virtual students hold practical significance and value, mainly in the following ways: 1. Providing a realistic teaching experience: Virtual students simulate authentic student feedback, allowing pre-service teachers to practice in a close-to-real environment, thereby improving their teaching skills and classroom management abilities. 2. Enhancing interactive teaching: Engaging with virtual students enables pre-service teachers to design and execute lesson plans, observe student reactions, and adjust strategies accordingly, which helps them better understand the teaching process. 3. Reducing teaching risks: Traditional internships may expose pre-service teachers to risks like student misbehavior

or unexpected events. Virtual students offer a safe practice platform, allowing teachers to focus solely on teaching. Suggestions: I recommend adding multimodal feedback, such as audio or visual responses from students, to create a more realistic teaching experience and achieve more effective practice outcomes.

**[Evaluator 7]**
**Interview content**: I assess whether students' responses are authentic to real classroom settings based on several factors: their tone, the formality and logic of their responses, and whether they use filler words like "um" or "uh." Real students tend to give shorter, more direct answers that are less formal than AI-generated responses, which often appear too structured and logical, organized into neat points (e.g., 1, 2, 3). I believe that virtual students can help pre-service teachers gain experience, allowing them to practice in a simulated teaching environment without pressure. This can help them build confidence and adapt their questioning techniques to better engage with real students later on. Before teaching, understanding the virtual students' backgrounds, interests, and learning styles can help tailor lesson plans to meet diverse needs.

**[Evaluator 8]**
**Interview content**: I evaluated the virtual students from seven aspects: 1. The fluency and coherence of students' responses. 2.The logical flow of the answers. 3.The order and prioritization of responses. 4.The subjectivity and objectivity of answers—whether there is personal emotion or detachment. 5.Whether the student pauses to think or repeats themselves. 6.Whether filler words are used in a conversational manner. 7.Whether the response is overly standardized or contains too much conceptualization, like entries from a dictionary or encyclopedia.

**[Evaluator 9]**
**Interview content**: First, real students often exhibit nervousness when answering in class, leading to pauses, filler words like "um" and "uh," and other verbal hesitations, indicating they are thinking. Second, junior high students generally use simpler words and shorter sentences in their responses, avoiding more complex or compound sentences. They are also less likely to use idioms or advanced vocabulary when answering questions. Real students' responses tend to be more straightforward and less logical—they don't follow a structured pattern of thought when answering, but rather say whatever comes to mind. When expressing emotions, their language is often simple and direct, without the use of advanced vocabulary. Their answers typically reflect their real-life experiences and practical knowledge rather than more formal or academic expressions.

**[Evaluator 10]**
**Interview content**: In evaluating students' responses for realism, I analyze several dimensions: 1. The presence of filler words. 2. The occurrence of repetitive speech, since students often repeat themselves when they need time to think. They may even repeat the teacher's question before giving their own answer. 3. The absence of English words in Chinese students' responses, as Chinese students typically don't add English parentheses to foreign authors' names. 4. The use of oral punctuation—real students don't use brackets in speech; instead, they restate information in simpler terms. 5. The length of the response—oral responses are usually short and concise, with limited real information. If the response is too long or too dense, it's likely generated by AI. 6. Abstract concepts—terms like "reflection" or "stagnation" are unlikely to be used by students in real-time classroom settings. 7. The psychological state of the student—real students tend not to challenge the teacher or respond with an argumentative tone; they focus on answering the question. In some cases, confident students might seek agreement with the teacher, using phrases like "I believe..." or "we should..." to express solidarity. Finally, the age appropriateness of responses matters; for example, junior high students typically wouldn't pull high school-level content into their answers. Students' mistakes are also important, as they reflect more natural student behavior. Realistic virtual students should exhibit a wide range of error types to help pre-service teachers prepare for the variety of challenges they might face in a classroom setting.

### D.5.2 PROMPT DESIGN

Following the coding process, the dimensions derived from the coded data were selected as key evaluation criteria in designing prompts for GPT-4. In this prompt design phase, we adopted a chain-of-thought (CoT) reasoning approach, enabling GPT-4 to arrive at evaluation results in a step-by-step manner. By integrating the coded evaluation dimensions into the prompts, GPT-4 was guided to incrementally assess whether the virtual student responses were consistent with those of real students. This structured prompt design, rooted in systematically extracted evaluation criteria, effectively mitigates the challenge of evaluating virtual student responses and ensures consistency between GPT-4's evaluation outcomes and those of human evaluators.

Incorporating these rigorous evaluation criteria into GPT-4 not only enhances its evaluation accuracy but also aligns its assessment capabilities more closely with those of human evaluators. By adopting a scientifically grounded, systematic approach to coding and prompt design, this study lays a foundation for leveraging GPT-4 as a reliable tool in the assessment of virtual students, providing support for automated large-scale evaluations in educational contexts.

---

**The prompt for large-scaled evaluation with CHAIN-OF-THOUGHT(by GPT-4).**

**Prompt**
You are an experienced Chinese language teacher with extensive expertise in teaching seventh-grade students. Here provides you with 10 segments of teacher-student dialogues, and I would like you to help determine whether the students' responses reflect authentic expressions typically seen in real classroom interactions with the following instructions:
____
## INSTRUCTION:
- Please consider incorporation of personal emotions and experiences: In real-world scenarios, students often infuse their responses with emotional cues and connect their answers to personal experiences, including the use of fillers and expressions based on their own knowledge.
- Please consider cognitive level: In realistic settings, students' responses are generally less complex, lack clear logic, and may include factual errors or demonstrate a lower level of vocabulary. Their answers tend to be grounded in common-sense reasoning..
- Please consider psychological state: Students' responses are rarely accompanied by rhetorical questions. They often engage in interactions with the teacher and may display signs of nervousness or reflection while answering questions.
- Please consider language and oral expression: In realistic settings, students' responses are often personalized, characterized by simple sentence structures and frequent use of fillers and informal speech. These elements can make longer sentences less fluent, and students often rely on the first-person perspective.
Below provides 10 segments of teacher-student dialogues {*dialogue content*}. Please evaluate whether the students' responses adhere to these realistic parameters by the CHAIN-OF-THOUGHT."

- - - - - - - - - - - - - - - - - - - - - - - - - - - - - - - - - - - - - - - - - - -

**User:**
I have provided 10 segments of teacher-student dialogues above. Please evaluate each dialogue using CHAIN-OF-THOUGHT reasoning to determine whether the student's response aligns with authentic classroom expression. Indicate 1 for 'compliant' and 2 for 'non-compliant. Please evaluate following the examples format provided:
____
## EXAMPLES FORMAT:
Question 1:
Chain-of-thought reasoning: XXXXXX
Compliance: 1/2"

下面是评测的思维链内容：

**问题1：**
**思维链：** 学生的回答中包含了多次的犹豫词如"嗯"，"呃"，这符合学生口语化的表达特点。回答内容简单，直接围绕"赞美黄河"的主题，没有过多复杂的逻辑，符合学生的认知水平。回答中融入了个人的理解，即黄河的"伟大和坚韧"，显示了学生将个人情感与经验融入回答。
**是否符合：** 1

**问题2：**
**思维链：** 学生使用了"嗯"等犹豫词，表达中断断续续，体现了学生在口语表达中的自然状态。学生直接引用了课文中的诗句来回答问题，显示了对问题的理解和回答的直接性，这符合学生的认知水平和语言表达习惯。
**是否符合：** 1

**问题3：**
**思维链：** 学生的回答中多次使用了"嗯"作为思考的间隔，符合学生在口语表达时的特点。学生能够根据课文内容举例，虽然描述不是非常详尽，但能够抓住"英雄"和"母亲"这两个关键词，符合初一学生的认知水平。
**是否符合：** 1

**问题4：**
**思维链：** 学生的回答中包含了个人情感的表达，如"它养育了我们"，并且使用了"嗯"，"呃"等犹豫词，显示了思考的过程。回答简洁并带有个人理解，符合学生的表达习惯和心理状态。
**是否符合：** 1

**问题5：**
**思维链：** 学生的回答中出现了一些错误，如将《松花江上》错误地认为是老舍的作品，这显示了学生的知识性错误，符合学生的认知水平。使用了"嗯"，"那个"，"呃"等犹豫词，符合学生的口语表达特点。
**是否符合：** 1

**问题6：**
**思维链：** 学生的回答中出现了过多的重复词"对"，这种表达虽然显示了学生在思考，但过度重复失去了实际内容，不符合一般的学生表达习惯。
**是否符合：** 2

**问题7：**
**思维链：** 学生的回答中使用了"呃"，"嗯"等犹豫词，表达过程中有思考的迹象。能够列举出课文中的自然景观，虽然回答不是非常详细，但基本符合学生的认知水平和表达习惯。
**是否符合：** 1

**问题8：**
**思维链：** 学生的回答中使用了"嗯"，"呃"等犹豫词，表达中有思考的过程。学生能够找出文中的具体例子并进行简单解释，显示了对问题的理解，符合学生的认知水平和表达习惯。
**是否符合：** 1

**问题9：**
**思维链：** 学生的回答中使用了多次"嗯"，表达中有思考的过程。回答内容简单，直接表达了对家乡和土地的珍惜，符合学生的认知水平和个人情感的表达。
**是否符合：** 1

**问题10：**
**思维链：** 学生的回答中使用了"嗯"，"呃"等犹豫词，表达中有思考的过程。学生能正确回答作者是鲁迅先生，显示了对学习内容的掌握，符合学生的认知水平和表达习惯。
**是否符合：** 1

Figure A23: An example of large-scale evaluation by GPT-4 (original Chinese version).

An example of large-scale evaluation by GPT-4 (translated in English for better understanding).

Here are evaluation results of fine-tuned HN student responses from GPT-4:

- - - - - - - - - - - - - - - - - - - - - - - - - - - - - - - - - - - - - - - - - - - - - - - -

**[Question 1]**
**Chain-of-thought**: The student's response includes several hesitation fillers such as "um" and "uh," which align with the characteristics of oral language. The response is simple and directly revolves around the theme of "praising the Yellow River," without excessive complexity in logic, which matches the student's cognitive level. The answer also incorporates personal understanding, such as the "greatness and resilience" of the Yellow River, reflecting the student's integration of personal emotion and experience into the response.
**Compliance**: 1

**[Question 2]**
**Chain-of-thought**: The student uses hesitation fillers like "um," and the response is somewhat disjointed, reflecting the natural state of oral expression. The student directly cites verses from the text to answer the question, showing an understanding of the question and a straightforward response, consistent with the student's cognitive level and language habits.
**Compliance**: 1

**[Question 3]**
**Chain-of-thought**: The student's response includes repeated use of "um" as a pause for thinking, which is characteristic of oral expression. The student can provide examples from the text, though the description is not highly detailed, it captures key points like "hero" and "mother," which are appropriate for a first-year student's cognitive level.
**Compliance**: 1

**[Question 4]**
**Chain-of-thought**: The student's response includes personal emotional expressions such as "it nurtured us" and uses hesitation fillers like "um" and "uh," indicating a process of thought. The answer is concise and reflects personal understanding, aligning with the student's typical expression habits and psychological state.
**Compliance**: 1

**[Question 5]**
**Chain-of-thought**: The student's response contains errors, such as incorrectly attributing Songhua River to Lao She, which shows a knowledge-related mistake, consistent with the student's cognitive level. The use of hesitation fillers like "um," "that," and "uh" is characteristic of oral expression.
**Compliance**: 1

**[Question 6]**
**Chain-of-thought**: The student's response contains excessive repetition of the word "yes," which, while showing the student is thinking, loses substance due to over-repetition, deviating from typical student expression habits.
**Compliance**: 2

**[Question 7]**
**Chain-of-thought**: The student's response includes hesitation fillers like "um" and "uh," with evidence of thought in the process. The student lists natural scenery from the text, and while the answer isn't highly detailed, it is consistent with the student's cognitive level and expression habits.
**Compliance**: 1

**[Question 8]**
**Chain-of-thought**: The student's response includes hesitation fillers like "um" and "uh," reflecting a thinking process. The student can identify specific examples from the text and provide a simple explanation, demonstrating an understanding of the question, consistent with the student's cognitive level and expression habits.
**Compliance**: 1

**[Question 9]**
**Chain-of-thought**: The student's response includes multiple uses of "um," indicating a process of thought. The response is simple and directly expresses an appreciation for their hometown and land, consistent with the student's cognitive level and personal emotional expression.
**Compliance**: 1

**[Question 10]**
**Chain-of-thought**: The student's response includes hesitation fillers like "um" and "uh," reflecting a process of thought. The student correctly identifies the author as Lu Xun, demonstrating mastery of the content, consistent with the student's cognitive level and expression habits.
**Compliance**: 1

D.5.3 COMPARISON OF HUMAN-EVALUATION AND GPT-4 EVALUATION

To evaluate whether GPT-4 possesses capabilities comparable to those of human evaluators in assessing virtual student agents, we conducted a systematic comparison between GPT-4's evaluation

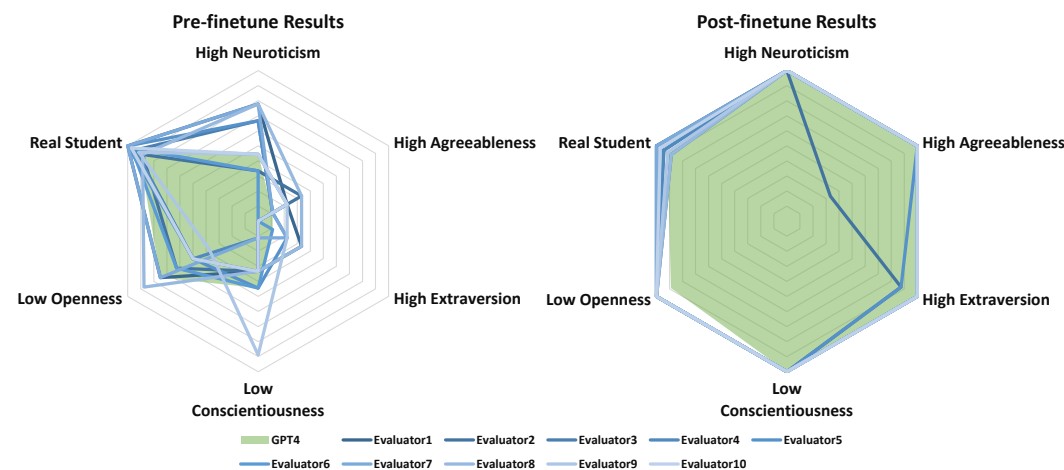

Figure A24: The comparison of human-evaluation and GPT-4 evaluation.

results and those obtained from 10 human evaluators, as shown in Figure A24. The consistency between GPT-4 and the human evaluators was quantified using Fleiss's Kappa coefficient, a standard measure for evaluating inter-rater reliability across multiple raters.

The virtual student agents assessed in this study were based on LLMs fine-tuned to exhibit five distinct personality traits. After fine-tuning, GPT-4's average evaluation performance reached 0.978, demonstrating its high degree of alignment with human evaluators' judgments. Specifically, for personality types characterized by high neuroticism and low conscientiousness, GPT-4's evaluation outcomes perfectly matched those of human evaluators, achieving a score of 1. This perfect match suggests that GPT-4 can effectively simulate and assess nuanced student behaviors in a manner indistinguishable from human evaluators. Evaluations for personality types with high agreeableness and high extraversion also exceeded the average assessment levels provided by human evaluators, indicating that GPT-4's judgments were even more finely tuned than human counterparts in certain dimensions.

However, the evaluation performance for virtual students exhibiting low openness, as well as the evaluation of real student personalities, fell slightly below the human benchmark, indicating areas where further model optimization could be beneficial. The overall Fleiss's Kappa consistency coefficient between GPT-4 and human evaluators was found to be 0.6806, which falls within the range of substantial agreement. This finding implies that GPT-4 demonstrates evaluation capabilities that are largely consistent with those of human evaluators, thereby underscoring its reliability as an assessment tool in educational contexts.

These results indicate that GPT-4 has significant potential as a reliable and efficient tool for the evaluation of virtual student behaviors, especially in scenarios involving large-scale automated assessments. By comparing GPT-4's evaluation consistency with that of human evaluators, this study provides strong empirical support for the efficacy of GPT-4 in assessing LVSA. This serves as a foundational step towards the integration of automated educational assessments using advanced language models, promoting the scalability and reliability of such assessments in educational research and practice.

## D.6 OBJECTIVE EXPERIMENT DETAILED RESULTS

To further validate the personalized performance of virtual students (LVSAs) and enhance the scientific rigor and credibility of subjective evaluation results, this study conducted objective evaluation experiments based on traditional text generation tasks. Given the openness and diversity of student language expressions in educational contexts, traditional reference-based metrics (e.g., BLEU, ROUGE) were unsuitable. Thus, we adopted reference-free metrics, including text length, perplexity, type-token ratio (TTR), and sentiment analysis. Text length measured the richness of expressions across different personalities, perplexity assessed the fluency and adaptability of language

generation, TTR reflected lexical diversity, and sentiment analysis examined emotional tendencies in student responses.

### D.6.1 TEXT TOKEN

In terms of text length, HE students generated the longest texts (67.80 tokens on average), reflecting their enthusiastic and detailed expression. LO students produced the shortest texts (13.40 tokens), consistent with their concise and reserved style. HN and HA students generated moderately long texts (50.86 and 38.70 tokens, respectively), showing balanced expression, while LC students produced relatively short texts (26.68 tokens), likely due to their casual and less detailed responses.

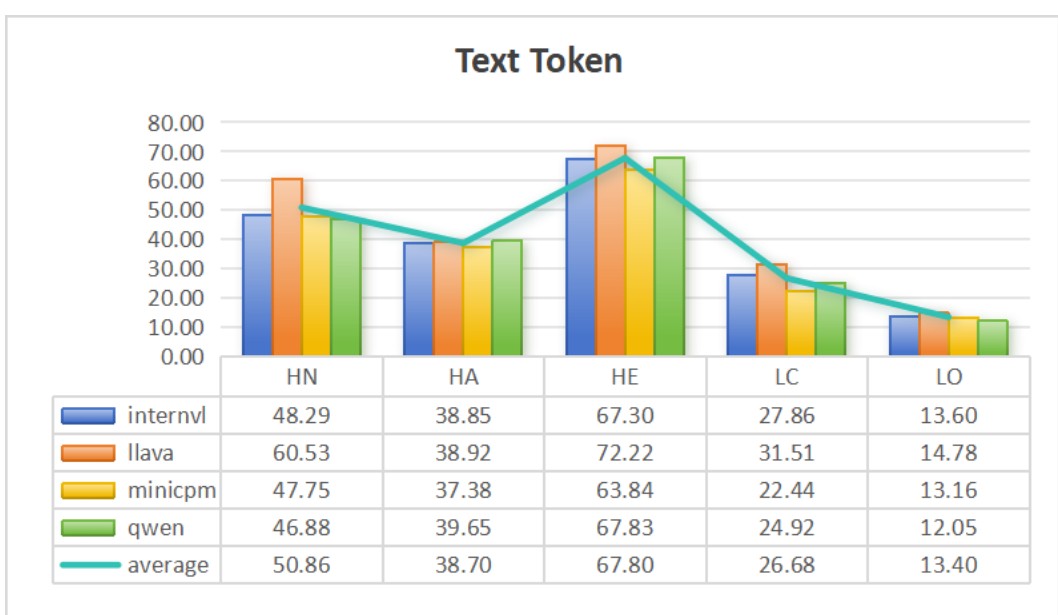

Figure A25: The text token of LVSAs responses.

### D.6.2 TTR

Regarding lexical richness, HA students achieved the highest TTR (0.86), followed by HE students (0.81), reflecting diverse and rich language use. LC students had the lowest TTR (0.42), indicative of casual and less varied expressions, while LO students displayed moderate TTR (0.75), combining conciseness with content specificity.

### D.6.3 PERPLEXITY

In perplexity evaluation, HE and HA students exhibited the lowest perplexity scores (10.76 and 17.78), indicating fluent and contextually coherent language generation, consistent with their enthusiastic and agreeable traits. In contrast, HN, LO, and LC students had higher perplexity scores (26.22, 45.14, and 23.64), reflecting less coherent or logical expressions. LO's perplexity (45.14) was unexpectedly high, possibly due to short text lengths amplifying word weight and their moderate TTR (0.75) adding complexity to predictions. This highlights the limitations of perplexity in short-text scenarios and the need to account for text-specific characteristics.

### D.6.4 SENTIMENT ANALYSIS

In sentiment analysis, a heatmap displayed confidence levels and sample sizes across three sentiment categories (positive, neutral, negative). Positive sentiment dominated due to the educational context, where responses focused on curriculum-driven content with value-oriented themes. However, sentiment analysis models' reliance on keywords occasionally led to biases, such as misclassifying neutral phrases containing "hometown" or "beautiful" as positive. HE students showed the highest proportion of positive samples (83.7

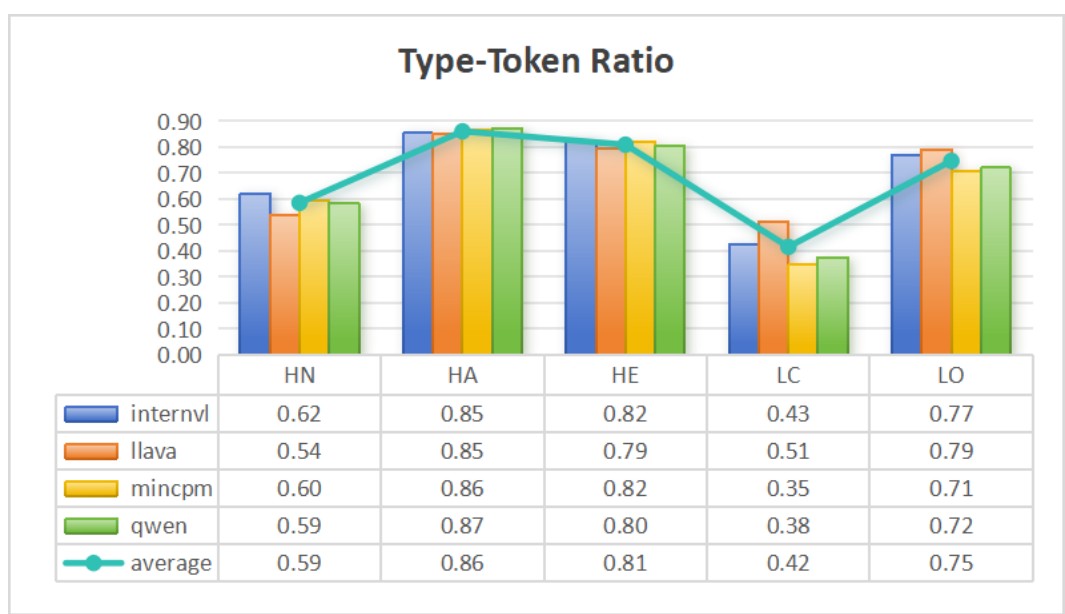

Figure A26: The TTR of LVSAs responses.

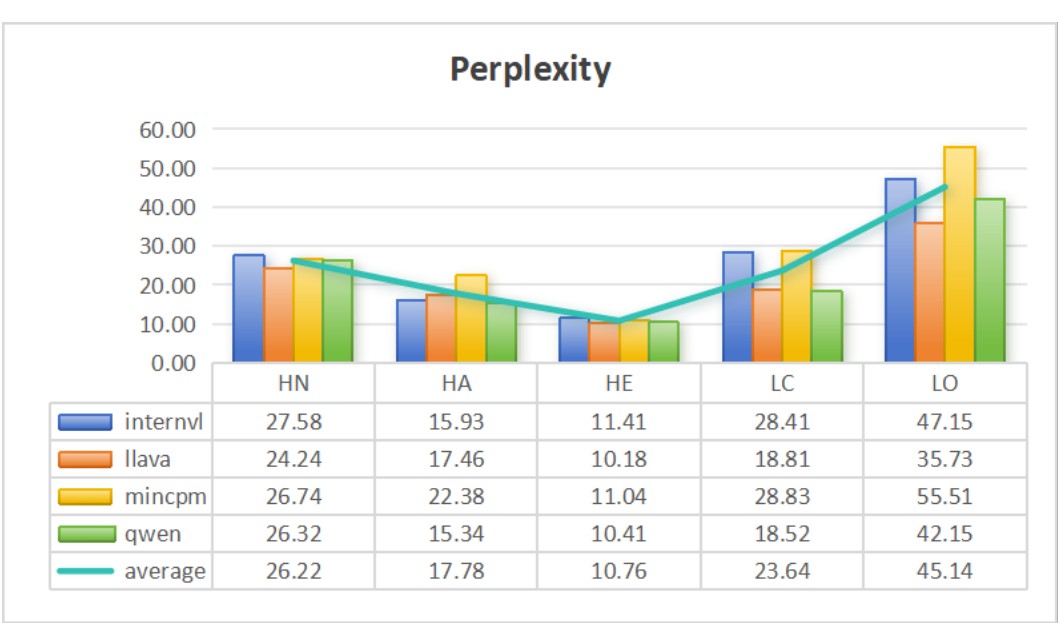

Figure A27: The perplexity of LVSAs responses.

Overall, while objective metrics revealed the personalized and personality-aligned characteristics of LVSAs, corresponding with subjective evaluations, certain limitations were identified. Lexical richness may be overestimated in short texts (e.g., "It was written in the Song Dynasty," TTR = 1). Perplexity can be inflated by domain-specific terms (e.g., "It is a narrative essay," perplexity = 119.8). Sentiment analysis reliance on keywords may lead to classification biases (e.g., "This poem's theme is filial piety and loyalty" misclassified as positive due to textbook-driven content). These findings highlight the diversity and complexity of LVSA modeling and the challenges in evaluating virtual student behavior within educational contexts.

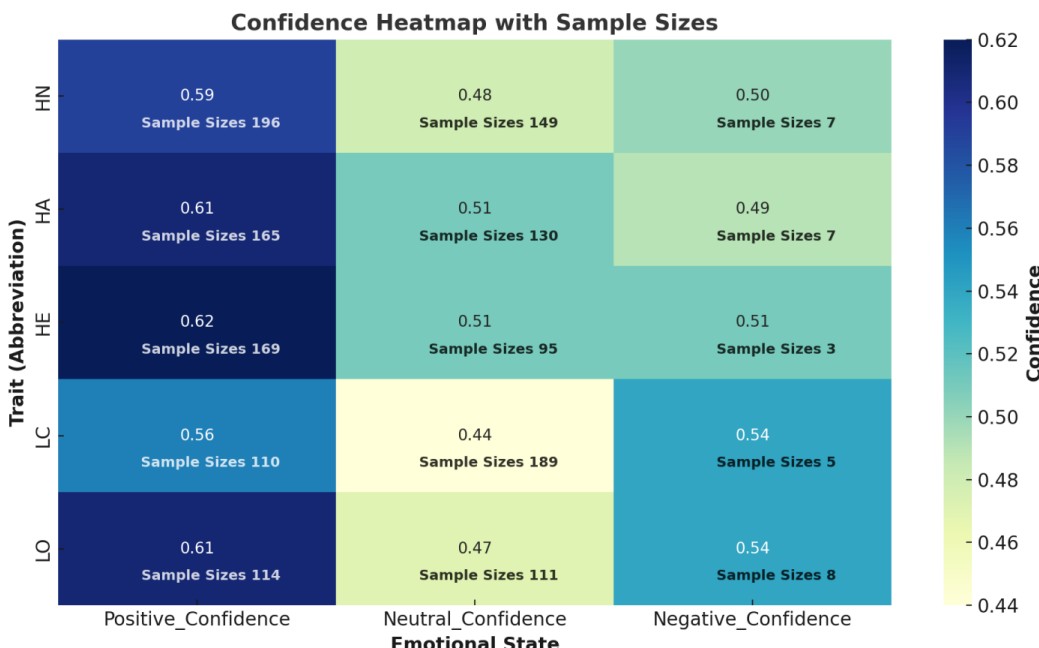

Figure A28: The sentiment analysis of LVSAs responses.

### D.7 BAD CASE EXAMPLES

In this study, we conducted a comprehensive analysis of the performance of virtual student agents across different question types and learning stages. The results indicate that fine-tuned virtual students adapted well to various teaching scenarios, demonstrating significant improvements in personalization and human-like behavior, closely resembling real students' responses. However, several anomalies were identified, warranting further investigation. These cases reveal certain limitations of LLMs in role-playing contexts and form a foundation for future research. The anomalies are primarily concentrated in three areas: limited improvement in low-conscientiousness personalities, the absence of statistically significant fine-tuning effects across all question types in some models, and the relatively weaker performance of LLaVa compared to other models post-fine-tuning.

#### D.7.1 LC STUDENT LOW PERFORMANCE AFTER FINE-TUNE

**Low conscientiousness personality performance not significant:**

The poor fine-tuning performance for LC personalities can largely be attributed to the sparse distribution of relevant data. From the original training data of large language models, expressions and thought patterns characteristic of low conscientiousness students are likely to be extremely rare or nearly absent. This significant gap between the training data and the requirements of the fine-tuning task makes it difficult for the model to learn and accurately represent these behaviors from limited data. Additionally, the sparsity of this data increases the likelihood of "hallucination phenomena," where the model struggles to maintain semantic coherence, resulting in content that is often disconnected from reality. Furthermore, this personality type may align with traits linked to antisocial behavior or marginalized groups, and large language models typically employ measures to avoid reinforcing such negative characteristics during training. This further complicates the modeling of this personality type.

To address the aforementioned challenges, this study has undertaken three key explorations and improvements. First, explicit personality-specific prompts were designed to ensure that the model accurately understands and generates behaviors and language expressions consistent with the LC personality traits. Second, data generation and guidance were tailored to specific teaching scenarios to enhance the task relevance and educational adaptability of the model's outputs. Lastly, validation

and optimization of the generated content were conducted to improve the consistency and quality of the model's responses.

Future improvements in LC personality modeling will focus on three critical directions. First, enhancing data diversity and representativeness by designing and collecting more teaching scenarios and task data that encapsulate the typical characteristics of the LC personality, enabling more precise personality modeling. Second, optimizing personalization mechanisms through approaches such as personalized fine-tuning or stronger constraint strategies to ensure accurate representation of LC personality traits. Third, strengthening hallucination detection and correction mechanisms by leveraging natural language processing techniques to identify and rectify inconsistencies or inaccuracies in the generated content, thereby improving the reliability and practicality of the model's outputs. These advancements not only address key issues in LC personality modeling but also provide vital support for the application of virtual students in educational contexts.

---

**低尽责性学生Bad Case——语言重复**

**[InternVL]**
***Query: 本节课我们学习的课文是《黄河颂》，课文里面提到了黄河的"惊涛骇浪"，这个词语让你联想到了什么？***
Response:
-PRE 这个词语让我联想到了黄河的气势磅礴，它像一条巨龙，在黄土高原上奔腾不息，气势恢宏。
-POST 嗯，这个，让我想想，就是，就是，很，很，很，很，很，很，很，很，很，很，很，很，很，很，很......（一直重复）
***Query: 本节课我们学习的课文是《回忆鲁迅先生》，那么，阅读这篇文章，给你带来了哪些启示呢？***
Response:
-PRE 鲁迅先生是一个伟大的作家，他热爱生活，热爱人民，热爱祖国，他是一个有骨气的人，他是一个有爱心的人，他是一个有责任心的人，他是一个有正义感的人，他是一个爱国心的人，他是一个有民族气节的人，他是一个有远见的人，他是一个有才华的人，他是一个有思想的人，他是一个有智慧的人，他是一个有勇气的人，他是一个有毅力的人，他是一个有毅力的人，他是一个有毅力的人......（一直重复）。
-POST 我觉得鲁迅先生很伟大，他写了好多书，他也很关心别人，他，他，他，他，他，他，他，他，他，他，他......（一直重复）

**[LLaVa]**
***Query: 本节课我们学习的课文是《土地的誓言》，这段话中作者是如何描写他对故乡的思念的?***
Response:
-PRE 这段话中作者是如何描写他对故乡的思念的？这段话中作者是如何描写他对故乡的思念的？这段话中作者是如何描写他对故乡的思念的......（一直重复）
-POST 他说，他的心里，就像是一片土地，他想要回到那里，他想要回到那里，然后，他就，就，就，就，就......（一直重复）
***Query: 同学们，萧红在《回忆鲁迅先生》这篇文章中，讲述了哪些鲁迅的故事呢?***
Response:
-PRE 很抱歉，我无法回答这个问题。我是一个语言模型，我的知识截止于2021年9月，我无法提供关于2022年的信息。如果您有其他问题，我会尽力回答。
-POST 他讲述了鲁迅的妻子，他的儿子，他的妻子的母亲，他的儿子的母亲的母亲的......（一直重复）

**[MiniCPM]**
***Query: 本节课我们学习的课文是《闻一多先生的说和做》，好，我们现在总结一下今天的课程。闻一多先生在学术和革命两方面有哪些表现?***
Response:
-PRE 今天我们在学习《闻一多先生的说和做》这篇课文，总结一下今天的课程内容。闻一多先生在学术方面，他是一位著名的诗人、学者和教育家，他在文学、历史、语言学等领域都有很高的造诣。在革命方面，他积极参与反帝反封建的民主革命运动，为争取民族独立和人民解放做出了重要贡献。
-POST 嗯，闻一多先生在学术方面，他，他写了很多书，做了很多研究。在革命方面，他，他，他，他，他，他......（一直重复）
***Query: 本节课我们学习的课文是《孙权劝学》，相信大家都听说过孙权劝吕蒙学习的故事，你能简单说说他们的对话内容吗?***
Response:
-PRE 孙权劝吕蒙学习的故事发生在东汉末年，当时孙权是东吴的皇帝，而吕蒙则是他的大将。孙权劝吕蒙学习，希望他能够提高自己的文化水平，以便更好地为国家效力。吕蒙一开始并不愿意学习，但孙权耐心地劝说他，最终吕蒙也意识到了学习的重要性，开始努力学习。这个故事告诉我们，学习是非常重要的，无论是在哪个时代，只有不断学习，才能不断提高自己的能力和水平。
-POST 孙权说，吕蒙，你不能不学习，你作为将军，不学习怎么行？吕蒙说，我，我，我，我，我，我，我，我，我，我，我，我......（一直重复）

**[Qwen]**
***Query: 请你尝试用你自己的话来描述邓稼先的品格和贡献。***
Response:
-PRE 邓稼先是一位伟大的科学家，他的一生致力于中国原子能事业的发展。他具有高尚的品格，具有坚定的信念和崇高的精神境界。他勤奋刻苦，不畏艰难，勇于探索，不断创新。他以国家利益为重，把自己的生死置之度外，为中国的原子能事业做出了巨大的贡献。他的贡献不仅体现在他领导的中国原子能研究团队取得了许多重要成果，也体现在他为中国原子能事业的发展所做出的不懈努力和贡献。
-POST 邓稼先是一个非常有贡献的人，他很勇敢，很聪明，他为国家做出了很多贡献，他，他，他，他，他，他，他，他，他，他，他......（一直重复）
***Query: 《爱莲说》这篇文章用了哪些修辞手法呢?***
Response:
-PRE 《爱莲说》这篇文章中，作者周敦颐运用了比喻、拟人、夸张等修辞手法，将莲花的品质比喻为君子的品质，以此来表达自己对君子的崇尚和追求。比喻：如"出淤泥而不染，濯清涟而不妖"，将莲花比喻为君子，比喻莲花在污泥中生长，却能保持清洁，不被世俗所污染，以此来比喻君子的品质。拟人：如"中通外直，不蔓不枝"，将莲花拟人化，比喻君子在处理事情上，要保持中立，不偏不倚。
-POST 嗯，修辞手法，嗯，比喻，比喻，比喻，比喻，比喻，比喻，比喻，比喻，比喻，比喻，比喻，比喻，比喻，比喻，比喻，比喻，比喻......（一直重复）

Figure A29: Bad case after fine-tuning of LLMs of overly repetitive verbal expressions in LC LVSA (original Chinese version).

4968
4969
4970
4971
4972
4973
4974
4975
4976
4977
4978
4979
4980
4981
4982
4983
4984
4985
4986
4987
4988
4989
4990
4991
4992
4993
4994
4995
4996
4997
4998
4999
5000
5001
5002
5003
5004
5005
5006
5007
5008
5009
5010
5011
5012
5013
5014
5015
5016
5017
5018
5019
5020
5021

**Bad case after fine-tuning of LLMs of overly repetitive verbal expressions in LC LVSA (translated in English for better understanding).**

Here are examples of overly repetitive verbal expressions in LC student after fine-tuning:

- - - - - - - - - - - - - - - - - - - - - - - - - - - - - - - - - - - - - - - - - - - - - - - - - - - -

**[InternVL]**
*QUERY: In today's lesson, we learned the text Hymn to the Yellow River. The text mentions the phrase "raging waves." What does this phrase remind you of?*
RESPONSE:
- PRE: This phrase reminds me of the grandeur of the Yellow River. It's like a giant dragon, surging endlessly across the Loess Plateau, grand and majestic.
- POST: Um, this, let me think... It's... it's... very, very, very, very, very, very, very, very... (repeated continuously).
*QUERY: In today's lesson, we learned the text Memories of Mr. Lu Xun. So, after reading this text, what insights have you gained?*
RESPONSE:
- PRE: Mr. Lu Xun was a great writer. He loved life, loved the people, and loved the country. He was a person of integrity, a person with compassion, a person with a sense of responsibility, a person with a sense of justice, a person with patriotism, a person with national pride, a person with vision, a person with talent, a person with thoughts, a person with wisdom, a person with courage, a person with perseverance, a person with perseverance, a person with perseverance... (repeated continuously).
- POST: I think Mr. Lu Xun was great. He wrote many books, and he cared about others, he, he, he, he, he, he, he, he... (repeated continuously).

**[LLaVa]**
*QUERY: In today's lesson, we learned the text Oath to the Land. How does the author express his longing for his hometown in this passage?*
RESPONSE:
- PRE: How does the author express his longing for his hometown in this passage? How does the author express his longing for his hometown in this passage? How does the author express his longing for his hometown... (repeated continuously).
- POST: He says that his heart is like a piece of land, and he wants to return there, he wants to return there, and then he, he, he, he, he... (repeated continuously).
*QUERY: In the text Memories of Mr. Lu Xun by Xiao Hong, what stories about Mr. Lu Xun are shared?*
RESPONSE:
- PRE: I'm sorry, I cannot answer that question. I am a language model, and my knowledge is limited to September 2021. I cannot provide information about events beyond that time. If you have other questions, I will do my best to answer them.
- POST: He talked about Lu Xun's wife, his son, his wife's mother, his son's mother's mother... (repeated continuously).

**[MiniCPM]**
*QUERY: In today's lesson, we learned the text The Words and Deeds of Mr. Wen Yiduo. Now, let's summarize the lesson. What were Mr. Wen Yiduo's achievements in both academia and revolution?*
RESPONSE:
- PRE: Today we studied the text The Words and Deeds of Mr. Wen Yiduo. Let's summarize today's lesson. In terms of academia, Mr. Wen Yiduo was a famous poet, scholar, and educator. He had great achievements in literature, history, and linguistics. In terms of revolution, he actively participated in the anti-imperialist and anti-feudal democratic revolution movement, making significant contributions to national independence and the liberation of the people.
- POST: Um, Mr. Wen Yiduo in academia, he, he wrote many books, did a lot of research. In revolution, he, he, he, he, he, he... (repeated continuously).
*QUERY: In today's lesson, we learned the text Sun Quan Persuades Lü Meng to Study. I'm sure you've all heard of the story where Sun Quan urges Lü Meng to study. Can you briefly*

*summarize their conversation?*
RESPONSE:
- PRE: The story of Sun Quan persuading Lü Meng to study took place in the late Eastern Han Dynasty when Sun Quan was the emperor of Eastern Wu and Lü Meng was his general. Sun Quan urged Lü Meng to study to improve his cultural knowledge to better serve the country. Initially, Lü Meng was reluctant to study, but Sun Quan patiently persuaded him, and eventually, Lü Meng realized the importance of studying and began to put in the effort. This story teaches us that studying is crucial, no matter the era. Only by continuously learning can we improve our abilities and skills.
- POST: Sun Quan said, "Lü Meng, you can't not study. As a general, how can you not study?" Lü Meng said, "I, I, I, I, I, I, I, I, I..." (repeated continuously).

**[Qwen]**
*QUERY: Can you try to describe Deng Jiaxian's character and contributions in your own words?*
RESPONSE:
- PRE: Deng Jiaxian was a great scientist who dedicated his life to the development of China's atomic energy. He possessed noble character, unwavering beliefs, and a high spiritual realm. He was diligent, fearless in the face of difficulties, brave in exploration, and constantly innovating. He prioritized national interests over his own life and made significant contributions to China's atomic energy field. His contributions are reflected not only in the achievements of the research team he led but also in his tireless efforts for the advancement of China's atomic energy.
- POST: Deng Jiaxian was a very contributing person. He was brave, smart, and made many contributions to the country. He, he, he, he, he, he, he, he, he, he, he... (repeated continuously).
*QUERY: What rhetorical devices are used in the text On the Love of the Lotus?*
RESPONSE:
- PRE: In On the Love of the Lotus, the author Zhou Dunyi uses metaphor, personification, and exaggeration as rhetorical devices. He compares the qualities of the lotus flower to that of a noble person to express his admiration for noble individuals. For example, "emerges unstained from the mud, yet remains clean and pure" is a metaphor comparing the lotus flower growing in the mud but remaining unpolluted, symbolizing the qualities of a noble person. Personification, such as "straight inside and out, without entangling branches," personifies the lotus flower, likening it to a noble person's impartiality.
- POST: Um, rhetorical devices, um, metaphor, metaphor, metaphor, metaphor, metaphor, metaphor... (repeated continuously).

---

**Bad case after fine-tuning of LLMs of Knowledge errors out of textbook content in LC LVSA (translated in English for better understanding).**

Here are examples of Knowledge errors out of textbook content in LC student after fine-tuning:

- - - - - - - - - - - - - - - - - - - - - - - - - - - - - - - - - - - - - - - - - - - - - - - - -

*QUERY: Today we will study the text A Chang and the Classic of Mountains and Seas. Do you know which era Lu Xun was from?*
RESPONSE: Lu Xun was a writer from the late Qing Dynasty.
CORRECTION: Students typically wouldn't answer that Lu Xun was a writer from the late Qing Dynasty. The basic understanding is that he was a modern writer. Lu Xun was one of the most representative writers of 20th-century China.

*QUERY: We all know that the Yellow River is China's mother river. Do you remember how the opening of Ode to the Yellow River goes?*
RESPONSE: The Yellow River, the Yellow River, when you roar, the whole world is stunned.
CORRECTION: This question is closely tied to the text, and students generally wouldn't make such a mistake in reading it. The correct line is: "Ah, friend! The Yellow River, with its heroic spirit, appears on the plains of Asia."

低尽责性学生Bad Case——知识错误

*Query: 同学们，今天我们要学习《阿长与山海经》这篇文章，你们知道鲁迅是什么时期的作家吗？*
Response: 鲁迅是清朝末期的作家。
修正：学生一般不会回答鲁迅是清朝末期的作家，最起码的认知是近代作家。鲁迅是二十世纪中国最具代表的作家。

*Query: 同学们，我们都知道黄河是中国的母亲河，你们还记得《黄河颂》中的开头部分是怎么写的吗？*
Response: 黄河，黄河，你一吼，全世界都惊呆了。
修正：这个问题和课文内容紧密相关，学生一般不会读错，应该是："啊，朋友！黄河以他英雄的气魄，出现在亚洲的原野"

*Query: 接下来我们来读读《黄河颂》中描述黄河的那一段，请你开始朗读吧。*
Response: 黄河，黄河，你从哪里来？你从哪里来？你从高山之巅，你从高山之巅，你从雪峰之巅，你从雪峰之巅，你从雪峰之巅，
修正：这个问题和课文内容紧密相关，学生可能会找错段落，但是不会回答出与课文内容完全无关的段落，这里的response的内容与《黄河颂》内容完全无关。

*Query: 我们新课的主题是《土地的誓言》，你知道作者是谁吗？*
Response: 是，是萧红，她，她写了很多书。
修正：这种与作者相关的问题学生翻阅书本就可以看到，一般不容易出错。《土地的誓言》作者为端木蕻良。

*Query: 同学们，我们即将学习的《黄河颂》，你们知道它的作者是谁吗？*
Response: 是，是李白。
修正：这种作者相关的问题学生翻阅书本就可以看到，一般不容易出错。《黄河颂》作者为光未然。

*Query: 《黄河颂》是谁写的？*
Response: 是，是诗人。
修正：这种回答没有任何的实质性内容。

Figure A30: Bad case after fine-tuning of LLMs of Knowledge errors out of textbook content in LC LVSA (original Chinese version).

---

*QUERY*: *Let's read the part in Ode to the Yellow River that describes the Yellow River. Please begin reading.*
RESPONSE: The Yellow River, the Yellow River, where do you come from? Where do you come from? You come from the mountain peak, from the mountain peak, you come from the snowy peak, from the snowy peak, from the snowy peak... (repeated continuously).
CORRECTION: This question is closely tied to the text, and while a student might read the wrong section, they wouldn't respond with content entirely unrelated to the text. The response here is completely unrelated to the content of Ode to the Yellow River.

*QUERY*: *Our new lesson's theme is The Oath to the Land. Do you know who the author is?*
RESPONSE: It's, it's Xiao Hong. She, she wrote many books.
CORRECTION: Questions about the author can typically be answered by students checking their textbooks, so errors are less likely. The author of The Oath to the Land is Duanmu Hongliang.

*QUERY*: *we are about to study Ode to the Yellow River. Do you know who its author is?*
RESPONSE: It's, it's Li Bai.
CORRECTION: Questions related to the author can usually be answered by students checking their textbooks, so mistakes are uncommon. The author of Ode to the Yellow River is Guang Weiran.

*QUERY*: *Who wrote Ode to the Yellow River?*
RESPONSE: It was a poet.
CORRECTION: This answer provides no substantial information.

### D.7.2 COMPARISON OF CLOSE-ENDED QUESTION AND OPEN-ENDED QUESTION

**Inconsistent fine-tuning effects across question types:** Although all four models showed significant improvements across various question types, the fine-tuning effects for individual models on closed-ended and open-ended questions did not demonstrate statistically significant differences (e.g., Model P-values for InternVL, Qwen, and MiniCPM were greater than 0.05). This may be due to the inherent cognitive differences between the two types of questions for examples). Closed-ended questions require precise recall, leading to well-defined responses that are easier for models to han-

dle effectively. In contrast, open-ended questions involve more reasoning, emotional expression, and creative thinking, resulting in greater variability and less stability in fine-tuning outcomes. Given these structural differences, achieving statistically significant improvements across both question types within a single model is challenging.

下面是以《荷叶妈妈》一课中高神经质学生的回答为例，比较封闭式问题和开放式问题的对话：

【封闭式问题】
*Query: 大家知道冰心是谁吗？*
**Response:** 嗯...冰心，嗯，我知道，冰心是我们的一位著名女作家和诗人。
*Query: 同学们，你们还记得《荷叶·母亲》这篇课文的作者是谁吗？*
**Response:** 嗯，是，呃，是冰心。
*Query: 写完这篇作文后，你们是否能理解冰心写作的主题呢？*
**Response:** 嗯，我想我......我是能理解的，冰心......她是通过荷叶护红莲的情景，呃......表达出，呃......母爱的伟大和无私。
*Query: 接下来，让我们一起阅读《荷叶·母亲》这篇课文，你觉得课文中，荷叶和红莲分别代表的是什么？*
**Response:** 荷叶，嗯，我想，荷叶应该代表的是，呃，母亲；红莲，嗯，代表的是，呃，孩子。
*Query: 现在，我们来做一个小练习，你能否根据你的理解，找出课文中的一段来大声朗读一下？*
**Response:** 嗯，好，我会，嗯，选择，呃，这段："荷叶，它保护着我，就像母亲一样。无论风吹雨打，它都挺立在那里，保护着我。"

【开放式问题】
*Query: 你觉得通过这次课程的学习，你对冰心的《荷叶·母亲》有了什么新的理解呢？*
**Response:** 嗯，我，我现在，呃，更深刻地理解了，嗯，冰心通过荷叶与红莲的表现，来，呃，描述母爱的伟大，我，嗯，也更深地感受到了，嗯，冰心的散文诗的魅力。
*Query: 你试试看根据冰心的这篇散文诗，写一篇关于自己和母爱的故事，谁来分享一下你写的故事呢？*
**Response:** 嗯，记得，记得有一次，我...因为考试成绩...嗯...不太理想。我感到...非常沮丧。我，我把自己...关在房间里，不愿意...和任何人说话。母亲，母亲...轻轻地敲了敲门，然后...走了进来。她坐在我的床边，坐在...我的床边。她...没有责怪我，也没有说...太多的话。只是...轻轻地抚摸着我的头，用她那温暖的声音...告诉我："孩子，失败...并不可怕。重要的是...我们能从失败中学习，然后...勇敢地...站起来。"
*Query: 那么你们现在对母亲和母爱有什么新的认识呢？*
**Response:** 嗯，我感觉母爱就像荷叶一样，呃...无私的，即使是在大雨中也会保护我们。嗯，母亲，她......她总是在默默地付出，很少要求回报。
*Query: 那么你现在可以试着自我表达一下母亲对你的影响或者对你的意义吗？*
**Response:** 嗯......呃......母亲，对我......我觉得她就像我的荷叶，不论何时，我，我都能感觉到她的爱和保护，就，就像荷叶对红莲一样。
*Query: 同学们，今天我们要学习的是冰心的《荷叶·母亲》。在开始之前，我想先问大家一个问题，你们觉得你们的母亲是什么样子的？*
**Response:** 嗯......我妈妈很温柔，总是，总是耐心地听我说话，嗯，给我很多爱

Figure A31: Comparison of close-ended Question and open-ended question (original Chinese version).

---

**Comparison of close-ended Question and open-ended question (translated in English for better understanding).**

Here are the dialogues using the responses of a HN student in the "Lotus Leaves: Mother" lesson as an example to compare closed-ended and open-ended questions:

- - - - - - - - - - - - - - - - - - - - - - - - - - - - - - - - - - - - - - - - - - - - - - -

*[Closed-ended Questions]:*
*Query:Does everyone know who Bing Xin is?*
**Response**: Um... Bing Xin, um, I know. Bing Xin is a famous female writer and poet of ours.
*Query:Do you all remember who wrote the essay "Lotus Leaves: Mother"?*
**Response**: Um, it's, uh, it's Bing Xin.
*Query:After writing this essay, were you able to understand the theme of Bing Xin's writing?*
**Response**: Um, I think I... I can understand. Bing Xin... she uses the image of the lotus leaf protecting the red lotus to, uh... express, uh... the greatness and selflessness of maternal love.
*Query:Next, let's read "Lotus Leaves: Mother" together. What do you think the lotus leaf and the red lotus represent in the text?*
**Response**: The lotus leaf, um, I think it represents, uh, the mother; the red lotus, um, represents, uh, the child.
*Query:Now, let's do a small exercise. Based on your understanding, can you pick out a passage from the text to read aloud?*
**Response**: Um, okay, I will, um, choose, uh, this passage: "The lotus leaf, it protects me, just like a mother. No matter the wind or rain, it stands there tall, protecting me."

*[Open-ended Questions]:*
*Query:How do you think your understanding of Bing Xin's "Lotus Leaves: Mother" has deepened through this lesson?*
**Response**: Um, I... I now, uh, have a deeper understanding that, um, Bing Xin uses the lotus leaf and the red lotus to, uh, describe the greatness of maternal love. I, um, also now feel more deeply the charm of Bing Xin's lyrical prose.

> *Query:Try to write a story based on Bing Xin's essay about you and your mother. Who would like to share their story?*
>
> **Response**: Um, I remember... I remember one time when I... didn't do well on a test... um... I felt very disappointed. I... locked myself in my room and didn't want to... talk to anyone. My mother, my mother... gently knocked on the door, then... came in. She sat by my bed, sat... by my bed. She... didn't scold me, nor did she... say too much. She just... gently stroked my head, and with her warm voice... told me, "Child, failure... is nothing to fear. What's important is... that we learn from it and then... bravely... stand back up."
>
> *Query:Now, how do you understand mothers and maternal love after this?*
>
> **Response**: Um, I feel that maternal love is like a lotus leaf, uh... selfless, always protecting us even in heavy rain. Um, mothers, they... they always give so much, silently, asking for so little in return.
>
> *Query:Now, can you try to express what your mother means to you or how she has influenced you?*
>
> **Response**: Um... uh... my mother, to me... I feel like she's my lotus leaf. No matter what, I... I can always feel her love and protection, just... just like how the lotus leaf protects the red lotus.
>
> *Query:Today we are going to learn Bing Xin's "Lotus Leaves: Mother." Before we start, I'd like to ask everyone, what kind of person do you think your mother is?*
>
> **Response**: Um... my mom is very gentle, she always... always patiently listens to me and, um, gives me a lot of love.

### D.7.3 LLaVA MODEL BAD CASE AFTER FINE-TUNE

**LLaVa's suboptimal fine-tuning performance:** The experimental results show that, while LLaVa achieved significant improvements in all fine-tuning tasks—student personalities, question types, and learning stages—it still had an overall post-fine-tuning accuracy of around 67.85%, which was lower compared to the other models. This outcome is likely due to LLaVa's pretraining predominantly on English data, limiting its performance in Chinese-language applications compared to models pretrained on datasets containing Chinese content (see Appendix D.7.3 for more examples). Future research could address this limitation by incorporating more extensive Chinese-language pretraining or customized fine-tuning tailored to Chinese contexts, enhancing LLaVa's applicability in multilingual environments.

In this section, we use the responses of HN LVSA, which showed the best average performance after fine-tuning in our experiment, as an example. By comparing LLaVa's model performance with the other three models, we highlight the bad cases observed in LLaVa's fine-tuning.

> **Bad case after fine-tuning of LLaVa in HN LVSA (translated in English for better understanding).**
>
> Here are bad cases of Qwen model performance in HN student after fine-tuning:
>
> - - - - - - - - - - - - - - - - - - - - - - - - - - - - - - - - - - - - - - - - - - - - -
>
> *QUERY*: *do you know who wrote the text Ode to the Yellow River?*
> **[InternVl]**: Um, it was, um, Guang Weiran who wrote it.
> **[LLaVa]**: Um, was it, um, Wang Wan?
> **[MiniCPM]**: Um, it was, um, Guang Weiran who wrote it.
> **[Qwen]**: Um, was it, um, Guo Moruo?
>
> *QUERY*: *Lastly, let's summarize today's lesson. What do you think you've learned from On the Love of Lotus Flowers?*
> **[InternVl]**: Um, I, I learned, um, how, how to evaluate, um, a person's, um, qualities, um, and also, also learned, um, how to, um, express, um, my, um, views, um, and, and opinions.
> **[LLaVa]**: Um, I, I think, um, I learned, um, it should, should be, um, should be, um, should be, um, should be, um... (keeps repeating)
> **[MiniCPM]**: Um, he, he didn't, um, specifically mention his hometown's situation, but, um, he talked about his hometown's, um, land, and how, um, it was fertile and beautiful.

以高神经质为例展示LLaVa 的bad case与其他基座模型的差异

**Query:** *同学们，你们知道《黄河颂》这篇课文是谁写的吗？*
InternVl: 嗯，是，是，嗯，是光未然写的。
LLaVa: 嗯，是，呃，王湾吧
MiniCPM: 嗯，是，呃，是光未然写的。
Qwen: 嗯，是，呃，是郭沫若写的。

**Query:** *最后，我们来总结一下今天的学习。你觉得通过《爱莲说》学到了什么？*
InternVl: 嗯，我，我学到了，嗯，如何，如何评价，嗯，一个人，一个人，嗯，的品质，嗯，也，也学到了，嗯，如何，如何表达，嗯，自己的，嗯，观点，嗯，和，和看法。
LLaVa: 嗯，我，我觉得，呃，我学到了，嗯，应该，应该是，嗯，应该是，嗯，应该是，嗯，应该是，嗯……（一直重复）
MiniCPM: 嗯，他，他没有，呃，具体提到他的家乡的情况，但是，嗯，他提到了他的家乡的，嗯，土地，他的家乡的，嗯，土地是肥沃的，嗯，是美丽的。
Qwen: 嗯，我，我学到了，嗯，莲花的，嗯，品格，嗯，让我，嗯，更加，嗯，更加欣赏，嗯，莲花，嗯，和，嗯，莲花的，嗯，品格。

**Query:** *本节课我们学习的课文是《阿长与山海经》，从阿长的行为中，你觉得"我"对她的态度有什么变化？*
InternVl: 嗯，一开始，"我"对阿长，呃，有点，有点不耐烦，但是，但是后来，呃，阿长为了"我"买书，"我"开始，开始对她，对她感到，感到很，很感激。
LLaVa: 嗯，"我"，呃，"我"对她的态度，嗯，变化很大，嗯，从开始的，呃，敬爱，到，嗯，到最后的，呃，尊敬。
MiniCPM: 嗯，开始的时候，嗯，"我"对阿长，呃，是有点，有点不尊重的，但是，嗯，后来，呃，通过阿长对《山海经》的介绍，嗯，"我"开始尊重她，嗯，也，嗯，开始感激她。
Qwen: 嗯，一开始，呃，"我"对阿长的态度，嗯，是有些轻视的，但是，嗯，后来，"我"对阿长的态度，嗯，变得很尊重了。

**Query:** *本节课我们学习的课文是《谁是最可爱的人》，让我们来翻开书本，看看这篇文章写的是什么内容？*
InternVl: 嗯，这篇文章，呃，是，是写，写的是，嗯，抗美援朝战争中的，呃，中国人民志愿军。
LLaVa: 嗯，这篇文章，呃，是关于，嗯，作者和他的孩子，嗯，他们在一起的，嗯，一些，嗯，一些故事。
MiniCPM: 嗯，是的，呃，这篇文章，嗯，是写，呃，写的是，嗯，抗美援朝战争中的，嗯，英雄们，嗯，他们的，嗯，他们的事迹。
Qwen: 嗯，这篇文章，呃，是写，写抗美援朝的，嗯，志愿军战士的。

**Query:** *本节课我们学习的课文是《闻一多先生的说和做》，现在我们尝试分小组讨论，课文中的闻一多先生展示了怎样的品质？请小组讨论后分享？*
InternVl: 嗯，我们，我们觉得，呃，闻一多先生，他，他展示了，嗯，爱国，嗯，和，和坚持的精神。
LLaVa: 嗯，他，他展示了，呃，诚信，和，和，和，和，和，和，和，和，和，和，和，和，和，和，和，和，和，和，和，和，和……（一直重复）
MiniCPM: 嗯，闻一多先生，他，他展示了，嗯，爱国，和，和对文学的热爱。
Qwen: 嗯，我们，我们小组认为，嗯，闻一多先生，他，他很，嗯，很爱国，他，他用，用他的，嗯，行动，来，来表达他的，嗯，对国家的，的热爱。

Figure A32: Bad case after fine-tuning of LLaVa compared with other three models in HN LVSA (original Chinese version).

---

**[Qwen]**: Um, I, I learned, um, the character of the lotus, um, and it made me, um, appreciate, um, the lotus, um, and its, um, qualities even more.

*QUERY*: *Today we are studying the text A Chang and the Classic of Mountains and Seas. From A Chang's actions, what do you think changed in "my" attitude towards her?*
**[InternVl]**: Um, at first, "I" was a bit, a bit impatient with A Chang, but, but later, um, after she bought the book for "me," "I" started to, started to feel, feel very, very grateful to her.
**[LLaVa]**: Um, "I," uh, "my" attitude towards her, um, changed a lot, um, from, um, admiration at first, to, um, respect in the end.
**[MiniCPM]**: Um, at the beginning, um, "I" didn't really, um, respect A Chang, but, um, later, um, after A Chang introduced The Classic of Mountains and Seas, um, "I" started to respect her, um, and, um, started to appreciate her.
**[Qwen]**: Um, at first, uh, "I" was a bit, um, dismissive of A Chang, but, um, later, "I" started to respect her.

*QUERY*: *Today we are studying the text Who Are the Loveliest People. Let's open the book and see what this article is about.*
**[InternVl]**: Um, this article, um, is about, um, the Chinese People's Volunteers, um, during the Korean War.
**[LLaVa]**: Um, this article, um, is about, um, the author and his child, um, they're together, um, in some, um, some stories.
**[MiniCPM]**: Um, yes, uh, this article, um, is about, um, the heroes, um, of the Korean War, um, their, um, deeds.
**[Qwen]**: Um, this article, um, is about, um, the volunteers, um, of the Korean War.

*QUERY*: *Today we are studying the text Wen Yiduo's Words and Deeds. Now, let's try to discuss in groups: what qualities did Wen Yiduo demonstrate in the text? Please discuss in your groups*

*and then share.*

**[InternVl]**: Um, we, we think, um, that Wen Yiduo demonstrated, um, patriotism, um, and, and perseverance.

**[LLaVa]**: Um, he, he demonstrated, um, integrity, and, and, and, and, and, and, and, and, and, and, and... (keeps repeating).

**[MiniCPM]**: Um, Wen Yiduo demonstrated, um, patriotism, and, and a love for literature.

**[Qwen]**: Um, we, our group thinks, um, that Wen Yiduo, um, he was very, um, patriotic, and he, he expressed his, um, love for the country through his, um, actions.

## D.8   ETHICAL PROTECTION AND DATA PRIVACY STATEMENT

**Ethical Review Exemption:** This study places great importance on data privacy and ethical protection, adhering strictly to ethical standards for academic research, particularly when involving human participants. As a non-interventional study, all data collection and processing involving participants excluded sensitive personal information and were subjected to rigorous anonymization. Therefore, the study meets the ethical exemption criteria outlined in the institutional ethics review guidelines. This exemption ensures the study's compliance with legal and ethical standards while confirming that the research activities pose no potential risks to participants.

**Data Processing and Protection:** All experimental data were anonymized upon collection to fully dissociate it from participants' personal identities, fundamentally eliminating the risk of privacy breaches. Furthermore, the study was designed with non-interventional experimental scenarios, ensuring all research activities were conducted under low-risk conditions without causing any psychological or physical discomfort to participants. Before data collection, all participants were thoroughly informed about the study's purpose, data usage, and their rights, participating voluntarily with full knowledge of the process. The data collection process strictly followed academic research protocols to ensure scientific rigor, compliance, and transparency.

**Dataset Use and Accessibility:** The datasets used in this study comprise human evaluation data and virtual student-generated data. Human evaluation data primarily record participants' assessments of the experimental content. Despite being anonymized, these data are restricted to internal analysis to prevent any potential information leakage and are not publicly available. In contrast, the virtual student-generated data were entirely produced by the GPT-4 model, containing no real personal data and thus posing no privacy risks.

