# OpenReview forum: "Students Rather Than Experts: A New AI for Education Pipeline to Model More Human-like and Personalised Early Adolescences"
_ICLR.cc/2025/Conference — Submitted to ICLR 2025_

### Official Review · Reviewer_yWBj · 2024-10-23

**Soundness:** 3
**Presentation:** 2
**Contribution:** 3
**Rating:** 6
**Confidence:** 5

**Summary:**

++ I have increased the scores after the clarifications from the Authors.

This paper introduces a novel approach to using Large Language Models (LLMs) for simulating virtual students, instead of the more common focus on AI-driven teacher models. The authors propose a framework called SOE (Scene-Object-Evaluation) for generating LLM-based Virtual Student Agents (LVSA). This framework is applied within educational contexts, particularly for teacher training. The key contributions include:

1.Theoretical Framework: A comprehensive model for constructing LVSA based on personalized teacher-student interaction data.
2.Human and GPT-4 Evaluations: The integration of human subjective evaluation with GPT-4's capabilities to assess the authenticity of virtual student agents.
3.Experimental Validation: Extensive experiments confirm the feasibility of generating human-like, personalized virtual student agents for improving teacher training.

**Strengths:**

Originality: The paper presents a novel shift from teacher-centric AI systems to virtual student simulations, which can greatly impact teacher training and educational AI systems. By focusing on virtual students, it provides a fresh and creative problem formulation, opening new avenues in AI4Education research.

Quality: The paper demonstrates a well-structured methodology, from developing a dataset of teacher-student interactions to fine-tuning LLMs and validating their outputs with human and GPT-4 evaluations. The theoretical and operational frameworks are well thought out, ensuring rigorous scientific contribution.

**Weaknesses:**

Limited Real-world Data Application: While the framework is well-developed, the fine-tuning process relies on datasets sourced from controlled environments, which may not fully capture the complexity of real-world classroom dynamics. Future work should consider using more diverse and naturalistic data for better generalization.


Evaluation Focus: The evaluation primarily centers on language comprehension and emotional simulation. However, cognitive challenges like problem-solving or more complex reasoning skills, which are critical to student behaviors, are underexplored. This may limit the scope of the virtual students' realism in broader educational contexts.

**Questions:**

Handling Cognitive Diversity: Can the authors expand on how their system handles more complex cognitive behaviors beyond basic language learning? For example, how well can LVSA simulate problem-solving tasks or deal with creative thinking in open-ended questions?

Mitigating LLM Hallucinations: What specific strategies can be employed to minimize hallucinations in virtual student responses? Would incorporating additional real-world classroom data or using more advanced fine-tuning methods help in this regard?

---

> ### Author Response · Authors · 2024-11-25
> **Response to Reviewer yWBj's concern about the use of real-world data in the SOE pipeline**
>
> ***W1: While the framework is well-developed, the fine-tuning process relies on datasets sourced from controlled environments, which may not fully capture the complexity of real-world classroom dynamics.***
>
> **Response**: Thanks for recognizing the SOE pipeline and for the valuable suggestions regarding the incorporation of real-world data. In our current study, we did consider using more authentic classroom data. **However, real-world classrooms often rely heavily on traditional teaching methods, where student responses are limited and lack diversity.** This makes it difficult to meet the requirements for balanced sampling. Additionally, in the context of pre-service teacher training, LVSA need to exhibit distinct personality traits to help teachers adapt to diverse student needs. The available real-world data is therefore highly limited, which led us to develop and validate the model using data generated in controlled environments.
>
> **It is worth mentioning that after reading your comments, we have been inspired to engage in deeper discussions as a team**, particularly regarding how real-world data could be integrated into future research. We plan to take the following steps in subsequent studies:
>
> - **Selecting Representative Personality Traits**: Based on five key personality traits, we aim to choose representative students from various disciplines, cultural backgrounds, and age groups.
> - **Collecting High-Quality Teacher-Student Interaction Data**: We plan to invite students and teachers to participate in one-on-one teaching sessions and collect interaction data during these sessions. These natural teaching interactions will be recorded and annotated to provide more realistic data for virtual student modeling.
> - **Enhancing Data Diversity and Coverage**: Gradually expanding data sources to include more classroom formats and educational scenarios, thereby further improving the practicality and generalizability of the SOE framework.
>
> We understand that translating research conducted under laboratory conditions into technology applicable to real-world education poses many challenges. The reviewer's feedback has provided us with valuable direction, which will guide us in better integrating and leveraging real-world data in future studies. Thanks again for the valuable suggestions. We look forward to translating these insights into tangible results in subsequent research.

---

> ### Author Response · Authors · 2024-11-25
> **Response to Reviewer yWBj's concern about simulating problem-solving and creative thinking in open-ended questions**
>
> ***W2&Q1: Can the authors expand on how their system handles more complex cognitive behaviors beyond basic language learning?***
>
> **Response**: Thanks for the reviewer's attention to the complex cognitive behaviors, such as problem-solving and creative thinking, which are essential to assessing the realism of virtual students and their applicability across diverse educational contexts. As noted in common concern 3, we fully agree that these abilities are critical and welcome this opportunity to clarify and expand on how our study addresses these aspects.
>
> In this study, we designed the SOE pipeline with an emphasis on the importance of cognitive development in areas like problem-solving and creative thinking. Consequently, we included question types as a key evaluation dimension to reflect the virtual students' ability to operate at different cognitive levels (**see Sec 4.1 & Appendix B.2.2**):
>
> - **Problem-Solving Ability**: This aspect primarily focuses on cognitive levels of application and analysis. We designed task-based questions closely related to the content of the lesson, requiring virtual students to engage in reasoning and analysis. For instance, when discussing reading material, virtual students with high extraversion were able to provide clear and logical answers based on the text.
>
>   - **Example Question**: “Duanmu Hongliang used many vivid words to express his feelings for his hometown. Can you find some examples and discuss them?”
>   - **Example Response**: “Sure, teacher. For example, “My hometown is in the northeast, with its dark, fertile soil, golden grains, red maple leaves, and white snow.” These descriptions make me feel his deep love for his hometown.”
>
> - **Creative Thinking Ability**: This aspect evaluates students at the evaluation and creation levels of cognition. We designed open-ended questions to encourage students to express creative ideas, such as reinterpreting the theme of a passage or continuing a story. Highly extraverted virtual students demonstrated proactive responses.
>
>   - **Example Question**: “Now, let’s see whose imagination is the most creative. Based on the text, can you describe the image of the Yellow River in your own words?”
>   - **Example Response**: “Sure, Teacher. In the poem, the Yellow River is depicted as a great mother who nourishes the entire Chinese land. Its surging waves symbolize the resilience and perseverance of the Chinese people, and its grandeur reflects their greatness and pride.”
>
> More detailed response examples at different learning stages are added in **Appendix D.2.2**. Through these designs, we have preliminarily validated the potential of LVSA to simulate problem-solving and creative thinking in complex cognitive behaviors.
>
> Our primary goal in this study is to propose a scientific and systematic SOE pipeline to address challenges in modeling and evaluating LVSA throughout the teaching process. Although comprehensive analysis of LVSA abilities is not the main focus at this stage, our experimental data suggests that LVSA already shows representative potential in complex cognitive behaviors, such as problem-solving and creative thinking.
>
> In future work, we plan to further expand this aspect through the following steps:
>
> - **Refining the Cognitive Evaluation Framework**: Drawing on cognitive taxonomy theories (e.g., Bloom's taxonomy), we aim to design more fine-grained question types to comprehensively cover virtual students' performance in complex cognitive behaviors.
> - **Analyzing the Relationship Between Personality and Cognitive Abilities**: Investigate how virtual students with different personality traits perform in problem-solving and creative tasks, to further improve the modeling accuracy of LVSA.
> - **Expanding Multidisciplinary Scenarios**: Extend evaluation tasks to other disciplines (e.g., solving math problems, designing scientific experiments) to test LVSA s cognitive simulation capabilities across subject domains.
>
> We greatly appreciate the valuable suggestions regarding complex cognitive tasks. Through the evaluation dimension of question types, this study has preliminarily validated the potential of virtual students in problem-solving and creative thinking. In the future, we will conduct further in-depth analysis to continually enhance the simulation capabilities and educational value of LVSA.

---

> ### Author Response · Authors · 2024-11-25
> **Response to Reviewer yWBj's concern about mitigating hallucinations**
>
> ***Q2: What specific strategies can be employed to minimize hallucinations in virtual student responses? Would incorporating additional real-world classroom data or using more advanced fine-tuning methods help in this regard?***
>
> **Response**: Thanks for raising the issue of mitigating hallucinations in the content generated by LVSA. Hallucination is a common and critical challenge in applying LLMs. We have implemented various strategies to minimize such issues and plan to further enhance these measures in future work, which has also been added into **Sec 6**.
>
> Below, we provide a detailed description of our current strategies and future improvement directions.
>
> **1. Current Strategies in This Study**
>
> In this research, we employed the following strategies to mitigate hallucination issues:
> - **Explicit Personality Trait Prompt Design** (see Appendix B.2.2): We emphasized incorporating detailed descriptions of personality traits in the prompts. This helps guide the model to generate content consistent with the specified personality traits.
> - **Contextualized Content Generation for Teaching Scenarios** (see Appendix C.2.2): By linking generation tasks with specific teaching contexts, we directed the model to produce relevant and practical teaching content, thereby reducing irrelevant or misaligned hallucination content.
> - **Post-Processing and Verification of Generated Content** (see Appendix C.1.2): All model outputs were verified to ensure consistency and logical coherence by expert revision. This step ensures that the generated content adheres to teaching standards and accurately reflects realistic student behaviors.
>
> While these methods have reduced hallucinations to a certain extent, we acknowledge that discrepancies between generated content and actual classroom contexts may still occur.
>
> **2. Insights and Improvements for Future Work**
>
> Based on your suggestions, we believe the following directions can further mitigate hallucination issues:
> - **Incorporating Real Classroom Data**: We plan to collaborate with schools to collect high-quality teacher-student interaction data from actual classroom scenarios. This will enhance the diversity and authenticity of the data, providing stronger constraints on the model's outputs.
> - **Optimizing Personalized Modeling Mechanisms**: We aim to explore and implement more precise personalization modeling techniques, such as reinforcement learning or multi-objective fine-tuning, to introduce stronger personality constraints and ensure the model accurately simulates behaviors of various complex personalities.
> - **Strengthening Monitoring and Correction of Hallucinated Content**: We plan to develop specialized mechanisms for monitoring and correcting hallucinated content. For instance, natural language processing techniques can be employed to detect and rectify inconsistencies or illogical content. Dedicated detection and correction mechanisms for hallucinations will further enhance the reliability and coherence of generated content.
>
> We place great importance on addressing hallucination issues and will continue exploring and implementing effective solutions in this area. We believe that these measures will significantly improve the quality of virtual student models and their effectiveness in educational applications. Thanks again for the valuable suggestions, which will help us better tackle this challenge in future research.

---

### Official Review · Reviewer_3QkY · 2024-10-27

**Soundness:** 2
**Presentation:** 2
**Contribution:** 2
**Rating:** 5
**Confidence:** 4

**Summary:**

This work presents an AI4Education framework, termed SOE (Scene - Object - Evaluation), to construct LVSA (LLM-based Virtual Student Agents). It contributes a new Chinese education dataset and evaluates four existing LLMs which are fine-tuned in this dataset in simulating virtual students with specific traits. A Turing test is used to evaluate whether LLMs can generate such human-like virtual students. Specifically, GPT-4 is used for large-scale automatic evaluation and a small group of real humans are recruited for small-scale evaluation.

**Strengths:**

1. This work presents a LLMs-based agent evaluation pipeline including data preparation, LLM fine-tuning, and virtual student evaluation.

2. A new large dataset is contributed to study the virtual student agents mimicked by LLMs.

3. A comprehensive experiment is conducted to evaluate the performance of student agents powered by LLMs.

4. This paper presents sufficient supplemental materials for the readers to understand the prompts.

5. Overall, it is easy to read and understand this paper, with clear arguments.

**Weaknesses:**

W1. One of my main concerns is the novelty in model/framework design. The proposed SOE framework looks more like a dataset processing and LLM testing pipeline. However, this is no specific model design or new prompt structure design. The authors simply instruct the LLMs to mimic students with different traits based on the big five rationale. The LLMs used from existing work are directly fine-tuned on a new dataset. There are no unique insights in prompt design or fine-tuning strategies, compared with previous work. Moreover, the incorporation of big five theory for agent personality is also widely utilized in existing work such as [1]. Therefore, this is not a novelty nor contribution.

W2. Another main concern is that I think this work cannot really support its claim in line 532 "resembled real student behavior" and line 533 “simulating realistic student roles”. I agree with line 053 “virtual student agents must replicate a broader range of human behaviors”. But such replication needs labels/ground truth to demonstrate and quantify the accuracy of such replication. A simple Turing test cannot really support that, and lots of existing work has explored LLMs in passing Turing test [2] and a new Turing experiment [3]. That said, Turing test can show that your agent simulation is human-like, but that does not necessarily mean that your agent simulation is realistic among students’ real behaviors, which needs more convincing ground truth for comparison. But this work does not have a real ground truth for comparison.

Specifically, I also do not understand why the five kinds of agents (HN, HA, LO, HE, LC) are combined with real students as the six one together for evaluation in appendix figure A18. Do the authors mean that human evaluators are asked to distinguish one of the six types? I’m confused because at first the authors say that this work aims to simulate student traits. So I thought real students serve as the labels instead of an additional type in addition to the five traits. Such Turing test setting can only show that humans cannot distinguish whether the expressions are from agents or real students. This is not surprising considering existing work showing the performance of LLMs in the Turing test such as [2]. However, it cannot show that such simulation is realistic because there is no ground truth for comparison.


W3. I’m also concerned regarding the limited generalization and evaluation. This work only uses one Chinese dataset from the authors. No public datasets or English datasets are used. It is uncertain whether the performance of this framework can have good generalization.

W4. Moreover, another important concern is the lack of a real baseline. The evaluation simply compares pre-finetuning with post-finetuning. It is not surprising that finetuning in the targeted dataset can result in better performance. But there is not a real baseline to show the unique advantage of the propose SOE framework, compared with existing approaches.

W5. Furthermore, using GPT-4 for automatic evaluation is not convincing. It looks like the authors are using one LLM to evaluate other LLMs’ performance. No ground truth is used for more convincing evaluation. Line 373 mentioned the high inter-coder reliability just proves that the two students have consistent opinions in annotation. But it cannot prove that GPT 4 is a convincing automatic evaluation tool to replace real human annotators to perform automatic evaluation.

W6. In Appendix D.5.3 and figure A18, since GPT4 can “effectively simulate and assess nuanced student behaviors” (from line 4236), why not directly use GPT4 for student trait simulation? Why do you use other LLMs to perform such simulation and use GPT 4 to serve as the evaluator? Furthermore, in line 4244, “GPT-4 and human evaluators was found to be 0.6806, which falls within the range of substantial agreement”. Why 0.6806 is acceptable and why 0.6806 is within the range of substantial agreement? Who defines the agreement? Is there a standard rule for the value range?

W7. I’m really confused about what the evaluation score means in section 5.4. What the score refers to? What does 100% mean in line 416? Does it refer to the accuracy for human evaluators or GPT 4 to distinguish different types of agent traits? Similarly, for different learning stages in line 463 and different question types in line 482, what does the percent % mean? Does it refer to the accuracy or any metrics, to measure what?

W8. Line 478: “These findings suggest that LVSA effectively adapts to different learning stages, offering comprehensive support for pre-service teacher training. Virtual students enable teachers to practice and refine instructional strategies across all teaching phases, enhancing skill development throughout the entire instructional process.” These assertions are over-claimed. If you want to show its effectiveness to help teachers, then you need to conduct a real teaching experiment to show it, which is more convincing. Otherwise, such assertions sound weak.

W9. Line 334: No ethic information (such as IRB approval) about recruiting human participants for evaluation and releasing collected datasets to the public. Potential ethic concerns.

W10. 115 samples are too limited for human evaluation. The number of questions used for human evaluation in Appendix Figure A11 and table A3 is too limited. It is also unclear why you only use 40 samples or 35 samples per setting, which is quite confusing. Why all five fatigue test items come from the real student responses? Why not use 2 items from each group of fine-tuned models, direct inference, and real students?

W11. For section 5.2 human Turing test, simply showing the average results of all human evaluators is not a standard nor convincing approach. A formal statistical analysis such as ANOVA is needed.

Reference:

[1]. CGMI: Configurable General Multi-Agent Interaction Framework, Shi et al. 2023

[2]. Does GPT-4 pass the Turing test? Jones et al. 2023

[3]. Using Large Language Models to Simulate Multiple Humans and Replicate Human Subject Studies, ICML 2023

**Questions:**

All questions are listed in the weakness above.

**Details Of Ethics Concerns:**

No ethic information (such as IRB approval) about recruiting human participants for evaluation and releasing collected datasets to the public. Potential ethic concerns.

---

> ### Author Response · Authors · 2024-11-25
> **Response to Reviewer 3QkY's concern about the lack of innovation or contribution in the SOE pipeline (Part I)**
>
> ***Q1: The proposed SOE framework looks more like a dataset processing and LLM testing pipeline. Moreover, the incorporation of Big Five theory for agent personality is also widely utilized in existing work.***
>
> **Response**: Thanks for the question. In addressing these concerns, we have placed special emphasis on the interdisciplinary nature and challenges of this study, **particularly its integration of education, psychology, social sciences, and AI technologies**.
>
> We would like to further elaborate on the core contributions of the SOE framework, which have also been recognized by reviewers yWBj and rtkz. Reviewer yWBj noted that “*The paper presents a novel shift from teacher-centric AI systems to virtual student simulations and provides a fresh and creative problem formulation, opening new avenues in AI4Education research.*” Similarly, reviewer rtkz stated that “*This paper brings a high degree of originality by shifting the focus from expert or teacher-oriented AI simulations to student simulations. The contributions align well with the ICLR community’s interest in novel applications of AI, particularly in education and social simulation, and could stimulate further research into AI-driven, personalized learning simulations.*”
>
> **Below, we detail how the SOE framework employs social science research methodologies alongside AI technologies to tackle the complexities of virtual student simulations and their evaluations**.
>
> **1. Research Challenges and Core Contributions of the SOE Framework**
>
> This study transcends mere experimental validation of LLMs by addressing the significant challenge of modeling and evaluating virtual students through the innovative SOE (Scene-Object-Evaluation) framework. Unlike traditional LLM applications, this framework integrates social science theories, such as the Big Five personality traits and student cognitive development theories, with advanced AI fine-tuning techniques like LLM LoRA. This integration is critical for simulating personalized and human-like behaviors of virtual students throughout the entire educational process. **Our research necessitates a profound understanding of these social theories and the precise translation of complex psychological and behavioral concepts into computable data for algorithmic processing.** By adopting a multi-disciplinary approach, we have refined our research methodology to incorporate diverse academic perspectives effectively.
>
> - **In-Depth Thinking Before Research Begins**:
>   - **Careful Selection of Validation Scenarios**: Language learning was selected due to its observable, open, and diverse nature, making it ideal for validating the SOE pipeline's theoretical foundation and practical applicability.
>   - **Analyzing the Challenges of Modeling Virtual Students**: Student behaviors are highly diverse and open-ended, raising critical questions such as:
>     - What diverse personality traits do students exhibit?
>     - What cognitive development stages do students undergo during learning?
>     - How do students with different personality traits behave?
>
> - **Interdisciplinary Approach to Modeling Pipeline Development**:
> The SOE pipeline integrates social sciences and AI to propose a modular framework bridging conceptual and operational theories. Its modular design (Scene-Object-Evaluation) supports diverse functionalities, as noted in common concern 1.
>
> **To tackle interdisciplinary evaluation challenges, we combined methodologies from social sciences (empirical analysis), psychology (surveys), education research (structured interviews and coding), LLM studies (subjective evaluation), and NLP (objective metrics) to develop a comprehensive evaluation system:**
>
> - **Challenges in Evaluating Virtual Students**: Spontaneous and varied student language lacks fixed ground truth, making traditional accuracy-based methods unsuitable.
>   - **Systematic Subjective Evaluation**: Expert interviews and content analysis encoded evaluation dimensions, ensuring consistency and rigor, while GPT-4 enabled efficient large-scale evaluations.
>   - **Objective Evaluation Supplements**: Incorporating metrics like text length, perplexity, TTR, and sentiment analysis validated subjective evaluation results from multiple perspectives. However, these metrics have limitations in capturing the complexity of student language behaviors in educational scenarios. These limitations underscore the necessity of incorporating subjective evaluation methods.

---

> > ### Author Response · Authors · 2024-11-25
> > **Response to Reviewer 3QkY's concern about the lack of innovation or contribution in the SOE pipeline (Part II)**
> >
> > **2. Application of the Big Five Personality Traits and Innovation**
> >
> > The Big Five personality theory, widely used in generative AI research, was adopted for its scientific framework to systematically simulate adolescent students' personality traits. **This approach enhances research generalizability and reduces biases in personality modeling**. Below are the reasons for selecting and applying the Big Five traits in this study:
> >
> > - **Reasons for Choosing the Big Five Personality Theory**:
> >   - **Universality and Scientific Basis**: The Big Five personality theory is one of the most widely accepted personality theories in psychology. It provides a comprehensive and structured framework for analyzing individual differences. Its universality makes it an ideal tool for understanding human behavior in interdisciplinary research.
> >   - **Alignment With Educational Psychology**: Compared to other psychological theories, such as Cattell’s personality theory, which is overly complex, or Maslow’s hierarchy of needs, Allport’s value theory, and Atkinson’s achievement motivation theory, **which primarily focuses on psychological factors other than personality traits**, the Big Five personality theory can be more directly applied to simulate student behavior and interactions in educational scenarios.
> >
> > - **Strategies for Selecting Personality Traits**:
> >   - **Goal-Oriented Trait Selection**: Among the ten high and low levels of traits within the Big Five framework, **we did not make random selections. Instead, we carefully selected five traits with high distinguishability and strong relevance to educational psychology**. This selection emphasizes traits that significantly influence student learning behavior and classroom interactions.
> >   - **Challenges and Opportunities**: **By focusing on specific traits, such as low conscientiousness (LC), we explored and addressed challenges in simulating these complex traits in virtual environments**, such as hallucination issues and ethical safety triggers. These efforts not only reveal current limitations in LLM-based role modeling but also point to future improvement directions.
> >
> > Our research applies recognized psychological theory and introduces an innovative method for modeling and evaluating virtual students, blending educational principles with AI technology. This interdisciplinary approach fosters personalized, adaptive education and marks a significant scientific advancement.
> >
> > In Summary, **the primary difficulty and core innovation of our work lie in utilizing AI technologies within interdisciplinary research to address the challenges of modeling and evaluating virtual students in educational scenarios.**  Although we employed existing LLMs for fine-tuning, our innovation lies in **how these technologies were applied to specific educational psychology scenarios, particularly in simulating complex personality traits and educational interactions**. The modular design of the SOE pipeline allows flexible adjustments to accommodate diverse educational needs, which is uncommon in traditional LLM applications. Additionally, while we leveraged existing technologies for experimentation, the key innovation lies in our approach to accurately simulate and evaluate students with different personality traits and enhance the model’s educational applicability and psychological fidelity through fine-tuning strategies.
> >
> > We trust this clarification underscores the SOE framework's innovative aspects and the intricate challenges and novel solutions of our interdisciplinary research. We appreciate your feedback and are committed to further advancing educational technology through ongoing research and innovation.

---

> > ### Comment · Reviewer_3QkY · 2024-11-26
> > **Overall**
> >
> > Overall, I'd like to say thanks to the authors for the efforts to address the concerns and limitations. Three main concerns are still not solved yet:
> >
> > 1. Framework novelty: there is no foundational novelty or breakthrough in LLMs' reasoning for student simulation. The framework looks more like a data processing pipeline that integrates several data structure modules. Although several new concepts are introduced, they do not contribute to the core model contribution.
> >
> > 2. Lack of baselines: there is no baseline comparison to show the unique advantage of the proposed framework.
> >
> > 3. Generalization: only one dataset from the authors is evaluated. No public datasets are used for evaluation.
> >
> > Therefore, I will maintain my score.

---

> > > ### Author Response · Authors · 2024-11-29
> > > **Overall response to Reviewer 3QkY**
> > >
> > > We sincerely thank the reviewer for their thorough review and valuable comments. We greatly appreciate the insightful questions raised, which have prompted us to further reflect on and improve our research. For the three core concerns you raised, we have provided detailed responses in the corresponding sections. Once again, we thank the reviewer for their careful reading and thoughtful guidance.
> > >
> > > **1. Innovation of the Framework**
> > >
> > > We understand the reviewer's focus on the innovation of the framework. We fully agree that innovations in interdisciplinary research often have multiple interpretations and perspectives, and we remain open to further discussion and exploration. We have addressed this concern in the section on the framework’s innovation, emphasizing that the SOE framework is not merely an algorithmic innovation but a new approach that integrates interdisciplinary theories and large language model technologies. This approach provides a novel solution to the challenges of virtual student modeling and evaluation in educational practice. **While our work differs from traditional AI algorithmic innovation, it combines quantitative research from AI with qualitative research from social science theories, and we believe it holds scientific value in its own right.**
> > >
> > > **2. Absence of Baseline Models**
> > >
> > > Regarding the reviewer’s comment on the lack of baseline comparisons, we understand the concern and have explained that existing benchmark models in the educational field predominantly rely on large language models using prompt engineering. Our study, however, focuses on the application of fine-tuning, which differentiates it from these baseline models. **We have made efforts to compare results from prompt engineering with those from fine-tuning and will add clarifications on this aspect in the final version paper.**
> > >
> > > **3. Generalization Validation**
> > >
> > > Regarding dataset validation, we understand the reviewer's focus on multi-dataset validation. While multi-dataset validation is not the primary focus of our work, the current emphasis is on verifying the adaptability of the SOE pipeline across different teaching stages and problem types. However, in response to the reviewer's keen interest, we have provided a detailed explanation of the uniqueness of the dataset we are using and the reasons why we could not select other publicly available datasets. **We also emphasize that, compared to cross-linguistic datasets, which mainly assess the language transferability of large models, hierarchical understanding presents a greater challenge in the context of virtual student modeling under the SOE pipeline.** Based on this, we argue that validating the pipeline's effectiveness in a Chinese-language environment provides strong evidence of its validity, and we expect it to be even more adaptable in a more general English-language context.
> > >
> > > We greatly appreciate the reviewer's questions and the profound insights they have provided. **We recognize that understanding interdisciplinary innovations is inherently subjective, and our research does not aim to achieve breakthroughs in algorithms but to offer a new research framework for virtual student modeling and educational evaluation in the AI for Education field.** Through discussions with the reviewer, we have further refined our work and clarified the direction of our research, which has been an invaluable process for us.
> > >
> > > **Once again, we thank the reviewer for the careful review and invaluable suggestions. We will continue to improve our paper based on your feedback and look forward to further discussions and refinements in the future.**

---

> > > > ### Comment · Reviewer_3QkY · 2024-12-02
> > > >
> > > > I appreciate the authors' efforts in explanation. I will maintain my adjusted score. Thanks!

---

> ### Author Response · Authors · 2024-11-25
> **Response to Reviewer 3QkY's concern about limitations of the Turing Test and the absence of ground truth comparisons**
>
> ***Q2: Such replication needs labels/ground truth to demonstrate and quantify the accuracy of such replication.***
>
> **Response**: Thanks for the insightful question regarding the limitations of the Turing Test and the absence of ground truth comparisons in this study.
>
> By "simulating real student behavior," we refer to generating responses that mimic the cognitive and emotional reactions of real students in educational scenarios, rather than perfectly replicating their exact behaviors. **Unlike traditional ground truth comparisons, our focus is on assessing whether virtual students can effectively engage in and contribute to educational processes, which has been added in Sec 5.2.**
>
> Considering the potential confusion caused by the term "replicate" in the phrase from Reviewer 053, "virtual student agents must replicate a broader range of human behaviors," we have revised the wording from "replicate" to "simulate." (**see Sec 1**)
>
> Below, we provide a detailed response.
>
> **1. Role of the Turing Test in This Study**
> - **Turing Test for Validating Human-Likeness**: The Turing Test in this study is not intended to prove the authenticity of virtual student behavior but serves as an evaluation method to verify whether virtual students achieve a human-like level in behavior and language generation.
> - **Related Work as Reference**: The Stanford Village study [1] assessed generative agents in a complex virtual environment using the Turing Test and a multi-dimensional subjective evaluation, without relying on fixed ground truth. Similarly, our study employs the Turing Test to evaluate whether virtual students' language generation and personality traits approximate human interactive behavior across multiple dimensions.
>
> **2. Lack of Ground Truth in This Study**
>
> Due to the complexity of language-based educational environments, we did not use fixed ground truth for evaluation. Instead, a multi-dimensional approach combined subjective methods (e.g., Turing Test) with objective metrics (e.g., perplexity, type-token ratio, sentiment analysis) to assess virtual student performance comprehensively.
> - **Challenges in Defining Ground Truth in Language-Based Experiments**: Language-based experimental scenarios naturally involve diversity and openness, making it difficult to define fixed answers or behavior patterns as ground truth compared to science-based tasks (e.g., mathematical calculations or solving physics problems).
> - **Alignment With Mainstream Practices in Agent Evaluation**: Similar to the Stanford Village study [1], we did not use fixed ground truth, relying on multi-dimensional methods like Turing Tests and human subjective evaluations to assess generative agents. To address the limitations of these approaches, we incorporated objective metrics such as perplexity, type-token ratio, and sentiment analysis to strengthen our conclusions.
>
> **3. Real Students in Appendix Figure A18**
>
> We understand the reviewer’s concern regarding the experimental design involving real students in Appendix Figure A18 and would like to provide further clarification:
>
> - The inclusion of real student responses in Appendix Figure A18 was intended to test whether evaluators could distinguish between virtual and real student responses without explicit labels. This was not meant to treat real students as an additional category but rather to increase the difficulty of the experiment, testing whether the virtual students’ responses were sufficiently human-like to be indistinguishable from those of real students.
> - **This setup reflects our effort to simulate the authenticity of student behavior, rather than directly using real student responses as ground truth**.
> - We recognize that this experimental setup may have caused some confusion, so we more clearly explain the experiment’s objectives and the specific role of real students in the design (see Sec 5.2).
>
> Thanks again for the valuable feedback. This study validates the human-likeness of virtual student behavior through the Turing Test and multi-dimensional subjective evaluation, supplemented by objective metrics. While the diversity and openness of language tasks limit the direct application of ground truth, our methods align with mainstream practices in the field of generative agent research. This work represents a critical step in validating the theoretical foundation and practical value of the SOE pipeline.
>
> [1] Park, J. S., O'Brien, J., Cai, C. J., Morris, M. R., Liang, P., & Bernstein, M. S. (2023, October). Generative agents: Interactive simulacra of human behavior. In Proceedings of the 36th annual acm symposium on user interface software and technology (pp. 1-22).

---

> ### Author Response · Authors · 2024-11-25
> **Response to Reviewer 3QkY's concern about the lack of public datasets evaluation on the SOE pipeline**
>
> ***Q3: This work only uses one Chinese dataset from the authors. No public datasets or English datasets are used.***
>
> **Response**: Thanks for the reviewer's concern regarding the use of a single dataset and the generalization ability of the SOE pipeline. We understand and respect your view on the importance of data diversity and comprehensive evaluation. **However, as this study is the first to systematically explore the use of fine-tuning techniques for virtual student modeling with LLMs, there are currently no publicly available datasets suitable for this task.**
>
> Below, we will explain in detail the research methods we chose and our plans.
>
> **1. Use of a Single Dataset and Its Challenges**
>
> Our study mainly used a self-constructed Chinese dataset due to the novelty of the research question. **This innovative perspective also leads to a lack of publicly available datasets that meet the specific needs of virtual student modeling.** Although this limits our ability to directly verify the generalization ability of the SOE pipeline on other datasets, this choice was made based on the following considerations:
>
> - **Data Suitability**: Existing public education datasets mainly focus on traditional teaching interactions and lack detailed annotations of the diverse behaviors of virtual students. This is insufficient for our study of specific behaviors and response patterns of virtual students.
> - **Innovation in Data Generation**: To overcome the limitations of existing datasets, we adopted a method using GPT-4 to simulate diverse student behaviors and dialogues. This not only provides a broader range of behavioral characteristics but also ensures high data quality and experimental controllability. This method has already been widely used in fine-tuning research for large models [2-3].
>
> **2. Generalization Ability of the SOE Pipeline**
>
> Although this study primarily relies on a Chinese dataset, the design of the SOE pipeline inherently supports adaptation to multilingual and multicultural environments as noted in common concern 1. The following key features demonstrate its generalization potential:
> - **Modular Design**: The SOE pipeline’s modular structure—comprising Scene, Object, and Evaluation—allows us to flexibly adjust the models and data processing methods according to the requirements of different educational scenarios and disciplines. This design facilitates future extensions to datasets in other languages, including English.
> - **Cross-Cultural Adaptability**: Although the current research focuses on the Chinese context, the pipeline's structural design enables easy adaptation to different cultural and educational systems. In the future, we plan to introduce multilingual datasets to validate and optimize the adaptability and effectiveness of the SOE pipeline in diverse cultural contexts.
>
> **3. Future Research Directions**
>
> Based on reviewer valuable feedback, we plan to broaden the diversity and coverage of datasets in future research, including:
> - **Incorporating Public Datasets**: We will explore using existing multilingual education datasets and adapt them as necessary to support cross-cultural research.
> - **Multi-Scenario Validation**: By applying the SOE pipeline to different educational scenarios and disciplines, we aim to further test and optimize its generalization ability and practical value.
>
> Thanks again for the suggestion. We look forward to enhancing the generalization ability of our research through these improvements and expansions, while better addressing the need for data diversity and model adaptability in the educational technology field.
>
> [2] Wang, Y., Kordi, Y., Mishra, S., Liu, A., Smith, N. A., Khashabi, D., & Hajishirzi, H. (2022). Self-instruct: Aligning language models with self-generated instructions. arXiv preprint arXiv:2212.10560.
>
> [3] Rohan Taori et al. Stanford Alpaca: An Instruction-following LLaMA model. https://github.com/tatsu-lab/stanford_alpaca. 2023.

---

> > ### Comment · Reviewer_3QkY · 2024-11-26
> >
> > Thanks for the reply. It does not make sense to me as well.
> >
> > First, I indeed did not see the unique novelty of the proposed dataset. In my view, it is just similar to most education datasets. Maybe the language is different in Chinese. However, there are also lots of Chinese datasets as well.
> >
> > Moreover, to really demonstrate the generalization ability, it is always more convincing to run a real experiment on additional datasets, instead of simply putting it into the future work.

---

> > > ### Author Response · Authors · 2024-11-29
> > > **Response to Reviewer 3QkY's concern about evaluation of other datasets**
> > >
> > > We sincerely appreciate the reviewer's inquiry. We fully understand your concern regarding the absence of validation using multiple datasets. Indeed, multi-dataset validation is a crucial step in assessing the generalization and robustness of AI algorithm models. **However, the SOE pipeline proposed in this study is not a traditional AI algorithm model, but rather an operational framework designed for application in educational internships and social simulation experiments.**
> > >
> > > At the early stages of our research, we actively sought publicly available datasets. However, **datasets containing teacher-student dialogues that align with the Big Five personality framework are extremely rare.** Existing Chinese dialogue and educational datasets are not relevant to the experimental scenarios of this study, making them unsuitable for direct application in our experiments.
> > >
> > > Furthermore, we believe that cross-dataset generalization tests are more focused on evaluating the language transferability of large language models, a capability that has already been extensively studied and validated. **The core challenge we aim to address is the development of hierarchical understanding capabilities in virtual students across different scenarios. This hierarchical understanding requires virtual students to move from superficial comprehension to contextual interpretation and, ultimately, to a deeper understanding of the emotional themes expressed in the text.** We have already conducted systematic experimental validation using our custom-built dataset, demonstrating the hierarchical understanding abilities of virtual students in text comprehension tasks and across different teaching stages. **This validates that the SOE pipeline can enable personalized, hierarchical understanding.** For datasets in different languages, the language transferability of large language models can further optimize this capability.
> > >
> > > A detailed response to this concern is provided in **Parts V and VI of the rebuttal summary box**. We again thank the reviewer for the thoughtful question.

---

> ### Author Response · Authors · 2024-11-25
> **Response to Reviewer 3QkY's concern about the lack of real baseline for comparison**
>
> ***Q4: There is not a real baseline to show the unique advantage of the proposed SOE framework, compared with existing approaches.***
>
> **Response**: We appreciate the reviewer's interest in baseline comparisons. We understand the concerns raised by the reviewer regarding the lack of a “true baseline” in the current assessment. **Due to the innovative and specialized nature of our research, there are currently no directly comparable existing works or widely accepted baselines that can provide a comprehensive comparison.** Our study aims to develop a novel SOE (Scene-Object-Evaluation) pipeline specifically designed for virtual student modeling, filling a gap in this field and offering a robust baseline for future research.
>
> Below, we clarify the logic behind the design of this study and explain why the current evaluation method is reasonable and scientifically sound.
>
> **The primary goal of this research is to propose a systematic SOE pipeline to address the challenges in virtual student modeling and evaluation, rather than optimizing the performance of a single algorithm.** The core contribution of the SOE pipeline lies in providing a generalizable framework that includes Scene (context setting), Object (student modeling), and Evaluation (multi-dimensional assessment) components. Through this framework, we systematically explore LVSA behaviors across different teaching stages, personality traits, and question types, offering a new paradigm for virtual student research. **Importantly, the SOE pipeline integrates pedagogical evaluation logic with AI research, accommodating both educational and technical needs.** This research paradigm is unprecedented in the existing literature and holds significant originality and academic value. This unique contribution has also been acknowledged in Reviewer rtkz's comments. **While adopting publicly available baselines is common in traditional AI research, the lack of comparable frameworks or general datasets for virtual student modeling currently limits broader comparisons.**
>
> To address this limitation, we used non-fine-tuned large language models as a baseline for comparison. This approach is widely accepted in the field [4]. **The performance of non-fine-tuned models not only reflects the improvement achieved through fine-tuning but also validates the modeling difficulty of the target task and the effectiveness of the fine-tuning method.** Comparing performance pre and post fine-tuning is a key step in verifying the pipeline's validity and refining its functionality, rather than simply comparing algorithm performance. Using non-fine-tuned models as a baseline aligns with current academic standards and is a reasonable choice. **It is worth emphasizing that the outputs generated by non-fine-tuned large language models are not random but are produced using carefully designed instructions via prompt engineering.** This approach allows us to preliminarily demonstrate the improvements achieved by the SOE pipeline’s fine-tuning strategy in educational scenarios.
>
> **The SOE pipeline’s strength lies in its modularity and extensibility rather than in optimizing a single performance metric.** To better highlight the distinctiveness and contribution of the SOE pipeline, our experiments employed multi-dimensional evaluation methods to verify its support for virtual student modeling throughout the teaching process. During this process, we focused on addressing the challenge of evaluating subjective responses from virtual students, rather than emphasizing model performance on existing benchmark tasks.
>
> [4] Zhang, R., Han, J., Liu, C., Gao, P., Zhou, A., Hu, X., ... & Qiao, Y. (2023). Llama-adapter: Efficient fine-tuning of language models with zero-init attention. arXiv preprint arXiv:2303.16199.

---

> > ### Comment · Reviewer_3QkY · 2024-11-26
> >
> > Thanks for the reply. It does not make sense to me. If there is no closely related baseline, then you should at least use a relatively related baseline. Otherwise, it can't demonstrate the superiority of the proposed framework.

---

> > > ### Author Response · Authors · 2024-11-29
> > > **Response to Reviewer 3QkY's Concern About Baseline Comparison**
> > >
> > > We sincerely thank the reviewer for the valuable feedback. We understand the concern regarding the comparison with baseline models and would like to further clarify our considerations regarding the experimental design.
> > >
> > > We would like to emphasize that **although our study does not directly employ a relevant baseline model, we have made efforts to approximate existing work on virtual student modeling based on prompt engineering during the pre-fine-tuning phase.** Since this aspect was not explicitly addressed in the manuscript, **we will add a clarification in the article stating, "The 'pre' refers to a baseline comparison with existing prompt-engineering-based virtual student modeling research, where prompt engineering was used."** In response to the reviewer's concerns about baseline comparison, we provide the following detailed explanation:
> > >
> > > **1. Existing Work Primarily Uses Large Language Models with Prompt Engineering, Not Fine-Tuning**
> > >
> > > Regarding baseline models in related work, we have made efforts to find relevant benchmarks for our method. However, most existing large language model-based prompt-engineering approaches (such as those involving GPT-3 or GPT-4 prompts) do not involve fine-tuning [13-15]. The SOE framework we propose is based on fine-tuning for virtual student modeling. Therefore, the existing baseline approaches cannot be directly compared to our method, as they primarily rely on prompt engineering without incorporating detailed model fine-tuning. **In our experiment, we used prompt engineering to perform preliminary experiments, enabling a comparison with the proposed SOE pipeline.** However, this process was not fully detailed in the manuscript. **Following the reviewer’s suggestion, we will include a more comprehensive explanation in the final version of the paper, describing how prompt engineering was used for model guidance before fine-tuning and how it compares to the existing pipeline.**
> > >
> > > **2. The Uniqueness of the Research Results in the Lack of Suitable Baseline Comparisons**
> > >
> > > Current educational field benchmark models lack comprehensive modeling of student behavior complexity (e.g., personality differences, cognitive levels, etc.). Since the SOE framework focuses on multi-dimensional virtual student modeling (e.g., personality traits, cognitive levels) and integrated educational assessment, existing baseline models are not fully applicable in this regard. Furthermore, these models do not propose detailed pipeline processes or directly open-source their work, making it impossible for us to compare with existing baseline models.
> > >
> > > The core innovation of the SOE framework lies in providing an operational, scalable research framework for virtual student modeling and educational assessment, integrating interdisciplinary theories to support personalized modeling and evaluation across various educational scenarios. Therefore, **the contribution of this study is more focused on presenting a new research methodology and approach, rather than purely optimizing algorithms or comparing with baseline models.**
> > >
> > > We again appreciate the reviewer's thorough review of our work, and we hope that our responses above address your concerns.
> > >
> > > [13] Zhang, Z., Zhang-Li, D., Yu, J., Gong, L., Zhou, J., Liu, Z., ... & Li, J. (2024). Simulating classroom education with llm-empowered agents. arXiv preprint arXiv:2406.19226.
> > >
> > > [14] Lee, U., Lee, S., Koh, J., Jeong, Y., Jung, H., Byun, G., ... & Kim, H. Generative Agent for Teacher Training: Designing Educational Problem-Solving Simulations with Large Language Model-based Agents for Pre-Service Teachers.
> > >
> > > [15] Markel, J. M., Opferman, S. G., Landay, J. A., & Piech, C. (2023, July). Gpteach: Interactive ta training with gpt-based students. In Proceedings of the tenth acm conference on learning@ scale (pp. 226-236).

---

> ### Author Response · Authors · 2024-11-25
> **Response to Reviewer 3QkY's concern about the effectiveness of GPT-4 in automatic evaluation**
>
> ***Q5: line 373 cannot prove that GPT 4 is a convincing automatic evaluation tool to replace real human annotators to perform automatic evaluation.***
>
> **Response**: Thanks for raising questions about the validity of using GPT-4 for automatic evaluation. To ensure the scientific rigor and reliability of our experimental results, the design process of our experiments incorporates social science research paradigms, which may have caused some misunderstanding regarding the consistency analysis in our evaluations.
>
> Below, we will clarify all consistency-related experiments in our study and explain why GPT-4 can serve as a reliable tool for large-scale automatic evaluation.
>
> **1. Consistency Analysis in Evaluations**
> Our study includes three levels of consistency experiments, involving both human evaluators and GPT-4, to ensure scientific and objective evaluation results:
> - **Consistency Among 10 Human Evaluators**:
>   - **Objective**: To verify whether the assessments of "whether virtual students exhibit human-like traits" by 10 human evaluators are consistent, eliminating randomness.
>   - **Method**: Fleiss’s Kappa coefficient was used for statistical analysis, proving the consistency among human evaluators and ensuring the reliability of evaluation conclusions.
> - **Consistency Between Two Content Coders**:
>   - **Objective**: To extract scientifically grounded evaluation dimensions used in GPT-4 prompts by coding the evaluation process of human evaluators.
>   - **Method**: Two coders were invited to analyze the interview data of the evaluators, and their consistency was calculated to prevent personal bias in coding the evaluation dimensions (a step consistent with the content analysis method in social science research).
>   - **Outcome**: 4 primary and 15 secondary evaluation dimensions were extracted to design the GPT-4 prompts, ensuring that GPT-4’s evaluations were scientifically based.
> - **Consistency Between 10 Human Evaluators and GPT-4**:
>   - **Objective**: To verify whether GPT-4's automatic evaluation aligns with human evaluators, thus assessing its reliability as an evaluation tool.
>   - **Method**: A Fleiss’s Kappa analysis was conducted to compare human evaluators’ and GPT-4’s results, showing a high degree of consistency. This confirms GPT-4’s applicability as a tool for large-scale evaluations to improve efficiency.
>
> **Clarification**: The reviewer may have misunderstood that we only analyzed coder consistency. In fact, GPT-4’s evaluation capability was validated against 10 human evaluators, with a consistency score confirming its suitability for large-scale evaluations. (Details on Fleiss’s Kappa are provided in Response to Q6.)
>
> **2. Reasons for Using GPT-4 for Automatic Evaluation**
> - **Evaluation Needs**: The complexity of educational scenarios and the diversity of language tasks make defining a single ground truth impossible, mirroring the complexity of real student behavior. This underscores the need for the SOE pipeline and a tool to process large-scale data efficiently and provide rapid feedback.
> - **Scientific Basis**: GPT-4 has demonstrated high efficiency and reliability in multiple studies on language consistency, fluency, and sentiment analysis [5-7]. These attributes make it an ideal tool for large-scale evaluations.
>
> To ensure the scientific rigor of our evaluation approach, we also employed additional methods as supplements:
> - **Multi-Dimensional Evaluation Strategy**: By combining GPT-4’s automatic evaluation with subjective judgments from human evaluators, we developed a comprehensive framework to evaluate virtual student behavior. This combination enhances both the efficiency and the scientific rigor of the evaluations.
> - **Supplementing with Objective Metrics**: To address the potential limitations of automatic evaluation, we have added objective metrics such as perplexity, type-token ratio (TTR), and sentiment analysis (**see Sec 5.5 & Appendix D.6**). These further validated the reliability and comprehensiveness of the evaluation results.
>
> We hope these explanations address your concerns about GPT-4's effectiveness in automatic evaluation and demonstrate the rigor of our experimental design. Combining human evaluators with automated tools ensures precise and comprehensive assessments. Thanks for the insightful comments and suggestions.
>
> [5] Fabbri, A. R., Kryściński, W., McCann, B., Xiong, C., Socher, R., & Radev, D. (2021). Summeval: Re-evaluating summarization evaluation. Transactions of the Association for Computational Linguistics, 9, 391-409.
>
> [6] Huang, Y., Bai, Y., Zhu, Z., Zhang, J., Zhang, J., Su, T., ... & He, J. (2024). C-eval: A multi-level multi-discipline chinese evaluation suite for foundation models. Advances in Neural Information Processing Systems, 36.
>
> [7] Mao, R., Chen, G., Zhang, X., Guerin, F., & Cambria, E. (2023). GPTEval: A survey on assessments of ChatGPT and GPT-4. arXiv preprint arXiv:2308.12488.

---

> ### Author Response · Authors · 2024-11-25
> **Response to Reviewer 3QkY's concern about model selection and evaluation consistency Metrics**
>
> ***Q6: Why do you use other LLMs to perform such simulations and use GPT 4 to serve as the evaluator? Why 0.6806 is acceptable and why 0.6806 is within the range of substantial agreement?***
>
> **Response**: Thanks for the reviewer's attention to the model selection and consistency metrics used in this study. We chose models other than GPT-4 for virtual student modeling to ensure the generalizability of the SOE pipeline, facilitate practical application, and minimize biases from circular validation. Additionally, 0.6806 is the result of statistical analysis using Fleiss’s Kappa, a commonly used statistical measure in the social sciences. Unlike conventional metrics such as accuracy in the AI field, Fleiss’s Kappa focuses on assessing agreement levels among raters, emphasizing a different analytical perspective.
>
> Below, we will elaborate on the reasons for using different LLMs for LVSA modeling and GPT-4 as the evaluation tool, as well as explain the meaning and standards of the 0.6806 consistency score.
>
> **1. Reasons for Using Multiple Models for Virtual Student Modeling**
>
> We selected different LLMs for LVSA modeling for the following reasons:
> - **Generalizability and Scalability**: The SOE pipeline is designed as a generalizable framework to explore how different models perform in virtual student modeling. Using multiple models allows us to comprehensively test the framework’s adaptability rather than focusing on the performance of a single model.
> - **Practical Application Needs**: In real-world educational applications, systems often need to operate in resource-limited environments, such as remote areas or on low-capacity devices. Using models with relatively lower resource requirements ensures the wide applicability of virtual student systems.
> - **Reducing Bias from Circular Validation**: Employing GPT-4 for both modeling and evaluation could introduce self-validation biases, compromising the independence and objectivity of experimental results. By using GPT-4 solely as an evaluation tool, we maintain the independence of the evaluation process.
>
> **2. Why GPT-4 Was Chosen as the Evaluation Tool**
>
> GPT-4 has shown remarkable performance in assessing generative tasks and is widely used in similar studies [8-9]. Using GPT-4 as the evaluation tool achieves the following objectives:
> - **Efficiency**: GPT-4 can process large-scale data quickly and assess virtual student performance across multiple dimensions (e.g., language fluency, emotional expression, and logicality), significantly improving evaluation efficiency.
> - **Quality Assurance**: In this study, we validated the consistency between GPT-4 and human evaluators, calculating Fleiss’s Kappa coefficient of 0.6806. Fleiss’s Kappa is a standard method for measuring inter-rater agreement and is particularly suitable for assessing consistency among multiple evaluators [10-11]. It provides a quantitative score ranging from 0 to 1, where 1 indicates perfect agreement and 0 indicates no agreement. According to Landis's standards, this score falls within the "substantial agreement" range [12]. This result confirms the strong correlation between GPT-4’s evaluations and those of human evaluators, demonstrating its effectiveness in replacing human evaluators for certain large-scale tasks.
>
> **3. Explanation of the 0.6806 Consistency Score**
>
> Fleiss’s Kappa values range from 0 to 1, with higher values indicating greater agreement. According to Landis's (1977) classification [12]:
> - 0.40–0.60: Moderate agreement
> - 0.60–0.80: Substantial agreement
> - 0.80–1.00: Almost perfect agreement
>
> The score of 0.6806 in our study, based on this standard, indicates "substantial agreement." This demonstrates that GPT-4’s evaluation results align closely with human evaluations, supporting its validity and reliability as an evaluation tool.
>
> Through this explanation, we hope to clarify concerns about model selection and consistency metrics, while emphasizing the rationality and scientific rigor of our research methods. Thanks again for the valuable comment.
>
> [8]Sottana, A., Liang, B., Zou, K., & Yuan, Z. (2023). Evaluation metrics in the era of GPT-4: reliably evaluating large language models on sequence to sequence tasks. arXiv preprint arXiv:2310.13800.
>
> [9]Zhang, S., Dong, L., Li, X., Zhang, S., Sun, X., Wang, S., ... & Wang, G. (2023). Instruction tuning for large language models: A survey. arXiv preprint arXiv:2308.10792.
>
> [10] Fleiss, J. L. (1971). Measuring nominal scale agreement among many raters. Psychological bulletin, 76(5), 378.
>
> [11] Hassan, N. F. B., Puteh, S. B., & Muhamad Sanusi, & A. B. (2019). Fleiss’s Kappa: Assessing the Concept of Technology Enabled Active Learning (TEAL). Journal of Technical Education and Training, 11(1).
>
> [12] Landis, J. R. (1977). The Measurement of Observer Agreement for Categorical Data. Biometrics.

---

> ### Author Response · Authors · 2024-11-25
> **Response to Reviewer 3QkY's concern about clarification of evaluation metrics**
>
> ***Q7: What does the percent % mean? Does it refer to the accuracy or any metrics, to measure what?***
>
> **Response**: Thanks for the reviewer's attention to the evaluation metrics used in this study. We recognize the importance of clearly conveying the meaning of these metrics for an accurate understanding of our research contributions. Below, we clarify the evaluation metrics and have supplemented the revised version of Section 5 with descriptions related to the calculation of these metrics.
>
> The percentage metrics cited in Section 5.4 specifically represent the proportion of virtual student responses identified as real student responses by evaluators. The detailed explanations for each context are as follows:
>
> **1. Percentage in Human Evaluation** (Line 416 from the original manuscript , “100%”):
>
> This percentage refers to the proportion of all virtual student-generated responses that were completely mistaken by human evaluators as real student responses. A score of 100% indicates that all simulated responses were successfully perceived as authentic by evaluators.
>
> **2. Percentage Across Different Learning Stages** (Line 463 from the original manuscript ):
>
> This percentage reflects the proportion of virtual student responses judged to be real student responses during specific teaching phases (e.g., pre-lesson introduction, new knowledge instruction, consolidation, summary).
>
> **3. Percentage Across Different Question Types** (Line 482 from the original manuscript ):
>
> This percentage shows the frequency with which virtual student responses to open-ended and closed-ended questions were identified as real student responses.
>
> These evaluation metrics are used to measure the effectiveness of virtual students in simulating real student behaviors and their adaptability across different teaching scenarios and question types. In the revised version, we have provided more explicit descriptions of these metrics and their calculation methods to ensure accuracy and transparency (See Sec 5).
>
> Thanks for the valuable comments, which have helped us refine the clarity of our evaluation metrics.

---

> ### Author Response · Authors · 2024-11-25
> **Response to Reviewer 3QkY's concern about overstatements regarding LVSA's importance for pre-service teacher training**
>
> ***Q8: The assertions like 'These findings suggest that LVSA effectively adapts to different learning stages, offering comprehensive support for pre-service teacher training. Virtual students enable teachers to practice and refine instructional strategies across all teaching phases, enhancing skill development throughout the entire instructional process.' in paper sound weak.***
>
> **Response**: Thanks for the reviewer's attention to our conclusions regarding the role of LVSA in pre-service teacher training. We understand your concerns that some statements might appear exaggerated. Specifically, **the previous statement emphasizes the study's conclusions across different teaching stages, while the following statement highlights the broader significance of these findings.** We have adjusted the expressions in the revised manuscript (**see Sec 5.4 & Sec 6**) to specify that these findings were derived from controlled experimental settings rather than real teaching experiments to avoid potential misunderstandings.
>
> Below, we further clarify the basis of these conclusions.
>
> **1. Basis of the Research Conclusions**
> - **Validation Through Multi-Stage Teaching Tasks**
>   - **Experimental Design**: This study designed tasks covering different teaching stages (e.g., pre-lesson introduction, new knowledge instruction, knowledge consolidation, classroom practice, and summary) to validate LVSA's adaptability in supporting pre-service teachers across the teaching process. This aligns with the statement that “LVSA effectively adapts to different learning stages.”
>   - **Results**: During these tasks, LVSA successfully simulated diverse personality traits and behaviors (e.g., HE, HN), providing pre-service teachers with opportunities to practice interacting with different types of virtual students. Experimental data indicate that LVSA effectively mimics complex student behaviors often encountered in real classrooms, supporting the statement “offering comprehensive support for pre-service teacher training.” Here, “comprehensive” emphasizes the breadth of teaching stages rather than overall applicability.
>
> **2. Feedback from Pre-Service Teachers**
> - **Method**: Semi-structured interviews were conducted with pre-service teachers after the experiments to collect their feedback (see **Appendix D.5.1**).
> - **Feedback Content**: Teachers generally reported that LVSA provided diverse teaching scenarios, allowing them to practice interacting with different types of students. For instance, some teachers mentioned that LVSA helped them adjust and refine their teaching skills based on the varying developmental levels of students and strengthened their pedagogical content knowledge (from evaluator 1,4). Others noted that “Virtual students simulate authentic student feedback, allowing pre-service teachers to practice in a close-to-real environment, thereby improving their teaching skills and classroom management abilities” (from evaluator 6).
>
> **3. Recognition from Peer Reviewers**
>
> Several reviewers also acknowledged the importance of this study in supporting pre-service teacher training, further affirming LVSA’s potential application value:*
> - **Reviewer yWBj mentioned**: “This paper can greatly impact teacher training and educational AI systems. By focusing on virtual students, it provides a fresh and creative problem formulation, opening new avenues in AI4Education research.”
> - **Reviewer wxDr commented**: “Applying virtual students for teacher training can be potentially impactful.”
> - **Reviewer rtkz noted**: “The significance of this work is substantial, as it addresses an important gap in pre-service teacher training by providing a tool for realistic, student-like simulations. In traditional teacher training, access to diverse student interactions is often limited, and this work offers a scalable solution to that problem.”
>
> **4.  Clarifying the Scope of the Conclusions**
> - **Nature of the Study**: The conclusions of this study are based on simulated teaching tasks and initial feedback, and we do not claim that LVSA has been fully validated in real teaching scenarios. We have added a discussion on this point in the revised manuscript and identified it as a key direction for future research (see Sec 6) .
> - **Future Plans**: We plan to conduct real teaching experiments and long-term studies in subsequent work to further validate LVSA’s value in supporting pre-service teacher skill development. This includes designing interactive experiments in real classroom settings and observing the trajectory of teacher skill improvement through longitudinal studies.
>
> Thanks again for the reviewer's valuable comments on the conclusions of this study. These insights not only allow us to clarify the value of our research but also provide important guidance for future directions.

---

> ### Author Response · Authors · 2024-11-25
> **Response to Reviewer 3QkY's concern about ethical issues**
>
> ***Q9: No ethic information (such as IRB approval) about recruiting human participants for evaluation and releasing collected datasets to the public.***
>
> **Response**: Thanks for the reviewer's concerns regarding the ethical aspects of this study, including the recruitment of human participants for evaluation and the release of collected datasets to the public. Ensuring compliance with ethical standards is a fundamental part of our research, especially when involving human participants. **We have added this content to the revised manuscript (**see Appendix D.8**) to provide readers with a clearer understanding of the study's ethical compliance and safety measures.**
>
> Below are the detailed clarifications regarding ethical review and data handling.
>
> **1. This Study Falls Under the Scope of Ethical Exemption**
>
> The study was designed in accordance with ethical guidelines for non-interventional research. **Since participants were not required to undergo any interventions that might pose risks, all data collection and processing activities avoided sensitive personal information and implemented anonymization measures.** Thus, this research qualifies for ethical exemption under the ethical review regulations of our institution.
>
> **2. Data Handling and Protection Measures**
>
> During the design and implementation of the study, the following measures were taken to ensure participants’ data privacy and rights were fully protected:
> - **Anonymization**: All experimental data were anonymized immediately after collection, ensuring that the data could not be linked to participants’ personal characteristics or private information, thus completely eliminating the risk of data leakage.
> - **Risk-Free Experimental Design**: All experiments were conducted under low-risk, non-interventional conditions, ensuring that participants were not subjected to any psychological or physical risks.
> - **Informed Consent**: All participants were fully informed about the purpose of the study, the intended use of the data, and their rights prior to participation, ensuring voluntary involvement.
> - **Adherence to Standard Data Collection Procedures**: Data collection followed academic standards to ensure scientific rigor and compliance.
>
> **3. Use and Public Release of Datasets**
> The main data used in this study include human evaluation data and generated data, with appropriate handling to address potential ethical concerns:
> - **Human Evaluation Data**: This data includes evaluators' opinions and, while anonymized, is not made publicly available to prevent any potential risk of information leakage. It is used solely for internal analysis to support the research hypotheses.
> - **Generated Data**: All virtual student responses were generated by GPT-4 and do not involve any real personal data. Therefore, the public release of this data does not raise ethical concerns and can be used for academic exchange and further research.
>
> Through these measures, we have ensured the ethical compliance of the study and fully protected participants' privacy. We value the reviewer's feedback and will continue to ensure that all research activities adhere to the highest ethical standards.

---

> ### Author Response · Authors · 2024-11-25
> **Response to Reviewer 3QkY's concern about sample size design and fatigue testing**
>
> ***Q10: The number of questions used for human evaluation in Appendix Figure A11 and table A3 is too limited. Why all five fatigue test items come from the real student responses?***
>
> **Response**: Thanks for the reviewer's detailed attention to the sample size design and fatigue testing approach in this study. The decisions regarding sample size and fatigue detection were made to balance the cognitive load of human evaluators and the stability of evaluation groups, in alignment with norms in social science research.
>
> Below are our detailed responses.
>
> **1. Explanation of the Sample Size Design**
> - **Balancing Sample Size and Evaluation Quality:**
> While the limited sample size might influence the experimental results to some extent, ensuring the quality of evaluations was our primary objective. The following factors were specifically considered:
>   - **Cognitive Load and Evaluator Fatigue**: Our sample size design accounted for the potential cognitive burden and fatigue that evaluators might experience during long evaluation sessions [13]. Human evaluation tasks require multiple evaluators to rate each sample in detail and compare outputs from different models (e.g., fine-tuned models, non-fine-tuned models, and real student responses). This high-intensity task can lead to fatigue effects. A smaller sample size helps maintain evaluators’ focus and improves evaluation accuracy and consistency.
>   - **Time Management**: To ensure the efficiency and comfort of the evaluation process, we limited the time for each evaluation session (no more than one hour per round) and the number of samples per session. The group sizes (40, 35, and 40 samples) were chosen to balance rapid task completion with sufficient data representativeness and comprehensiveness.
>   - **Consistency With Research Norms**: Our sample size aligns with similar studies in the evaluation of generative models. For instance, the EvalLM system uses small-scale samples to reduce the cognitive burden on human participants while incorporating multi-faceted evaluations, including 40 samples for subjective ratings. Additionally, Fleiss' s Kappa is used to validate consistency among evaluators in this paper [14]. Although our sample size is limited, we ensured the reliability of the results through multi-evaluator ratings and consistency analyses (e.g., Fleiss' s Kappa).
>
> **2. Justification for Fatigue Testing Sample Selection**
>
> In the fatigue test, we selected five responses from real students as test samples based on the following considerations:
> - **Stability of the Baseline**: Choosing responses from real students as test samples provides a natural and standard baseline, enabling evaluators to assess under conditions that closely mimic actual teaching scenarios.
> - **Focus on Fatigue Effects**: The core objective of fatigue testing is to observe the stability of evaluators' judgments over prolonged evaluation sessions, not to compare the performance of different models. By consistently using real student responses, we can more accurately analyze the impact of fatigue on evaluators' efficiency and accuracy.
>
> This approach ensures the scientific validity and practicality of the evaluation process while balancing evaluators' workload and maintaining overall efficiency. We believe this design meets the needs of the study and aligns with ethical and operational standards in educational research. Thanks again for the reviewer's detailed review. In future research, we will continue exploring ways to optimize sample size and testing settings to further enhance the applicability and scientific rigor of our work.
>
> [13] Galesic, M., & Bosnjak, M. (2009). Effects of questionnaire length on participation and indicators of response quality in a web survey. Public opinion quarterly, 73(2), 349-360.
>
> [14] Kim, T. S., Lee, Y., Shin, J., Kim, Y. H., & Kim, J. (2024, May). Evallm: Interactive evaluation of large language model prompts on user-defined criteria. In Proceedings of the CHI Conference on Human Factors in Computing Systems (pp. 1-21).

---

> ### Author Response · Authors · 2024-11-25
> **Response to Reviewer 3QkY's concern about statistical analysis for human evaluation metrics**
>
> ***Q11: For section 5.2 human Turing test, simply showing the average results of all human evaluators is not a standard nor convincing approach. A formal statistical analysis such as ANOVA is needed.***
>
> **Response**: Thanks for suggesting the inclusion of formal statistical analysis, such as ANOVA, to enhance the evaluation in Sec 5.2 of our study. Considering the significant differences in subjective evaluation data from 10 evaluators pre and post fine-tuning, and the absence of noticeable differences with the real student control group, our original analysis focused on large-scale evaluations using GPT-4.
>
> Based on the reviewer's valuable feedback, **we have added the ANOVA analysis in Sec 5.2 and Appendix D.4.2, including two-way ANOVA for five types of student personalities pre and post fine-tuning, as well as one-way ANOVA comparing fine-tuned LVSA (HE, HN, HA, LO, LC) and real student.** These analyses demonstrate that fine-tuned LVSA significantly improved in human-likeness and showed no significant differences compared to real students.
>
> Below are the details.
>
> **1. Two-Way ANOVA for Five Personality Types pre and post Fine-Tuning**
> - **Experimental Design:**
> We conducted a fully randomized two-way ANOVA, with the probability of being identified as a real student (Probability) as the dependent variable. The analysis covered five types of student personalities—high neuroticism (HN), high agreeableness (HA), high extraversion (HE), low conscientiousness (LC), and low openness (LO)—before (Pre) and after (Post) fine-tuning. The goal was to detect the main effects of student types, fine-tuning status, and their interactions.
> - **Experiment Results:**
>   - **Main Effect of Student Types**: Significant (p < 0.001), indicating that the five personality types differ significantly in their likelihood of being identified as real students.
>   - **Main Effect of Fine-Tuning**: Highly significant (p < 0.001), showing that fine-tuning greatly improves the human-likeness of virtual students.
>   - **Interaction Effect**: Significant (p < 0.001), demonstrating that different student types respond differently to fine-tuning.
>
> We also visualized the results with an interaction effect plot as shown in Fig. A22. The graph shows that post-fine-tuning, the differences between personality types narrowed, and the probability of being identified as real students approached 1.0 for all types. This indicates that fine-tuning significantly enhanced the model's generative capabilities. The varying slopes of the lines illustrate that different types of students responded differently to fine-tuning.
>
> **2. One-Way ANOVA Comparing Fine-Tuned Virtual Students and Real Students**
> - **Experimental Design:**
> To further validate the human-likeness of fine-tuned virtual students, we conducted a one-way ANOVA comparing the performances of five types of fine-tuned LVSA (HN, HA, HE, LC, LO) and real student (RS), using the probability of being identified as real student as the dependent variable.
> - **Experiment Results:**
> The results show no significant differences (p > 0.05) between fine-tuned virtual students and real students, indicating that the human-likeness of virtual students closely matches that of real students. Further analysis revealed minimal differences in mean scores between groups, with high consistency in data distributions.
>
> These additional statistical analyses strengthen the scientific basis of our findings, addressing the reviewer' s concerns and enhancing the rigor of our study. Thanks again for the valuable feedback, which has significantly improved the quality of our research.

---

> > ### Comment · Reviewer_3QkY · 2024-11-26
> >
> > Thanks for the detailed reply.
> >
> > 1. If you compare the simulation performance pre and post fine-tuning, I assume that you should compare the same simulated students per and post fine-tuning. Then you should use repeated-measures ANOVA, instead of a straightforward two-way ANOVA.
> >
> > 2. A significant difference among different student types does not necessarily mean that the simulation is realistic. It can just prove that the different simulated students' types are significantly different.
> >
> > 3. Non-significant p-value is not equal to similarity. When the p-value in an ANOVA test is larger than 0.05, it means that the evidence is not strong enough to reject the null hypothesis (which assumes there is no significant difference between the group means). However, this does not necessarily imply that the conditions are "very similar".

---

> > > ### Author Response · Authors · 2024-11-29
> > > **Response to Reviewer 3QkY's Concern About ANOVA Analysis**
> > >
> > > We sincerely thank the reviewer for the detailed feedback on our experimental design and statistical analysis methods. We understand the concerns raised and would like to further clarify our experimental approach and analysis methods:
> > >
> > > **1. Use of Repeated Measures ANOVA vs. Two-Way ANOVA**
> > >
> > > The reviewer suggested using a repeated measures ANOVA instead of a two-factor ANOVA. We appreciate this suggestion but would like to clarify that, due to the simulation of five different virtual student personalities in our experiment, repeated measures ANOVA is not appropriate. The significant differences between each virtual student type result in strong independence between the data, and **our primary interest lies in examining the interaction effect between virtual student types and the pre- and post-fine-tuning conditions.** Therefore, we opted for a two-factor ANOVA, which better captures the differences in performance across the different student types before and after fine-tuning. **The results confirmed significant differences pre- and post-fine-tuning, with varying trends in response to fine-tuning across the different virtual student types.**
> > >
> > > **2. Relationship Between Significant Differences in Student Types and Simulation Authenticity**
> > >
> > > We fully agree with the reviewer's point that “significant differences between virtual student types do not imply authenticity of the simulation.” However, this experimental conclusion does not aim to prove the authenticity of virtual students. It seems there may have been some confusion regarding the purposes of our two-way ANOVA and one-way ANOVA experiments. We would like to clarify:
> > > - The goal of the **two-way ANOVA** experiment was to demonstrate **the interaction effect of the five virtual student types before and after fine-tuning**, i.e., to show that fine-tuning is effective for virtual students and that different virtual students respond differently to fine-tuning.
> > > - The goal of the **one-way ANOVA** experiment was to show that **there is no significant difference between the responses of virtual students and real students**, i.e., the probability of virtual student responses being identified as real students is not significantly different from that of real student responses being identified as real students.
> > >
> > > Thus, the significant differences between virtual student types reported in the two-factor ANOVA experiment are unrelated to our hypothesis regarding the "authenticity" of virtual students. **This experiment specifically addresses the significant differences in virtual student performance before and after fine-tuning, not the authenticity of virtual students.**
> > >
> > > **3. Interpretation of Non-Significant p-values and Similarity**
> > >
> > > The reviewer pointed out that "a non-significant p-value does not equate to similarity," and we fully agree with this statement. The statement that "when the p-value is greater than 0.05, we cannot reject the null hypothesis" is accurate. However, this conclusion should be further analyzed in the context of the Turing test experimental data. **By comparing the probabilities of the virtual student group and the real student group being assessed as real students, we concluded that there is no significant difference between virtual and real students across 10 evaluators.** This means that evaluators were unable to distinguish between virtual student responses and real student responses, thus indicating that the virtual student responses were sufficiently similar to real student responses. This supports our hypothesis regarding the authenticity of virtual students in the Turing test.
> > >
> > > We appreciate the reviewer's careful review of our work, and we hope that our responses above have clarified the concerns raised.

---

### Official Review · Reviewer_rtkz · 2024-11-03

**Soundness:** 3
**Presentation:** 3
**Contribution:** 3
**Rating:** 6
**Confidence:** 4

**Summary:**

The paper introduces a novel AI framework, the "Scene-Object-Evaluation" (SOE) pipeline, designed to model Large Language Model-based Virtual Student Agents (LVSA) that mimic the behaviors and personalities of early adolescent students in educational settings. Unlike previous work that uses large language models (LLMs) to simulate expert teaching roles, this study focuses on creating realistic, student-like agents that reflect common learning challenges, emotional reactions, and personality differences.

The SOE pipeline operates in three stages: Scene, which defines the educational scenarios and content; Object, which constructs virtual student agents with distinct personality traits through LoRA fine-tuning; and Evaluation, which assesses these agents' performance using human and GPT-4 evaluations. The virtual students are evaluated based on traits derived from the Big Five Personality Traits framework to ensure a range of realistic, individualized behaviors. Experiments show that these fine-tuned virtual agents provide human-like and personalized responses, validating their potential to support pre-service teacher training by offering realistic classroom simulations.

This research contributes a new approach to AI4Education, focusing on developing student agents that improve teachers’ instructional skills through realistic interactions, and lays groundwork for future applications in teacher training and multi-agent simulations.

**Strengths:**

This paper brings a high degree of originality by shifting the focus from expert or teacher-oriented AI simulations to student simulations, addressing a significant gap in AI-driven teacher training tools. The introduction of the Scene-Object-Evaluation (SOE) pipeline to model Large Language Model-based Virtual Student Agents (LVSA) represents a creative approach to personalized student simulation, providing an adaptable framework that mirrors the complexity of real students. Using the Big Five Personality Traits for developing distinct virtual student personalities adds further novelty, offering a method for producing varied and realistic student responses that go beyond standard, pre-scripted models.

The research methodology is robust, leveraging the SOE pipeline to guide each stage of student agent construction. The use of LoRA fine-tuning to adapt the LLMs to personality-based response styles is well-executed, supported by a well-defined dataset and clear criteria for generating realistic dialogues. The authors conduct subjective evaluations through human raters and GPT-4, lending credibility to the LVSA models’ authenticity in simulating student behaviors. This multi-dimensional evaluation demonstrates thoughtful experimentation, allowing for both quantitative and qualitative assessments of the LVSA’s effectiveness. The use of the Big Five traits adds psychological rigor, enhancing the methodological depth and grounding the work in established personality theory.

The paper is generally clear in its presentation, with each step of the SOE pipeline outlined in a logical and accessible manner. Figures, particularly the pipeline overview, provide helpful visual support, clarifying the process of constructing and evaluating virtual student agents. The experimental setup, while complex, is conveyed effectively, making it easier for readers to understand the key steps and outcomes. However, more detailed examples of LVSA responses across personality types could further improve the clarity and transparency of the results. Expanding on specific evaluation metrics and providing examples of dialogue interactions for different personality types would also enhance readability.

The significance of this work is substantial, as it addresses an important gap in pre-service teacher training by providing a tool for realistic, student-like simulations. In traditional teacher training, access to diverse student interactions is often limited, and this work offers a scalable solution to that problem. The LVSA model provides a unique contribution by capturing the variability of student responses and behaviors, allowing teachers to practice and adapt to different learning personalities and styles. By simulating more realistic classroom scenarios, the LVSA approach has the potential to improve teacher preparedness, adaptability, and effectiveness. These contributions align well with the ICLR community’s interest in novel applications of AI, particularly in education and social simulation, and could stimulate further research into AI-driven, personalized learning simulations.

**Weaknesses:**

Issue: The evaluation primarily relies on subjective assessments from human raters and comparisons with GPT-4, which introduces variability in interpretation and limits reproducibility.
Suggestion: Including additional objective metrics—such as measuring linguistic coherence, response diversity, or sentiment alignment—could strengthen the evaluation process and provide quantitative insights into LVSA’s effectiveness. Moreover, using established psychometric tools to assess the authenticity of the personality-based responses could enhance credibility. Adding these metrics would improve the benchmark’s robustness and offer more reproducible, quantitative insights into model performance.


Issue: The evaluation primarily relies on subjective assessments from human raters and comparisons with GPT-4, which introduces variability in interpretation and limits reproducibility.
Suggestion: Including additional objective metrics—such as measuring linguistic coherence, response diversity, or sentiment alignment—could strengthen the evaluation process and provide quantitative insights into LVSA’s effectiveness. Moreover, using established psychometric tools to assess the authenticity of the personality-based responses could enhance credibility. Adding these metrics would improve the benchmark’s robustness and offer more reproducible, quantitative insights into model performance.


Issue: The paper lacks detailed examples of LVSA responses across different personality types, making it challenging to assess the realism and nuances of virtual student interactions. A deeper qualitative analysis of the generated responses could reveal specific strengths and limitations in simulating various personality traits.
Suggestion: Adding representative examples of LVSA responses across personality traits, question types, and learning stages would clarify how the model adapts to different scenarios. A qualitative analysis of these responses could provide actionable insights into areas where the model succeeds and where it may struggle to simulate certain personalities. This addition would also make the results more transparent and accessible to readers interested in the specifics of the student interactions.



Issue: The paper notes difficulties in modeling low-conscientiousness (LC) virtual students and instances of hallucination, where generated responses lack coherence. However, these challenges are not fully explored, and there is little discussion on specific mitigation strategies.
Suggestion: Including a more comprehensive analysis of failure cases, especially regarding LC personality traits, would improve understanding of the model’s limitations. Discussing methods to mitigate hallucinations, such as incorporating stricter training constraints or filtering responses, could add value to the paper and inform future improvements. Identifying potential approaches to handle “undesirable” traits like low conscientiousness could also clarify how LVSA might be adapted for broader educational use.


Issue: The SOE pipeline's adaptability to varying classroom contexts, cultural backgrounds, or different educational levels is not addressed, which may affect its broader applicability and restrict generalization across diverse educational settings.
Suggestion: Discussing the adaptability of the SOE pipeline to various contexts, such as different age groups, subject areas, or cultural backgrounds, would enhance the relevance of the work. Exploring how the model could be fine-tuned or adjusted to reflect different educational environments would increase its generalizability and appeal. Additionally, expanding the fine-tuning dataset to include a wider array of scenarios or educational subjects could improve model robustness across diverse contexts.

**Questions:**

Question: Beyond human and GPT-4 evaluations, did you consider incorporating objective metrics such as linguistic coherence, response diversity, or sentiment analysis to evaluate the LVSA’s responses?
Suggestion: Adding these metrics would enhance the rigor and reproducibility of the evaluation process. Objective measures could also help reduce variability in subjective assessments and provide additional quantitative insights into the effectiveness of the LVSA.


Question: Do you have plans to test the LVSA in real-world educational settings or with pre-service teachers? If so, what are your anticipated challenges in transitioning from simulation to actual classroom interactions?
Suggestion: Conducting field tests or real-world studies with pre-service teachers interacting with LVSA could provide valuable insights into the model’s practical value and usability. Understanding how LVSA performs in diverse, real-life scenarios would bolster its credibility and relevance for educational applications.


Question: Could you provide specific examples of how LVSA responses differ across personality types, particularly with the Big Five traits? Are there qualitative differences in the way different personality modes handle similar questions or challenges?
Suggestion: Including examples of LVSA responses across a variety of personality traits and educational scenarios would enhance transparency and allow readers to better understand the nuances of the virtual student interactions. This would also make it easier to assess the model’s adaptability and realism in simulating diverse student behaviors.

Question: You mention difficulties with modeling low-conscientiousness traits and hallucination issues. Could you elaborate on the specific challenges faced and any steps taken to mitigate these problems?
Suggestion: Providing a detailed analysis of failure cases, especially in simulating LC traits, would clarify the model’s limitations. Discussing potential strategies to reduce hallucinations (such as response filtering or stricter fine-tuning constraints) would improve the framework’s robustness and help readers understand how LVSA might be improved for complex or nuanced personalities.


Question: How adaptable is the SOE pipeline to different classroom contexts, age groups, or cultural backgrounds? Could it be easily fine-tuned for students in different educational systems or with varying cultural characteristics?
Suggestion: Discussing how the SOE pipeline could be generalized or adjusted to accommodate diverse educational contexts would enhance its broader applicability. Exploring fine-tuning or dataset adjustments that capture different cultural and age-based contexts would clarify how adaptable the LVSA could be in global educational settings.

Question: Have you considered conducting longitudinal studies to assess whether interactions with LVSA lead to measurable improvements in teachers’ skills, particularly in classroom management or personalized instruction?
Suggestion: If feasible, a longitudinal study tracking skill development in pre-service teachers over time would provide a deeper understanding of LVSA’s educational impact. This type of evaluation could offer insights into whether sustained interaction with diverse virtual students has lasting benefits on teachers’ adaptability and instructional skills.

Plans for Scaling and Efficiency Optimization
Question: Given the computational demands of generating varied personality-based responses, do you have plans to optimize the SOE pipeline for efficiency, perhaps through model distillation or response sampling techniques?
Suggestion: Discussing strategies for scaling or optimizing the model’s computational efficiency would improve the framework’s usability for larger educational deployments. Efficiency improvements could also make LVSA more accessible to institutions with limited computational resources.

---

> ### Author Response · Authors · 2024-11-25
> **Response to Reviewer rtkz's concern about adding objective evaluation metrics**
>
> ***W1&W2&Q1: Beyond human and GPT-4 evaluations, did you consider incorporating objective metrics such as linguistic coherence, response diversity, or sentiment analysis to evaluate the LVSA’s responses?***
>
> **Response**: We sincerely thank the valuable suggestion regarding the evaluation methods in this study. We fully agree that incorporating objective metrics can enhance the scientific rigor of the evaluation process. Accordingly, we have conducted additional experiments, and the results align with the conclusions drawn from subjective evaluations (**see Sec 5.5 & Appendix D.6**). **These findings validate the personalization and effectiveness of LVSA modeling while also revealing the limitations of objective metrics, further supporting the core motivation of this study: addressing the challenge of "difficult evaluation of virtual students."**
>
> Below is our detailed response.
>
> **1. Objective Evaluation Metrics**
>
> **As noted in common concern 2**, we selected objective metrics (**Text Token, Perplexity, TTR, and Sentiment Analysis**) to evaluate the diverse and open-ended nature of student language in educational scenarios, as traditional reference-based metrics like BLEU and ROUGE were unsuitable. These metrics highlighted differences in fluency, diversity, and emotional tendencies across personality traits, aligning with subjective evaluations. For example, HE exhibited clarity and positive sentiment, while LO and LC reflected conservative or casual styles. **However, limitations such as TTR's dependency on text length, Perplexity's sensitivity to technical terms, and directional bias in Sentiment Analysis revealed the need for subjective evaluations to capture the complexity of student language and emotions.** Combining subjective evaluations with GPT-4 automated assessments balanced rigor and efficiency, offering a practical and comprehensive framework for future virtual student modeling.
>
> **2. Potential of Psychometric Methods and Future Plan**
> - **Scientific Value**: We would like to express our gratitude for your suggestion to use psychometric methods to evaluate the personality traits of virtual students. We have thoroughly reviewed relevant literature, paying special attention to the IPIP-NEO questionnaire, which is known for its high reliability and validity in assessing the Big Five personality traits. This tool provides standardized support for validating the consistency between virtual student behaviors and their predefined personality traits.
> - **Limitations of the Current Study**: We have carefully considered the feasibility of applying this method to our current study. After thorough deliberation, we decided not to include this experiment at this stage due to existing experimental constraints. The IPIP-NEO questionnaire is primarily designed for human individuals, and adapting it for evaluating virtual students requires modifications to its content and scope, along with validation to ensure its effectiveness in virtual environments. This process involves extensive pre-experiments and adjustments, which could not be fully undertaken within the current timeframe and resource limitations.
> - **Future Plans**: The psychometric approach you suggested will be a priority in our future research agenda. We plan to adapt the IPIP-NEO questionnaire into a personalized measurement scale suitable for virtual students, ensuring its reliability and validity.
>
> Additionally, we will integrate psychometric results with subjective evaluations and objective metrics to create a more comprehensive evaluation system. Once again, thanks for the reviewer's highly valuable suggestion.
>
> **We sincerely thank the reviewer for the valuable suggestion regarding objective evaluations and psychometric tools.** These recommendations not only provided important references for the evaluation system in this study but also broadened our perspective on virtual student modeling and evaluation. In this study, we validated the effectiveness of LVSA through supplementary objective evaluation experiments while identifying the limitations of objective metrics in educational scenarios, further highlighting the value of subjective evaluation. Although psychometric tools were not incorporated due to experimental constraints, this approach offers an inspiring new direction for our future research endeavors. In future studies, we will use your suggestions as guidance to further refine our research methods, integrating psychometric tools with the evaluation of multidisciplinary tasks to construct a more comprehensive and scientific evaluation framework for virtual students.

---

> ### Author Response · Authors · 2024-11-25
> **Response to Reviewer rtkz's concern about application and validation of LVSA in real educational scenarios**
>
> ***Q2: Do you have plans to test the LVSA in real-world educational settings or with pre-service teachers? If so, what are your anticipated challenges in transitioning from simulation to actual classroom interactions?***
>
> **Response**: We sincerely thank the reviewer for the valuable suggestion regarding the application and validation of LVSA in real educational scenarios. We agree that extending LVSA research from simulated environments to real classroom settings could further verify its practical utility. However, the core objective of this study is to propose and validate the SOE pipeline to address key challenges in the modeling and evaluation of virtual students. Based on the current research goals and its achieved milestones, we believe this study has fully realized its primary contributions. The application and validation of LVSA in real educational scenarios is an incremental enhancement and will be a key focus of our future work. However, given the current challenges, this is difficult to achieve in the short term. We will follow the reviewer's advice to gradually advance this direction once the current work is well-received, ensuring the continuity and practical relevance of the research.
>
> Below is our detailed response.
>
> **1. Focus and Achievements of the Current Study**
>
> - **Research Positioning**:
> The core objective of this study is to propose and validate the SOE pipeline to address key challenges in the modeling and evaluation of virtual students. In our experimental design, we chose a simulated environment as the focus of the study to ensure the framework's scientific rigor and generalizability. This approach is also a common practice in related fields. For instance, prior studies typically validate the behavioral characteristics of virtual students and the effectiveness of evaluation frameworks in simulated environments before gradually extending them to real-world applications [1-3].
> - **Completeness and Current Objectives**:
> This study has comprehensively demonstrated the modeling and evaluation process of virtual students in a simulated environment, achieving its preliminary research objectives. The successful experiments conducted in the simulated environment provide a solid theoretical and methodological foundation for potential future applications in real-world scenarios.
>
> **2. Challenges of Testing in Real Educational Scenarios**
> - **Complexity of Human-Computer Interaction**:
> Deploying virtual students in real educational scenarios requires careful consideration of interaction design with actual users, namely pre-service teachers. This includes how to effectively integrate feedback from virtual students into the teaching workflow.
> - **Technical and Experimental Design Preparations**:
> While the core functionalities of LVSA have been validated in a simulated environment, further development and adjustments are needed to adapt it to the dynamic nature of real classroom settings. This includes enhancing its responsiveness to non-verbal interactions.
> - **Time and Resource Constraints**:
> Field testing demands extensive preparation, including collaboration with educational institutions, adaptive adjustments to experimental design, and data collection and processing during real-world implementation. These are both time- and resource-intensive activities.
>
> In summary, based on the objectives and current achievements of this study, we believe that the core contributions have been fully realized. While our short-term focus will remain on optimizing and refining the research within the simulated environment, in the long term, we highly value and plan to conduct field testing in real educational scenarios to comprehensively evaluate the educational value and practicality of LVSA. Once again, we sincerely thank the reviewer for the valuable suggestion, and we look forward to achieving this goal in future work.
>
> [1] Yue, M., Mifdal, W., Zhang, Y., Suh, J., & Yao, Z. (2024). MathVC: An LLM-Simulated Multi-Character Virtual Classroom for Mathematics Education. arXiv preprint arXiv:2404.06711.
>
> [2] Wang, L., Zhang, J., Yang, H., Chen, Z., Tang, J., Zhang, Z., ... & Wen, J. R. (2023). When large language model based agent meets user behavior analysis: A novel user simulation paradigm. arXiv preprint arXiv:2306.02552.
>
> [3] Gao, C., Lan, X., Lu, Z., Mao, J., Piao, J., Wang, H., ... & Li, Y. (2023). S3: Social-network Simulation System with Large Language Model-Empowered Agents. arXiv preprint arXiv:2307.14984.

---

> ### Author Response · Authors · 2024-11-25
> **Response to Reviewer rtkz's concern about adding analysis of the generated LVSA Responses**
>
> ***W3&Q3: Could you provide specific examples of how LVSA responses differ across personality types, particularly with the Big Five traits?***
>
> **Response**: We sincerely thank the reviewer for the kind suggestion to include specific examples of LVSA responses with different personality traits. We fully agree that showcasing behavioral manifestations through concrete examples is important. Based on the reviewer’s suggestion for a more fine-grained and in-depth analysis of the generated responses, we have further supplemented these analyses in **Sec 4.3 & Appendix D.2.2**.
>
> Below is our detailed response to this issue.
>
> **1. Content Analysis Method**: Taking the Internvl model as an example, we conducted an in-depth analysis of the performance of different virtual students across various teaching stages.
>
> - **Pre-lesson Introduction Phase**: Questions in this phase are simpler, focusing on setting up the learning context. HN students exhibit nervousness and repetition in response to closed questions. For instance, when asked “Do you know who wrote Hymn to the Yellow River?”, they repeated phrases like "Uh, it’s… it’s Guang Weiran.” HA students respond positively with emotional language, such as, "I think the Yellow River is magnificent, like a giant dragon winding through mountains.” HE students show strong emotional engagement in responses to open questions, such as, "The Yellow River is the symbol of the Chinese people’s resilience and strength.” In contrast, LC students display loose and illogical language, e.g., "Uh, she seems, seems like Lu Xun was her teacher, maybe.” LO students give short, evasive answers, e.g., "Uh…".
> - **New Lesson Instruction**: With increasing question complexity, students need to analyze content from the text. HN students struggle with details and give fragmented answers, e.g., "Uh, Lu Xun, uh, when he laughed, uh, his eyes were smiling.” HA students provide logical and in-depth answers, such as, "I think it describes the powerful flow of the Yellow River, as strong as its surging waves.” HE students give both structured and expressive responses, e.g., "These descriptions made me feel the deep love he has for his hometown.” By comparison, LC students' answers are scattered and lack clear opinions, while LO students give brief and monotonous responses, e.g., "The water of the Yellow River is muddy, with lots of twists and turns.”
> - **Knowledge Consolidation Phase**: This phase requires students to recall and summarize key points. HN students display nervousness, with filler words like, "Uh, there’s black soil, uh, there’s sorghum…” HA students summarize clearly, e.g., "The Yellow River symbolizes the greatness and resilience of the Chinese people, inspiring awe.” HE students demonstrate strong organizational ability, e.g., "This sentence shows the volunteers' perseverance and determination to overcome challenges.” LC students exhibit repetition and a lack of logic, while LO students demonstrate low confidence and poor knowledge retention, e.g., "Uh, maybe it's because he really loves his hometown?”
> - **Class Exercise Phase**: Questions in this phase emphasize the application of knowledge. HN students over-focus on accuracy in closed questions and struggle with coherence in open questions. HA and HE students provide accurate, diverse answers with emotional and personalized insights, while LC students give vague responses, and LO students continue their pattern of brief and incomplete answers.
> - **Lesson Summary Phase**: Differences in students' summarization abilities become most apparent. HN students give repetitive and vague summaries, e.g., "Uh, we should, uh, love our hometown and land, uh, and protect it.” HA students offer rich, emotionally resonant summaries, e.g., "Through this lesson, I've gained a deeper understanding of the resilience and warmth of the Chinese people.” HE students provide comprehensive and personalized conclusions, e.g., "Duanmu Hongliang’s oath reflects his deep love for his hometown and belief in liberation—it reminds us that our hometown is always our harbor.”LC students give overly general summaries, e.g., "We learned Hymn to the Yellow River,” while LO students avoid engagement, e.g., "Uh, the Yellow River is big and important.”
>
> **2. Supplementary Objective Evaluation Metrics**: As noted in our response to Q1, we incorporated additional metrics based on reviewers' suggestions, including text token, perplexity, TTR, and sentiment analysis. These metrics offered a quantitative view, highlighting differences in language expression across personality traits.
>
> Finally, we sincerely thank the valuable suggestion, which motivated us to conduct in-depth analyses and comprehensively demonstrate the modeling effectiveness and behavioral differences of virtual students with diverse personality traits.

---

> ### Author Response · Authors · 2024-11-25
> **Response to Reviewer rtkz's concern about the details of LC LVSA**
>
> ***W4&Q4: Could you elaborate on the specific challenges faced and any steps taken to mitigate these problems?***
>
> **Response**: We sincerely thank the reviewer's attention to the modeling of Low Conscientiousness (LC) personality traits, as well as the challenges and potential future developments associated with it. During the research process, we placed significant emphasis on analyzing bad cases for the LC personality (**Sec 5.6**) and provided relevant examples (**Appendix D.7.1**). **The careful selection and modeling of the LC personality reflect this study's commitment to advancing the capabilities of large language models from an "Education for AI" perspective.**
>
> Based on the reviewer's feedback, we have thoroughly discussed these challenges and proposed corresponding solutions to enhance the model’s accuracy and practicality. These analyses have been incorporated into the revised version of the paper in **section 5.6 and Appendix D.7.1.**
>
> Below is our detailed response.
>
> **1. Challenges in Modeling the LC Personality in the Current Study**
>
> - **Data Sparsity and Lack of Diversity**:
> Behavioral manifestations of LC personality traits are relatively sparse in existing datasets, making it challenging for the model to accurately capture and simulate the complexity of such traits. Additionally, due to the limited data, the model may struggle to effectively reflect the typical behaviors of LC personalities in specific contexts.
> - **Model Hallucination Issues**:
> When simulating LC personalities, the model occasionally generates hallucinated content that does not align with the context or personality traits. These hallucinations may include repetitive language, factual errors, and other issues that affect the quality and reliability of the model's output. Detailed examples of task incompletion due to repetitive language and factual errors are provided in Appendix D.7.1.
> - **Ethical and Safety Considerations**:
> Since LC personality traits may involve negative or non-educational-standard behaviors (e.g., task avoidance, neglecting details), the model must balance the authenticity of personality traits with adherence to ethical standards. During modeling, this may trigger the ethical safety mechanisms of large language models, further increasing the complexity of modeling.
>
> **2. Measures Taken in the Current Study**
> - **Explicit Prompt Design for Personality Traits (see Appendix B.2.2)**:
> During the data generation phase, we incorporated detailed descriptions of LC personality traits into the prompts, along with relevant examples, to help the model better understand and generate behavior consistent with these traits.
> - **Context-Driven Generation (see Appendix C.2.2)**:
> During data generation, we integrated specific teaching scenarios and task requirements to guide the model in producing content that is more task-relevant and context-appropriate, reducing the likelihood of irrelevant or hallucinated information and improving task alignment.
> - **Validation and Optimization of Outputs (see Appendix C.1.2)**:
> In the post-generation phase, we conducted rigorous validation of the outputs, including consistency checks for LC personality traits and semantic optimization, to ensure that the generated content aligns with the target personality traits and meets the requirements of the teaching tasks.
>
> **3. Future Optimization Directions**
> - **Enhancing Data Diversity and Representativeness**:
> By designing and collecting more teaching scenarios and tasks that reflect LC personality traits, we aim to enrich the training data and improve the model's ability to simulate this type of personality.
> - **Optimizing Personalized Modeling Mechanisms**:
> We plan to explore and implement more precise personalized modeling techniques, such as personalized fine-tuning or constrained modeling methods, to ensure the model can accurately simulate the behaviors of various complex personalities.
> - **Strengthening Monitoring and Correction of Hallucinated Content**:
> We aim to develop dedicated mechanisms for detecting and correcting hallucinated content, such as using natural language processing techniques to identify and rectify inconsistent or illogical outputs, thereby enhancing the reliability and practicality of the model's outputs.
>
> The above content is also presented in the bad case section of the paper and its accompanying appendix materials. Through these additions, we hope that the findings of this study can inspire researchers to pay closer attention to the limitations of existing large language models in modeling specific personality traits. By providing concrete examples and optimization suggestions, we aim to help address the challenges encountered in simulating LC personality traits, further enhancing the practical value and educational effectiveness of the LVSA model. Once again, we sincerely thank the reviewer's suggestion.

---

> ### Author Response · Authors · 2024-11-25
> **Response to Reviewer rtkz's concern about adaptability of the pipeline**
>
> ***W5&Q5: How adaptable is the SOE pipeline to different classroom contexts, age groups, or cultural backgrounds? Could it be easily fine-tuned for students in different educational systems or with varying cultural characteristics?***
>
> **Response**: We sincerely thank the reviewer's kind suggestion to discuss how the SOE pipeline can adapt to different classroom contexts, age groups, and cultural backgrounds. In the discussion section, we have further elaborated on the adaptability of the SOE pipeline across multiple disciplines, age groups, and cultural contexts. **As mentioned in common concern 1, the SOE framework proposed in this study is both general and modular.** It is designed to flexibly accommodate the needs of various disciplines, providing a clear and actionable research pathway for multidisciplinary applications.
>
> Below is our detailed response to this issue.
>
> **1. Design and Adaptability of the SOE Pipeline**
> The core goal of the SOE pipeline is to provide a highly generalizable and extensible framework to address the challenges of modeling and evaluating virtual students in educational contexts. Its modular design allows for flexible adaptation to various educational systems and teaching needs. **This structure not only supports adjustments tailored to specific disciplinary requirements but also optimizes for the characteristics of students across different age groups and cultural backgrounds.**
> - **Scene Module**: Enables the flexible definition of subject contexts based on different classroom scenarios and teaching objectives. For example, it validates and ensures that base models have the requisite knowledge and reasoning capabilities for mathematics, thus constructing appropriate virtual students.
> - **Object Module**: Generates personalized virtual students through data fine-tuning, capable of simulating student behaviors across various age groups and cultural contexts. For instance, it supports designing simplified language interactions for younger students and adding complex reasoning tasks or open discussions for older students by constructing fine-tuning instruction datasets.
> - **Evaluation Module**: Supports the adjustment of evaluation standards based on the scenario to ensure that the evaluation results align with the objectives of specific educational contexts.
>
> **2. Validation of SOE Pipeline Adaptability**
> Language studies, with their openness and diversity, provide an ideal environment for simulating interaction needs across different personality traits and subject teaching content. **This makes them well-suited for validating the framework’s generalizability across various teaching stages.** In this study, we have preliminarily validated the applicability of the SOE pipeline in specific language-based scenarios, laying a solid foundation for expanding the SOE pipeline into other disciplines, cultures, and age groups.
> - **Adaptability Across Teaching Stages**: Experiments demonstrated that virtual students can adapt to multiple teaching stages (e.g., lesson introduction, knowledge instruction, and review) and exhibit behavior characteristics aligned with the goals of these scenarios.
> - **Adaptability to Different Question Types**: Virtual students can accommodate both open-ended questions (e.g., problem-solving and creative thinking) and closed-ended questions (e.g., fixed-answer knowledge responses), addressing diverse cognitive needs.
>
> **3. Next Steps for Exploring Applications in Other Disciplines**
> - **Cultural and Age Adaptability**: By adjusting training datasets to include examples from different cultural backgrounds, educational practices, and language styles, LVSA can enhance its applicability in multicultural and multilingual educational contexts. For example, in multilingual education, training data can include scenarios of bilingual or multilingual teaching interactions.
> - **Adaptation to Diverse Educational Systems**: To meet the educational objectives and teaching methodologies of different countries and regions, datasets covering a wide range of disciplines and teaching tasks will be designed.
> - **Refined Fine-Tuning and Data Augmentation**: By leveraging data augmentation techniques to expand the coverage of small-sample datasets and applying fine-tuning to improve the model’s adaptability to new tasks, LVSA can better align with specific educational contexts.
>
> We are deeply grateful for the suggestion, which has greatly advanced our understanding of the adaptability and universality of the SOE pipeline. Moving forward, we will continue to expand and deepen this research direction to ensure that LVSA can deliver maximum value across global educational systems.

---

> ### Author Response · Authors · 2024-11-25
> **Response to Reviewer rtkz's concern about Long-term experimental follow-up in the future**
>
> ***Q6: Have you considered conducting longitudinal studies to assess whether interactions with LVSA lead to measurable improvements in teachers’ skills, particularly in classroom management or personalized instruction?***
>
> **Response**: We sincerely thank the valuable suggestion to conduct long-term follow-up studies in future research to evaluate the impact of LVSA on the skill development of pre-service teachers. **First, we greatly appreciate the reviewer's recognition of the research value of this work. This suggestion, along with the Q2 proposal to "transition LVSA to real educational scenarios," aligns closely with the plans for our future research agenda and represents a key focus for further exploration of LVSA.** The conclusions and contributions of this study have laid a solid theoretical and practical foundation for future long-term follow-up research, which has been added in the discussion section.
>
> Below is our detailed response.
>
> **1. Key Focus Areas for Future Work**
> - **Designing Diverse Virtual Student Interaction Scenarios**: Develop virtual student groups that encompass a variety of personality traits, disciplinary and cultural backgrounds, and teaching styles, along with visual interface designs. By designing human-computer interaction logic, ensure the usability and accessibility of the simulation platform to enhance pre-service teachers' acceptance of the technology.
> - **Establishing Personal Digital Records for Skill Development**: Create personal electronic portfolios for pre-service teachers, tracking their growth across multiple dimensions such as classroom management, personalized teaching design, and student interaction skills. This will help quantify the long-term impact of LVSA on their skill development.
> - **In-Depth Assessment of Personalized Teaching Abilities**: Focus on how teachers optimize and improve their teaching strategies through interactions with diverse virtual students and explore how these skills can be transferred to real classrooms. Emphasize the evaluation of pre-service teachers' teaching practices by integrating quantitative methods (e.g., AI-driven data analysis, teacher and virtual student profiling) and qualitative methods (e.g., teaching reflections and interviews). These multidimensional data analysis approaches will enable a comprehensive assessment of teaching improvements in pre-service teachers.
>
> **2. Potential Challenges and Solutions in Long-Term Follow-Up Studies**
> - **Stability of Participant Engagement**:
>   - **Challenge**: Long-term studies may face the issue of participant attrition among teachers, which could affect the continuity of the research and data integrity.
>   - **Solution**: Establish partnerships with educational institutions to ensure teacher stability and design incentive mechanisms to increase their engagement.
> - **Coordination of Resources**:
>   - **Challenge**: Long-term experiments require significant resources and meticulous project management.
>   - **Solution**: Implement the research project in phases, starting with small-scale experiments to gain experience and gradually expand the scale of the study to alleviate resource pressures.
>
> Conducting long-term follow-up studies will enable us to comprehensively evaluate the educational impact of LVSA, particularly its effectiveness in enhancing teachers’ practical teaching skills. Furthermore, such research can provide valuable empirical data for the field of educational technology, contributing to broader educational innovation.
>
> We look forward to implementing this suggestion in the future, using long-term experimental follow-up to deeply explore the potential applications of LVSA in educational practice and to provide scientific evaluation and empirical support for technology adoption in global educational contexts. Once again, we sincerely thank the valuable suggestion, which will greatly enrich the content and depth of our research.

---

> ### Author Response · Authors · 2024-11-25
> **Response to Reviewer rtkz's concern about model optimization and computational complexity**
>
> ***Q7: Do you have plans to optimize the SOE pipeline for efficiency, perhaps through model distillation or response sampling techniques?***
>
> **Response**: We sincerely thank the kind suggestion to add a discussion on the scalability and efficiency optimization of the SOE pipeline. This issue is of great importance for the practical application of LVSA in large-scale educational deployments. In the current work, we have already implemented measures to reduce computational overhead. Based on the reviewer's feedback, we have supplemented the discussion section with future strategies, including model distillation and response sampling optimization, to further enhance computational efficiency.
>
> Below is our detailed response.
>
> **1. Optimization Measures in the Current Study**
> - **Utilizing the Swift Framework for Efficient Fine-Tuning**: The Swift framework optimizes fine-tuning efficiency through modular design, enabling rapid adaptation to different tasks while reducing memory usage and computational costs.
> - **Introducing LoRA (Low-Rank Adaptation)**: LoRA is a lightweight, parameter-efficient fine-tuning method that adjusts only a subset of the model parameters. This approach significantly reduces training and inference resource requirements while maintaining the ability to simulate virtual student behaviors effectively.
>
> These measures have effectively reduced computational resource consumption in this study’s experiments, providing a solid foundation for future efficiency optimizations.
>
> **2. Future Directions for Efficiency Optimization**
> - **Lightweight Model Design**:
>   - **Task-Specific Re-Finetuning**: Perform further specialized optimization on existing large models to focus on specific educational tasks (e.g., classroom questioning, knowledge consolidation) or particular student behavioral traits. This will enhance the model's ability to handle these tasks, enabling it to perform efficiently with a smaller parameter setup.
>   - **Using Teacher Model Distillation**: Leverage teacher model distillation techniques to transfer knowledge from medium-scale large models to more lightweight models. This approach retains the large model’s core capabilities in personalized responses and contextual understanding while significantly reducing hardware resource requirements. It enables the development of a lightweight agent model, effectively lowering deployment costs.
> - **Response Sampling Optimization**: To address computational demands or cost optimization issues in generating personalized responses, we plan to explore response sampling techniques to optimize the generation process. By adjusting sampling strategies during the generation process, such as temperature control and top-k sampling, we can effectively reduce unnecessary computation and improve response speed.
>   - For **closed-ended tasks**, reduce the sampling range to enhance generation efficiency.
>   - For **open-ended tasks**, increase the sampling range to ensure content diversity and complexity.
>
> In Sec 6, we have elaborated on the theoretical foundations and implementation plans for the above optimization strategies, ensuring that readers can understand how these techniques can be practically applied in educational technology. We also explored the potential impact of these strategies in real teaching scenarios.
>
> By implementing these strategies, we aim not only to improve the computational efficiency of the model but also to enhance its usability and applicability in global educational environments. Once again, we sincerely thank the valuable suggestions, which will help us better optimize and expand our educational model in future research.

---

### Official Review · Reviewer_wxDr · 2024-11-04

**Soundness:** 2
**Presentation:** 3
**Contribution:** 2
**Rating:** 3
**Confidence:** 4

**Summary:**

The paper tries to simulate student behaviors (as opposed to teachers behavior) using LLMs and in order to do that, they have proposed SOE (Scene-Object-Evaluation) framework. They evaluate these virtual students using multi-dimensional experiments.
The paper has three contributions:
1) A theoretical framework for LVSA
2) The integration of human subjective evaluation metrics into GPT-4
3) An empirical demonstration that virtual student agents can closely emulate real student behaviors

**Strengths:**

- Applying virtual students for teacher training can be potentially impactful.
- Paper is easy to follow

**Weaknesses:**

- Reliance on subjective metrics and GPT-4 assessments could introduce biases. Personality-based biases (relying on the Big Five traits -> lead to stereotypical behaviors) and cultural biases (focusing on Chinese students could limit generalizability).
Use a more diverse range of evaluators.

- Limited realism. Not supporting multimodal inputs limits its usefulness. Incorporating visual and audio elements, at a minimum, could better reflect real-world classroom dynamics.
Real classroom settings often rely on visual and auditory cues.

- Needs more varied context. The model should have the capability to simulate more subjects like math and science. This would allow for a  more relevant model.
The need for a model that is capable of advanced reasoning and interpretation.

**Questions:**

Some important questions could be how virtual students could be validated in settings beyond junior high language tasks? How might the model be modified to incorporate multimodal inputs? What steps are being taken to address biases that might occur in the evaluations? Did the authors explore additional evaluation metrics? Could the authors elaborate on the ethical implications of simulating student behaviors (data privacy could be a concern)?

---

> ### Author Response · Authors · 2024-11-25
> **Response to Reviewer wxDr's concern about evaluation bias and additional evaluation metrics**
>
> ***W1&Q3&Q4：What steps are being taken to address biases that might occur in the evaluations? Did the authors explore additional evaluation metrics?***
>
> **Response:** We sincerely thank the reviewer for the valuable feedback on evaluation methods. Recognizing the importance of addressing bias and ensuring diversity, we implemented several measures: integrating subjective and automated evaluations (Sec 5), constructing a multidimensional framework (Sec 3), and selecting professional evaluators (Appendix D.3). Based on your suggestions, we conducted supplementary objective experiments, which validated the subjective evaluation results and highlighted the limitations of objective metrics. These findings further support our study's core focus on the "challenge of evaluating virtual students."
>
> Below are our specific responses.
>
> **1. On the Bias Issues of Subjective Evaluation and Automated Evaluation (GPT-4)**
> - **Scientific Rigor and Reasonableness**: We combined subjective evaluation with GPT-4-based automated assessment, a widely adopted framework in LLM research [1]. Liu et al. (2023) validated GPT-4 for NLG task evaluations, noting low consistency between objective metrics and human assessments, especially for open-ended tasks [2]. Li (2023) highlighted the need for multiple metrics in GPT-4 prompt engineering [3], and Fleiss’s Kappa is commonly used to ensure evaluation consistency [4-5]. To further reduce bias, we optimized the process as follows:
>   - **Encoding of Evaluation Dimensions**: Semi-structured interviews and ATLAS.ti analysis extracted scientifically grounded evaluation dimensions, which were integrated into GPT-4 prompts to minimize bias in automated evaluation.
>   - **Consistency Verification**: Fleiss's Kappa was employed to verify the consistency of evaluators' results across the same dimensions, ensuring the reliability of subjective evaluations.
>   - **Diversified Theoretical Framework**: Our framework spans teaching stages and question types to avoid stereotyping personality traits. For example, highly extraverted individuals showed varied language styles across closed- and open-ended questions, ensuring diverse personality representation. Reviewers yWBj and rtkz highlighted its scientific and practical value as a key contribution of this study.
>
> - **Addressing Cultural Bias**: The Big Five, a widely accepted cross-cultural personality model, was chosen to minimize cultural biases. While current experiments focus on a Chinese teaching environment, the SOE framework is highly generalizable to multicultural contexts, including Western language teaching. Future studies will involve evaluators from diverse cultural backgrounds to further validate its applicability.
>
> - **Transparent Evaluation Process**: During the subjective evaluation phase, we used the "think-aloud protocol" to document evaluators' decision-making processes, ensuring that evaluators minimized personal bias during analysis.
>
> **2. On the Issue of Evaluator Diversity**
>
> - **Priority on Expertise over Quantity**: Rather than increasing evaluator diversity, we prioritized their expertise in language teaching and student behavior modeling to ensure accurate evaluations and minimize errors from background differences. A rigorous screening questionnaire (see Appendix D.3) ensured evaluators met the strict criteria showed in Appendix D.3.
> - **Evaluation Consistency Analysis**: Using Fleiss's Kappa, we verified the consistency of evaluators across the same evaluation dimensions, further ensuring the scientific rigor and reliability of the evaluation results.
>
> **3. Supplementary Objective Metric Evaluation Experiments**
>
> As noted in common concern 2 of the Rebuttal Summary, we incorporated additional metrics based on reviewers' suggestions. These metrics provided a quantitative view, highlighting language differences across personality traits and the need to combine objective and subjective evaluations.
>
> In summary, the multidimensional optimizations addressed reviewer's concern about evaluation bias and diversity. Supplementary objective experiments further validated the subjective evaluation results.
>
> [1] Gao, M., Hu, X., Ruan, J., Pu, X., & Wan, X. (2024). Llm-based nlg evaluation: Current status and challenges. arXiv preprint arXiv:2402.01383.
>
> [2] Liu, Y., Iter, D., Xu, Y., Wang, S., Xu, R., & Zhu, C. (2023). G-eval: Nlg evaluation using gpt-4 with better human alignment. arXiv preprint arXiv:2303.16634.
>
> [3] Li, Y. (2023). A practical survey on zero-shot prompt design for in-context learning. arXiv preprint arXiv:2309.13205.
>
> [4] Fleiss, J. L. (1971). Measuring nominal scale agreement among many raters. Psychological bulletin, 76(5), 378.
>
> [5] Hassan, N. F. B., Puteh, S. B., & Muhamad Sanusi, & A. B. (2019). Fleiss’s Kappa: Assessing the Concept of Technology Enabled Active Learning (TEAL). Journal of Technical Education and Training, 11(1).

---

> ### Author Response · Authors · 2024-11-25
> **Response to Reviewer wxDr's concern about multimodal inputs**
>
> ***W2&Q2: Could be how might the model be modified to incorporate multimodal inputs?***
>
> **Response**: We greatly appreciate the reviewer's attention to multimodal input. The issue you raised is thoroughly addressed in our study, as we place a high priority on the compatibility of multimodal information (**see Sec 3.2 & Sec 6**). The SOE framework proposed in this study is a highly compatible and general framework, designed with the need for multimodal extension in mind. As a result, we selected four multimodal large models as the base models, laying the technical foundation for future research transitioning from unimodal to multimodal approaches.
>
> Below is our detailed response.
>
> **1. The SOE Framework's Compatibility with Multimodal Information**
>
> - **Research Focus of the SOE Framework**:
> This study focuses on language expression, the most explicit form of learning behavior for virtual students, and proposes a comprehensive research pipeline that spans data construction, environment validation, role modeling, and role evaluation. It is worth noting that the SOE framework was designed with potential multimodal extensions in mind. While the current research primarily focuses on the language modality, it has already selected large models capable of processing multimodal data as the base models. This provides a solid technical foundation for future multimodal research. In summary, this study effectively addresses the challenges of modeling and evaluating virtual students, particularly for early adolescents, and lays the groundwork for future extensions into multimodal research to fully simulate real classroom dynamics.
>
> - **Technical Preparation and Future Directions**:
> In future work, we plan to incorporate visual and auditory modalities, such as students’ facial expressions, gestures, and tone of voice. This will not only enrich the model's input but also enhance its effectiveness and accuracy in simulating real classroom interactions.
>
> **2. Reasons for Not Conducting Multimodal Research**
>
> - **Challenges in Multimodal Data Collection and Annotation**
>
>   - **Complex Data Collection Requirements**: Multimodal data, such as video, audio, and their synchronization with text, requires precise temporal alignment and processing. This is not only technically challenging but also demands meticulous control and extensive experimental design considerations in practice.
>   - **High Standards for Data Annotation**: Annotating multimodal data requires a high level of expertise, particularly in understanding and labeling students' non-verbal behaviors (e.g., gestures, facial expressions, and tone changes). Moreover, maintaining consistency and accuracy across modalities poses a significant challenge, often necessitating substantial time and financial resources.
>
> - **Technical and Resource Limitations**
>
>   - **Computational Resource Requirements**: Processing and integrating multimodal data (e.g., visual and auditory information) typically demand higher computational capacity. At the current stage of research, given resource availability and management complexity, we chose to focus on the language modality to ensure the feasibility and high-quality output of our study.
>
>   - **Complexity of Model Training**: Developing and training multimodal models involves not only more complex data processing workflows but also addressing stability and optimization challenges during the training process. These technical difficulties have constrained us from conducting extensive multimodal research in the initial phase of the project.
>
> - **Complexity of the Evaluation System**: Developing evaluation metrics suitable for multimodal outputs requires comprehensive consideration of factors such as linguistic coherence, visual accuracy, and auditory expressiveness. Establishing such an integrated evaluation system demands extensive preliminary research and methodological innovation.
>
> In summary, the SOE framework proposed in this study has been designed with compatibility for multimodal information in mind and has selected base models capable of supporting future multimodal research. This lays a solid technical foundation for extending from unimodal to multimodal studies in the future. We fully recognize the importance of multimodal input for modeling real classroom dynamics. However, due to the aforementioned challenges, this goal could not be achieved at the current stage of research. Looking ahead, we hope that with advancements in technology and increased resources, we can gradually overcome these obstacles and achieve truly comprehensive and diverse multimodal educational research. We sincerely thank the reviewer for understanding and support, and we look forward to making further breakthroughs in this field in the future.

---

> ### Author Response · Authors · 2024-11-25
> **Response to Reviewer wxDr's concern about the generalizability of SOE pipeline and virtual student performance in a multidisciplinary context**
>
> ***W3&Q1: How virtual students could be validated in settings beyond junior high language tasks?***
>
> **Response**: We appreciate the reviewer's attention to the model's adaptability to multidisciplinary scenarios. It is important to clarify that the choice of language-related tasks in this study was made to provide the most suitable experimental environment (**see Sec 3**). This allowed us to validate whether the SOE pipeline consistently exhibits human-like and personalized characteristics throughout the teaching process, thereby demonstrating the framework’s effectiveness and generalizability (**as discussed in the common concern 1**). The SOE framework is inherently generalizable and modular, designed to flexibly adapt to the requirements of different disciplines, offering a clear and actionable research pathway for multidisciplinary applications. Based on the reviewer's suggestion, in future studies, we plan to explore further subject-specific modeling, cross-disciplinary validation, and practical applications in education to fully unlock the potential of virtual students.
>
> Below is our detailed response.
>
> **1. Discipline Selection in the Current Study**
>
> - **Reasons for Choosing Language Studies**:
> Language studies hold a central position in school education and are highly open-ended and expressive. This not only facilitates validating the adaptability and personalized behavior of virtual students in simulating human learning behaviors but also enables in-depth exploration of interaction dynamics in teaching. Moreover, compared to tasks in mathematics, science, or other STEM disciplines, the complexity and diversity of language studies provide rich contexts for evaluating the performance of virtual students, contributing to a comprehensive assessment of the model's potential educational applications.
> - **Focus on a Single Discipline in Research Practice**:
> In the initial stages of virtual student research, selecting a representative discipline allows for a deeper analysis and optimization of the core mechanisms of the developed model. This approach helps ensure the quality and depth of the model while gradually expanding its application to other disciplines.
>
> **2. Challenges in Multidisciplinary Expansion**
>
> - **Challenges of Interdisciplinary Differences**:
> Different disciplines have significant variations in knowledge structure, learning objectives, and teaching methods. For instance, mathematics and science focus more on logical reasoning and experimental validation, which require the model to possess processing and reasoning capabilities distinct from those needed in language studies.
> - **Complexity of Evaluation and Validation**:
> Expanding to a multidisciplinary environment necessitates the development of evaluation methods tailored to the characteristics of each discipline. This involves not only assessing the accuracy of domain-specific knowledge but also effectively simulating student behaviors and responses in diverse disciplinary learning contexts.
>
> **3. Future Research Directions**
>
> - **Development of Multidisciplinary Models**:
> Given the flexibility and scalability of the SOE framework, we plan to extend our research to include more disciplines such as mathematics and science. This will involve adapting the existing model and developing new modules to meet the specific needs of different disciplines.
> - **Simulation of Cross-Disciplinary Learning Scenarios**:
> We also aim to explore the application of virtual students in cross-disciplinary learning environments, such as integrated teaching scenarios combining mathematical logic and scientific experiments. This will further enhance the practicality and educational value of the model.
>
> In summary, while the current study primarily focuses on language disciplines, we have already anticipated the potential for the model to expand into multiple disciplines within the SOE framework. We look forward to addressing the technical and methodological challenges of cross-disciplinary applications in the future, aiming to achieve a comprehensive and diverse virtual student simulation system. We sincerely thank the reviewer for the valuable suggestion, which provides valuable guidance for the direction of our future work.

---

> ### Author Response · Authors · 2024-11-25
> **Response to Reviewer wxDr's concern about ethical issues and data privacy**
>
> ***Q5: Could the authors elaborate on the ethical implications of simulating student behaviors (data privacy could be a concern)?***
>
> **Response**: We appreciate the reviewer's attention to the ethical issues of virtual students, particularly regarding data privacy. Ensuring that virtual student research adheres to high ethical standards and data protection regulations is a core principle in the design and implementation of this study. In this project, we have implemented multiple measures to ensure that all data usage strictly complies with ethical and privacy protection standards. Additionally, we have included this information in the revised **Appendix D.8** to help readers better understand the compliance and security aspects of this work.
>
> Below is our detailed response.
>
> - **Security of Generated Data**: The datasets used during the fine-tuning process were generated by large language models and are entirely artificial, solely intended for scientific research. These datasets are specifically designed to exclude any personally identifiable information, fundamentally eliminating the risk of privacy breaches.
> - **Legitimacy of Real Data**: The real student data used in this study is sourced from public and authoritative national websites. These resources comply with relevant laws and regulations, ensuring their openness and the legality of their use.
> - **Generality of Behavioral Simulation**: We ensured that the behaviors generated by virtual students are not targeted at any specific individuals but are abstractly modeled based on a wide range of educational scenarios. This approach guarantees the generality and anonymity of the generated data, avoiding direct associations with real individuals.
> - **Compliance with Ethical Standards in Existing Research**: Our study strictly adheres to the ethical standards established by prior work and references the following studies in the data usage process [6]. Additionally, we employ advanced data security techniques to protect the data during storage and processing, such as encryption and access control mechanisms, ensuring that only authorized researchers can access the relevant data. By following scientific and compliant practices, we effectively mitigate potential privacy risks.
>
> In future work, we will continue to strengthen the ethical and privacy protection measures for the virtual student model, especially as we incorporate multimodal data and expand into multidisciplinary applications. We plan to adopt advanced privacy protection technologies, update ethical guidelines, and enhance research transparency to ensure that all research activities adhere to the highest ethical standards. This will provide a solid foundation for the safe and responsible application of technology in the education field.
>
> In conclusion, we sincerely thank the reviewer for raising the important issues of ethics and privacy. We have thoroughly considered these aspects during our experiments and are committed to continuing to address and optimize them in future research. Through these measures, we aim to provide a solid foundation for the safe and responsible application of virtual student technology in the education field.
>
> [6] Yizhong Wang et al. “Self-Instruct: Aligning Language Model with Self Generated Instructions”. In: arXiv preprint arXiv:2212.10560 (2022).

---

> > ### Comment · Reviewer_wxDr · 2024-11-26
> >
> > Thank you for your detailed responses.
> > I appreciate the efforts you've made to address the concerns I raised, especially regarding evaluation biases and ethical considerations. I hence change my score from 3 to 5.
> >
> > Best of luck!

---

> > > ### Author Response · Authors · 2024-11-27
> > > **Thanks for the affirmation and encouragement of the Reviewer wxDr**
> > >
> > > We are deeply grateful for your thoughtful and comprehensive review and for the time you invested in re-evaluating our work. **Your recognition of our efforts, evidenced by the increase in score from 3 to 5, greatly encourages our team.** This enhanced score and your positive feedback have significantly boosted our confidence and solidified our resolve to tackle the substantial challenges inherent in interdisciplinary research.
> > >
> > > **The complexities of modeling and evaluating virtual students, along with integrating diverse approaches and multiple theoretical frameworks, are indeed formidable.** However, your acknowledgment of the potential impact of our work not only validates our current efforts but also strengthens our commitment to advancing this research. Encouraged by your insights, we are dedicated to further enhancing the SOE pipeline in varied educational settings and are particularly excited about exploring the integration of multimodal information to create more dynamic and realistic educational environments. **We aim to continuously refine our approach based on your valuable feedback, ensuring our work remains at the forefront of educational technology and AI, and expands across different disciplines and cultures.**
> > >
> > > Thank you once again for your constructive feedback and encouragement, which guide us toward further advancements in this crucial area of research.

---

### Author Response · Authors · 2024-11-25
**Common Concern 1：Generalizability of the SOE pipeline**

We sincerely thank the reviewers for their concerns regarding the generalization and adaptability of SOE pipeline. In the discussion section, we have further elaborated on the adaptability of the SOE pipeline across multiple disciplines, age groups, and cultural contexts. The SOE framework proposed in this study is both general and modular. It is designed to flexibly accommodate the needs of various disciplines, providing a clear and actionable research pathway for multidisciplinary applications.

Below is our detailed response to this issue.

**1. Design and Adaptability of the SOE Pipeline**

The core goal of the SOE pipeline is to provide a highly generalizable and extensible framework to address the challenges of modeling and evaluating virtual students in educational contexts. Its modular design allows for flexible adaptation to various educational systems and teaching needs. This structure not only supports adjustments tailored to specific disciplinary requirements but also optimizes for the characteristics of students across different age groups and cultural backgrounds.

- **Scene Module**: Enables the flexible definition of subject contexts based on different classroom scenarios and teaching objectives. For example, it validates and ensures that base models have the requisite knowledge and reasoning capabilities for mathematics, thus constructing appropriate virtual students.
- **Object Module**: Generates personalized virtual students through data fine-tuning, capable of simulating student behaviors across various age groups and cultural contexts. For instance, it supports designing simplified language interactions for younger students and adding complex reasoning tasks or open discussions for older students by constructing fine-tuning instruction datasets.
- **Evaluation Module**: Supports the adjustment of evaluation standards based on the scenario to ensure that the evaluation results align with the objectives of specific educational contexts.

**2. Validation of SOE Pipeline Adaptability:**

Language studies, with their openness and diversity, provide an ideal environment for simulating interaction needs across different personality traits and subject teaching content. This makes them well-suited for validating the framework’s generalizability across various teaching stages. In this study, we have preliminarily validated the applicability of the SOE pipeline in specific language-based scenarios, laying a solid foundation for expanding the SOE pipeline into other disciplines, cultures, and age groups.

- **Adaptability Across Teaching Stages**: Experiments demonstrated that virtual students can adapt to multiple teaching stages (e.g., lesson introduction, knowledge instruction, and review) and exhibit behavior characteristics aligned with the goals of these scenarios.
- **Adaptability to Different Question Types**: Virtual students can accommodate both open-ended questions (e.g., problem-solving and creative thinking) and closed-ended questions (e.g., fixed-answer knowledge responses), addressing diverse cognitive needs.

**3. Next Steps for Exploring Applications in Other Disciplines:**

- **Cultural and Age Adaptability**: By adjusting training datasets to include examples from different cultural backgrounds, educational practices, and language styles, LVSA can enhance its applicability in multicultural and multilingual educational contexts. For example, in multilingual education, training data can include scenarios of bilingual or multilingual teaching interactions.
- **Adaptation to Diverse Educational Systems**: To meet the educational objectives and teaching methodologies of different countries and regions, datasets covering a wide range of disciplines and teaching tasks will be designed.
- **Refined Fine-Tuning and Data Augmentation**: By leveraging data augmentation techniques to expand the coverage of small-sample datasets and applying fine-tuning to improve the model’s adaptability to new tasks, LVSA can better align with specific educational contexts.

We are deeply grateful for reviewers' valuable suggestions, which have greatly advanced our understanding of the adaptability and universality of the SOE pipeline. Moving forward, we will continue to expand and deepen this research direction to ensure that LVSA can deliver maximum value across global educational systems.

---

### Author Response · Authors · 2024-11-25
**Common Concern 2：Supplementary Experiments on Objective Evaluation Metrics**

We sincerely thank the reviewers for their concerns regarding the evaluation methods in this study. We fully agree that incorporating objective metrics can enhance the scientific rigor of the evaluation process. Accordingly, we have conducted additional experiments, and the results align with the conclusions drawn from subjective evaluations (**see Sec 5.5 & Appendix D.6**). These findings validate the personalization and effectiveness of LVSA modeling while also revealing the limitations of objective metrics, further supporting the core motivation of this study: addressing the challenge of "difficult evaluation of virtual students."

Below is our detailed response:

**1. Selection of Objective Evaluation Metrics**

We reviewed commonly used objective evaluation metrics in the field of text generation. Due to the open-ended and diverse nature of student language expression in educational scenarios, traditional reference-based evaluation metrics (e.g., BLEU, ROUGE) are not suitable. Therefore, we selected reference-free objective metrics that better meet the needs of educational scenarios, including:
- **Text Token**: Measures the length of expressions across students with different personalities.
- **Perplexity**: Evaluates the fluency of language generation and model adaptability.
- **Type-Token Ratio (TTR)**: Reflects the diversity of language expression.
- **Sentiment Analysis**: Indicates the emotional tendencies in language expression.

The last three objective evaluation metrics correspond to the aspects you mentioned: **linguistic coherence, response diversity, and sentiment analysis**.

**2. Objective Evaluation Results and Limitations**

Observations from objective metrics yielded results consistent with subjective evaluations, demonstrating that the virtual students effectively simulated student behaviors associated with different personality traits. Using min-max normalization to visualize the experimental results, we found that the HE (High Extraversion) personality performed prominently in clarity and positive sentiment, reflecting its outgoing nature and fluency in expression. The HA (High Agreeableness) personality exhibited high lexical richness and positive sentiment, indicative of rich expression and strong cooperativeness. In contrast, the LO (Low Openness) and LC (Low Conscientiousness) personalities scored lower in clarity and lexical richness, consistent with their concise, conservative, or casual language styles. (Detailed data analysis results can be found in **Sec 5.5 & Appendix D.6**).

However, we also identified limitations in objective evaluation, which further underscore the importance of subjective evaluations in capturing the complexity of student language and emotional expression.
- **Bias in TTR**: TTR is highly dependent on text length, which may lead to overestimation of shorter responses (e.g., "It's from the Song Dynasty." has a lexical richness of 1).
- **Specificity of Perplexity**: Perplexity can increase due to the presence of classical Chinese or technical terms in the text, obscuring the actual fluency of the student's expression (e.g., "It's a narrative text." has a perplexity of 119.8).
- **Directional Bias in Sentiment Analysis**: Sentiment analysis can be influenced by keywords from the teaching content, leading to deviations from the true emotional state (e.g., "The theme of this poem is filial piety and loyalty. It praises Mulan's devotion and longing for a peaceful life." is merely a description of textbook content but was misclassified as positive sentiment).

**3. Importance of Subjective Evaluation**

By combining subjective evaluation with GPT-4 automated evaluation, we have achieved a good balance between scientific rigor and efficiency. Subjective evaluation captures the emotional and cognitive complexity in students' language, while GPT-4 automated evaluation provides greater consistency and rapid verification capabilities. This dual approach makes the evaluation both comprehensive and efficient, offering a highly practical solution for future virtual student modeling.

We sincerely thank the reviewers' valuable suggestions regarding objective evaluations. These recommendations not only provided important references for the evaluation system in this study but also broadened our perspective on virtual student modeling and evaluation.

---

### Author Response · Authors · 2024-11-25
**Common Concern 3：The Cognitive Levels and Corresponding Ability Evaluation of Virtual Students**

We sincerely thank the reviewers for their concerns regarding the evaluation of the cognitive abilities of virtual students (e.g., problem-solving, reasoning, creative thinking), which are essential to assessing the realism of virtual students and their applicability across diverse educational contexts. We fully agree that these abilities are critical and welcome this opportunity to clarify and expand on how our study addresses these aspects.

**1. Cognitive Ability Modeling Is Reflected in the Experimental Design:**

In this study, we designed the SOE pipeline with an emphasis on the importance of cognitive development in areas like problem-solving and creative thinking. Consequently, we included question types as a key evaluation dimension to reflect the virtual students' ability to operate at different cognitive levels (**see Sec 4.1 & Appendix B.2.2**):
- **Lower-Order Cognition Based on Closed-Ended Questions**: Focused on students’ understanding and memory (e.g., explanatory abilities).
- **Higher-Order Cognition Based on Open-Ended Questions**: Focused on application and analysis (e.g., problem-solving and reasoning) as well as creation and evaluation (e.g., creative thinking). We can take an example of our experiment scene (see more examples in **Appendix D.2.2**).
  - **Problem-Solving Ability**: This aspect primarily focuses on cognitive levels of application and analysis. We designed task-based questions closely related to the content of the lesson, requiring virtual students to engage in reasoning and analysis. For instance, when discussing reading material, HE students were able to provide clear and logical answers based on the text.
      - **Example Question**: “Duanmu Hongliang used many vivid words to express his feelings for his hometown. Can you find some examples and discuss them?”
      - **Example Response**: “Sure, teacher. For example, “My hometown is in the northeast, with its dark, fertile soil, golden grains, red maple leaves, and white snow.” These descriptions make me feel his deep love for his hometown.”
  - **Creative Thinking Ability**:  This aspect evaluates students at the evaluation and creation levels of cognition. We designed open-ended questions to encourage students to express creative ideas, such as reinterpreting the theme of a passage or continuing a story. HE virtual students demonstrated proactive responses.
      - **Example Question**: “Now, let’s see whose imagination is the most creative. Based on the text, can you describe the image of the Yellow River in your own words?”
      - **Example Response**: “Sure, Teacher. In the poem, the Yellow River is depicted as a great mother who nourishes the entire Chinese land. Its surging waves symbolize the resilience and perseverance of the Chinese people, and its grandeur reflects their greatness and pride.”

**2. Research Focus on Modeling More Human-Like and Personalized LVSA**:

In educational scenarios, students display diverse traits and behaviors, with varying cognitive abilities. **Rather than focusing solely on advanced capabilities, we prioritize modeling and assessing human-like, diverse virtual student behaviors. The priority lies in evaluating whether the base model can support virtual student modeling in the whole teaching stage and give effective feedback for pre-service teachers.** Although comprehensive analysis of LVSA abilities is not the main focus at this stage, our experimental data suggests that LVSA already shows representative potential in complex cognitive behaviors, such as problem-solving and creative thinking.

In future work, we plan to further expand this aspect through the following steps:
- **Refining the Cognitive Evaluation Framework**: Drawing on cognitive taxonomy theories (e.g., Bloom's taxonomy), we aim to design more fine-grained question types to comprehensively cover virtual students' performance in complex cognitive behaviors.
- **Analyzing the Relationship Between Personality and Cognitive Abilities**: Investigate how virtual students with different personality traits perform in problem-solving and creative tasks, to further improve the modeling accuracy of LVSA.
- **Expanding Multidisciplinary Scenarios**: Extend evaluation tasks to other disciplines (e.g., solving math problems, designing scientific experiments) to test LVSA cognitive simulation capabilities across subject domains.

We greatly appreciate the reviewers' valuable suggestions regarding complex cognitive tasks. Through the evaluation dimension of question types, this study has preliminarily validated the potential of virtual students in problem-solving and creative thinking. In the future, we will conduct further in-depth analysis to continually enhance the simulation capabilities and educational value of LVSA.

---

### Author Response · Authors · 2024-11-25
**Rebuttal Summary**

We would like to express our heartfelt gratitude to all reviewers and the Area Chair (AC) for their thorough evaluation and thoughtful feedback on this study, which have greatly enhanced the academic value of our work and encouraged critical reflection. The reviewers have highlighted the following key contributions of our paper:

- **Research contributions align with ICLR's focus**: This study aligns with ICLR’s interest in innovative AI applications in education and social simulation, showcasing potential to inspire future research (Reviewer rtkz).
- **Originality and innovation**: By shifting from teacher-centered AI to virtual student simulations, this study pioneers a new direction in AI4Education and introduces a novel approach to personalized student modeling (Reviewers rtkz, yWBj).
- **Significant educational value**: The application of virtual students in pre-service teacher training addresses the lack of student interaction in traditional training, enhancing teachers' readiness, adaptability, and effectiveness (Reviewers wxDr, rtkz, yWBj).
- **Comprehensive and rigorous research methodology**: The SOE (Scenario-Object-Evaluation) pipeline integrates psychological theories and AI fine-tuning to systematically tackle virtual student modeling and evaluation with scientific rigor (Reviewers rtkz, 3QkY, yWBj).
- **Robust experimental validation**: A large-scale dataset and comprehensive experiments validate virtual student performance, with supplementary materials providing deeper insights (Reviewer 3QkY).
- **Clear and comprehensible writing**: The visualization of the SOE framework and structured experimental design make the research accessible and easy to understand (Reviewers wxDr, rtkz, 3QkY).
- **Distinct interdisciplinary characteristics**: This study integrates education, psychology, and AI, adding depth to AI applications in education (Reviewers rtkz, yWBj).

Based on the reviewers' invaluable suggestions, we have made extensive revisions to the paper, with all changes highlighted in blue. The key revisions are as follows:

**1. Expanded Experiments and Evaluations**

- **Objective Evaluation Experiments:** To complement subjective evaluation, four objective metrics—Text Token, Perplexity, TTR, and Sentiment Analysis—were introduced and added in Sec 5.5 and Appendix D.6, along with visualizations of the objective evaluation results (fig. 7, fig. A25-A28). These experiments collectively validate the personalization and human-like attributes of LVSA modeling.
- **ANOVA Experiments:** One-way and two-way ANOVA analyses were added to Sec 5.2, Appendix D.4.2, Table A7, and fig. A22. The results confirm significant statistical differences before and after fine-tuning, with no significant differences between virtual students and real students in human evaluations.
- **Fine-Grained Analysis of Virtual Student Responses Across Teaching Stages:** Specific examples of responses from five types of virtual students across different teaching stages and question types were added to Sec 4.3 and Appendix D.2. Using content analysis, these responses were examined in depth to reveal how they differentiate and reflect the unique personality traits of virtual students.

**2. Supplemented Details in the Manuscript**

- **Refined Abstract and Introduction:** Added description of objective evaluation: "we conduct multi-dimensional evaluation experiments that integrate both subjective human evaluations and objective metrics."
- **Refined Discussion on LC Personality:** In Sec 5.6 and Appendix D7.1, further detailed analyses of the LC personality are provided.
- **Adjustments to the Discussion Section:** In Sec 6, descriptions of the SOE framework’s generalization capability have been added, along with future directions.
- **Supplemented Description of Evaluation Metrics:** In Sec 5, a clarification was added: “The metrics are based on the probability of virtual responses being judged by humans or GPT-4 as resembling real student responses.”
- **Clarification of Real Students’ Role in the Turing Test:** In Sec 5.2, it was emphasized that “Instead of using a fixed ground truth, real students served as a control group to assess the human-like characteristics and role-playing effectiveness of LVSA in educational interactions.”
- **Clarification of Experimental Conclusions and Their Boundaries:** In Sec 5.4 and Sec 6, it was clarified that the current results are based on controlled experimental settings and human feedback.

**3. Introduction of Ethical Protection**

An ethical protection and data privacy statement was added in Appendix D.8.

We sincerely thank the reviewers for their valuable suggestions, which greatly improved our paper’s quality and impact. We believe the revised manuscript is now more rigorous and compelling. Below, we address three common concerns first, followed by point-by-point responses to specific comments.

---

> ### Comment · Reviewer_3QkY · 2024-11-27
> **Adjusted Score**
>
> I'd like to say thanks to the authors for the comprehensive improvement. I just checked other responses. Although my concern about model novelty and generalization is not solved yet, other concerns such as automatic evaluation and clarification are well-addressed. Therefore, I'd like to increase my score from 3 to 5.
>
> Good luck!

---

> ### Author Response · Authors · 2024-11-29
> **Response to Reviewer 3QkY's Concern About the Pipeline Novelty and Generalization (Part I: Innovation of the SOE Pipeline)**
>
> We sincerely thank the reviewer for the thorough reexamination of our research. Your deep insights into the intersection of artificial intelligence and educational sciences have provided invaluable suggestions. Your decision to raise our score from 3 to 5 is greatly encouraging to our team. **We highly value your feedback and have taken extra time to reorganize our responses to address your concerns in detail. We believe that the complexity of interdisciplinary research implies that researchers from diverse academic backgrounds may interpret "novelty" differently.** This has motivated us to refine our presentation to better align with a broad academic context and diverse research needs. Your feedback has been instrumental in helping us clarify the contributions of this study and precisely articulate its innovations, demonstrating its potential for future applications.
>
> Below is our detailed response to your comments:
>
> **1. Innovation of the SOE Pipeline**
>
> Regarding the innovation concerns raised by the reviewer, we fully understand and appreciate your feedback. You referred to the "model novelty and generalization" and considered the SOE pipeline as a traditional AI model. However, we would like to clarify that, unlike traditional AI models that primarily focus on algorithmic innovation, this study is situated within the AI for Social Science domain. **Research in this area particularly requires the integration of interdisciplinary theories with AI agents to develop more human-like research pipelines [1]. The inclusion of these theories is not only significant for data construction but also enhances the model's interpretability, fairness, and generalization ability [2], offering a foundation upon which future research can build.**
>
> This work follows this paradigm to address the challenges of modeling and evaluating virtual students by proposing the SOE pipeline. The core innovation of this pipeline lies not in algorithmic advances but in **its combination of interdisciplinary theories to provide a practical research paradigm. This includes the introduction of theoretical perspectives and the proposal of a mixed-method approach for evaluating the simulation quality of large language models, combining both subjective and objective evaluation methods.** This approach supports future research in virtual student construction, role modeling, and educational/social simulation experiments. Our pipeline integrates experimentally validated theories and methods across the design, modeling, and evaluation stages. **Furthermore, the modular design ensures the framework's scalability and generalization across diverse educational and social simulation environments.**
>
> In the subsequent response box, we will address each of your comments on the novelty of our work in detail.
>
> [1] Xu, R., Sun, Y., Ren, M., Guo, S., Pan, R., Lin, H., ... & Han, X. (2024). AI for social science and social science of AI: A survey. Information Processing & Management, 61(3), 103665.
>
> [2] Radford, J., & Joseph, K. (2020). Theory In, Theory Out: How social theory can solve problems that machine learning can’t. arXiv preprint arXiv:2001.03203.

---

> > ### Author Response · Authors · 2024-11-29
> > **Response to Reviewer 3QkY's Concern About the Pipeline Novelty and Generalization (Part II: Reasoning Ability)**
> >
> > ***Reviewer's Concern on Lack of Breakthrough in LLM’s Reasoning Ability***
> >
> > We appreciate the reviewer's attention to our work, particularly the thoughtful consideration of LLM reasoning abilities. We fully understand and agree with the importance of reasoning in large language models and virtual student modeling. In this study, **reasoning ability is systematically integrated into the SOE framework across various modules**.
> >
> > Below, we elaborate on how each module of the SOE pipeline addresses reasoning ability:
> >
> >   - **Scene Module (Subject Scene Selection and Reasoning Validation):**
> >
> > When designing the Scene module, we carefully considered the selection of subject areas. Unlike fields such as STEM, where reasoning is based on objective, rule-based facts, language-based disciplines emphasize reasoning grounded in semantics and contextual understanding (text comprehension). **This reasoning process is often facilitated by teachers through guided questioning at various stages of instruction, reflecting the Socratic method, where continuous questioning and dialogue aim to enhance students' reasoning abilities.** In contrast to STEM subjects, where fixed answers exist, language scenarios see students with different personality traits providing varied interpretations and expressions. **This aligns with the goal of this study to model personalized and human-like virtual students and is a significant manifestation of the challenges in modeling and evaluation.** Therefore, in this module, we preliminarily validated the reasoning ability of base models through text comprehension tasks and selected four base models suitable for this educational context.
> >
> >   - **Object Module (Instruction Fine-Tuning Data and Reasoning Representation):**
> >
> > In the Object module, we constructed a scientifically grounded teacher-student dialogue fine-tuning dataset within a reasonable theoretical framework. **A key focus of this process was the transition from surface-level imagery to deep metaphorical reasoning, an essential element in language-based contexts.** For example, in the dialogue example shown in Figure A8, the teacher asks, “In the final paragraph, 'The two goldfish are still swimming, and the chrysanthemums are blooming,' what do you think the chrysanthemums and the goldfish symbolize?” Here, “chrysanthemums” and “goldfish” serve as surface imagery, requiring students to use context and the teacher’s explanation to infer their deeper metaphorical significance. This process represents the embodiment of reasoning ability based on text comprehension. In constructing the dataset, we not only focused on students’ basic understanding but also systematically built the incremental development of their reasoning abilities. **Teacher questions across different teaching stages and problem types guided students from shallow cognition to deeper reasoning, providing virtual students with a rich array of language and reasoning scenarios.**
> >   - **Evaluation Module (Fine-Grained Evaluation and Reasoning Validation):**
> > In the Evaluation module, **fine-grained assessments across different teaching stages and problem types further validated the reasoning abilities of virtual students.** For example:
> >       -  **Pre-Lesson Introduction Phase**: Virtual students mainly provide simple answers based on understanding and memory. For instance, when asked, "Today, we’re going to study Shi Tiesheng’s Autumn Thoughts. Does anyone know who Shi Tiesheng is?"  a student responds, "Shi Tiesheng, he's a, a writer, right, right, he wrote I and the Temple of Earth." Such responses validate the virtual students’ basic grasp of the text and background knowledge.
> >       -  **Class Practice Phase**: The questions are designed to encourage deeper reasoning based on text comprehension. For example, when asked, "The two goldfish are still swimming, and the chrysanthemums are blooming, what do you think the chrysanthemums and the goldfish symbolize?" The student responds, "Um, the chrysanthemums probably represent the mother, the mother's love. And the goldfish, maybe they represent, uh, life, life continuing." These responses require students to engage in deeper reasoning and metaphorical interpretation based on context.
> >
> > Through this fine-grained evaluation, **we can clearly verify whether virtual students can meet the cognitive demands of different teaching stages and provide more appropriate feedback for pre-service teachers.** This also highlights the generalization capabilities of the SOE pipeline in educational contexts.

---

> > > ### Author Response · Authors · 2024-11-29
> > > **Response to Reviewer 3QkY's Concern About the Pipeline Novelty and Generalization (Part III: The Role of Data in the SOE Pipeline)**
> > >
> > > ***Reviewer's Concern on the Framework Appears as a Data Processing Pipeline***
> > >
> > > We understand the reviewer's concern about the SOE pipeline being perceived as a "data processing pipeline" and would like to clarify the core contribution of this framework. While we acknowledge the critical role of data processing within the SOE framework, **it is essential to highlight that data-driven approaches have become foundational to AI development, particularly in the AI for Science domain**, where data design, processing, and validation are integral to scientific applications [3]. **As articulated by Li Feifei's team at Stanford University in their 2022 Nature Machine Intelligence article, AI is shifting from a model-driven to a data-driven approach, marked by the introduction of ImageNet. The creation of suitable datasets and data processing workflows has become one of the greatest challenges in AI development and evaluation [4]**. Hence, the data-driven research paradigm is not only applicable to AI algorithm development but also forms the core logic for AI applications in scientific fields.
> > >
> > > In this context, the inclusion of detailed data processing steps in the design of the SOE pipeline reflects the scientific rigor that underpins the pipeline. It is understandable that the reviewer sees our work as a "data processing pipeline." However, **we emphasize that the core contribution of the SOE framework lies not only in the dataset itself but in the introduction of a data-driven framework that integrates the practical needs of education to propose a novel pipeline and validation method.**
> > >
> > > In other words, **the dataset serves as a necessary foundation for creating the experimental environment, not the central contribution of the research.** The design of the SOE pipeline goes beyond simply stitching together various data processing modules. Instead, **it addresses the complexities and diversity of virtual student modeling and evaluation through the modular design of Scene, Object, and Evaluation.** Each module, from data construction to experimental validation, has been carefully designed and tested. The core goal is to operationalize and scientifically model personalized virtual students and multi-dimensional evaluations in educational contexts through an interdisciplinary research platform.
> > >
> > > [3] Xu, Y., Wang, F., An, Z., Wang, Q., & Zhang, Z. (2023). Artificial intelligence for science—bridging data to wisdom. The Innovation, 4(6).
> > >
> > > [4] Liang, W., Tadesse, G. A., Ho, D., Fei-Fei, L., Zaharia, M., Zhang, C., & Zou, J. (2022). Advances, challenges and opportunities in creating data for trustworthy AI. Nature Machine Intelligence, 4(8), 669-677.

---

> > > > ### Author Response · Authors · 2024-11-29
> > > > **Response to Reviewer 3QkY's Concern About the Pipeline Novelty and Generalization (Part IV: The Role of  Social Science Theories in the SOE Pipeline)**
> > > >
> > > > ***Reviewer’s Concern on Theoretical Concepts Do Not Contribute to the Core Model***
> > > >
> > > > We understand the reviewer's concern regarding the application of theoretical foundations to the core contribution of this research.
> > > >
> > > > Below, we respond to your concerns by explaining the motivations, challenges, and core contributions of the theoretical research incorporated into this study:
> > > >
> > > > - **Motivation for Introducing Theoretical Frameworks**
> > > > As previously mentioned, the integration of interdisciplinary theories not only plays a crucial role in constructing the dataset but also enhances the model's interpretability, fairness, and generalization ability. The theoretical frameworks used in this study, including Bloom's Taxonomy of Cognitive Domains, the Big Five Personality Theory, and educational foundation theories, are all well-established, widely accepted, and scientifically validated. **With nearly 60 years of academic development, these theories are universally applicable and scientifically grounded, making them ideal for meeting the needs of the SOE framework across multidisciplinary, multicultural, and diverse educational contexts.** They are especially crucial for avoiding cultural bias and improving the framework’s generalization capacity.
> > > >
> > > > - **Challenges in Integrating Theoretical Frameworks**
> > > >   - **Balancing AI Paradigms and Social Science Theories**: Interdisciplinary research requires balancing the quantitative AI paradigms with the qualitative research methods of social science theories. **AI models require quantifiable and computable data, whereas social science theories necessitate more interpretive and analytical approaches.** In designing the SOE pipeline, we carefully balanced these two distinct paradigms, **ensuring the computational requirements of AI were met while respecting the interpretability of social science theories.**
> > > >   - **Transforming Unquantifiable Educational Theories into Operational Frameworks**: Many educational theories, such as cognitive learning theories and personality theories, are abstract and difficult to quantify. Therefore, we needed to find appropriate frameworks and methods to operationalize these theories for application in virtual student modeling.
> > > >   - **Selecting Theories for Broad Applicability**: The fields of education and psychology are rich with theories, each offering some degree of value. However, choosing the most universally applicable theories that align with the goals of the model is crucial. We selected those theories that could be widely applied across different disciplines and cultural backgrounds, ensuring the SOE framework’s broad applicability and generalization capacity.
> > > >
> > > > - **Core Contributions of Theoretical Concepts in the SOE Pipeline**
> > > >   - **Bloom's Taxonomy**: As one of the most classic and universally applicable cognitive theories, Bloom's Taxonomy provides a clear hierarchical framework for modeling student learning progress across various educational settings. **This theory enhances the interpretability and generalizability of virtual student cognition modeling.**
> > > >   - **Big Five Personality Theory**: The Big Five personality theory, applied in virtual student modeling, helps avoid cultural bias and is widely applicable across diverse student backgrounds. This theory allows for a more realistic simulation of student personality traits, **making virtual student behavior more diverse and flexible, thus improving personality diversity and model interpretability.**
> > > >   - **Educational Foundation Theories**: In terms of teaching theory, whether in the design of teaching stages or question types, we selected theories that are adaptable to multi-disciplinary and multicultural contexts, **ensuring the disciplinary suitability of the SOE pipeline.**
> > > >
> > > > The theoretical concepts introduced in this study, which have been rigorously tested and analyzed over nearly 60 years, have been scientifically validated for their universality and applicability. **By integrating interdisciplinary theories from education and psychology, while accommodating both the AI quantitative research paradigm and the qualitative research paradigm of social science theories, we have ensured that the pipeline is both operational and scientifically grounded.** This integration also enhances the model’s interpretability, generalization, and cross-disciplinary adaptability.
> > > >
> > > > Therefore, **one of the core contributions of the SOE framework is that it provides scientific theoretical support for virtual student modeling through interdisciplinary theories.** These theories are critical in addressing the complexity of virtual student modeling and enhancing the model’s depth and breadth of application. We firmly believe that this framework will provide an important theoretical foundation and methodological guidance for future research in educational AI and virtual student modeling.

---

> > > > > ### Author Response · Authors · 2024-11-29
> > > > > **Response to Reviewer 3QkY's Concern About the Pipeline Novelty and Generalization (Part V: Generalization of the SOE pipeline regarding dataset smilarity)**
> > > > >
> > > > > **2. Generalization of the SOE Framework**
> > > > >
> > > > > We fully understand the reviewer's concern regarding the lack of validation using multiple datasets. While multi-dataset validation is indeed a key step in testing the generalization and robustness of AI models, the SOE pipeline proposed in this study is not a traditional AI algorithm model. Rather, **it is an operational framework designed for applications in educational internships and social simulation experiments**. In our study, **generalization focuses more on the adaptability and flexibility of the framework, particularly in supporting teaching internships across different educational stages and providing personalized, human-like feedback for various question types.** These capabilities have been validated through our experiments, and the generalization aspect was addressed in response to common concern 1.
> > > > >
> > > > > The reviewer’s suggestion to validate the framework using multiple datasets is more aligned with viewing the SOE pipeline as an AI algorithm model, which diverges from the objectives of our research. **Our framework is not aimed at optimizing a single algorithm but rather serves as a comprehensive research tool intended to provide solutions across interdisciplinary and educational contexts**. While multi-dataset validation is crucial for AI algorithm models, it is not the core of our work. Nonetheless, we are happy to engage further with the reviewer’s suggestions and address each point in detail:
> > > > >
> > > > > ***Reviewer’s Concern on Similarity of the Dataset to Existing Educational Datasets***
> > > > >
> > > > > The reviewer pointed out that our dataset "is just similar to most education datasets. Maybe the language is different in Chinese. However, there are also many Chinese dataset available."
> > > > >
> > > > > - We acknowledge the existence of various educational datasets, such as the SQuAD reading comprehension dataset created by Stanford [5], the NCES dataset by the U.S. National Center for Education Statistics [6], and the Chinese evaluation dataset C-eval [7]. However, **these datasets typically evaluate model competency in subject-specific contexts or predict students' academic performance. They often focus on fine-tuning large educational models resembling experts, which does not align with our study's goals.** The dataset we have built is not only for educational context evaluation but also supports the personalized modeling of virtual students, incorporating cognitive levels and personality traits.
> > > > >
> > > > > - Additionally, while there are many Chinese dialogue datasets [8-10], they primarily consist of conversations from films or social media, involving adults rather than adolescents, and thus do not match our educational context. **The dataset in our study, built on a scientific theoretical framework, addresses the detailed needs of personalized modeling and ensures that teacher-student dialogues avoid cultural biases while covering all stages of teaching.** This design better supports comprehensive teaching internships for pre-service teachers.
> > > > >
> > > > > [5] Rajpurkar, P., Jia, R., & Liang, P. (2018). Know what you don't know: Unanswerable questions for SQuAD. arXiv preprint arXiv:1806.03822.
> > > > >
> > > > > [6] NCES. https://nces.ed.gov/
> > > > >
> > > > > [7] Huang, Y., Bai, Y., Zhu, Z., Zhang, J., Zhang, J., Su, T., ... & He, J. (2024). C-eval: A multi-level multi-discipline chinese evaluation suite for foundation models. Advances in Neural Information Processing Systems, 36.
> > > > >
> > > > > [8] Chinese-chatbot-corpus. https://github.com/codemayq/chinese-chatbot-corpus
> > > > >
> > > > > [9] Wang, Y., Ke, P., Zheng, Y., Huang, K., Jiang, Y., Zhu, X., & Huang, M. (2020). A large-scale chinese short-text conversation dataset. In Natural Language Processing and Chinese Computing: 9th CCF International Conference, NLPCC 2020, Zhengzhou, China, October 14–18, 2020, Proceedings, Part I 9 (pp. 91-103). Springer International Publishing.
> > > > >
> > > > > [10] Zhang, Z., Li, J., Zhu, P., Zhao, H., & Liu, G. (2018). Modeling multi-turn conversation with deep utterance aggregation. arXiv preprint arXiv:1806.09102.

---

> > > > > > ### Author Response · Authors · 2024-11-29
> > > > > > **Response to Reviewer 3QkY's Concern About the Pipeline Novelty and Generalization (Part VI: Generalization of the SOE pipeline regarding multiple datasets)**
> > > > > >
> > > > > > ***Reviewer’s Concern on Demonstrating Generalization Using Multiple Datasets***
> > > > > >
> > > > > > The reviewer mentioned that "to really demonstrate the generalization ability, it is always more convincing to run a real experiment on additional datasets, instead of simply putting it into future work.”
> > > > > >
> > > > > > We understand the reviewer’s suggestion to further validate the pipeline’s generalization using multiple datasets. **Indeed, we initially sought publicly available datasets. However, we found that datasets with teacher-student dialogues suitable for Big Five personality modeling are extremely rare**. Given our study’s need for fine-grained personalized modeling, including personality modeling, cognitive levels, and problem-type designs, we discovered that very few existing datasets fully meet these needs. Furthermore, many educational datasets are subject to ethical protection protocols, and some are restricted to internal research use only, preventing us from using them for validation.However, **our research does not rely solely on the diversity of datasets to demonstrate generalization. Instead, it focuses on a data-driven framework and methodology with cross-context, cross-disciplinary adaptability.**
> > > > > >
> > > > > > Moreover, we want to emphasize that cross-dataset generalization testing is more concerned with evaluating the language transferability of large language models, a capability that has already been extensively validated in the literature [11-12]. We believe **the core challenge in our work lies in overcoming the hierarchical understanding capabilities of virtual students in different scenarios. This hierarchical understanding requires virtual students to progress from superficial comprehension to deeper contextual understanding, ultimately interpreting the emotional themes conveyed in texts—an understanding that relates closely to their personality traits.** In our custom-built dataset, we have already conducted systematic experimental validations centered around this core need, demonstrating the hierarchical understanding abilities of virtual students in text comprehension tasks and across various teaching stages. For example, students with high extraversion and agreeableness exhibit more prominent comprehension abilities, while students with low conscientiousness and openness show mixed responses (see Appendix D.2.2). **We have demonstrated that the SOE pipeline can support the expression of personalized, hierarchical understanding abilities in our dataset.** For datasets in different languages, the language transferability of large language models can better adapt and optimize these abilities.
> > > > > >
> > > > > > Through the above points, we hope to clarify the reviewer's concerns about the dataset and generalization capabilities. **We emphasize that the SOE pipeline's objective is not to innovate AI algorithm models but to serve as an interdisciplinary, multi-scenario educational research tool.** It provides scientifically grounded, actionable solutions across various educational stages, question types, and subject backgrounds. The diversity of datasets is not the core focus of our research; **what truly matters is the framework’s adaptability and flexibility and whether it can effectively support personalized modeling and evaluation tasks for virtual students.**
> > > > > >
> > > > > > We again thank the reviewer for their valuable feedback. We will continue to refine and strengthen the framework’s generalizability and applicability. Should there be any further questions, we welcome continued discussion.
> > > > > >
> > > > > > [11] Chowdhery, A., Narang, S., Devlin, J., Bosma, M., Mishra, G., Roberts, A., ... & Fiedel, N. (2023). Palm: Scaling language modeling with pathways. Journal of Machine Learning Research, 24(240), 1-113.
> > > > > >
> > > > > > [12] Ahuja, K., Diddee, H., Hada, R., Ochieng, M., Ramesh, K., Jain, P., ... & Sitaram, S. (2023). Mega: Multilingual evaluation of generative ai. arXiv preprint arXiv:2303.12528.

---

### Meta-Review · Area_Chair_pg1R · 2024-12-22

**Metareview:**

This paper aims to model LLM-based Virtual Student Agents (LVSA) that mimic the behaviors and personalities of early adolescent students in educational settings. Reviewers recognized that this paper studies a quite interesting problem and the new framework is well motivated. However, reviewers also raised some major concerns on contributions, technical details, model justifications, evaluations, etc. Although some of the concerns have been addressed by the authors during the rebuttal and discussion stage, reviewers still found that some issues cannot be easily addressed by a minor revision, such as novelty and generalization ability. Overall, the current version of this work is not ready for publication at ICLR.

**Additional Comments On Reviewer Discussion:**

Reviewers raised some major concerns about novelty and contributions, technical details, model justifications, evaluations, etc. The authors have provided very detailed responses during the rebuttal stage, which have addressed some of the concerns. However, some major issues still remain, such as novelty and generalization ability.

---

### Decision · Program_Chairs · 2025-01-22

Reject